# Quantum Field Theory Anomalies
# in Condensed Matter Physics[*]

R. Arouca[1,2,3], Andrea Cappelli[4] and T. H. Hansson[5]

**1** Department of Physics and Astronomy, Uppsala University, Uppsala, Sweden
**2** Institute for Theoretical Physics, Utrecht University, the Netherlands
**3** Instituto de Física, Universidade Federal do Rio de Janeiro, Brazil
**4** INFN, Sezione di Firenze, Italy
**5** Fysikum, Stockholm University, Stockholm, Sweden

April 14, 2022

## Abstract

**We give a pedagogical introduction to quantum anomalies, how they are calculated using various methods, and why they are important in condensed matter theory. We discuss axial, chiral, and gravitational anomalies as well as global anomalies. We illustrate the theory with examples such as quantum Hall liquids, Fermi liquids, Weyl semi-metals, topological insulators and topological superconductors. The required background is basic knowledge of quantum field theory, including fermions and gauge fields, and some familiarity with path integral and functional methods. Some knowledge of topological phases of matter is helpful, but not necessary.**

[*]Partially based on lectures given at the Spring 2020 Delta ITP Advanced Topics Course at Utrecht University and at the PhD school of the University of Florence.
email: rodrigo.arouca@physics.uu.se, andrea.cappelli@fi.infn.it, hansson@fysik.su.se

# 1   Introduction

## 1.1   Overview

The history of anomalies in quantum field theories (QFT) is long and interesting. In this introductory section we shall mention some of the most important concepts and give a brief outline of the lectures. The account is far from complete and will zoom in on the parts that are of most importance for condensed matter physics.

A symmetry in field theory is said to be anomalous if one cannot quantize the theory without violating it. If a continuous symmetry has an anomaly the corresponding current is no longer conserved at the quantum level. Such non-conservation can have important consequences in that physical processes that are forbidden by selection rules derived from the symmetry in question, are in fact allowed because of an anomaly. Historically, the first instance was the axial anomaly, whose existence explains the rapid decay of the neutral pi-meson into two photons, $\pi^0 \to \gamma\gamma$, in agreement with experiments. In this and other cases, anomalies are important to understand the physics described by well-defined QFTs, as they leave characteristic fingerprints in the form of exact and universal effects.

However, if an anomaly violates the conservation of a current that is coupled to a dynamical gauge field, it destroys the gauge invariance, and usually makes the whole theory inconsistent. This is the case of the chiral gauge coupling of weak interactions in the Standard Model of elementary particles, where anomaly cancellation gives important constraints on the particle content.

Anomalies often occur in QFTs with fermions, and in a strict sense only in continuum theories with massless (gapless) particles. However, if a gapless system has an anomaly, this affects the physics also when the system is perturbed to develop a gap. A continuum QFT has an infinite number of degrees of freedom even in a finite volume. As a consequence, when calculating physical quantities, typically in perturbation theory, one often encounters infinities that must be regulated by introducing a high-energy (short-distance) cut-off. This can be done in various ways, such as dimensional regularization, introducing Pauli-Villars regulators or

formulating the theory on a lattice. In important cases, there are symmetries that are violated by any regularization – they are anomalous.

This clash between symmetry and regularization can be seen in the perturbative loop expansion of QFTs involving fermions coupled to external gauge fields. Returning to the important case of the pion decay as an example, the axial current, $J_5^\mu$, which is directly related to the $\pi^0$ field, is no longer conserved but satisfies $\partial_\mu J_5^\mu = c\, e^2 \vec{E} \cdot \vec{B}$ where $\vec{E}$ and $\vec{B}$ are the electromagnetic fields related to the emitted photons. The anomalies are, however, very peculiar loop effects. Although a regularization is needed for the calculations, the results are finite at one loop level and do not get corrections from higher loops. In the example we just gave, this amounts to the constant $c$ having the exact value $1/(2\pi^2)$.

Anomalies are usually associated with relativistic fermions, as in the case of the decay of the $\pi^0$ meson, but Lorentz invariance is not a prerequisite in condensed matter applications, although in many cases a relativistic dispersion relation provides a simple model that captures the essence of the system in question. One example where anomalies are important is the Luttinger liquid which describes electrons in various one-dimensional systems such as nanowires. Other examples, of great current interest, are the Weyl semi-metals, which are three-dimensional materials where bands cross to form "Dirac cones".

Later we will discuss both these examples, but it is now fair to ask why anomalies are at all relevant in these systems which have a finite (although very large) number of degrees of freedom? The quick answer is that in order to give a consistent description of the original many-body quantum mechanical system, the effective continuum theory must be such that the anomalies cancel. More precisely, it is only the total anomalies in conserved currents that have to vanish, meaning that contributions resulting from different fields should cancel. For example, in the case of the Luttinger liquid there are different currents describing right and left moving electrons, and only when taken together do the anomaly in the electric current vanishes.

In condensed matter physics, many experimental techniques are based on subjecting a system to classical electric and magnetic fields and measuring the response. An important theoretical tool to describe this is the effective response action, $S_{\text{eff}}[A]$, which is obtained by coupling the system to a background gauge potential $A_\mu$ and then integrating out the matter fields. Various response functions can then be determined from $S_{\text{eff}}[A]$. It turns out that in several important cases, the low-energy response is robust, and in a certain sense "geometric", *i.e.* independent of the detailed dynamics. The deep reason for this is that the response is determined by anomalies which are related to topological invariants that, by nature, are insensitive to small deformations of the Hamiltonian including the addition of disorder. The integrated anomaly in four dimensions $\int \vec{E} \cdot \vec{B}$ is an example of topological quantity.

Another concept that you will encounter in these lectures is that of an effective low energy theory. These are dynamical QFTs that encode not only the response to external fields, but also the effective low energy dynamics. An early example from high energy physics, is the chiral pion theory that captures the low energy phenomenology of quantum chromodynamics. An important example in condensed matter physics that we shall discuss in some detail is the hydrodynamic theory of quantum Hall liquids, where the effective degrees of freedom are gauge fields. We shall also encounter a special kind of effective field theories that only encode topological response, these are the topological quantum field theories (TQFT). Anomalies in a microscopic theory, put severe restrictions on the low energy theories, in that these have to reproduce the anomalies at the low energy scale – these are the t' Hooft anomaly matching conditions.

Most condensed matter systems studied in the laboratory have boundaries,[1] and anomalies are often crucial in order to understand the connection between bulk and boundary. It turns

---

[1]Exceptions do occur in one and two dimensions where one can have closed loops and surfaces.

out that the boundary might be described by an anomalous theory, which by itself would be inconsistent, typically by not conserving the electrical current. This is, however, compensated by a flow of current from the bulk. The bulk theory is not anomalous, but has an action that violates gauge invariance, and thus current conservation, at the boundary. This cancellation of a quantum anomaly by a classical current in one extra dimension is called "anomaly inflow" or the "Callan-Harvey mechanism".

The first, and most well known, example in which anomaly inflow takes place is the quantum Hall effect: The system is gapped in the two-dimensional bulk but hosts low-energy boundary excitations that are massless one-dimensional fermions. This system also provides a paradigm for understanding other topological phases of matter, in different dimensions and possessing different symmetries. Their gapless boundary excitations are again characterized by specific forms of anomalies and corresponding topological invariants.

As a consequence, anomalies provide a classification of topological phases of matter. This can be compared to the "ten-fold way" classification based on the properties of free fermions in band systems, and in many cases, as for topological insulators, the two approaches match. In others, as for topological superconductors, they differ in the presence of fermion interactions. Since anomalies can be exactly determined even in this case, they provide the correct characterization, and thus are very important for getting a complete understanding of topological phases of matter.

There are other ways, beside perturbation theory, to calculate anomalies and some will be presented in these lectures. A very important method is based on path integrals, where the anomalies appear since the integration measure cannot be regulated in a way that respects all symmetries of the classical Lagrangian. The simplest, and most intuitive, approach to anomalies is by regularizing the infinite Dirac sea of relativistic fermions, and this gives a direct way to understand why anomalies involve simple numerical factors like the $1/(2\pi^2)$ mentioned earlier. This is much harder to comprehend in the context of perturbation theory. A closely related, but more general, method is via the study of the spectral flow of eigenvalues of the Dirac operator: the related index theorems reveal a connection to certain topological invariant quantities made out of the gauge field, as well as the background metric if the systems are put on a curved manifold.

Concerning the latter, one would think that curved spaces would only be of interest in high energy physics or string theory, but they are also relevant in condensed matter physics. As originally shown by Luttinger, thermal transport can be related to gravitational response. This is an important tool for characterizing topological phases of matter, especially in cases where there is no conserved $U(1)$ current, as for superconductors.

The anomalies we talked about so far are all about non-conservation of currents related by Noether's theorem to continuous symmetries. In addition to these, there are global anomalies that can violate discrete symmetries like parity or time reversal. In these cases, the partition function is not invariant under the discrete symmetries, and also not under "large" gauge transformations. Important examples are topological insulators and superconductors in three dimensions. We shall discuss global anomalies in the last three sections.

## 1.2 Outline of the lectures and reading instructions

The chiral anomaly is introduced in Sect. 2, through the discussion of $(1 + 1)$-dimensional fermions and the Luttinger model – how it originates from the regularization of the Dirac sea, its relation to the monopole charge and the associated flow of energy eigenvalues.

Sect. 3 shows anomalies at work in the fractional quantum Hall effect. We discuss the bulk effective field theory of Chern-Simons type, and derive the conformal field theory of edge excitations, starting from free fermions, and then including interactions with the help

of bosonic currents (chiral Luttinger liquid). We explain how the anomaly characterizes the Hall current, and thus make a case for the exactness of the Hall conductance. Next we show that a gravitational anomaly can occur in the presence of a metric background. This implies a nonconservation of the matter current that corresponds to heat flow, and an exact result for the thermal Hall conductivity.

In Sect. 4 we derive anomalies by a perturbative expansion in the gauge coupling. This is the historical approach in particle physics, and we briefly discuss the application to $\pi^0$ decay. The spectral flow argument is extended to higher dimension, and in Sect. 5 we introduce the Fujikawa method for calculating anomalies in the path-integral formulation. The section is concluded with some general comments on the importance of anomalies.

The use of anomalies in three-dimensional condensed matter systems is reviewed in Sect. 6, which describes anomaly governed low-energy properties of Dirac and Weyl materials such as the chiral magnetic effect and other electromagnetic responses.

In Sect. 7, we describe general mathematical aspects of gauge and gravitational anomalies in even dimensions by explaining the relevant topologically invariant quantities that are involved. These results are later used in Sect. 10. Sect. 8 provides an extended analysis of condensed matter systems on curved backgrounds, and in particular the role of gravitational anomalies in the fractional quantum Hall effect. Sect. 7 and 8 are more technical and require some familiarity with curves space calculus, which is briefly reviewed in App. E, as a help for the reader.

In Sect. 9 we discuss effective theories for symmetry protected topological states, dwelling in some detail on topological insulators and topological superconductors in three space dimensions, and explaining the nature of their fermionic boundary excitations.

The study of these and other topological states requires addressing global anomalies, a concept that is explained in Sect. 10, in terms of the non-invariance of the partition function under "large" gauge transformations and discrete symmetry transformations. We analyze several examples involving topological insulators and superconductors in various dimensions.

Sect. 11 explains the anomalous origin of the gravitational Chern-Simon action that has physical applications in the quantum Hall effect and other systems. Sect. 10 and 11 are the most technical parts of the lectures, but we tried to simplify them as much as possible, limiting the presentation to the key physical ideas.

Finally, the concluding section gives a list of the most important points made in the lectures.

A number of appendices provide technical discussions that complement the analysis in the text: App. A summarizes our spinor conventions; App. B describes the effective theory of general fractional quantum Hall states; App. C details some steps in the derivation of the three-dimensional spectral flow. App. D spells out the two definitions of anomalous currents that are possible in the presence of both vector and axial vector gauge fields. This rather technical aspect has nonetheless important physical applications. The last, App. E provides an overview of differential geometry and curved space formulas that are used in the text, complemented with brief discussions of the main concepts behind them.

We emphasize that this is not a review paper, but a set of lecture notes covering a wide range of topics, with anomalies as the overreaching theme. Thus we have not attempted to provide a comprehensive bibliography. In addition to fundamental works, we have referenced accessible review papers and books, and also some recent work that points students to new and exciting directions.

This rather long exposition can be read at different levels of ambition and with different focus. A first reading can be limited to Sect. 2-6. Starting from the simple case of $(1 + 1)$-dimensional fermions and the application to edge excitations in the quantum Hall effect, one can progress to perturbative and path-integral analyses of the anomaly. Then, the application to Weyl semi-metals is rather straightforward. In the table below, we show how the different

sections are interconnected to help the readers to select the parts of most interest to them.

| Section | 2 | 3 | 4 | 5 | 6 | 7 | 8 | 9 | 10 | 11 |
|---|---|---|---|---|---|---|---|---|---|---|
| Needs sections | | 2 | 2 | 2, 4 | 2, 4 | 4, 5 | 2, 3 | 2, 3 | 2, 3, 4, 7, 9 | 2, 3, 8, 10 |

## 1.3 List of acronyms

| | |
|---|---|
| APS | Atiyah-Patodi-Singer (theorem) |
| BdG | Bogoliubov-de Gennes (Hamiltonian) |
| C | Charge conjugation |
| CFT | Conformal (invariant) Field Theory |
| CS | Chern-Simons (action) |
| D | Spacetime dimension |
| d | Space dimension |
| $(3+1)$ | Three space dimensions and time (Minkowskian) |
| FQH, FQHE | Fractional quantum Hall (effect) |
| P | Parity (symmetry) |
| PCAC | Partially conserved axial current |
| QCD | Quantum Chromodynamics |
| QFT | Quantum field theory |
| QH, QHE | Quantum Hall (effect) |
| SPT | Symmetry protected topological (phase) |
| TI | Topological insulator |
| T, TR | Time reversal (symmetry) |
| TO | Topological ordered (phase) |
| TQFT | Topological quantum field theory |
| TSC | Topological superconductor |
| WSM | Weyl semi-metal |

# 2 The axial anomaly in (1+1) dimensions and the infinite hotel

In this section we start by discussing the field theory for fermions in (1+1) dimensions and showing how to regularize the infinities related to the filled Dirac sea. We then derive the axial anomaly, first by a direct calculation and then by a simple and intuitive method based on spectral flow. Next we apply what we learned to an important physical system, namely the Luttinger liquid, and we end the section with a first look at the topological aspect of anomalies, to which we shall return in greater detail later.

## 2.1 Two dimensional fermions

Our first, and simplest, example of an anomalous theory is given by free relativistic fermions in $(1+1)$ dimensional Minkowski space,

$$\mathcal{L}_f = \bar{\psi}(x) i \slashed{D} \psi(x) = \bar{\psi}(x) \gamma^\mu (i \partial_\mu + e A_\mu) \psi(x), \tag{2.1}$$

where $x^\mu = (t, x)$, $\mu = 0, 1$, is the space-time coordinate, $A_\mu = (A_0, A_x)$ the electromagnetic potential and we set $\hbar = c = 1$. Throughout these lectures the covariant derivative $D_\mu$ is set for the electron charge $-e$ and the positive constant $e$ is incorporated in the definition of $A_\mu$. Our conventions for the Dirac equation are summarized in App. A.

This Lagrangian is invariant both under the vector gauge transformation,

$$\psi(x) \to e^{i\lambda(x)}\psi(x), \qquad A_\mu \to A_\mu + \partial_\mu\lambda(x), \qquad (2.2)$$

and the global axial transformation,

$$\psi(x) \to e^{i\gamma^3\xi}\psi, \qquad (2.3)$$

where $\gamma^3 = \gamma^0\gamma^1$, obeying $\gamma^{3\dagger} = \gamma^3$ and $(\gamma^3)^2 = 1$. By Noether's theorem, it follows that both the vector current $j_\mu = \bar\psi\gamma_\mu\psi$ and the axial current $j_\mu^5 = \bar\psi\gamma_\mu\gamma^3\psi$ are conserved.[2] Since $\{\gamma^3, \gamma^\mu\} = 0$ it follows that if $\psi$ solves $i\slashed{D}\psi = 0$, so does $\gamma^3\psi$. We can thus label the eigenfunction by their eigenvalue of $\gamma^3$ which is just a sign, i.e. $\gamma^3\psi_\pm = \pm\psi_\pm$, with,

$$\psi_\pm = P_\pm\psi = \frac{1}{2}(1 \pm \gamma^3)\psi_\pm, \qquad (2.4)$$

where $P_\pm$ are projectors. Written in these components, the Lagrangian decouples into two independent expressions,

$$\mathcal{L}_f = \psi_+^\dagger(iD_0 + iD_x)\psi_+ + \psi_-^\dagger(iD_0 - iD_x)\psi_-, \qquad (2.5)$$

which associates $\psi_+$ and $\psi_-$ to right and left moving excitations, respectively. These components are called "chiral", borrowing the name from the 4D case, where they actually correspond to the different property of "handedness" of particles, i.e. whether the spin in parallel or antiparallel to the momentum [1].

To quantize, we choose the gauge condition $A_0 = 0$ and read off the equal time anticommutation relations from the time-derivative terms,

$$\{\psi_+(x, t), \psi_+^\dagger(y, t)\} = \{\psi_-(x, t), \psi_-^\dagger(y, t)\} = \delta(x - y), \qquad (2.6)$$

and the corresponding Hamiltonian is,

$$H = \int dx \left[ -\psi_+^\dagger(i\partial_x + A_x)\psi_- + \psi_-^\dagger(i\partial_x + A_x)\psi_- \right]. \qquad (2.7)$$

Now define the theory on a circle with circumference $L = 2\pi R$, and impose antiperiodic and periodic boundary conditions on the fermions and the gauge field, respectively,[3]

$$\psi_\pm(0, t) = -\psi_\pm(L, t), \qquad A_x(0, t) = A_x(L, t). \qquad (2.8)$$

Since there is no magnetic field in one space dimension, any spatial dependence in $A_x(t, x)$ can be removed by a suitable gauge transformation [2]. Note also that a spatially constant field $A_x(t) = \text{const.}$ does have physical significance since in a gauge $A_0 = 0$, the electric field is simply $E^x = F_{0x} = \partial_t A_x$. From (2.7) we have $\mathcal{E}^{(\pm)} = \pm(p - A_x)$, and since $p$ is quantized on the circle, we get the spectrum,

$$\mathcal{E}_{n_+}^{(+)} = (n_+ - \frac{1}{2})\frac{1}{R} - A_x, \qquad \mathcal{E}_{n_-}^{(-)} = -(n_- + \frac{1}{2})\frac{1}{R} + A_x, \qquad n_\pm \in \mathbb{Z}, \qquad (2.9)$$

where the sign of the half unit offset is chosen for future convenience. To define the ground state we must as usual fill up the Dirac sea to some chemical potential (or Fermi energy) as shown in the left panel of Fig. 1 (assuming $A_x(t) = 0$ at $t = 0$).

Let us define the charge operators for left and right fermions, or alternatively the vector and axial charges,

$$\hat{Q}_\pm = \int_0^L dx\, \bar\psi_\pm\gamma^0\psi_\pm, \qquad \hat{Q} = \int_0^L dx\, \bar\psi\gamma^0\psi, \qquad \hat{Q}_5 = \int_0^L dx\, \bar\psi\gamma^0\gamma^3\psi, \qquad (2.10)$$

where $\hat{Q} = \hat{Q}_+ + \hat{Q}_-$ and $\hat{Q}_5 = \hat{Q}_+ - \hat{Q}_-$.

---

[2]We shall use the notation $j_\mu^5 = \bar\psi\gamma^\mu\gamma^{2n+1}\psi$ for the axial current in any even spacetime dimension.

[3]Antiperiodic (Neveu-Schwarz) boundary conditions are standard for fermions.

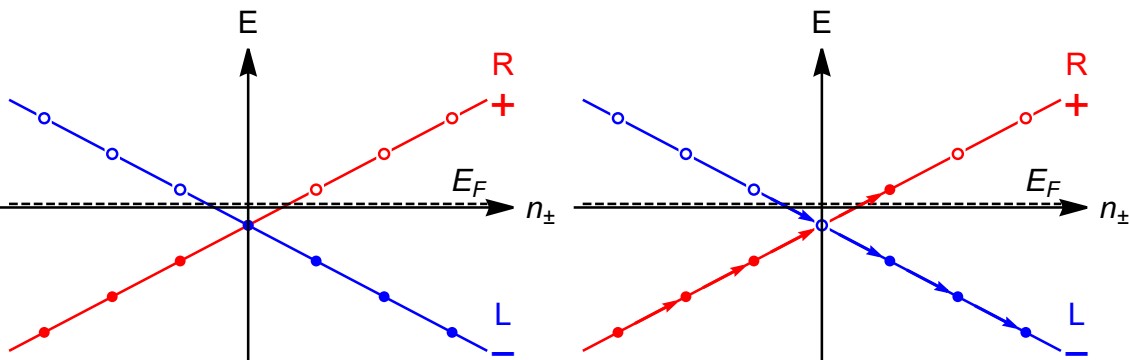

Figure 1: Energy levels of the Dirac fermion and Fermi energy $E_F$. Left panel: Ground state filling. Right panel: Filling after adiabatic insertion of one flux quantum, with arrows showing the spectral flow.

## 2.2 Regularization by point-splitting

We expand the fields $\psi_\pm$ in plane wave eigenmodes as follows,

$$\psi_\pm(x,t) = \sum_{n\in\mathbb{Z}} \varphi_{\pm,n}(x,t)\,\hat{c}_{\pm,n}, \tag{2.11}$$

$$\varphi_{\pm,n}(x,t) = \frac{1}{\sqrt{L}}\exp\left(\frac{i}{R}(n\mp\frac{1}{2})(x\mp t)\pm iA_x t\right), \tag{2.12}$$

where $\hat{c}_{\pm,n}$ are the usual anticommuting annihilation operators. Putting the Fermi energy to zero, $E_F = 0$, we get the following expressions for the expectation values of the right and left moving charges in the ground state $|\Omega\rangle$:

$$Q_+ = \langle\Omega|\hat{Q}_+|\Omega\rangle = \sum_{n=-\infty}^{0}\int_0^L dx\,\varphi_{+,n}^\dagger(x)\varphi_{+,n}(x),$$

$$Q_- = \langle\Omega|\hat{Q}_-|\Omega\rangle = \sum_{n=0}^{\infty}\int_0^L dx\,\varphi_{-,n}^\dagger(x)\varphi_{-,n}(x). \tag{2.13}$$

These are formal expressions, since, as written, they are infinite sums of ones. A first try to make sense of this would be to bound the value of $n$, to regularize the sums. This does not work: a cutoff on the momentum $p_x$, a gauge non-invariant quantity, would violate gauge symmetry, or equivalently, vector charge conservation. However, since the divergence at high momenta is due to having two quantum fields at the same point, we can regularize the sums by a gauge invariant point-splitting. Let us verify that the following prescription does the job,

$$\psi_\pm^\dagger(x,t)\psi_\pm(x,t) \;\rightarrow\; \psi_\pm^\dagger(x,t)\exp\left(-i\int_x^{x+\varepsilon}dx'A_x(x')\right)\psi_\pm(x+\varepsilon,t)\,. \tag{2.14}$$

For a spatially constant potential, the phase factor is simply $\exp(i\varepsilon A_x)$, and we get the regularized versions of (2.13),

$$Q_+^r = \sum_{n=0}^{\infty}\exp\left[-i(n+\frac{1}{2})\frac{\varepsilon}{R}-i\varepsilon A_x\right]\,, \qquad Q_-^r = \sum_{n=0}^{\infty}\exp\left[i(n+\frac{1}{2})\frac{\varepsilon}{R}-i\varepsilon A_x\right]\,. \tag{2.15}$$

Note that the expressions in the exponents are just the energies of the eigenmodes, and thus gauge invariant. Summing the geometric series, and extracting the leading terms in the limit

$\varepsilon \to 0$, we get,[4]

$$Q^r_+ = \frac{R}{i\varepsilon} - RA_x \,, \qquad\qquad Q^r_- = -\frac{R}{i\varepsilon} + RA_x \,, \qquad (2.16)$$

giving $Q = 0$ and $Q_5 = 2R/i\varepsilon - 2RA_x$.

After subtracting the divergent term, we can check whether the two charges are conserved at the quantum level in the presence of the background gauge field,

$$\dot{Q}(t) = 0, \qquad\qquad \dot{Q}_5(t) = -2R\dot{A}_x = -\frac{1}{\pi}\oint dx\, E^x(x,t). \qquad (2.17)$$

Note that the last expression on the right-hand side has been rewritten in gauge invariant form, since a non-vanishing contribution $-\partial_x A_0$ to $E^x$ as well as a gauge transform of $A_x$ would vanish when integrated around the circle with periodic boundary conditions. Using translational invariance, (2.17) can be written in the following Lorentz invariant form, using $E^x = F_{0x}$ and $\varepsilon_{0x} = -\varepsilon^{0x} = 1$,

$$\partial_\mu J^\mu = 0, \qquad\qquad \partial_\mu J^\mu_5 = \frac{1}{2\pi}\varepsilon^{\mu\nu}F_{\mu\nu}. \qquad (2.18)$$

These expressions show that the regularization of the infinite Dirac sea leads to the violation of a classical conservation law, that of the axial current. This is the axial anomaly in $(1+1)$ dimensions.

A natural question to ask at this point is how much this result depends on the regularization method. Let us consider another possible choice, where the fermion is subjected to an axial gauge field background $A_{5\mu}$, with the coupling $J^\mu_5 A_{5\mu}$. Then, taking the same steps as above but with the point-splitting,

$$\psi^\dagger(x,t)\psi(\mathrm{r}) \;\to\; \psi^\dagger(x,t)\exp\left(-i\gamma^3\int_x^{x+\varepsilon} dx' A_{5x}(x')\right)\psi(x+\varepsilon,t), \qquad (2.19)$$

which respects the axial gauge symmetry (2.3), we get,

$$Q^r_+ = \frac{R}{i\varepsilon} - RA_{5x}, \qquad\qquad Q^r_- = -\frac{R}{i\varepsilon} - RA_{5x}. \qquad (2.20)$$

So with this regularization, $Q_5$ is independent of $A_5$ but $Q$ is not; the axial current is conserved but not the vector charge,

$$\partial_\mu J^\mu = \frac{1}{2\pi}\varepsilon^{\mu\nu}F_{5\mu\nu}, \qquad\qquad \partial_\mu J^\mu_5 = 0. \qquad (2.21)$$

In conclusion we have found that a regularization preserving one symmetry leads to violation of another. How to choose one among two incompatible symmetries is dictated by the physical problem and amounts to carefully fixing the renormalization conditions. Once this is done, the anomaly is unambiguously determined. That anomalies can be shuffled around by using different cutoff procedures is something we shall encounter several times in the following.

You might, and with all rights, wonder if there could not be some regularization that would completely remove all anomalies. That this is not possible is an important result in QFT, which

---

[4]Strictly speaking the series does not converge, but you could add a small imaginary part to $\epsilon$ to remedy this. Alternatively we can just cut off the series with the gauge invariant factor $\exp(-\tau|(p_n + eA_x)|)$, since any gauge invariant regularization of the sum should be equally good. This last prescription is similar to the "heat kernel" regularization [1].

in its most elegant formulation uses deep mathematical results in differential geometry. We will touch on this important subject in Sect. 7.

Finally, a remark about the physical interpretation of the vector fields $A$, and $A_5$: In condensed matter systems you should think of $A$ as the usual electromagnetic potential which couples to the electric current which is always conserved. The axial field $A_5$, on the contrary, is a non-dynamical "emergent gauge potential", and there is no fundamental principle that requires the axial current to be conserved. We will return to the physical interpretation of $A_5$ in Sect. 6 and App. D discusses aspects of the mixed anomalies where both vector and axial (background) fields are present. The full effective theory for $A$ also includes a kinetic term for the vector field $A$ which eventually derives from the microscopic Maxwell theory but is modified by material effects such as electric polarizability. For many phenomena discussed in these lectures, however, we can neglect the photons, and consider $A$ as a background field just as $A_5$. In a couple of cases, such as in the Luttinger model in Sect. 2.5, we will include the electromagnetic interaction in the guise of an instantaneous Coulomb potential.

## 2.3 Spectral flow and the infinite hotel

From the previous section one would naturally conclude that anomalies are ultraviolet effects since they depend on how you regularize the high-energy end of the spectrum. There is, however, another approach which reveals the infrared nature of the anomalies. In condensed matter this is all important, since all the relevant field theories are effective low energy descriptions that should not much care about the "ultraviolet completion". The key to understanding the infrared face of anomalies is the concept of spectral flow.

We start by noting that on a circle, even a spatially constant $A_x$ has a physical significance since it can provide a nontrivial gauge invariant phase factor,

$$W[A_x] = e^{i \oint dx A_x} , \qquad (2.22)$$

to an electron that makes a full turn around the circle.

Our next observation is that there is a concrete way to change $A_x$. Let us suppose that the $(1+1)$-dimensional theory is a part of a higher-dimensional system: The integral $i \oint dx A_x$ can be changed by inserting a long solenoid inside the circle and varying the magnetic flux in time, for $-T < t < T$ to insert one flux (here and in the following, this means one quantum of flux), so called "flux insertion". If $T$ is large enough, the process is adiabatic and the system at all times remains in the ground state. The field,

$$A_x = -\frac{f(t)}{R}, \qquad \text{with} \qquad f(-T) = 0 , \qquad f(T) = 1 , \qquad (2.23)$$

is obtained by adding a single flux through a surface $D_2$ bounded by the circle $S_1$,

$$\Phi(T) - \Phi(-T) = \int_{-T}^{T} dt \int_{D_2} \partial_t B_z = -\int_{-T}^{T} dt \int_{S_1} E^x = 2\pi \int_{-T}^{T} dt \, \partial_t f(t) = \Phi_0 , \qquad (2.24)$$

where $\Phi_0 = h/ec$ is the flux quantum (after restoring the electric charge and all units).

Note that the entire process, going from $t = -T$ to $t = T$, amounts to a gauge transformation, $A_x \to A_x + \partial_x \lambda$, with $\lambda = -x/R$, that is not itself periodic in space but keeps $W[A_x]$ periodic, since $A_x$ appears at the exponent. This is a so-called "large" gauge transformation that *does* change the physics, in the following way.

Being a gauge transformation respecting the boundary conditions, the spectrum of the $(1+1)$-dimensional theory at $t = -T$ and $t = T$ are equal. On the other hand, upon integrating

the axial anomaly (2.17), we find,

$$\Delta Q_5 = \int_{-T}^{T} dt \, \dot{Q}_5 = 2 \left( f(T) - f(-T) \right) = 2, \qquad \Delta Q = 0. \qquad (2.25)$$

This means that the ground state changes in the process, in that a right-moving fermion is appearing from the Fermi sea while a left-moving one is diving down, as illustrated in Fig. 1. Actually, each level in the right-moving (left-moving) Dirac spectrum is mapped into the next (previous) level, carrying the electron with itself, if present: this is the so-called spectral flow. In conclusion, the flux insertion has caused an anomalous non-conservation of the chiral charges![5]

A popular analogy is that of the "infinite hotel"[6]: In a hotel with an infinite number of rooms, there is always space for a new guest even if all rooms are taken. The manager just moves the guest in room one to room two, the person in room two moves to room three, and so on. Now, room one is free for the new guest. This relabeling amounts to the process we just described. Obviously this trick does not work in a finite hotel ...

The spectral flow is an adiabatic process that, as a whole, amounts to a gauge transformation, but where the intermediate states are physically distinct. This argument shows that, from a physics point of view, the axial anomaly is an infrared effect; it is about what is happening at the Fermi surface. Thus, the axial anomaly has a Janus face. The physical aspect, *viz.* the non-conservation of a current, is the infrared face, while the need to regularize in a way consistent with the symmetries is the ultraviolet one. We will next explain that this double nature of the anomaly is deeply rooted in its topological origin.

## 2.4   Anomalies and topology – a first look

A recurrent theme of these lectures will be the close connection between anomalies and various topological invariants. This is a very deep and interesting topic by itself, but it is also of paramount importance to understand topological phases of matter which is at the frontline of current research in condensed matter physics. [7]

We now return to the (1+1)D fermions, and rephrase our previous discussion in a mathematically more precise way, which will be of use later when we discuss anomalies in higher dimensions. For this we shall look more carefully at a gauge transformation that gives rise to a spectral flow ($A_\mu = (A_t, A_x)$),

$$A_\mu(t = -T) = (0,0), \qquad A_\mu(t = T) = \left( 0, -\frac{n}{R} \right), \qquad \mu = 0, x,$$

$$A_\mu(T) - A_\mu(-T) = -\partial_\mu \lambda, \qquad \lambda(x) = n \frac{x}{R}. \qquad (2.26)$$

We already argued that this is an acceptable gauge transformation since it leaves the spectrum invariant, but another, and more fundamental, observation is that the group element $u(x) = \exp(i\lambda(x)) \in U(1)$ is periodic on the circle. From a mathematical point view, $u(x)$ is a map from the spatial circle to the group manifold of $U(1)$, that is the manifold of phase factors, which is also a circle. Such functions from $x \in S_1$ to $u(x) \in U(1) \sim S_1$ are loops that wind in the "target space" and can be divided into classes that cannot be continuously deformed into each other: those that do not wind, that wind once in one direction, twice, and so on. These

---

[5]Note that this is not in contradiction with the Byers-Yang theorem that states that if a system consists entirely of particles with charge $e$, there is no way to detect whether there are an integer number of fluxes through a puncture in the plane [3].

[6]This name seems to have been coined by H.B. Nielsen in reference to Hilbert's infinite hotel paradox [4] [5].

[7]For a general review on topological aspects in condensed matter physics, see [6].

classes form a group by addition that is called the first homotopy group $\pi_1(U(1)) = \mathbb{Z}$. The example given by (2.26) has winding number $n$.

To proceed, we return to the adiabatic flux insertion (2.23). For large $T$ the ramp function $f(t)$ can be chosen to vary very slowly near the extremes $t = \pm T$, so that $F_{\mu\nu}$ effectively vanishes. We can then identify the field configurations at $t = \pm T$, so that the spacetime manifold becomes $S_1 \times R \rightarrow S_1 \times S_1$ *i.e.* a torus. Integrating the gauge invariant form (2.18) of the anomaly over this torus, we get,

$$\Delta Q_5 = \int_{-T}^{T} dt \int_0^L dx\, \partial_\mu J_5^\mu = \frac{1}{2\pi} \int_{S_1 \times S_1} d^2x\, \varepsilon^{\mu\nu} F_{\mu\nu} = 2I[A] = 2n\,. \tag{2.27}$$

The quantity $I[A]$ is quantized as an integer,

$$I[A] = \frac{1}{4\pi} \int_{S^1 \times S^1} d^2x\, \varepsilon^{\mu\nu} F_{\mu\nu} = n \in \mathbb{Z}\,, \tag{2.28}$$

and is usually referred to as the monopole charge, while it is called the first Chern class in the mathematical literature.

$I[A]$ is a very special type of functional of the gauge field. It is a topological invariant, *i.e.* it is independent of any smooth deformation $A_\mu(x) \rightarrow A_\mu(x) + \delta A_\mu(x)$. These cannot alter the integer value which only depends on the global properties of the field configuration. Let us understand how the quantization comes about. Since the integrand is a total derivative, a naive use of Stokes theorem would imply that the integral (2.28) over a compact surface vanishes. However, the gauge field $A_\mu$ is not well defined over the entire surface, and the theorem must be applied with care. Remembering the discussion of the gauge transformation (2.26), $A_\mu$ is double valued at $t = \pm T$, the difference being a non-trivial gauge transformation $u(x) \in U(1)$ with winding number $n \in \mathbb{Z}$. The use of Stokes' theorem then gives,

$$I = \frac{1}{2\pi} \int_{S_1 \times S_1} \varepsilon^{\mu\nu} F_{\mu\nu} = \frac{1}{2\pi} \int_{S_1} (A_\mu(-T) - A_\mu(T))\, dx^\mu = \frac{1}{2\pi} (\lambda(2\pi R) - \lambda(0)) = n\,. \tag{2.29}$$

Note that, any local deformations of the gauge field $A_\mu(x, t)$, for $-T < t < T$, do not affect its value as anticipated.

In conclusion, we have shown that the integrated form of the anomaly is related to a topological invariant of the gauge field, the first Chern class, that takes integer values, owing to the homotopy group $\pi_1(U(1)) = \mathbb{Z}$. The spectral flow with $n$ right-moving electrons pumped out of the Dirac sea (and left-moving holes pumped in) is realized precisely when the gauge field configuration is that of a charge $n$ monopole inside the space-time cylinder (compactified to a torus). It is apparent that the anomaly is only sensitive to the global infrared properties of the underlying theory.

Finally we should comment on the connection between the adiabatic flux insertion (2.24) of the previous section, and the purely (1+1)-dimensional approach just discussed, leading to (2.27). First you should notice that topology is important also in the flux insertion argument, since the winding number is a topological invariant, just as the Chern class. Secondly, the Yang-Byers quantization of magnetic flux in three dimensions [3] is the counterpart to the topological electric flux quantization in the (1+1) dimensional case. For physical phenomena that takes place at boundaries, it is rather natural to think in terms of embeddings in a higher dimensional space, but in the following discussion of other systems we shall explore both approaches.

## 2.5  Connection to the Luttinger model

In condensed matter physics, fermion systems in one dimension can be realized in various ways, such as quantum wires, carbon nano-tubes or edge states in a quantum Hall droplet. All these systems have a rich and fascinating phenomenology, but here we shall just mention one aspect of the theory which is intimately connected to the anomaly we derived in the previous sections.

The way relativistic fermions emerge in a condensed matter framework is illustrated in Fig. 2. The left, and more realistic picture, represents a single parabolic band in a one-dimensional crystal. It should however be clear that the low-lying excitations in this model are the same as those of the $(1+1)$ dimensional Dirac theory (2.1), shown on the right part of the figure. That the former has a finite number of electrons filling the band, and the other an infinite Dirac sea should not change the physics close to the Fermi points. The second model, which was originally studied by Luttinger, is given by the Tomonaga-Luttinger Hamiltonian,

$$H = \sum_{\alpha=\pm 1} v_F \int dx \, \psi_\alpha^\dagger (i\alpha \partial_x - k_F) \psi_\alpha - \frac{1}{2} \int dx dx' \rho(x) V(x-x') \rho(x') \,, \qquad (2.30)$$

where $\alpha$ labels the two Fermi points, $v_F$ is the Fermi velocity, and $\rho$ the total density, $\rho = \rho_+ + \rho_- = \psi_+^\dagger \psi_+ + \psi_-^\dagger \psi_-$. In addition to the kinetic term, there is a density-density interaction. This model can be studied using perturbation theory, renormalization group methods, and most importantly with the powerful method of bosonization, to be described in the next section.

Let us anticipate that the Fourier components for the chiral density operators $\rho_{\pm,n}$, with $k = (n+1/2)/R$ on a circle of length $2\pi R$, can be shown to satisfy,

$$[\rho_{\pm,n}, \rho_{\pm,n'}] = \pm n \delta_{n+n',0} \,, \qquad (2.31)$$

which is known as current algebra ($U(1)$ Kac-Moody algebra [7]). Just as the axial anomaly, this is a quite surprising result. Normally we think that the density should commute with itself! Indeed, upon expressing $\rho$ in terms of fermionic creation and annihilation operators $c_n$ and $c_n^\dagger$, we would naively find that the commutator vanishes. However, a finite term is generated when the UV regularization of the densities is taken into account.[8] As will be described in Sect. 3.3, this is a manifestation of the anomaly.

We already stressed that the anomaly can also be understood as an infrared effect, so we expect that the anomalous commutator could also be derived is a similar manner. To do this, we start from the one-dimensional (non-chiral) density and current operators in first quantization, $\rho(x) = \sum_{n=1}^N \delta(x-x_n)$ and $J(x) = \sum_{n=1}^N [p_n \delta(x-x_n) + \delta(x-x_n) p_n]/2m$. Using the quantum mechanical commutators, $[x_n, p_m] = i\hbar \delta_{mn}$, we get the non-relativistic current algebra relations,

$$[\rho(x), J(x')] = -i\frac{\hbar}{m} \rho(x) \partial_x \delta(x-x') \,, \qquad (2.32)$$

and $[\rho(x), \rho(x')] = [J(x), J(x')] = 0$. Now assume that there are only small excitations around both the left-moving and right-moving Fermi points, and write $\rho = \rho_0 + \rho_+ + \rho_-$ and $J = v_F(\rho_+ - \rho_-)$. Here $\rho_0$ is a constant background density which is much larger than the

---

[8]See *e.g.* [8].

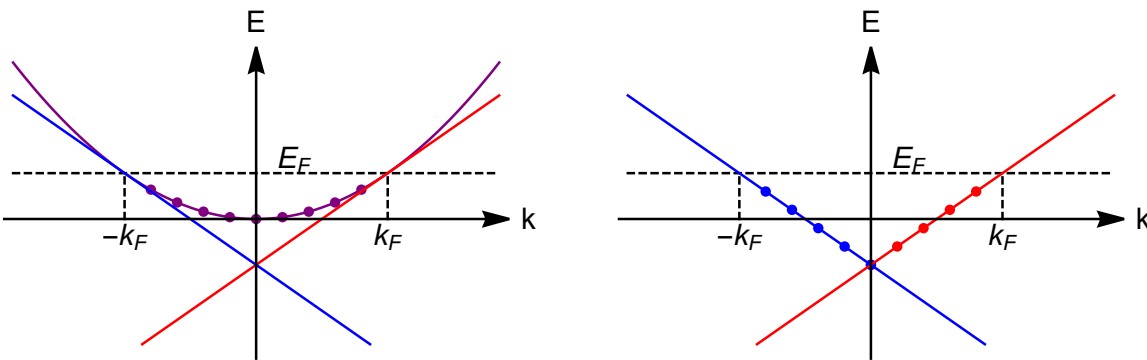

Figure 2: Left panel: Dispersion relation for a parabolic band, with the Fermi energy $E_F$ and momenta $\pm k_F$. The dots indicate filled levels in the ground state. The red and blue lines are linear approximations near Fermi points. Right panel: Linear dispersion relation for the Luttinger model, with filled levels.

perturbations $\rho_+$ and $\rho_-$. With this we have,

$$
\begin{aligned}
[\rho_+(x),\rho_+(x')] &= \frac{1}{4}\left[\rho(x)+\frac{J(x)}{v_F},\rho(x')+\frac{J(x')}{v_F}\right] \\
&= -i\frac{\hbar}{2v_F m}\rho(x')\partial_x\delta(x-x') \\
&= -i\frac{1}{2\pi}\partial_x\delta(x-x').
\end{aligned}
\tag{2.33}
$$

In going to the last line we have substituted $\rho_0$ for $\rho(x)$ on the right, since it is assumed to be much larger than $\rho_+$ and $\rho_-$. We also used the values $\rho_0 = k_F/\pi = mv_F/(\pi\hbar)$ for spinless one-dimensional fermions. Finally, Fourier transforming (2.33) gives (2.31).

Note that just as in the "infinite hotel" derivation of the spectral flow, we only considered effects close to the Fermi surface. To show the connection to the chiral anomaly, we couple the right moving charge to electromagnetism. In the axial gauge $A_x = 0$, this coupling is $H = -\int dx\, A_0\rho_+$, and we can use Heisenberg's equation to get,

$$
\begin{aligned}
\partial_t Q_+ &= -\frac{1}{i}\int dx\,dy\,[\rho_+(x),A_0(y)\rho_+(y)] \\
&= \frac{1}{2\pi}\int dx\,dy\,\partial_x\delta(x-y)A_0(y) = -\frac{1}{2\pi}\int dx\, E_x\,.
\end{aligned}
\tag{2.34}
$$

Therefore, the charge commutator (2.33) implies the chiral anomaly (2.34).

# 3 Anomalies and effective field theories of the Quantum Hall Effect

## 3.1 Introduction

The best known example of a topological phase of matter is the quantum Hall effect [9]. A two-dimensional electron system is realized in a layered semiconductor, *e.g.* Gallium Arsenide, and subjected to an orthogonal magnetic field $B$.

A typical experiment, which is schematically shown in Fig. 3, is performed in very clean samples at a very low temperatures ($T \sim 10$ mK) and high magnetic field ($B \sim 10$ Tesla).

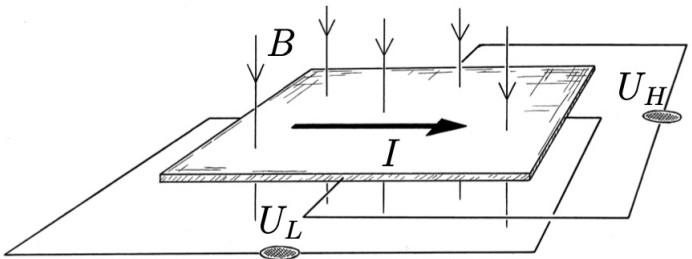

Figure 3: Quantum Hall bar with circuits to measure the longitudinal and transverse voltage drops (Figure thanks to Sören Holt).

The transverse (Hall) resistance shows extremely stable plateaus when $B$ is varied over a sizeable range. To analyze this remarkable phenomenon, one must first understand that in in 2d, the Hall resistance $R_H = U_H/I$ (using the notation of the figure) equals the Hall resistivity $\rho_H$ which is a material property independent of the geometry [9]. On the plateaus, the corresponding Hall conductivity takes the values,

$$J_i = \sigma_H \varepsilon_{ij} E^j, \qquad \sigma_H = \nu \frac{e^2}{h}, \qquad \nu = 1, 2, \ldots, \frac{1}{3}, \frac{2}{5}, \ldots. \tag{3.1}$$

In this equation, $J_i, E^i, i = 1, 2$, are the in-plane current and electric field, respectively, $h$ is the Planck constant; $\nu$ is the so-called filling fraction, the density of electrons in units of a filled Landau level as explained below. At the same time, the longitudinal conductance vanishes, indicating a gap in the spectrum. The prominent plateaus occur for integer filling and for the fractional values of the Laughlin series, $\nu = 1/p$, with $p = 3, 5, \ldots$. Experiments show remarkably precise values, $\Delta\sigma_H/\sigma_H \sim 10^{-9}$, that are independent of the sample details. As will be explained below, this is due to the invariance of $\sigma_H$ under both continuous deformations of the sample, and the interaction between the electrons; $\sigma_H$ is a topological quantum number.

The main physical picture for the quantum Hall effect, due to Laughlin [10] [9], is that the electrons form an incompressible quantum fluid, characterized by a constant density in the bulk and a gap that forbids density waves. This type of ground state can most easily be understood for non-interacting electrons completely filling one or more Landau levels. In this case, the Coulomb interaction can be neglected and the gap is the Landau level spacing, *i.e.* the cyclotron energy $\hbar\omega = \hbar e B/m_e c \gg k_B T$ ($m_e$ the mass of the electron).

Let us recall some properties of one-particle states in the Landau levels [11]. For polarized electrons, the Zeeman term can be disregarded,[9] and the Hamiltonian is,

$$H = \frac{1}{2m_e}(\mathbf{p} + \mathbf{A})^2, \tag{3.2}$$

within our conventions for the electron charge. The planar momentum is $\mathbf{p} = (p_1, p_2)$ and the downward magnetic field in Fig. 3 is realized by the vector potential $\mathbf{A} = B/2(x_2, -x_1)$, in the so-called symmetric gauge. We introduce two independent pairs of creation-annihilation operators,

$$a = \frac{z}{2} + \overline{\partial}, \qquad a^\dagger = \frac{\overline{z}}{2} - \partial, \qquad [a, a^\dagger] = 1,$$

$$b = \frac{\overline{z}}{2} + \partial, \qquad b^\dagger = \frac{z}{2} - \overline{\partial}, \qquad [b, b^\dagger] = 1, \tag{3.3}$$

---

[9]In relevant materials, the magnetic dipole moment is high enough for the Zeeman gap to be much larger than the cyclotron gap.

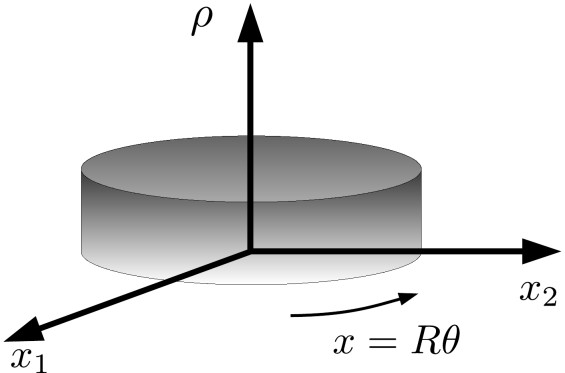

Figure 4: Ground-state electron density (3.9) in the geometry of the disk with radius $R$; the coordinate along the edge is $x = R\theta$.

expressed in the complex coordinate, $z = x_1 + ix_2$ and derivative $\partial = \partial/\partial z$. In these, and many of the following expressions, we set the magnetic length,[10] $\ell_B = \sqrt{2\hbar c/eB}$, as well as $c$, $e$, $\hbar$ and $m_e$, equal to one, so $\omega = B = 2$. The Hamiltonian and angular momentum $M$ can then be rewritten as,

$$H = \omega\left(a^\dagger a + \frac{1}{2}\right), \qquad M = \left(b^\dagger b - a^\dagger a\right). \tag{3.4}$$

The single particle wave functions $\psi_{n,m}(z,\bar{z})$ are labeled by a Landau level index $n = 0, 1, \dots$ and an angular momentum $m = -n, -n+1, \dots$. The energies are degenerate with respect to $m$ as a consequence of the translation invariance of the cyclotron orbits. In the following we shall limit ourselves to the first level, where the wavefunctions are,

$$\psi_{0,m}(z,\bar{z}) = e^{-z\bar{z}/2}\frac{z^m}{\sqrt{\pi m!}}. \tag{3.5}$$

One can easily check that these states are localized on the classical cyclotron orbits of radius $r_m = \ell_B\sqrt{m}$, in that the area bounded by the $m$-th orbit encloses $m$ fluxes $\Phi_0$ (equal to $2\pi$ in our units).

Let us consider the Hall system in the geometry of a disk of radius $R$. Then the angular momentum has an upper bound, approximatively given by the largest orbit $m_{\text{Max}} = R^2$, so the Landau degeneracy is equal to the number of enclosed fluxes, $\mathcal{D} = BA/\Phi_0$. Filling all the available levels determines the average density $\rho_0 = \mathcal{D}/A$ and the Hall conductivity $\sigma_H = \rho_0 ec/B = e^2/h$, corresponding to filling $\nu = 1$ in (3.1).

The $\nu = 1$ ground state wave function $\psi$ is the Slater determinant of the single particle states (3.5), which can be written as the Vandermonde determinant [12],

$$\psi(z_1, \cdots, z_N) = e^{-\sum_{i=1}^{N}|z_i|^2/2}\prod_{\substack{i<j=1}}^{N}\left(z_i - z_j\right), \tag{3.6}$$

where $z_i$ are the positions of the $N$ electrons.

The profile of the ground state density is most easily obtained using second-quantized expressions. The non-relativistic field operator is,

$$\hat{\Psi}(\vec{x}, t) = \sum_{m=0}^{\infty}\psi_{0,m}(\vec{x})\hat{c}_m, \tag{3.7}$$

---

[10]Another common convention is $\tilde{\ell}_B = \sqrt{\hbar c/eB} \to 1$, as in Ref. [12].

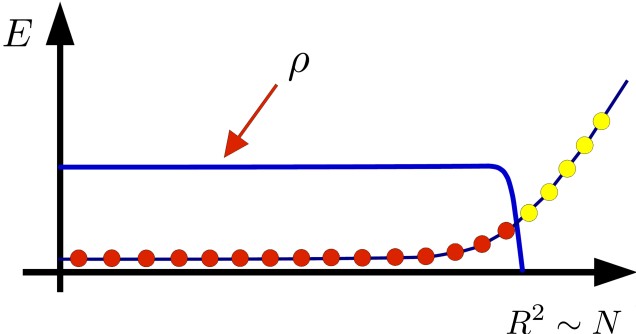

Figure 5: Energy levels of the first filled Landau level as a function of angular momentum $m \propto r^2$ (red filled, yellow empty levels). Near the edge, $m \sim N$, $r^2 = R^2 \sim N$, the confining potential breaks the degeneracy leading to a kind of Fermi surface in coordinate space. The radial shape of the density $\rho(r^2)$ is also shown.

where $\vec{x} = (x_1, x_2)$, $\hat{c}_m$ is the fermionic annihilation operator and the ground state energy $\omega/2$ has been discarded. The $\nu = 1$ ground state $|\Omega\rangle$ is built over the Fock vacuum $|0\rangle$, as follows,

$$|\Omega\rangle = \hat{c}_0^\dagger \hat{c}_1^\dagger \dots \hat{c}_{N-1}^\dagger |0\rangle. \tag{3.8}$$

The expectation value of the density $\hat{\rho} = \hat{\Psi}^\dagger \hat{\Psi}$ then becomes,

$$\langle\Omega|\hat{\rho}(r)|\Omega\rangle = \sum_{m=0}^{N-1} |\psi_{0,m}(\vec{x})|^2 = \frac{1}{\pi} e^{-r^2} \sum_{m=0}^{N-1} \frac{r^{2m}}{m!}, \qquad r = |\vec{x}|, \tag{3.9}$$

as shown in Fig. 4. This describes a droplet of fluid with constant density $\rho_o = 1/\pi$ in the interior, which is rapidly falling to zero at the edge within a few magnetic lengths.

## 3.2 Edge excitations

Fig. 5 shows a radial profile of the density, where the one-particle states are represented by dots. A confining potential at the edge of the system prevents the electrons from escaping. Such a potential breaks the Landau level degeneracy so the states will be filled up to a Fermi energy and empty above it.

Thus, the quantum Hall droplet has a Fermi surface, or rather Fermi point, but in coordinate space! Following the steps of Sect. 2.5, we can linearize the energy of the one-particle states around the "Fermi angular momentum" $m_F \sim R^2$,

$$\varepsilon(m) \sim \frac{\nu_F}{R}(m - R^2 - \mu), \qquad R \to \infty, \tag{3.10}$$

where $\nu_F$ is the Fermi velocity and $\mu$ is the chemical potential to be determined later. This corresponds to a relativistic dispersion relation, $\varepsilon(k) = \nu_F(k - k_F)$, where $k = m/R$ is the one-dimensional momentum of waves propagating along the boundary circle. The resulting $(1+1)D$ chiral, relativistic particle is called a Weyl fermion. This theory is a chiral version of the Luttinger model discussed in Sect. 2.5.

Let us verify that the Landau electrons near the edge indeed behave as a relativistic fermions. For this we consider the Landau orbitals (3.5) in the combined limit [13],

$$m = R^2 + m', \quad |m'| < R, \qquad r = R + r', \quad |r'| = O(1), \qquad R \to \infty. \tag{3.11}$$

This corresponds to looking at a finite region of a few magnetic lengths around the boundary in the limit of large droplet sizes. In this limit, the angular momentum range is in the linear part of the spectrum (3.10). Using the Stirling approximation in the Landau wavefunctions (3.5), we obtain,

$$\psi_{0,R^2+m'}(r,\theta) = \mathcal{N} \frac{e^{i(R^2+m')\theta}}{\sqrt{2\pi R}} \, e^{-\left(r'-\frac{m'}{2R}\right)^2} \left(1 + O\left(\frac{1}{R}\right)\right), \tag{3.12}$$

where the normalization constant is $\mathcal{N} = (2/\pi)^{1/4}$. In this expression, one recognizes the wave function for the $(1+1)$-dimensional Weyl fermion, $\psi_{m'}(x) = e^{ikx}/\sqrt{2\pi R}$, with $k = m'/R$ and $x = R\theta$. Note also that edge states with $m'$ in the range (3.11) are superposed within a distance of one magnetic length, up to $1/R$ corrections. Thus the radial dependence is inessential and can be neglected.

Therefore, in a suitable kinematic range, the Landau level field operator $\hat{\Psi}$ (3.7), becomes,

$$\hat{\Psi}(R\theta, t) \sim e^{i(R^2+\mu)\theta} \hat{\psi}_-(x, t),$$
$$\hat{\psi}_-(x, t) = \sum_{m=-R^2}^{\infty} \frac{1}{\sqrt{2\pi R}} \exp\left[\frac{i}{R}(m-\mu)(x - v_F t)\right] \hat{c}_m. \tag{3.13}$$

In this expression, the energy dependence (3.10) has been included and the indexing of the Fock operators has been redefined by $\hat{c}_{R^2+m} \to \hat{c}_m$. So, if we set $v_F = 1$ and $\mu = 1/2$ (antiperiodic boundary conditions) we recover, up to an overall phase, the expression for the relativistic fermion field (2.11) discussed in Sect. 2.2

## 3.3  Current algebras and conformal field theory – a primer

The Weyl fermion theory provides a simple example for introducing $(1+1)$-dimensional conformal field theory (CFT) [14], and, before proceeding, we provide some background material on conformal symmetry.

Massless theories, such as the Weyl fermion, do not involve any dimensionful parameter and therefore the action (2.1) is invariant under scale transformations, or dilatations, $x^\mu \to \lambda x^\mu$ and $\psi \to \lambda^{-1/2}\psi$. Conformal invariance amounts to a local extension of scale invariance where infinitesimal dilatations are independently chosen at any point. In $(1+1)$ dimensions, this generalization is possible under the generic assumptions of unitarity, local degrees of freedom and a local Hamiltonian. Conformal transformations of the metric are given by local scale factors $g_{\mu\nu} \to \lambda(x)g_{\mu\nu}$ which can be realized as follows: In Euclidean flat coordinates, the line element can be written,[11]

$$ds^2 = dx_0^2 + dx_1^2 = dz d\bar{z}, \qquad z = x_0 + ix_1, \tag{3.14}$$

in terms of complex coordinates. Thus, analytic coordinate changes, $z \to w$,

$$z = z(w), \qquad ds^2 = \left|\frac{dz}{dw}\right|^2 dw d\bar{w} = 2g_{w\bar{w}} \, dw d\bar{w}, \tag{3.15}$$

realize the conformal transformation of the metric with $\lambda(x) = |dz/dw|^2$. Note that the conformal symmetry is much more powerful than just scale invariance because it involves an infinite number of parameters that characterize the analytic function $z = z(w)$. Note also the factorization of the transformation in $z, \bar{z}$, which, after continuation to Minkowski space, are the natural coordinates to describe right- and left-moving particles, respectively.

---

[11]See App. E for an introduction.

Since conformal transformations are analytic reparameterizations, for infinitesimal changes,

$$z \to z + \varepsilon(z), \qquad \varepsilon(z) = \sum_{n=-\infty}^{\infty} \varepsilon_n z^{n+1}, \tag{3.16}$$

the transformation of functions reads,

$$f(z) \to f(z + \varepsilon(z)) = \left[ 1 + \sum_{n=-\infty}^{\infty} \varepsilon_n z^{n+1} \partial_z \right] f(z), \tag{3.17}$$

which amounts to having the following (classical) generators and algebra,

$$L_n = -z^{n+1} \partial_z, \qquad [L_n, L_m] = (n-m) L_{n+m}, \tag{3.18}$$

which is the famous Virasoro algebra [14].

The stress-energy tensor $T_{\mu\nu}$ plays a central role in conformal field theories. It expresses the response of the theory to a varying background metric $g_{\mu\nu}$,

$$\delta S[\psi, g_{\mu\nu}] = -\frac{1}{2} \int d^2 x \sqrt{g} \, T_{\mu\nu} \delta g^{\mu\nu}, \tag{3.19}$$

similarly to currents being responses to background gauge fields.

Coordinate reparameterizations, $x^\mu \to x^\mu + \varepsilon^\mu(x)$, correspond to the metric variations $\delta g^{\mu\nu} = D^\mu \varepsilon^\nu + D^\nu \varepsilon^\mu$, where $D_\mu$ is the covariant derivative with respect to the metric, while conformal mappings (3.15) are instead given by $\delta g^{\mu\nu} = -\delta\lambda(x) g^{\mu\nu}$. Using (3.19), we see that invariance of the action with respect to these transformations implies the following conditions,

$$D_\mu T^{\mu\nu}(x) = 0, \qquad T^\mu_\mu(x) = 0, \tag{3.20}$$

expressing (covariant) momentum conservation and conformal invariance, respectively.

Again there are anomalies that violate these symmetries at the quantum level. The lack of conservation is captured by the gravitational anomaly [15],

$$D^\mu T_{\mu\nu} = \frac{c}{48\pi} \partial_\nu \mathcal{R}, \qquad T^\mu_\mu = 0, \tag{3.21}$$

and the non-vanishing trace by the conformal (or trace) anomaly,

$$D^\mu T_{\mu\nu} = 0, \qquad T^\mu_\mu = -\frac{c}{24\pi} \mathcal{R}, \tag{3.22}$$

where $\mathcal{R}$ is the scalar curvature of the background metric. The coupling to gravity and its anomalies will be further discussed in Sect. 8, and our conventions for curved-space calculus are summarized in App. E. In the following we mention some basic facts.

As in earlier cases, the two expressions (3.21) and (3.22) are related; by using different regularizations, *i.e.* adding counterterms to the effective action, the anomaly can be moved from one law to the other, but cannot be eliminated completely. The dimensionless universal number $c$ is the "central charge" which characterizes the conformal theory. In the case of chiral theories, there are two independent central charges, $(c, \bar{c})$, one for each chiral component.[12] This is due to the factorization of conformal transformations (3.15) in independent analytic and anti-analytic reparameterizations, $z = z(w)$ and $\bar{z} = \bar{z}(\bar{w})$. For example, the Weyl and Dirac fermions have $(c, \bar{c}) = (1, 0)$ and $(c, \bar{c}) = (1, 1)$, respectively.

---

[12]The expressions of anomalies given in (3.21) and (3.22) are only valid for $c = \bar{c}$. The general case will be discussed in Sect. 8.

Of course, many more things could be said about CFTs and the interested reader can find extensive reviews in the literature, at different level of sophistication: for the beginner, we already suggested [14], a modern standard references is [7], and a comprehensive account is given in [16].

The CFT approach is algebraic, *i.e.* it is based on obtaining correlation functions and observable quantities from the study of representations of the Virasoro Algebra. Deep mathematical results on the representations for interacting theories have made it possible to find exact solutions of models with very interesting dynamics.

In these lectures we shall, however, only consider the Luttinger model, also called the compactified boson in the CFT literature, which is a free theory where all results can be obtained rather easily, and which also describes interacting fermions through the method of bosonization. Below we shall explain how this can be used to describe the fractional charge and statistics of the edge excitations in the Laughlin quantum Hall states.

To continue the analysis of the $\nu = 1$ edge excitations we define the Fourier modes of density and energy (hereafter, we suppress hats on operators) as,

$$\rho_n = \int_0^{2\pi R} dx\, e^{-ixn/R}\, \psi_-^\dagger(x,t)\psi_-(x,t) = \sum_{k=-\infty}^{\infty} :c_{k-n}^\dagger c_k:, \qquad (3.23)$$

$$L_n = R\int_0^{2\pi R} dx\, e^{-ixn/R}\, \psi_-^\dagger(x,t)(-i\partial_x)\psi_-(x,t)$$

$$= \sum_{k=-\infty}^{\infty}\left(k - \frac{n+1}{2}\right):c_{k-n}^\dagger c_k:, \qquad (3.24)$$

where we again put $\mu = 1/2$. The stress tensor is a two-dimensional symmetric and traceless matrix and therefore possesses two independent components that correspond to the two chiral terms in the fermionic Hamiltonian (2.7). One chiral component appear in (3.24), so the $L_n$ operators are the stress-tensor Fourier modes which generate the conformal transformations. They will turn out to be the quantum version of the classical generators (3.18).

In Sect. 2.2 we showed that bilinears of the fields require a regularization in the presence of an infinite Dirac sea, and used the point-splitting method. In this section, we shall use another technique called normal ordering $:\cdots:$, that works as follows: We fill the Fermi sea up to an edge, which corresponds to the $k=0$ state after the momentum relabeling (3.11). This ground state satisfies the conditions,

$$c_k|\Omega\rangle = 0, \qquad k > 0,$$
$$c_k^\dagger|\Omega\rangle = 0, \qquad k \leq 0. \qquad (3.25)$$

Normal ordering amounts to putting $c_k$ to the right of $c_k^\dagger$ for $k > 0$, and the other way around for $k \leq 0$,

$$:c_k^\dagger c_k: = -:c_k c_k^\dagger: = c_k^\dagger c_k, \qquad k > 0,$$
$$:c_k^\dagger c_k: = -:c_k c_k^\dagger: = -c_k c_k^\dagger, \qquad k \leq 0, \qquad (3.26)$$

while no conditions are imposed for unequal indices. Once implemented in the definition of $\rho_0$ and $L_0$ in (3.23) and (3.24), this prescription removes the infinities in the ground-state values, therefore,

$$\rho_0|\Omega\rangle = L_0|\Omega\rangle = 0. \qquad (3.27)$$

Furthermore, we have:

$$\rho_n|\Omega\rangle = L_n|\Omega\rangle = 0, \qquad n > 0, \qquad (3.28)$$

while for $n < 0$ these operators create particle-hole excitations by pulling electrons out of the filled Dirac sea.

In Sect. 2.5, we have already derived the commutator between the $\rho_n$'s by analyzing what happens close to the Fermi points. We now complement this by deriving the full algebra of the $\rho_n$'s and $L_n$'s using a ultraviolet regularization. These two methods give the same result which demonstrates that the anomalous commutators have both an infrared and an ultraviolet aspect.

Using Fock space anticommutators, we find,

$$[\rho_n, \rho_m] = \sum_{k=-\infty}^{\infty} c_{k-n}^{\dagger} c_{k+m} - \sum_{k=-\infty}^{\infty} c_{k-n-m}^{\dagger} c_k = n\delta_{n+m,0}. \tag{3.29}$$

The two sums in the r.h.s. cancel each other after shifting the summation variable in the second term. However, for $m = -n$, the Fock operators have the same index so we need the normal ordering (3.26), which is not translation invariant in $k$. Upon enforcing it, there remains the factor $\delta_{n+m,0}\left(\sum_{k\leq n} 1 - \sum_{k\leq 0} 1\right) = n\delta_{n+m,0}$; thus, the non-vanishing commutator originates from a finite offset of normal orderings. The same argument can be applied to the commutators involving $L_n$. The resulting algebras are found to be [17],

$$[\rho_n, \rho_m] = n\delta_{n+m,0}, \tag{3.30}$$

$$[L_n, \rho_m] = -m\rho_{n+m}, \tag{3.31}$$

$$[L_n, L_m] = (n-m)L_{n+m} + \frac{c}{12}n(n^2-1)\delta_{n+m,0}, \qquad c = 1. \tag{3.32}$$

The first relation is the current algebra for the generators $\rho_n$, already introduced in Sect. 2.5; the third expression is the Virasoro algebra (3.18) realized in the fermionic Fock space. Note the additional $c$-number term in the right-hand side of the algebra, which is the so-called central extension. This is a quantum addition to the algebra, that follows from the conformal anomaly; actually, the coefficient $c$, called central charge, is the same as the parameter in Eqs. (3.22) (see App. E for the proof of this statement). Note that in the Dirac theory, there is a corresponding set of commutation relations for the other chirality, with central charge $\bar{c} = 1$.

Let us also mention a third derivation of the commutation relations (3.30) that, like the one in Sect. 2.5, does not rely on shifts of summation indices in (3.29). Consider the expectation value of the density commutators, and impose the ground state conditions (3.28) as well as their Hermitian conjugates, $\langle\Omega|\rho_n = 0$, $n \leq 0$, (recall $\rho_n^{\dagger} = \rho_{-n}$), to obtain,

$$\langle\Omega|[\rho_n, \rho_m]|\Omega\rangle = \delta_{n+m,0}\langle\Omega|\rho_n\rho_{-n}|\Omega\rangle = \delta_{n+m,0}n\langle\Omega|\Omega\rangle, \qquad n > 0. \tag{3.33}$$

In these equations, we used the definition (3.23) and the Fock-space conditions (3.25). This calculation can be extended to $\langle\Omega|L_n L_{-n}|\Omega\rangle$ and $\langle\Omega|L_n\rho_{-n}|\Omega\rangle$, $n > 0$, showing that the anomalies are uniquely determined once the ground-state conditions are fixed; they depend not only on the algebra but also on the norm of the Hilbert space.

To summarize this section: The conformal field theory of the Weyl fermion is characterized by central charges $(c, \bar{c}) = (1, 0)$. The states in the theory can be organized into representations of the algebras (3.30) – (3.32), whose central terms are due to chiral and conformal anomalies. The ground state $|\Omega\rangle$ is the bottom state of one representation, characterized by the conditions (3.27), (3.28). Other states are the particle-hole excitations,

$$|\{n_i\}, \Omega\rangle = \rho_{-n_k}\cdots\rho_{-n_2}\rho_{-n_1}|\Omega\rangle, \qquad n_k \geq \cdots n_2 \geq n_1 > 0, \tag{3.34}$$

that form an infinite tower of states. Charged excitations, with some net number of particles or holes, correspond to other representations. For example, for charge $Q = n > 0$ the bottom

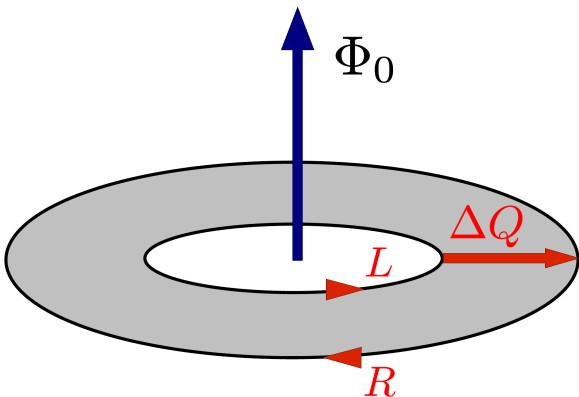

Figure 6: Laughlin flux argument: the adiabatic insertion of a flux quantum $\Phi_0$ inside the annulus moves a charge $\Delta Q = 1$ from the inner to the outer edge.

state is $|n\rangle = c_1^\dagger c_2^\dagger \cdots c_n^\dagger |\Omega\rangle$, satisfying,

$$\rho_0|n\rangle = |n\rangle, \qquad L_0|n\rangle = \frac{n^2}{2}|n\rangle, \tag{3.35}$$

together with the conditions (3.28), as readily obtained from the Fock space expressions (3.25). Note that the $L_0$ eigenvalue by definition gives the energy of the charged excitation within the linear relativistic spectrum. The tower of particle-hole excitations has energies $n^2/2 + \sum_k n_k$. From the Hamiltonian (3.24) it follows that this is the original Landau level angular momentum, *i.e.* $z\partial/\partial z \to -i\partial/\partial\theta$. Therefore, $L_0$ also measures the angular momentum of the edge excitations.

## 3.4 Hall current and the chiral anomaly

Having established that the edge excitations in the quantum Hall state are described by the chiral (1+1)-dimensional fermion, let us explain the consequences of the chiral anomaly in this context. Consider the annulus shown in Fig. 6, which is a convenient geometry for describing the Hall currents. The outer edge harbors a chiral fermion, and the inner boundary hosts one of opposite chirality. Together they can be viewed as the two components of a Dirac fermion, although they are spatially separated.

Let us recall the spectral flow in the formulation in Sect. 2.3. Upon adiabatically inserting one unit of flux in the hole of the annulus the spectrum returns to itself, but one electron is removed from the left-moving Dirac sea and one is added to the right-moving sea, so effectively one electron has moved from the inner to the outer edge,

$$\Delta Q = \Delta Q_R = -\Delta Q_L = \frac{\Phi(T) - \Phi(-T)}{\Phi_0} = 1\,. \tag{3.36}$$

Physically, the magnetic flux insertion induces an azimuthal electric field $E_x$, which in turn generates a radial Hall current $I = 2\pi\sigma_H d\Phi/dt$. Therefore, we have shown that the spectral flow of edge states corresponds to the charge transport by the $\nu = 1$ Hall current! From the perspective of spectral flow, the bulk Hall current appears as a collective radial motion of electrons, each one moving from one orbital to the next – the only difference with respect to the relativistic case is that the two Dirac seas are joined at the bottom into a single Fermi sea.

This reformulation in terms of the anomaly brings in an extra bonus. As stressed in Sect. 2.4, the integrated anomaly is proportional to a topological number, the first Chern class, which takes integer values. This correspondence provides a profound explanation for the robustness and universality of the quantization of the Hall conductance. For example, it is independent of small local fluctuations of the density and/or the magnetic field, impurities, lattice defects, *etc.,* which explains the remarkable experimental precision.

Let us remark that the original Laughlin argument [18] for the exactness of the integer Hall conductance is basically the same as the spectral flow just described, but is formulated in the context of the non-relativistic Landau problem, and supplemented by an important discussion of the effect of impurities that is crucial to understand real quantum Hall devices [9].

We remark that there is no anomaly in the whole system: the charge non-conservation at the two edges of the annulus is compensated by the classical Hall current flowing radially. The apparent violation only appears in the effective relativistic theory that considers just a single edge. The general mechanism of a quantum anomaly (edge) canceled by a classical current in one dimension higher (bulk) is called "anomaly inflow" [19], and we shall return to this phenomenon at several occasions in the context of the so called bulk-boundary correspondence in topological states of matter.

In the pivotal work of Niu *et al.* [20], the Hall conductivity was also related to a topological quantity characterizing the non-relativistic Landau levels, namely the first Chern class of the Berry phase acquired by wave functions during the adiabatic flux insertion. The two arguments of the anomaly in the edge theory and of the Berry phase of wavefunctions are equivalent for the integer Hall effect. However, for interacting electrons, as in the fractional Hall effect, the anomaly argument generalizes most easily.[13] It will be our first example of how to use anomalies to characterize topological phases of interacting matter.

## 3.5 Fractional Hall states and Chern-Simons effective theory

We now discuss some features of the fractional Hall effect at the Laughlin filling factors $\nu = 1/k$ = $1/3, 1/5, \ldots$. Before deriving the conformal field theory of the edge excitations, we first briefly review the theoretical description of the bulk system in terms of wave functions and effective low-energy field theory.

### 3.5.1 The Laughlin wave function

For fractional fillings, the free electron system in the lowest Landau level is highly degenerate, so the existence of a gap is a non-perturbative effect due to the Coulomb repulsion among the electrons. Laughlin argued that the ground state is again an incompressible fluid with constant density and gapful density wave excitations [9]. He proposed the following ansatz wave function,

$$\psi(z_1, \cdots, z_N) = e^{-\sum_{i=1}^{N} |z_i|^2/2} \prod_{i,j=1, i<j}^{N} \left(z_i - z_j\right)^k , \qquad \nu = \frac{1}{k} . \qquad (3.37)$$

This wave function minimizes the kinetic energy by being entirely in the lowest Landau level, and has low Coulomb energy due to the $k^{th}$ power of the Vandermonde determinant (3.6) that effectively keeps the electrons apart; since $k$ is odd, (3.37) is fully antisymmetric.

The Laughlin wave function has several "magic" properties that are difficult to explain in terms of one-particle states, because it is an involved superposition of Slater determinants [22]. Thus other methods must be used, and an important technique, again due to Laughlin, is the

---

[13]For a discussion of the Berry phase argument, see [20] and [21].

mapping to a two-dimensional plasma [9]. Here the idea is that the norm of the wave function,

$$\|\psi\|^2 = \int \prod_{i=1}^{N} d^2 z_i \, \exp\left(-\beta \, \mathcal{H}_{\text{plasma}}\right),$$

$$\mathcal{H}_{\text{plasma}} = k^2 \sum_{i<j}^{N} \log|z_i - z_j|^2 - k \sum_{i}^{N} |z_i|^2, \qquad \beta = \frac{1}{k}, \qquad (3.38)$$

can be thought of as the partition function of a classical two-dimensional Coulomb plasma of particles with charge $k$ subject to a uniform neutralizing background. (Verify this for yourself using that the two-dimensional Laplace operator can be written $\nabla^2 = 4\partial_z \partial_{\bar{z}}$.).

It is known that when the "temperature" of this "analogous" plasma is low, $k \ll 70$, it is in a "screening phase" where the Coulomb potential is dynamically screened over a range $O(\ell)$. In the original electronic system, this corresponds to the electrons forming an incompressible liquid with density $\rho_0 = \nu/\pi = 1/\pi k$, and with a gap to density (or sound) wave excitations. The lowest charged excitations are vortices in the fluid which correspond to having terms $k^{-1} \sum_{i}^{N} (z_i - \eta)$ in $\mathcal{H}_{\text{plasma}}$ where $\eta$ is the position of the vortex. This term depletes the plasma at $\eta$, and describes a particle with fractional charge $e/k$. With a bit of work one can also establish that the wavefunction acquires a non-trivial phase $\exp(i\theta)$, with $\theta = \pi/k$, when two such vortices are exchanged, thus demonstrating that they obey fractional exchange statistics [23]. Such particles are called anyons [24]. These and other properties of the Laughlin wave function have been confirmed both numerically and experimentally, as described *e.g.* in [12, 25].

### 3.5.2 The hydrodynamic Chern-Simons theory

The experimentally observed precision and universality in quantum Hall systems, suggest the existence of an effective field theory, just as in the description of critical phenomena [14].

Since the quantum Hall liquids are topological states of matter with a bulk gap, the low energy theory should lack dynamical bulk degrees of freedom, while still coding for quantized conductance, fractionalized excitations and gapless edge modes. Such a "topological" field theory can be derived, or rather guessed, from these properties together with the relevant symmetries [26].

To do so, we first express the conserved matter current $J^\mu$, which describes excitations of the incompressible fluid, in terms of a "dual" vector potential $a_\mu$,

$$J^\mu = \frac{1}{2\pi} \varepsilon^{\mu\nu\rho} \partial_\nu a_\rho, \qquad (3.39)$$

which by construction satisfies $\partial_\mu J^\mu = 0$, and is invariant under the gauge transformation, $a_\mu \to a_\mu + \partial_\mu \varphi$. Since $a_\mu$ parameterizes a current it is referred to as a hydrodynamic gauge field, and is not to be confused with the background electromagnetic field $A_\mu$.

We now claim that the effective hydrodynamic action is,

$$S_{\text{eff}}[a;A] = -\frac{k}{4\pi} \int d^3 x \, \varepsilon^{\mu\nu\rho} a_\mu \partial_\nu a_\rho + \frac{1}{2\pi} \int d^3 x \, A_\mu \varepsilon^{\mu\nu\rho} \partial_\nu a_\rho + \int d^3 x \, \mathcal{J}_\mu a^\mu, \qquad (3.40)$$

and go on to motivate the individual terms. The first is the celebrated Chern-Simons action,

$$S_{CS}[a] = \frac{1}{4\pi} \int d^3 x \, \varepsilon^{\mu\nu\rho} a_\mu \partial_\nu a_\rho, \qquad (3.41)$$

which is gauge invariant and dominates at low energy because it has one derivative less than the Maxwell term. The coupling constant $k$ is called the "level", and should take integer values for the proper definition of the theory. [14]

---

[14]This quantization is related to that of the monopole charge (2.28), see Sect. 2.3 in [27].

Fixing the gauge $a_0 = 0$, the action reduces to a symplectic form, $S \sim \int a_1 \dot{a}_2 \sim \int p\dot{q}$, and the Hamiltonian vanishes. There are no propagating "photons" associated to this gauge theory: the Chern-Simons action only describes global effects that occur in the Hall system at energies below the bulk gap. Furthermore, it violates time reversal (TR) and parity (P) invariance, as is pertinent for describing quantum Hall systems. The properties of this theory will be discussed in more detail later.

The second term in (3.40) is just the electromagnetic potential coupling to the current (3.39), and since the action is quadratic in $a$ it can be integrated out using the equations of motion to get the response action,

$$S_{\text{eff}}[A] = \frac{1}{4\pi k} \int d^3x \, \varepsilon^{\mu\nu\rho} A_\mu \partial_\nu A_\rho \, , \tag{3.42}$$

having neglected the last term in (3.40). From (3.42) we can calculate the electromagnetic charge and current by,

$$J_{\text{ind}}^\mu = \frac{\delta S_{\text{eff}}[A]}{\delta A_\mu} = \frac{1}{2\pi k} \varepsilon^{\mu\nu\rho} \partial_\nu A_\rho, \tag{3.43}$$

that are induced by background variations and correspond to the response of the system. Written in components, they read,

$$\rho_0 = \sigma_H B \, , \qquad J_{\text{ind}}^i = \sigma_H \varepsilon^{ij} E_j \, , \qquad \sigma_H = \frac{1}{2\pi k} \, . \tag{3.44}$$

These expressions reproduce the desired results for charge density and conductance if we identify the level number $k$ of the Chern-Simons action with the $k$ in the Laughlin wave function.

To understand the meaning of the last term in (3.40), consider the equation of motion in the presence of a static source $\mathcal{J}^0$,

$$\frac{1}{2\pi} b = \frac{1}{2\pi k} B + \frac{1}{k} \mathcal{J}^0 \equiv \rho_{em} \, , \tag{3.45}$$

where $b(B)$ is the magnetic field of $a(A)$ field and we used (3.44). Equation (3.45) shows that the charge of the current $\mathcal{J}$ is $1/k$ times the electric charge, so we can identify it with the quasiparticle current.

We also learn from (3.45) that a static quasihole with fractional charge $Q = n/k$, represented by the source $\mathcal{J}_0(x) = n\delta^{(2)}(x)$, generates a flux $b = n/k$ of the field $a_\mu$. Thus, when another quasi-hole with charge $m/k$ winds around it we expect an Aharonov-Bohm phase, and a calculation gives the braiding phase,[15]

$$2\theta = \frac{2\pi nm}{k} \, , \tag{3.46}$$

which by definition is twice the "exchange" phase that determines the quantum statistics [28].

These results show that the Chern-Simons theory (3.45) also describes excitations with fractional charge and fractional exchange statistics, in agreement with the Laughlin theory. Among the excitations, the electron has $Q = 1$ and statistics $\theta/\pi = k$, so the level number $k$ is odd consistent with Fermi statistics.

A final important remark: The Chern-Simons action (3.40) does not change in curved space, *i.e.* in the presence of a background metric $g_{\mu\nu}$, since the usual $\sqrt{g}$ factor in the measure of integration cancels against an inverse factor coming from the covariant form of the antisymmetric tensor $\varepsilon^{\mu\nu\rho}/\sqrt{g}$. Since the action does not couple to the metric, it is invariant under smooth changes of the geometry. In physical terms, this means independence on lattice defects, strain, and impurities.

---

[15]Some care is needed not to get a factor of 2 wrong when doing this.

## 3.6  Chiral Luttinger theory of edge excitations

In this section we show how the Chern-Simons effective action allows us to derive the $(1+1)$-dimensional theory of edge excitations also for fractional fillings. We shall then canonically quantize this theory and explore the consequences of chiral and conformal anomalies [17].

In the $a_0 = 0$ gauge, and in absence of $A_\mu$ background and sources, the equations of motion of the Chern-Simons action (3.41) imply $a_i = \partial_i \varphi$. In a disk $\mathcal{D}$, the action reduces to a boundary term ($\mathcal{C} = \partial \mathcal{D}$),

$$S_{\text{edge}} = -\frac{k}{4\pi} \int_{\mathcal{C}} d^2x \, \partial_x \varphi \, \partial_t \varphi \, , \tag{3.47}$$

where $d^2x \equiv dt \, dx$, with $x = r\theta$ the circle coordinate. This is just a symplectic form, that defines the canonical brackets as discussed later. The Hamiltonian is zero as expected, since (3.47) derives from the topological Chern-Simons action.

We should introduce a dynamics for $\varphi$, and the simplest choice is to add a term quadratic in the momenta, $H \sim \partial_x \varphi \partial_x \varphi$ to get the "chiral Luttinger liquid" [26],

$$S_{\text{edge}} = -\frac{k}{4\pi} \int_{C} d^2x \, \partial_x \varphi \, (\partial_t \varphi + v_F \partial_x \varphi) \, , \tag{3.48}$$

which is called "chiral boson" in the conformal field theory literature [29]. A simple physical way to derive the Hamiltonian in (3.48) is to assume a constant electric field $E_y = E$ transverse to the edge. For an edge excitation with profile $h(x)$, the density is $\rho(x) = \rho_0 h(x)$, with $\rho_0 = B/(2\pi k)$, and the electrostatic energy density $\mathcal{E}$ is,

$$\mathcal{E}(x) = \int_0^{h(x)} dy \, \rho_0 E y = E \rho_0 \frac{1}{2} h(x)^2 = \frac{E}{B} \pi k \, \rho(x)^2 = \pi k v_F \rho(x)^2 = \frac{k v_F}{4\pi} (\partial_x \varphi)^2 \, , \tag{3.49}$$

where we wrote the electrostatic potential $E y$, the drift velocity at the edge $v_F = E/B$ and expressed the density in terms of the scalar field (see (3.59) below),

$$\rho(x) = -\frac{1}{2\pi} \partial_x \varphi \, . \tag{3.50}$$

It also follows that the conserved electromagnetic current is,

$$J^\mu = -\frac{1}{2\pi} \epsilon^{\mu\nu} \partial_\nu \varphi \qquad (\mu, \nu = t, x) \, . \tag{3.51}$$

The equation of motion of the edge theory (3.48) is,

$$(\partial_t + v_F \partial_x) \partial_x \varphi = 0, \tag{3.52}$$

which shows that excitations are indeed gapless and chiral (we set the velocity $v_F = 1$ in the following). Note the residual gauge symmetry $\varphi \to \varphi + \text{const.}$. The Hamiltonian does not include a potential term because it would gap the edge excitations.

We can summarize the discussion so far by saying that the bulk gauge degrees of freedom have become dynamical at the edge. In the following, we canonically quantize the theory (3.48) and derive the axial and conformal anomalies (later we shall understand why this is possible in a bosonic theory). We also clarify the issue of gauge invariance of the whole system, made by bulk and edge.

As mentioned earlier, one important property of the scalar field $\varphi$ is that it is periodic, being the phase of the $U(1)$ gauge field,

$$u \in U(1) \, , \qquad u = e^{i\varphi} \, , \tag{3.53}$$

the period of $\varphi$ being $2\pi$. We shall temporarily consider the more general period $2\pi r$, where $r$ is the so-called compactification radius [7],

$$\varphi(x,t) \equiv \varphi(x,t) + 2\pi n r, \qquad n \in \mathbb{Z}. \tag{3.54}$$

Let us rescale the coordinates $t \to Rt$ and write $x = R\theta$, for simplicity. The field $\varphi(\theta)$ maps the edge circle $\theta \in (0, 2\pi)$ into another circle $\varphi \in (0, 2\pi r)$. Using the solutions of the field equations we can expand it as,

$$\varphi(\theta, t) = \varphi_0 - \alpha_0(\theta - t) + i \sum_{m \neq 0} \frac{\alpha_m}{m} \exp(im(\theta - t)), \tag{3.55}$$

where $\alpha_m^* = \alpha_{-m}$. Moreover, $\varphi_0 \equiv \varphi_0 + 2\pi r$ in order to satisfy (3.54). Note that the field expansion contains both chiral waves and "solitonic" modes $(\varphi_0, \alpha_0)$ that are nontrivial for compact fields.

The action (3.48) is first order in time-derivatives, and thus already in Hamiltonian form, $S \sim \int p\dot{q} - H(p,q)$, with the first term being the symplectic form. The way to quantize such a theory differs from the usual canonical procedure: Rather than dividing the phase space into coordinates and momenta, and imposing canonical quantization conditions, the commutators are obtained directly from the symplectic form. In our case, the detailed recipe, that is given in [29],[16] yields,

$$\left[\varphi(\theta, t), \partial_x \varphi(\theta', t)\right] = \frac{2\pi i}{k} \delta(\theta - \theta'). \tag{3.56}$$

Upon substituting the expansion (3.55), we obtain the following commutation relations among the modes,

$$[\varphi_0, \alpha_0] = \frac{i}{k}, \qquad [\alpha_n, \alpha_m] = \frac{n}{k}\delta_{n+m,0}. \tag{3.57}$$

The quantum operators $\varphi_0$, $\alpha_0$ and $\alpha_n$ act in a bosonic Fock space, whose ground state $|\Omega\rangle$ is defined by,

$$\alpha_n |\Omega\rangle = 0, \qquad n \geq 0. \tag{3.58}$$

In the previous section, we saw that the Weyl fermion corresponds to a conformal theory with central charge $c = 1$. We shall find that the chiral boson also has $c = 1$ and, furthermore, that it can describe the edge states at both integer and fractional fillings, for different values of the parameter $k$. We first observe that the $\alpha_n$ commutation relation (3.57) are the same as the current algebra relations (3.30) obeyed by the modes $\rho_n$ of the edge current in the Weyl fermion theory, for $k = 1$. Our definition of the edge charge is also consistent with the bulk expression (3.39) as follows,

$$Q = \int_{\mathcal{D}} dx^1 dx^2 J^0 = -\frac{1}{2\pi} \int_{\mathcal{C}} d\theta \, a_\theta = -\frac{1}{2\pi}(\varphi(2\pi) - \varphi(0)) = \alpha_0, \tag{3.59}$$

so we can identify $\alpha_n = \rho_n$ for all $n$.

The Virasoro generators are again introduced as the modes of the Hamiltonian and are therefore quadratic in the $\alpha_n$,

$$L_n = \frac{k}{2} \sum_{l=-\infty}^{\infty} : \alpha_{n-l}\alpha_l : . \tag{3.60}$$

---

[16]This method is very useful also for quantizing fermion and Chern-Simons actions which are also first order in time derivatives.

This expression requires normal ordering, consistent with the ground-state conditions (3.58), that amounts to setting the $\alpha_n$ with positive index to the right of those with negative index. This procedure only affects $L_0$, that becomes,

$$L_0 = \frac{k}{2}\alpha_0^2 + k\sum_{l=1}^{\infty} \alpha_{-l}\alpha_l. \tag{3.61}$$

Using the commutation relations (3.57) and paying attention to normal ordering, we can obtain the rest of the current algebra relations [17],

$$[L_n, \alpha_m] = -m\alpha_{n+m}, \tag{3.62}$$

$$[L_n, L_m] = (n-m)L_{n+m} + \frac{c}{12}n(n^2-1)\delta_{n+m,0}, \qquad c = 1. \tag{3.63}$$

It is apparent that these have the same form as in the Weyl fermion, thus showing that the chiral boson is a conformal field theory with central charge $c = 1$ and current-algebra symmetry. It is, however, more general than the fermion theory, owing to the freedom to choose $k \neq 1$.

## 3.7 Spectral flow and fractional Hall current

The minimal coupling of the chiral boson to the electromagnetic field amounts to adding the following term to the action (3.48),

$$S_{int} = -\int_{\partial C} d^2x\, J^\mu A_\mu = \frac{1}{2\pi}\int_{\partial C} d^2x\, (A_t\partial_x\varphi - A_x\partial_t\varphi), \tag{3.64}$$

where we used the expression (3.51) for the current and added a minus sign for the electron charge. The equation of motion becomes,

$$(\partial_t + \partial_x)\partial_x\varphi = -\frac{1}{k}E^x. \tag{3.65}$$

Integration over space, $x \in [0, 2\pi R]$, leads to charge non-conservation, i.e. to the chiral anomaly,

$$\partial_t Q = -\frac{1}{2\pi k}\int_C dx\, E^x. \tag{3.66}$$

We can now repeat the spectral flow argument and obtain, after the insertion of $n$ flux quanta $\Phi_0$,

$$\Delta Q = \frac{1}{4\pi k}\int_{S^1 \times S^1} d^2x\, \varepsilon^{\mu\nu}F_{\mu\nu} = \frac{n}{k}. \tag{3.67}$$

This charge variation corresponds to a Hall conductivity $\sigma_H = 1/(2\pi k)$, as seen from the perspective of edge physics. The result is again expressed in terms of the topological quantity given by the first Chern class of the gauge field (2.28), and is exact in the low-energy limit in which the effective field theory was derived. Note that this limit is fully justified since the adiabatic process occurs in the presence of a bulk gap. Furthermore, the charge non-conservation at the edge is compensated by the bulk radial current related to the anomaly inflow mechanism already explained for integer fillings in Sect. 3.4.

In conclusion, the chiral boson theory displays the chiral anomaly and generalizes the anomaly inflow mechanism to fractional fillings. It is rather remarkable that the exact quantization of the Hall current can be proven in interacting systems by using the effective field

theory approach and anomalies. Further exact results due to the gravitational anomaly will be discussed later. Finally note that restoring $\hbar$ in (3.67) we get $\Delta Q \sim \hbar$ as expected.

That the bosonic low-energy theory, for $k = 1$, reproduces the anomaly present in the original fermionic theory is an example of the 't Hooft anomaly matching conditions [30, 31]. These put important constraints on low-energy theories by requiring them to have the same anomalies as the microscopic theories from which they derive.

## 3.8 Excitation spectrum and bosonization of $(1+1)$-dimensional fermions

To determine the spectrum of allowed charged excitations in the bosonic theory, we must understand the quantization of the solitonic modes in (3.57). Since $\varphi_0$ is $2\pi r$ periodic, and $\varphi_0$ and $k\alpha_0$ are canonically conjugate, we have the quantization, $k\alpha_0 = n/r$, with $n \in \mathbb{Z}$. Thus, there are two periodicities,

$$\varphi(2\pi, t) = \varphi(0, t) + 2\pi r m - 2\pi \frac{n}{kr}, \qquad n, m \in \mathbb{Z}. \tag{3.68}$$

We require the two periods to be commensurable, then $kr^2$ takes rational values, $kr^2 = p/q$, with $p$ and $q$ coprime integers [17]. In the conformal field theory literature, this choice is called "rational compactification", because the spectrum of conformal dimensions contains a finite number of fractional values, in agreement with the observed critical exponents of two-dimensional statistical models [16]. Indeed, from the expressions (3.61) and (3.58), we have $L_0 = k\alpha_0^2/2 = n^2/(2kr^2) = n^2 q/(2p)$. In the application to the fractional Hall effect, we know that $r = 1$ and $k$ is integer, so we should identify $p = k$ and $q = 1$. Thus, the spectrum of charged excitations is given by,

$$\alpha_0 = \frac{n}{k}, \qquad n \in \mathbb{Z}. \tag{3.69}$$

We conclude that the fractional values (3.69) obtained by canonical quantization of the bosonic edge theory match the charges $Q = n/k$ found in the bulk Chern-Simons theory. Note that an anyonic bulk charge corresponds to a solitonic mode in the edge theory, which is another instance of the bulk-boundary correspondence.

As already explained in the case of the Weyl fermion, excitations of the conformal field theory form infinite towers of states. Each of them is characterized by a bottom state, called highest-weight state in the CFT literature [16], that actually is the ground state in the solitonic sector $\alpha_0$,

$$\alpha_0 \left| \frac{n}{k} \right\rangle = \frac{n}{k} \left| \frac{n}{k} \right\rangle, \qquad n \in \mathbb{Z}. \tag{3.70}$$

This state obeys $\alpha_n|\alpha_0\rangle = 0$, for $n > 0$, while its infinite tower of particle-hole excitations are obtained by repeatedly acting with $\alpha_n$, $n < 0$. The bottom state is also characterized by a value of $L_0$,

$$L_0 \left| \frac{n}{k} \right\rangle = \frac{n^2}{2k} \left| \frac{n}{k} \right\rangle. \tag{3.71}$$

Let us now verify that these angular momentum values reflect the anyonic statistics of the edge excitations, which is in agreement with the results for the bulk Chern-Simons theory. We first need the field that creates a charged excitation at a point on the edge. This is the so-called vertex operator, which is well known in conformal field theory [7],

$$V_n(z) =: \exp(in\varphi(z)) :, \tag{3.72}$$

Note that powers of the field $\varphi$ are well defined thanks to the normal ordering prescription for the $\alpha_n$ modes in (3.58). The commutation relation,

$$[\alpha_0, V_n(z)] = \frac{n}{k} V_n(z), \tag{3.73}$$

is easily derived since (3.58) implies $\alpha_0 \sim (-i/k)\partial/\partial\varphi_0$, and shows that the vertex operators indeed describe the insertion of a fractional charge $Q = \alpha_0 = n/k$ at the point $z = Re^{i\theta}$ on the boundary.

Next we compute the two point function of vertex operators. Using the mode expansion (3.55), and the ground state conditions (3.58), we get the correlator of two currents,

$$\langle\Omega|\partial_\theta\varphi(\theta,0)\,\partial_\chi\varphi(\chi,0)|\Omega\rangle = \frac{zw}{(z-w)^2}, \qquad z = Re^{i\theta},\ w = Re^{i\chi}, \tag{3.74}$$

and upon integration in $z$ and $w$ we obtain the field correlator, $\langle\varphi(\theta)\varphi(\chi)\rangle = -\log(z-w)$. We now have the elements to compute the vertex operator two-point function. By expanding the exponentials in power series and using the $\langle\varphi\varphi\rangle$ correlator for any pair of fields [14], we get,[17]

$$\langle V_n(z)V_n(w)\rangle = (z-w)^{n^2/k}. \tag{3.75}$$

Upon setting the two points opposite to each other on the edge, i.e. $z = we^{i\pi}$, and then moving both of them around by an angle $\pi$, we can realize the exchange of the two excitations in the plane. In this process, the edge correlator (3.75) acquires the phase $\exp(i\pi n^2/k)$, corresponding to the fractional statistics already found in the Chern-Simons theory.

Another fundamental property implied by the expression (3.75) is that the vertex operators with $n = \pm 1$, for $k = 1$, represent Weyl fermion fields, as follows,

$$\psi(\theta,t) \equiv V_{-1} = {:}\exp(-i\varphi(\theta,t)){:}, \qquad \psi^\dagger(\theta,t) \equiv V_1 = {:}\exp(i\varphi(\theta,t)){:}. \tag{3.76}$$

This is the bosonization of fermions in $(1+1)$ dimensions, an exact map between the two theories, that actually are two descriptions of the same Hilbert space of states [7]. We already saw that they have the same conformal charge $c = 1$ and realize the same chiral algebra relations (cf. (3.30) – (3.32) and (3.57), (3.63)). Also, the two theories (for $k = 1$) share the same set of representations of this algebra, and the partition functions and multi-point correlators are also found to be equal.

It has been shown that the correspondence extends to bosons at $k \neq 1$ and fermions with current-current interaction (four-fermions coupling) [32]. This relation is rather natural from the point of view of bulk Hall physics, since the Laughlin state describes interacting fermions, and the interaction extends to the edge [26]. The bosonic theory described in these sections has been extensively applied to the dynamics of edge excitations and has been important for interpreting many experiments [25]. In particular, the fractional charges (first observed in Refs. [33,34]), and the resonant tunneling and fractional statistics recently observed [35].

We close this section with a couple of remarks.

- The bosonization of fermions in (1+1) dimensions makes it clear how the chiral anomaly is manifested in a bosonic theory.

- In conformal theories involving both chiralities, i.e. $c = \bar{c}$, the correlation functions of chiral vertex operators generalize to,

$$\langle V_n(z)V_n(w)\rangle = (z-w)^{2h}(\bar{z}-\bar{w})^{2\bar{h}}, \tag{3.77}$$

---

[17]The experts of conformal field theory should excuse us for the little abuse of notation at this point.

where $(h, \bar{h})$ are the eigenvalues of $(L_0, \bar{L}_0)$. The sum $\Delta = h + \bar{h}$ is called the conformal (or scaling) dimension, because it gives the power-law behavior of the field under scale transformations. The difference, $s = h - \bar{h}$, is an angular momentum that is referred to as "conformal spin" in the CFT literature, or "orbital spin" in the context of quantum Hall physics.

## 3.9  Thermal currents and the gravitational anomaly

We now discuss the coupling of the conformal theory of edge excitations to a metric background. While a general analysis of the field theory response to deformations of the geometry will be presented in Sect. 8, here we shall discuss a simple physical consequence of the trace and gravitational anomalies introduced in (3.22) and (3.21). The notations for curved space calculus are summarized in App. E.

We consider theories that have both chiral and antichiral components and corresponding central charges $(c, \bar{c})$: The gravitational anomaly generalizing (3.21) is [36],

$$D^z T_{zz} = \frac{c}{48\pi} \partial_z \mathcal{R}, \qquad D^{\bar{z}} T_{\bar{z}\bar{z}} = \frac{\bar{c}}{48\pi} \partial_{\bar{z}} \mathcal{R}, \qquad T_{z\bar{z}} = T_{\bar{z}z} = 0, \tag{3.78}$$

where we use the complex Euclidean coordinates $z = x_0 + ix_1$, $\bar{z} = x_0 - ix_1$ that allow for the factorization of conformal transformations as explained in Sect. 3.3 (*cf.* (3.14) and (3.15)); the corresponding vector indices are raised and lowered with the metric, $g_{z\bar{z}} = g_{\bar{z}z} = 1/(g^{z\bar{z}})$ and $g_{zz} = g_{\bar{z}\bar{z}} = 0$ [7].

The non-conservation laws (3.78) imply that the stress tensor does not transform covariantly under conformal transformations, but acquires an additive term proportional to the central charge [7]. When mapping from the plane to another geometry, like the cylinder having one periodic coordinate, this additive term causes a non-vanishing ground state value, that is the source of the 1d Casimir effect [14].

Next consider the edge theory at finite temperature $T$, by taking the Euclidean time periodic, with period $\beta = 1/(k_B T)$. In the limit where the spatial period $R$ is much larger than $\beta$, the theory is effectively defined on a "time cylinder". In this geometry, the ground-state expectation value of the stress tensor gives the energy density,

$$\mathcal{E}(x) = \langle \Omega | T_{00}(x) | \Omega \rangle = \frac{\pi c}{12 v_F \beta^2}. \tag{3.79}$$

This result is very general as it holds for any chiral conformal theory with central charges $(c, 0)$, $v_F$ being the velocity of edge excitations.[18] In a chiral theory, the energy density $\mathcal{E}$ is proportional to the momentum density, $\mathcal{P}$, by $\mathcal{P} = v_F \mathcal{E}$. Furthermore, in the presence of antichiral excitations there is another contribution corresponding to (3.78), but parameterized by $\bar{c}$, that adds to the energy and subtracts from the momentum,

$$\mathcal{E}(x) = \frac{\pi k_B^2 T^2}{12 v_F}(c + \bar{c}), \qquad \mathcal{P}(x) = \frac{\pi k_B^2 T^2}{12}(c - \bar{c}). \tag{3.80}$$

In applications of conformal theories to statistical models, one generally deals with parity invariant theories where $c = \bar{c}$; in this case, $\mathcal{P}(x)$ vanishes and $\mathcal{E}(x)$ determines the specific heat by $c_V = \partial \mathcal{E}/\partial T$. In the quantum Hall effect, it is rather natural to have edge excitations with $c \neq \bar{c}$, implying a nonvanishing momentum density that is a matter, or thermal, edge current $J_T \equiv \mathcal{P}$, as we now explain.[19]

---

[18] The proof of this result is rather technical and is given in App. E.

[19] The specific heat of the edge excitations is actually negligible in comparison with the lattice phonon contribution.

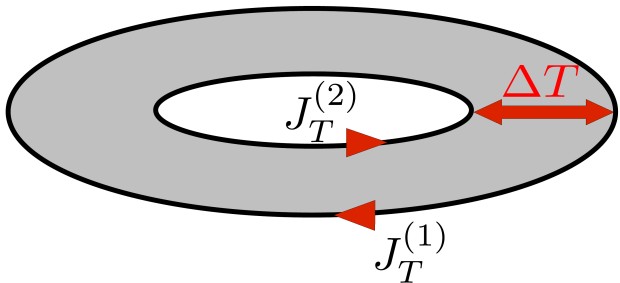

Figure 7: Annular geometry with the two edges at different temperatures and their thermal currents.

While the Laughlin states, are purely chiral, with $\bar{c} = 0$, other prominent plateaus belonging to the Jain series with filling fractions,

$$\nu = \frac{n}{2pn \pm 1}, \qquad n, p = 1, 2, \ldots, \tag{3.81}$$

have, in general, both chiralities. As explained in App. B, these states can be described as $n \geq 1$ filled "effective Landau levels". The edge excitations correspondingly possess $n$ branches: for the $(+)$ sign in (3.81), all branches are chiral, thus making a conformal theory with $(c, \bar{c}) = (n, 0)$. For the $(-)$ sign, there is one chiral branch and $n - 1$ antichiral branches of neutral excitations, corresponding to central charges $(c, \bar{c}) = (1, n - 1)$.

Let us consider the annular geometry of Fig. 7, and assume that the two edges are in thermal equilibrium. The expression (3.80) refers to one edge, say the outer one. At equilibrium the currents on the two edges add up to zero, $J_T^{(1)}(T) + J_T^{(2)}(T) = 0$. If we heat up one edge, there is a disequilibrium and a net heat flow,

$$\Delta J_T = J_T^{(1)}(T + \Delta T) + J_T^{(2)}(T) \sim \frac{\partial J_T^{(1)}}{\partial T} \Delta T = \kappa_H \Delta T. \tag{3.82}$$

Using (3.80), we find that the thermal Hall conductance $\kappa_H$ is [37] [36],

$$\kappa_H = \frac{\pi k_B^2 T}{6} (c - \bar{c}). \tag{3.83}$$

Note that the thermal current is orthogonal to the temperature gradient, just as the Hall current is orthogonal to the potential gradient.

We have shown that $\kappa_H$ is proportional to the gravitational anomaly (3.22); this is another example of how anomalies determine universal transport properties, thus measuring in a unique way a fundamental property of the underlying effective theory. The result is very general and applies to any conformal theory of quantum Hall edge excitations with specific central charges $(c, \bar{c})$.

**Experimental confirmations.** The thermal current has been measured and the results agree fairly well with the prediction (3.83). The experimental probe used in Ref. [38] has a geometry with four arms and corresponding edge currents, that can be switched on and off. One measures the power $\Delta P$ dissipated by the thermal currents flowing in $N$ edges,

$$\Delta P(N) = N \frac{\kappa_H T}{2} = N \frac{\kappa_0 T^2}{2} |c - \bar{c}|, \qquad \kappa_0 = \frac{\pi k_B^2}{6}, \tag{3.84}$$

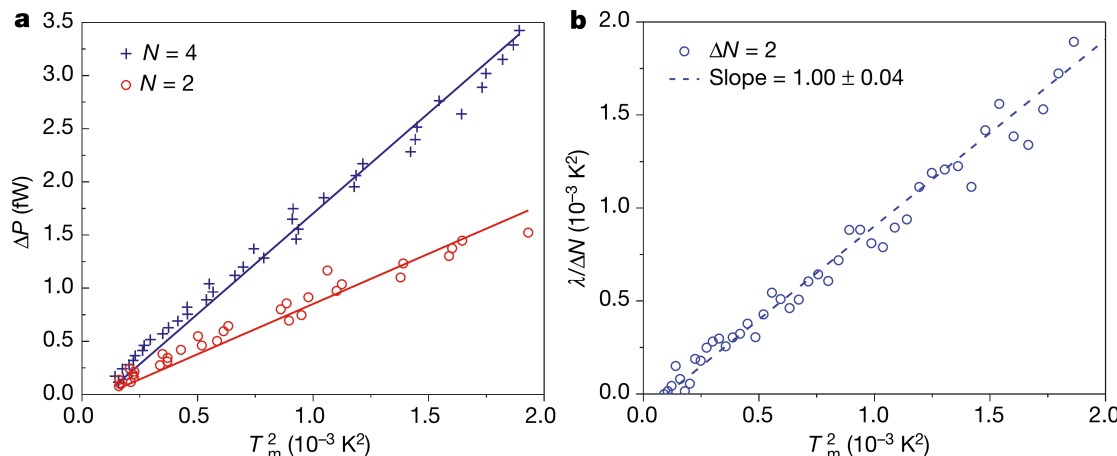

Figure 8: Measurement of the thermal current for $\nu = 1/3$ [38]: (a) Power dissipation $\Delta P$ as a function of $T^2$ for $N = 2, 4$ active edges (currents), compared with the theoretical prediction (3.84) for one chiral mode; (b) Difference of power dissipations for $N = 4$ and $N = 2$ edges, and linear fit for measuring the slope $|c - \bar{c}|$ in (3.85).

as well as the difference of powers $\lambda$ between measures with different number of active edges $N_1$ and $N_2$, suitably normalized,

$$\frac{\lambda}{\Delta N} = \frac{\Delta P(N_2) - \Delta P(N_1)}{\Delta N \kappa_0 / 2} = T^2 |c - \bar{c}|, \qquad \Delta N = N_1 - N_2 . \qquad (3.85)$$

This quantity is useful because the constant contribution by lattice phonons cancels out. Note that the experiment cannot determine the sign of the thermal current, *i.e.* of $(c - \bar{c})$.

The results for the Laughlin state at $\nu = 1/3$ are shown in Fig. 8, where one sees that the experiment measures $|c - \bar{c}| = 1.00 \pm 0.04$, in good agreement with the chiral Luttinger theory. The measures for filling fractions belonging to the Jain series (3.81) are reported in Fig. 9. In these cases, there can be negative (upstream) contributions to the current by the antichiral modes. For $\nu = 3/5$, for example, the theory predicts one chiral and two antichiral modes leading to $|c - \bar{c}| = |1 - 2| = 1$; the experimental result $1.04 \pm 0.04$ confirms the expectations. Other cases are also fairly well verified. For $\nu = 2/3$, there is a discrepancy between the predicted $c - \bar{c} = 1 - 1 = 0$ and measured $0.33 \pm 0.02$ values, which might be explained by the lack of thermal equilibration between the up- and down-stream modes [39].

Let us finally add some remarks.

- The determination of the thermal Hall conductance by the gravitational anomaly is a genuine result of the conformal theory of edge excitations. In Laughlin and Jain states, each edge mode contributes $\pm 1$ to the central charge, so the expression (3.83) reduces to a mode counting. However, non-trivial fractional values are found in edge theories involving excitations with non-Abelian fractional statistics, for example, $(c, \bar{c}) = (3/2, 0)$ for the Pfaffian state at $\nu = 5/2$, including a bosonic edge mode and a neutral Majorana fermion [40].

- The measurement of the thermal current is particularly interesting in cases where there are several competing theories for the same filling fraction, as for example at $\nu = 5/2$

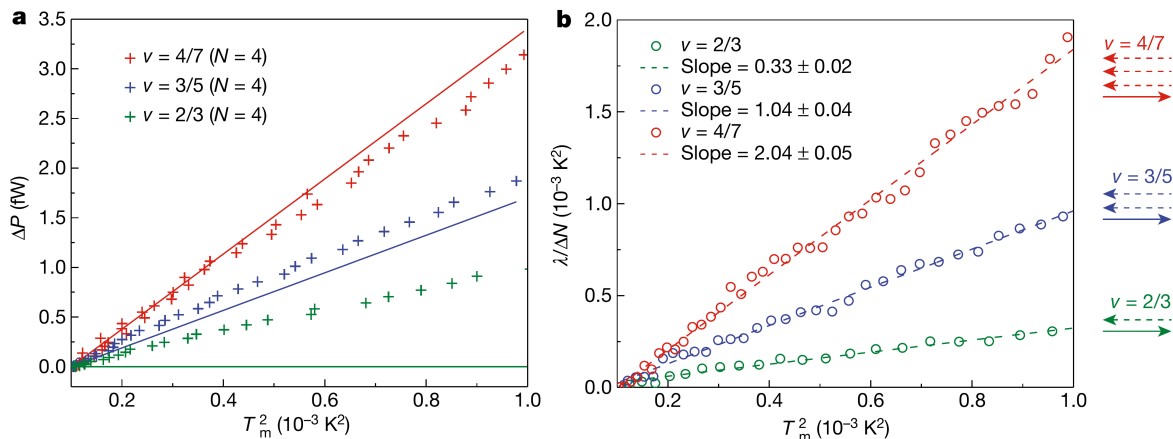

Figure 9: Measurement of thermal current for Jain states with both chiral and antichiral edge excitations at $\nu = 2/3, 3/5, 4/7$ [38]: (a) Power dissipation $\Delta P$ as a function of $T^2$ for $N = 4$ edges versus (3.84) with $|c - \bar{c}|$ predicted values for the up- and down-stream modes shown on the right; (b) Difference of power dissipations between $N = 4$ and $N = 2$ edges and linear fits determining the experimental $|c - \bar{c}|$ values in (3.85).

where two other theories have been proposed beside the Pfaffian state. Unfortunately, the measures of the thermal current are not yet definitive [39]. The determination of the correct theory could confirm the presence of non-Abelian anyons. It has been proposed that using them, one could model qubits that are decoherent-free due to their topological nature and realize a platform for quantum computations [41].

- The anomaly inflow, *i.e.* the compensation of the edge anomaly by a bulk current, also takes place in the gravitational case as discussed in Sect. 8.

- The spacetime integral of the trace anomaly (3.22) is again related to a topological invariant quantity: Actually, the integral of the scalar curvature $\mathcal{R}$ gives the so-called Euler characteristic,

$$\chi = \frac{1}{4\pi} \int_{\mathcal{M}} d^2 x \sqrt{g} \mathcal{R} = 2 - 2g - b = n_F - n_E + n_V. \qquad (3.86)$$

This integer counts the number of handles $g$ and boundaries $b$ of the two-dimensional surface $\mathcal{M}$, as well as the alternating sum of faces (F), edges (E) and vertices (V) in a triangulation of the surface [42].

# 4 Anomalies from perturbation theory

Although the method to calculate the axial anomaly described in Sect. 2 has a great pedagogical value, and can with some extra effort be generalized to higher spacetime dimensions [4], it cannot in any obvious way be used to calculate non-Abelian or gravitational anomalies. We now describe how to compute anomalies from perturbation theory using Feynman diagrams. Later we will return to the infinite hotel in the 4D.

## 4.1 2D anomalies from perturbation theory

We shall make the calculations in Euclidean space-time, and for this we need some notation and conventions ($D = d + 1$),

$$x \equiv x_E^\mu = (\vec{x}, ix^0) = (\vec{x}, x^D) \,, \qquad p \equiv p_\mu = (\vec{p}, -ip_0) = (\vec{p}, p_D) = -i(\vec{\partial}, \partial_D) \,. \qquad (4.1)$$

With the conventions given in App. A we write the Euclidean action of a Dirac fermion in the presence of the gauge backgrounds $A_\mu$ and $A_{5\mu}$ as follows,

$$S_E = \int dx_E \, \bar{\psi}(i\slashed{D}_E + im)\psi \,, \qquad (4.2)$$

where $i\slashed{D}_E = \gamma_E^\mu(i\partial_\mu + A_\mu + A_{5\mu}\gamma^{D+1})$ is an Hermitian operator, and $dx_E$ is the D-dimensional Euclidean measure (when there is no risk of confusion we will just write $dx$). We also included the mass for future reference.[20]

The quantization of the fermion field is defined by the path integral,

$$Z[A, A_5] = e^{-\Gamma[A, A_5]} = \mathcal{N} \int \mathcal{D}\bar{\psi}\mathcal{D}\psi \, e^{-S_E[\psi, \bar{\psi}, A, A_5]}, \qquad (4.3)$$

where the normalization constant is given by, $\mathcal{N} = 1/Z[0, 0]$. The quantity $\Gamma[A, A_5]$ gives the response of the quantum system to varying backgrounds. It is the same as $S_{\text{eff}}[A]$ (3.42) introduced in the previous section; both are called "effective actions", but condensed matter and particle physics papers often use different notations. In the following, we shall use both $S_{\text{eff}}[A]$ and $\Gamma[A]$ for this response action, adapting to the standard conventions of the topic discussed.

We now proceed to calculate the effective action $\Gamma[A, A_5]$ in two dimensions to quadratic order in the gauge fields. This is obtained by expanding the logarithm of the fermionic determinant, as follows,

$$\begin{aligned}
\Gamma[A, A_5] &= \text{Tr}\left[-\ln\left(i\slashed{\partial} - \slashed{A} - \slashed{A}_5\gamma^3\right) + \text{Tr}\ln(i\slashed{\partial})\right] = \text{Tr}\left[\ln\left(1 - \frac{1}{i\slashed{\partial}}(\slashed{A}_\mu + \slashed{A}_{5\mu}\gamma^3)\right)\right] \\
&= \frac{1}{2}\int \frac{d^2q}{(2\pi)^2}\left(A_\mu(q) + \tilde{A}_{5\mu}(q)\right)\Pi^{\mu\nu}(q)\left(A_\nu(-q) + \tilde{A}_{5\nu}(-q)\right) + \dots \,, \qquad (4.4)
\end{aligned}$$

where the term linear in the potentials vanished because of the gamma matrix trace. We also used the two-dimensional form of the Euclidean gamma matrices, i.e. the Pauli matrices, for expressing $\gamma^3\gamma^\mu = i\varepsilon^{\mu\nu}\gamma_\nu$, such that the axial field is expressed in terms of $\tilde{A}_5^\mu = i\varepsilon^{\mu\nu}A_{5\nu}$. In momentum space the polarization tensor is given by the Feynman integral ($m = 0$),

$$\Pi^{\mu\nu}(q) = \int \frac{d^2p}{(2\pi)^2}\frac{\text{tr}[\gamma^\mu \slashed{p} \gamma^\nu(\slashed{q} + \slashed{p})]}{p^2(p-q)^2} \,, \qquad (4.5)$$

as illustrated in Fig. 10. The loop integral is logarithmically divergent in the ultraviolet, and must be regulated. The normal procedure is to add a local counter term to $\Gamma[A, A_5]$ that removes the singularity, that is actually independent of the external momentum, for dimensional reasons. One finds,

$$\Pi^{\mu\nu}(q) = \frac{1}{\pi}\left(c\delta^{\mu\nu} - \frac{q^\mu q^\nu}{q^2}\right). \qquad (4.6)$$

In this expression, the constant $c$ expresses the freedom to adjust the subtraction by a finite

---

[20]Note that $(i\slashed{\partial} - im)(i\slashed{\partial} + im) = -\partial^2 + m^2 = p^2 + m^2$ which is positive definite.

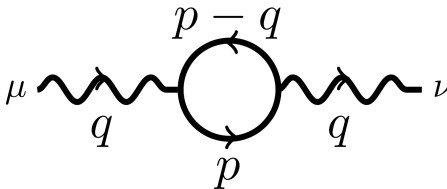

Figure 10: Feynman diagram for the 2D polarization tensor $\Pi_{\mu\nu}(q)$.

amount,[21] after removing the infinite part, it is a so-called choice of renormalization conditions that fully specify the renormalized quantities [1]. In general, this is dictated by some physical requirement. It is apparent that the effective action $\Gamma$ (4.4) cannot be simultaneously gauge invariant with respect to both the vector and axial fields. Vector gauge symmetry would require $c = 1$, *i.e.* a transverse polarization, while axial gauge invariance needs $c = 0$ (Note the two-dimensional identity $\varepsilon_{\mu\alpha} p^\alpha \varepsilon_{\nu\beta} p^\beta = p^2 \delta_{\mu\nu} - p_\mu p_\nu$). We thus have the same two options already encountered in the derivation of the anomaly by point-splitting of currents in Sect. 2.2.

The axial and vector currents are obtained by variation of the effective potential, respectively $J^\mu = \delta\Gamma[A, A_5]/\delta A_\mu$ and $J_\mu^5 = \delta\Gamma[A, A_5]/\delta A_{5\mu}$. Next we compute the divergence of these currents, adding a factor of $(i)$ for switching back to Minkowskian notation, as explained in App. A. With vector gauge invariance, choosing $c = 1$ and $A_{5\mu} = 0$, we find,

$$\partial_\mu J^\mu = 0, \qquad\qquad \partial_\mu J_5^\mu = \frac{1}{\pi} F, \qquad\qquad (4.7)$$

and with axial gauge invariance, choosing $c = 0$ and $A_\mu = 0$,

$$\partial_\mu J^\mu = \frac{1}{\pi} F_5, \qquad\qquad \partial_\mu J_5^\mu = 0, \qquad\qquad (4.8)$$

where $F = \varepsilon^{\alpha\beta} \partial_\alpha A_\beta \equiv E_x$ and $F_5 = \varepsilon^{\alpha\beta} \partial_\alpha A_{5\beta} \equiv E_{5x}$. (Recall that the electric and axial charges are included in the definition of gauge fields).

In conclusion, the perturbative calculation reproduces the earlier results for the 2D anomaly, (2.18) and (2.21), clearly showing that the anomaly depends on the choice of renormalization conditions, and that we can preserve at most one symmetry out of the two. In Sect. 6.1 we shall also discuss the case of "mixed anomalies" when both vector and axial fields are present.

## 4.2    4D anomalies from perturbation theory

We now very briefly outline how the 4D anomalies come about in the perturbative expansion.[22] A more detailed derivation will be given in the following section using path integrals.

Let us first anticipate which Feynman diagrams could contribute to the axial anomaly. As in two-dimensions, we want to calculate the vector and axial currents as functions of the gauge fields $A$ and $A_5$. Since $\partial_\mu J_5^\mu$ is a pseudoscalar quantity, we need the Levi-Civita tensor $\epsilon^{\mu\nu\sigma\lambda}$ for parity to work out: its indices should be contracted with gauge invariant quantities, so we expect the anomaly to be $\sim \epsilon^{\mu\nu\sigma\lambda} F_{\mu\nu} F_{\sigma\lambda}$. From this follows that the relevant diagrams must have one vertex with the axial or vector current insertion, and two vertices with insertions of $A$ and/or $A_5$. They correspond to the third order expansion of the fermionic determinant (4.4). The contributing "triangle diagrams" are shown in Fig. 11 .

Note that diagrams with an even number of $\gamma^5$ insertions vanish because of time reversal invariance, as dictated by Furry's theorem (you should consult your QFT textbook for this!). Thus, as shown in Fig. 11, all diagrams have either a single or three insertions of $\gamma^5$.

---

[21]Not to be confused with the central charge of the previous section.

[22]Details of this analysis can be found in [43]. A comprehensive discussion of 4D Dirac theory is given in [44].

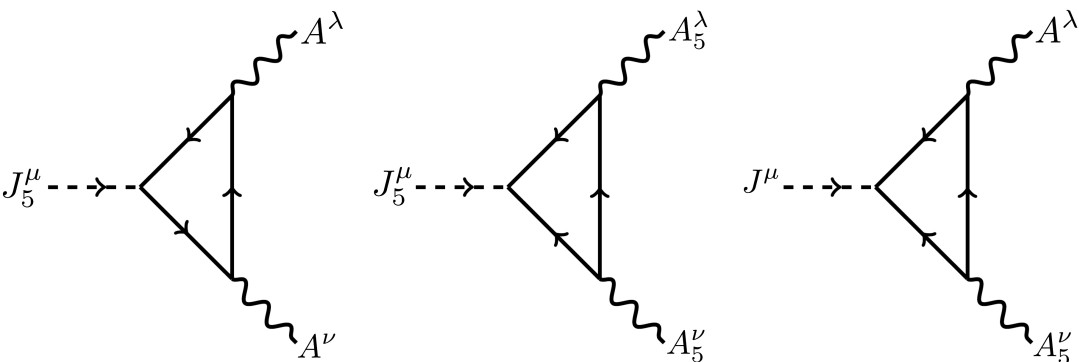

Figure 11: The two leftmost diagrams contribute to the axial current, while the third one is used to calculate the vector current. There are three more diagrams where the external $A$-lines are exchanged.

The Feynman amplitudes take the form,

$$\Pi^{\mu\nu\lambda}(q,k,p) \propto \int \frac{d^4 r}{(2\pi)^4} \frac{\text{tr}[\Gamma^\mu(\not{r}+\not{p})\Gamma^\lambda\not{r}\Gamma^\nu(\not{r}-\not{k})]}{(r+p)^2 r^2 (r-k)^2}, \tag{4.9}$$

where $\Gamma^\nu$ is either $\gamma^\mu$ or $\gamma^\mu\gamma^5$, for vector or axial-vector insertions, respectively. By commuting two $\gamma^5$ matrices and using $(\gamma^5)^2 = 1$ we conclude that all diagrams give the same contribution, although with different statistical factors. The actual evaluation of the integral is a bit tricky since it involves cancellation between linearly diverging terms. In fact, there is no unique answer – although the divergences cancel, the finite result depends on the details of the regularization employed. As in two dimensions, there is a freedom of finite counterterms in the effective action, that are polynomial in the momenta. A direct calculation using Minkowski signature gives [43],

$$\begin{aligned}
q_\mu \Pi^{\mu\nu\lambda} &= \frac{-i}{8\pi^2}(2c)\epsilon^{\nu\lambda\alpha\beta}k_\alpha p_\beta, \\
k_\nu \Pi^{\mu\nu\lambda} &= \frac{-i}{8\pi^2}(1-c)\epsilon^{\lambda\mu\alpha\beta}p_\alpha q_\beta, \\
p_\lambda \Pi^{\mu\nu\lambda} &= \frac{-i}{8\pi^2}(1-c)\epsilon^{\mu\nu\alpha\beta}q_\alpha k_\beta,
\end{aligned} \tag{4.10}$$

where the momentum $q$ flows into the vertex with the $\gamma^5$, and $c$ is the free parameter. The choice $c = 1$ amounts to conserving the vector currents at the other vortices, thus enforcing vector gauge invariance. Setting $A_5 = 0$ we get a contribution from the first diagram in Fig. 11, and using $q_\mu \to -i\partial_\mu$ *etc.* to go back to coordinate space one finally gets

$$\partial_\mu J^\mu = 0, \qquad \partial_\mu J_5^\mu = \frac{1}{16\pi^2}\varepsilon^{\mu\nu\alpha\beta}F_{\mu\nu}F_{\alpha\beta} = \frac{1}{8\pi^2}F_{\mu\nu}\tilde{F}^{\mu\nu}, \tag{4.11}$$

where $\tilde{F}_{\mu\nu} = \varepsilon_{\mu\nu\alpha\beta}F^{\alpha\beta}/2$ is the dual field strength. The result for both $A, A_5 \neq 0$, with contributions from all three diagrams, will be discussed in Sect. 6.1 and App. D: This is the case of so-called mixed anomalies, that requires a further analysis of the anomalous currents, and will be applied to the physics of Weyl semi-metals.

## 4.3 Decay of the neutral pi meson

Here we shall give a short account of how a quantum field theory anomaly for the first time was evoked to explain an observed phenomenon, the decay of the neutral pion $\pi^0$ into two

photons. In this historical excursus we shall assume some knowledge of strong interaction physics.

Before the advent of Quantum Chromodynamics (QCD), the non-Abelian gauge theory of strong interactions, theoretical approaches used effective field theories based on approximate symmetries and phenomenological parameters. The first discovered of these symmetries was the isospin $SU(2)_V \times SU(2)_A$ vector-axial symmetry, in which up and down quarks $u$ and $d$ form a isodoublet. The vector part is responsible for isospin conservation, while the axial symmetry is spontaneously broken. The corresponding three Goldstone bosons are identified as the pseudoscalar pions, $\pi^a = (\pi^+, \pi^0, \pi^-)$, in the account for their masses being much smaller than those of other mesons like $\rho$ and $\omega$. The vector and axial currents are exactly conserved in the limit of vanishing masses, $m_u = m_d = 0$ and $m_{\pi^a} = 0$, called "chiral limit".

The spontaneous symmetry breaking, means that the axial current $\tilde{J}_{5\mu}^a$, $a = 1, 2, 3$, which is the generator of $SU(2)_A$ transformations, does not leave the ground state invariant, but creates a Goldstone boson, leading to the expression [44],

$$\langle \Omega | \tilde{J}_{5\mu}^a(p) | \pi^b \rangle = -i f_\pi \delta^{ab} p_\mu, \tag{4.12}$$

where $|\pi^a\rangle$ is the pion state, $f_\pi \approx 93$ MeV is a phenomenological parameter, and the momentum factor $p_\mu$ signals the Goldstone nature of pions. Taking the derivative of (4.12) we can write a operator relation between the pion field and the divergence of the axial current,

$$\partial_\mu \tilde{J}_5^{\mu a}(x) = f_\pi m_\pi^2 \pi^a(x), \qquad (p^2 = m_\pi^2), \tag{4.13}$$

where $\langle \Omega | \pi^a(p) | \pi^b \rangle = \delta^{ab}$ is the pion field normalization. Equation (4.13) is known as the partially conserved axial current (PCAC) relation, since the current is only conserved in the chiral limit.

However, this spontaneous symmetry breaking argument overlooked the presence of the axial anomaly, and for this reason we used the notation $\tilde{J}_{5\mu}^a$ for the anomaly-free current. The PCAC relation should be modified[23] [45] as follows,

$$\partial_\mu \tilde{J}_5^\mu(x) = f_\pi m_\pi^2 \pi^0(x) + \frac{e^2}{8\pi^2} F_{\mu\nu} \tilde{F}^{\mu\nu} \tag{4.14}$$

by including the contribution of the anomalous Feynman diagram on the left in Fig. 11, according to (4.10) and (4.11).

The old PCAC relation (4.13) would imply the vanishing of $\pi^0$ decay, $\Gamma(\pi_0 \to \gamma\gamma) = 0$, in the chiral limit ($m_\pi = 0$), and even taking into account the non-vanishing mass, the value of $\Gamma$ was three orders of magnitude smaller than the observed value.[24]

The modified PCAC relation (4.14) leads to the following matrix element,

$$\langle \gamma\gamma | \pi^0(p) | \Omega \rangle = \frac{1}{m_\pi^2 f_\pi} \langle \gamma\gamma | \left( i p^\mu \tilde{J}_{5\mu}(p) - \frac{Q^2}{8\pi^2} F\tilde{F}(p) \right) | \Omega \rangle, \tag{4.15}$$

where the anomaly term creates the photons and the contribution of $\tilde{J}_\mu^5$ is now negligible ($Q$ is the charge of the particle circulating the loop). A standard field theory calculation leads to the decay rate [1, 46],

$$\Gamma(\pi_0 \to \gamma\gamma) = S^2 \frac{\alpha^2}{64\pi^3} \frac{m_\pi^3}{f_\pi^2} \approx 7.63 \text{eV}, \tag{4.16}$$

---

[23]We use the $U(1)$ current for the neutral component of the pion triplet.
[24]Weinberg gives a detailed discussion of this estimate in [46].

where $\alpha = e^2/4\pi$. Note the factor $S = 3[(2/3)^2 - (1/3)^2] = 1$ involving the charges $2e/3$ and $-e/3$ of the up and down quarks circulating the loop, the multiplicity 3 due to color of $SU(3)$ Quantum Chromodynamics and a minus sign due to the isospin wave function of $\pi^0$.

The good agreement with with the experimental value of $(7.7 \pm 0.6)\ eV$, clearly demonstrated that anomalies are in fact essential for understanding how symmetries are manifested and broken in quantum field theories.

*Historical note*: Tracing the early history of quantum anomalies, the first paper we know of that clearly describe the axial anomaly in 2D is by Ken Johnson [47]. The first, and arguably most important, papers with direct application to the $\pi^0$ decay were the ones by Bell and Jackiw [48], and Adler [45]. It is interesting to mention that the very first derivation was made by Steinberger [49] long before the discovery of quarks and QCD. He calculated a triangle graph with the proton in the loop, and got the same result as from (4.15). The reason is that the anomaly does not depend on the mass of the fermion in the loop, and the factor is $S = 1$ in (4.16) for a single charge $e$ proton, just as for the sum of the colored quarks. If, however, quarks circulate in the loop, the needed factor of 3 is an experimental confirmation of QCD having three colors. Another early calculation was done by H. Fukuda and Y. Miyamoto, [50]. These old papers preceded the development of renormalization in quantum field theory and did not offer a proper understanding of the axial anomaly.

## 4.4 The three-dimensional infinite hotel

As promised, we will rederive the 4D axial anomaly using a spectral flow argument. The identity $F_{\mu\nu}\tilde{F}^{\mu\nu} = 4\vec{E}\cdot\vec{B}$ tells us to look at the response of fermions for parallel electric and magnetic fields. More precisely, by assuming a constant magnetic field, $\vec{B} = -B\hat{z}$ and then adiabatically turning on an electric field in the same direction, we will have a situation similar to that of the 1d spectral flow in Sect. 2.3. We need the spectrum for the Dirac particle in a constant magnetic field. To proceed, we recall that the massless Dirac field can be separated into chiral components, $\psi = (1+\gamma^5)\psi/2 + (1-\gamma^5)\psi/2 = \psi_L + \psi_R$, by using the Weyl representation for the (Minkowski space) gamma matrices,

$$\gamma^0 = \begin{pmatrix} 0 & -1 \\ -1 & 0 \end{pmatrix}, \qquad \gamma^i = \begin{pmatrix} 0 & \sigma^i \\ -\sigma^i & 0 \end{pmatrix}, \qquad \gamma^5 = \begin{pmatrix} 1 & 0 \\ 0 & -1 \end{pmatrix}. \tag{4.17}$$

Since we are interested in the spectrum, we use the matrices $\alpha^i = \gamma^0\gamma^i$, and write the Dirac Hamiltonian as, $H_{Dirac} = \vec{\alpha}\cdot(\vec{p}+\vec{A})$. Specializing to the case of a magnetic field in the $z$-direction, and doing some manipulations of the Pauli matrices as explained in App. C, we obtain the following expression for the square of the Hamiltonian,

$$H_{Dirac}^2 = \begin{pmatrix} p_z^2 + (\vec{q}+\vec{A})^2 - B\sigma^z & 0 \\ 0 & p_z^2 + (\vec{q}+\vec{A})^2 - B\sigma^z \end{pmatrix}, \tag{4.18}$$

where $\vec{q}$ is the momentum in the $xy$-plane and $\vec{A}$ the two-dimensional vector potential describing the magnetic field. Next note that $(\vec{q}+\vec{A})^2$ is the Hamiltonian for a 2d non-relativistic particle with mass $m = 1/2$ moving in a magnetic field that gives the Landau levels discussed in Sect. 3.1, with energies $\varepsilon_n = \omega_c(n+\frac{1}{2}) = (2n+1)B$. Thus the energy eigenvalues of (4.18) are,

$$E_{p_z,n,s} = \pm\sqrt{p_z^2 + (2n+1)B + sB}, \qquad n = 0, 1, \ldots, \quad s = \pm 1, \tag{4.19}$$

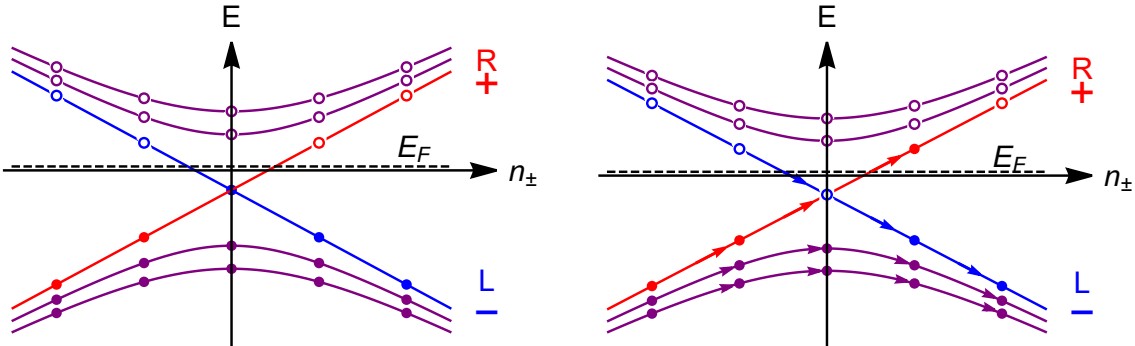

Figure 12: Left panel: Energy spectrum of the $3d$ infinite hotel with ground state filling. Right panel: Filling after insertion of one flux quantum with spectral flow shown by arrows.

where $s$ is the eigenvalue of $\vec{\sigma} \cdot \vec{B}/|\vec{B}| = s\sigma^z$. This spectrum is shown in Fig. 12. From (4.18) follows that the energy levels are doubly degenerate since $2n - 1 = 2(n-1) + 1$, with the exception of the $n = 0, s = -1$ case, corresponding to the lowest Landau level.

The spectral flow has significant effects only for the eigenvalues that cross the Fermi level $E_F$ under the adiabatic change of the potential; thus we only have to consider the linearly dispersing modes corresponding to $n = 0$ and $s = -1$, as illustrated in the right panel of Fig. 12. The mode with $E = p_z$ is right-moving, while the one at $E = -p_z$ is left-moving. For antiperiodic boundary conditions along the $z$-axis, the momentum is quantized as $p_z = (n_z - 1/2)/R$. So for these modes, the situation is identical to the one-dimensional spectral flow in Sect. 2.3, with charge density $\rho_0 = B/2\pi$ of the lowest Landau level. Thus, introducing the spatially constant potential $A_z(t)$, we get, in complete analogy with (2.17) and (2.25),

$$\dot{Q}_\pm = \mp R \mathcal{A}_{xy} \rho_0 \dot{A}_z(t) = \mp \frac{Vol}{2\pi} \rho_0 E^z \, , \tag{4.20}$$

where $\mathcal{A}_{xy}$ is the area in the $xy$-plane, and $Vol$ the volume. Since the system is homogeneous we can write the chiral currents (recalling that $B^z = -B$),

$$\partial_\mu J_\pm^\mu = \pm \frac{1}{4\pi^2} E^z B^z = \pm \frac{1}{4\pi^2} \vec{E} \cdot \vec{B} = \pm \frac{1}{16\pi^2} F\tilde{F} \, , \tag{4.21}$$

whose sum and differences reproduce the anomaly (4.11).

The spectral flow is moving an integer number of electrons in agreement with flux quantization. Let us verify that the normalizations are correct, the total charge displacement in the adiabatic process being, for each chirality,

$$\Delta Q_\pm = \int d^4x \, \partial_\mu J_\pm^\mu = \pm \frac{1}{32\pi^2} \int_{\mathcal{M}} d^4x \, \varepsilon^{\mu\nu\alpha\beta} F_{\mu\nu} F_{\alpha\beta} = n \in \mathbb{Z} \, . \tag{4.22}$$

This completes the infrared derivation of the anomaly for 4D fermions.

As in the two-dimensional case, the integrated anomaly in the r.h.s. is expressed by a topological invariant quantity, a gauge-invariant functional of the $A$ field taking integer values. This is called the second Chern class in the mathematical literature. While more technical aspects will be dealt with in Sect. 7, let us momentarily check its normalization. The integration is over a compact space-time manifold $\mathcal{M}$, the four-dimensional torus $S^1 \times S^1 \times S^1 \times S^1$, because we took periodic boundary conditions in space and in time, by identifying the points $t = -T$ and $t = T$, as explained in Sect. 2.4. The four-torus is actually the product of two two-torii,

such that the integration in (4.22) splits in two orthogonal fluxes, up to a factor of two for exchange symmetry. One then recovers the square of the monopole charge (first Chern class) (2.28), taking square integer values, as *e.g.* one. Therefore, the insertion of one flux quantum in each torus leads to the spectral flow of one electron per chirality, as required.

## 4.5 When are anomalies dangerous, and why?

Before moving on, we make some general comments about gauge theories. We already briefly discussed the physical meaning of vector and axial gauge fields, and mentioned that anomalies might make theories inconsistent. Here we shall expand on these points.

The notion of a local gauge symmetry is in fact misleading. A real symmetry operation, like a space rotation, takes you from one realization of a system to another, as *e.g.* when rotating a sphere. A gauge transformation does not change the state of the system but only changes the description. That there are different descriptions of the same state means that certain apparent degrees of freedom are redundant and thus not physical, such as the longitudinal and time-like components of photons. If one starts from a gauge theory and then adds terms that do not respect the local symmetry, such as $A_\mu A^\mu$ in quantum electrodynamics, one completely changes the theory. In this case the longitudinal photons become propagating, *i.e.* physical, and the ultraviolet behavior changes, which generically makes the theory non-renormalizable.

"Spontaneously broken" gauge symmetry is yet another concept prone to be misunderstood. Here the longitudinal gauge bosons indeed become physical by the Higgs mechanism, but no new degrees or freedom appear. Rather they are "repackaged", so that a degree of freedom in a scalar field appear in guise of a longitudinal photon.

Here it is important to understand that gauge theories can be realized in different phases. The electrodynamics in insulators is a gauge theory realized in the Coulomb phase, with propagating transverse photons. In a superconductor, electrodynamics is in a Higgs phase, meaning that there are no gapless photons at low energy.[25]

There is an important distinction between "dynamical" gauge fields and "external" gauge fields. We have discussed the chiral anomaly of $(1+1)$-dimensional fermions and found that it has interesting physical effects. The gauge fields were, by necessity, treated as classical background fields, because there are no photons in $(1+1)$ dimensions. In higher dimensions, on the contrary, we have the option to quantize the gauge fields or not. Let us discuss the consequences of having anomalies in the two cases.

Classical, or "external", gauge field do not describe physical degrees of freedom. Sometimes they are just auxiliary quantities that are used to calculate various responses. For example, the metric background can mimic the thermal response as well as the mechanical stress on materials. Background gauge fields are widely employed in condensed matter system. In these cases, anomalies cause no problems, but actually characterize the behavior of the system. Several examples of this have already been discussed. Needless to say, any realistic theory of nature must respect the experimentally verified conservation laws. For condensed matter physics this means that we must regularize in a way that leaves the electric current non-anomalous, even if we choose not to include photons in our theory.

Dynamical gauge fields in $d \geq 1$ have a kinetic term in the action and describe propagating degrees of freedom. The archetypical example is the electromagnetic field, but we also have gluon fields and fields related to the weak interaction. In condensed matter physics, it can also happen that "emergent" gauge fields describing low energy effects acquire dynamics. Anomalies in currents coupled to dynamical gauge fields destroy the gauge invariance, and, just as adding explicitly non-invariant terms to the action, drastically change the theory.

---

[25]In non-Abelian theories, and also in "compact" electrodynamics, there is also the possibility of a confining phase.

In the Standard Model of fundamental interactions, chiral anomalies of currents coupled to dynamical gauge fields do vanish. This cancellation occurs between the contributions of several particles with different (in general non-Abelian) charges. For an account of this see *e.g.* [1]. In more general theories, such as string theory, anomaly cancellations provide important consistency checks. In condensed matter systems, the only fundamental gauge symmetry is that of electromagnetism. There are however examples of "emergent gauge symmetries" which describe effective low energy degrees of freedom. In this case, anomalies have to be absent, or cancel, just as in the Standard Model.

In condensed matter systems there are relevant examples of theories with unavoidable chiral anomalies, such as a single right-moving fermion at the $(1+1)$ dimensional edge of the quantum Hall state, as discussed in Sect. 3.6. In this and other examples to be discussed in following chapters, charge conservation is preserved due to the presence of a compensating current flow from an attached bulk. This is the general mechanism that takes place at the boundary of various topological states of matter.

# 5   Anomalies from path integrals

## 5.1   The Schwinger model, and a path integral riddle

We shall start this chapter by presenting a paradox. Let us consider the Schwinger model, which is the one-dimensional Dirac theory in (2.1) coupled to dynamic electromagnetism,

$$\mathcal{L} = \bar{\psi}(i\slashed{\partial} + e\slashed{A})\psi - \frac{1}{4}F^2 , \tag{5.1}$$

where $F^2 = F_{\mu\nu}F^{\mu\nu} = 2E_x^2$. Without transverse dimensions, there are no photons, only electrons moving on a line interacting via a linear Coulomb potential (the solution of the one-dimensional Laplace equation). This means that any state with non-zero charge has an infinite energy, so that electrons cannot, even approximately, exist as widely separated particles. The analogy with quark confinement in 4D QCD is obvious, and the Schwinger model has been used to gain insights about this problem.

In 2D we can always decompose the vector potential as,

$$eA_\mu = \partial_\mu\lambda + \epsilon_{\mu\nu}\partial^\nu\xi . \tag{5.2}$$

The first term is just a gauge transformation while the second encodes the electric field, $E_x = -\partial^2\xi$.[26]

We pick the gauge $\partial_\mu A^\mu = 0$ by taking $\lambda = 0$ and perform the local axial transformation,

$$\psi \to e^{-i\xi(x)\gamma^3}\psi , \tag{5.3}$$

under which,

$$\bar{\psi}i\slashed{\partial}\psi \to \bar{\psi}i\slashed{\partial}\psi + (\partial_\mu\xi)\bar{\psi}\gamma^\mu\gamma^3\psi = \bar{\psi}i\slashed{\partial}\psi - \epsilon^{\mu\nu}(\partial_\nu\xi)\bar{\psi}\gamma_\mu\psi = \bar{\psi}i\slashed{\partial}\psi - \bar{\psi}e\slashed{A}\psi . \tag{5.4}$$

Substituting this in (5.1), it looks like the Coulomb force suddenly disappeared! Here you will presumably protest and say that this is a purely classical consideration and that one has to be much more careful in quantum field theory. The problem, however, apparently remains if one

---

[26]This is a special case of the Hodge decomposition which in general also includes gauge transformations that cannot be continuously deformed to the identity. These "large gauge transformations" are topological in nature, and in our case are precisely the ones discussed in Sect. 2.3 when the theory is defined on a circle.

uses the path-integral approach which tells us that all correlation functions can be obtained from the expression,

$$ Z[J, \eta, \bar{\eta}] = \mathcal{N} \int \mathcal{D}A\mathcal{D}\psi\mathcal{D}\bar{\psi}\, e^{-S_E + \bar{\eta}\psi + \bar{\psi}\eta + J^\mu A_\mu} \,. \tag{5.5} $$

Since this expression only involves classical variables, it seems that our problem remains.

The solution to this apparent paradox was given by Fujikawa [51] who showed that the measure $d\mu = \mathcal{D}\psi\mathcal{D}\bar{\psi}$ is not invariant under the chiral transformation(5.3), but picks up a dependence on the gauge field.

In the next section we shall derive this result, and its counterpart in 4D, and show how it is related to the anomaly derived in the previous section. But first we shall finish the story about the Schwinger model. To calculate how the finite axial transformation (5.3) affects the measure, we refer to the work [52]. Here we just give the result,

$$ d\mu \to e^{-\frac{e^2}{2\pi} \int d^2x\, A_\mu A^\mu} d\mu \,, \tag{5.6} $$

where we restored the charge $e^2$ which in 2D has the dimension of mass. Thus, picking the Lorenz gauge $\partial_\mu A^\mu = 0$, the action for the gauge field becomes,

$$ \mathcal{L}_{Sch} = -\frac{1}{2}A_\mu \left( \partial_\mu \partial^\mu - \frac{e^2}{\pi} \right) A^\mu \,, \tag{5.7} $$

which means that longitudinal photon, which was originally just a gauge freedom, is reborn as a free boson with the "Schwinger mass" $m^2 = \frac{e^2}{\pi}$. In the fermion theory, this boson is actually the particle-hole excitation discussed in the context of bosonization in Sect. 3.8.

Also, by adding a fermion mass term to (5.1) we get the "massive Schwinger model" which has very interesting topological features [53, 54].

## 5.2 Ward-Takahashi identities

Before moving to anomalies, we shall recall some general results about symmetries in quantum field theories. The symmetries manifest themselves in the so called Ward-Takahashi identities between Green's functions (Ward identities in short). The original one that you presumably learned about in QED, is a relation between the vertex function and the inverse propagator. Path integrals provide a simple and transparent way to derive these identities, and also show how they are modified by anomalies.

The starting point is the partition function (4.3), again with a background gauge field $A$ and sources for the fermions, keeping $A_5 = 0$,

$$ Z = e^{-\Gamma[A, \eta, \bar{\eta}]} = \mathcal{N} \int \mathcal{D}\bar{\psi}\mathcal{D}\psi\, e^{-S_E[A, \psi, \bar{\psi}] + \int dx\, (\bar{\eta}\psi + \bar{\psi}\eta)} \,. \tag{5.8} $$

The Euclidean action was given in (4.2). That a QFT has a symmetry means that $W$ is invariant under the corresponding symmetry transformations, *i.e.* $\delta W = 0$, that is implemented by a change of integration variables. Under an infinitesimal vector and axial transformation of the fermion fields, $\delta\psi = i(\lambda + \xi\gamma^{D+1})\psi$, $\delta\bar{\psi} = i\bar{\psi}(-\lambda + \xi\gamma^{D+1})$, we obtain the following expression inside the path integral,

$$ \lambda(x)\left[-\partial_\mu J^\mu + i(\bar{\eta}\psi - \bar{\psi}\eta)\right] + \xi(x)\left[-\partial_\mu J_5^\mu + 2m\bar{\psi}\gamma^{D+1}\psi + i(\bar{\eta}\gamma^{D+1}\psi + \bar{\psi}\gamma^{D+1}\eta)\right]. \tag{5.9} $$

Since $\lambda(x)$ and $\xi(x)$ are independent, the two corresponding local expressions in square brackets should vanish independently, as expectation values in the presence of the sources $\eta, \bar{\eta}$.

Taking functional derivatives with respect to $\eta(x)$ and $\bar{\eta}(x)$ of the path integral including (5.9), gives the Ward identities between multipoint correlators. The simplest ones, with no additional fermion fields, are obtained by setting the sources to zero, and read,

$$\partial_\mu J^\mu = 0 \qquad \text{and} \qquad \partial_\mu J_5^\mu = 2m\langle \bar{\psi}\gamma^{D+1}\psi\rangle . \tag{5.10}$$

Clearly there is something wrong with this argument since it appears that for $m = 0$ the axial current is conserved, and we know that this is not the case because of the anomaly. We now explain how this is resolved by carefully defining the integration measure.

## 5.3 Anomalies from the path integral measure

We learned in Sect. 5.1 that the path integral measure must be treated with care, and to be more precise, it must be properly defined. The problem is that the (continuum) path integral, even in a finite volume, is over an infinite number of degrees of freedom, so there are potential ultraviolet divergences. We shall closely follow the original treatment by Fujikawa [51].

Fujikawa's central insight was that in the above derivation of the Ward identities, we assumed that the functional integration measure was invariant under the symmetry transformations. To show that this is not the case, we note that in Euclidean space $i\slashed{D}$ is an Hermitian operator (see App. A), so the eigenvalues defined by,

$$i\slashed{D}\varphi_n(x) = \lambda_n \varphi_n(x) , \tag{5.11}$$

are all real. Now expand the spinors in terms of Euclidean 4D eigenfunctions, as follows,

$$\psi(x) = \sum_n \varphi_n(x)c_n , \qquad\qquad \bar{\psi}(x) = \sum_n \varphi_n^\dagger(x)\bar{c}_n , \tag{5.12}$$

obeying,

$$\int dx\, \varphi_n^\dagger(x)\varphi_m(x) = \delta_{mn} , \qquad\qquad \sum_n \varphi_n^\dagger(x)\varphi_m(y) = \langle x|y\rangle,$$

and define the fermionic path integral measure as,

$$d\mu = \mathcal{D}\bar{\psi}(x)\mathcal{D}\psi(x) = \prod_n d\bar{c}_n dc_n . \tag{5.13}$$

In any even spacetime dimension D, we can define the axial transformation,

$$\psi \rightarrow \psi' = e^{i\xi(x)\gamma^{D+1}}\psi = \sum_n c_n e^{i\xi(x)\gamma^{D+1}}\varphi_n(x) = \sum_n c_n' \,\varphi(x), \tag{5.14}$$

so we can solve for $c_n'$ by using the orthogonality of the $\varphi_n$,

$$c_m' = \sum_n \int dx\, \varphi_m^\dagger e^{i\xi(x)\gamma^{D+1}}\varphi_n c_n \equiv \sum_n T_{mn}c_n . \tag{5.15}$$

Recalling that the $c$ variables are Grassmann numbers, so that under a transformation $c \rightarrow ac$, the measure transforms as $dc \rightarrow a^{-1}dc$, we get,

$$\prod_n dc_n' = [\det T_{mn}]^{-1}\prod_n dc_n . \tag{5.16}$$

For infinitesimal $\xi$,

$$T_{mn} = \delta_{mn} + i\int dx\, \xi(x)\varphi_m^\dagger \gamma^{D+1}\varphi_n , \tag{5.17}$$

giving,

$$[\det T_{mn}]^{-1} = e^{-\text{Tr}\ln T_{mn}} = e^{-i\sum_n \int dx\, \xi(x)\varphi_n^\dagger \gamma^{D+1}\varphi_n}. \tag{5.18}$$

Combining this with an *equal* result from the antichiral part, we get,

$$d\mu \to d\mu\, e^{-2i\int dx\, \xi(x)\mathcal{A}(x)}, \tag{5.19}$$

where $\mathcal{A}$ is given by,

$$\mathcal{A}(x) = \sum_n \varphi_n^\dagger(x)\gamma^{D+1}\varphi_n(x). \tag{5.20}$$

This quantity involves an infinite sum and requires regularization. We can use the eigenvalues $\lambda_n$ of the Hermitian operator $i\slashed{D}$ to introduce a gauge invariant Gaussian cutoff as follows,[27]

$$
\begin{aligned}
\mathcal{A}(x) &= \lim_{M\to\infty}\sum_n \varphi_n^\dagger \gamma^{D+1} e^{-\left(\frac{\lambda_n}{M}\right)^2}\varphi_n = \lim_{M\to\infty}\sum_n \varphi_n^\dagger\gamma^{D+1} e^{-\left(\frac{i\slashed{D}}{M}\right)^2}\varphi_n \\
&= \lim_{M\to\infty}\langle x|tr\left[\gamma^{D+1} e^{\frac{\slashed{D}^2}{M^2}}\right]|x\rangle = \lim_{M\to\infty}\int \frac{d^D k}{(2\pi)^D} e^{ikx}\, tr\left[\gamma^{D+1} e^{\frac{\slashed{D}^2}{M^2}}\right]e^{-ikx},
\end{aligned}
\tag{5.21}
$$

where tr means summing over the spinor indices. The last identity follows by inserting a complete set of states, which we choose as the plane waves which are eigenfunctions of $i\slashed{\partial}$. To evaluate the integral, we rewrite $\slashed{D}^2$ as,

$$\slashed{D}^2 = \left[\frac{1}{2}\{\gamma^\mu,\gamma^\nu\} + \frac{1}{2}[\gamma^\mu,\gamma^\nu]\right]D_\mu D_\nu = D^2 + \frac{1}{2}\gamma^\mu\gamma^\nu[D_\mu,D_\nu] = D^2 - \frac{i}{2}\gamma^\mu\gamma^\nu F_{\mu\nu}, \tag{5.22}$$

and then expand the exponential and pick up the leading term in an expansion in $1/M^2$. In doing this we can also replace $D^2$ with $\partial^2$, which dominates the ultraviolet behavior. This is how far we can proceed in a general even space-time dimension, and we now specialize to D=2 and D=4.

For D=2, $\text{tr}[\gamma^3\gamma^\mu\gamma^\nu] = 2i\epsilon^{\mu\nu}$, and the momentum integral gives a factor $M^2/4\pi$. Thus, expanding the exponent to order $1/M^2$ and using (A.3), we get $\mathcal{A}(x)_{2D} = F(x)/2\pi$, with $F = \epsilon^{\mu\nu}\partial_\mu A_\nu$, and finally using (5.19),

$$d\mu \to d\mu \exp\left(-\frac{i}{\pi}\int dx\, \xi(x) F(x)\right), \qquad D = 2. \tag{5.23}$$

For D=4, $\text{tr}[\gamma^5\gamma^\mu\gamma^\nu\gamma^\sigma\gamma^\lambda] = -4\epsilon^{\mu\nu\sigma\lambda}$ so we must expand the exponent to second order in $F_{\mu\nu}$ for the trace not to vanish. In this case the momentum integral gives $M^4/16\pi^2$ and the anomaly,

$$\mathcal{A}(x)_{4D} = \frac{1}{16\pi^2}F(x)_{\mu\nu}\tilde{F}^{\mu\nu}(x), \tag{5.24}$$

and finally for the change in the measure,

$$d\mu \to d\mu \exp\left(-\frac{i}{8\pi^2}\int dx\, \xi(x) F(x)_{\mu\nu}\tilde{F}(x)^{\mu\nu}\right), \qquad D = 4. \tag{5.25}$$

(recalling that $\tilde{F}_{\mu\nu} = \varepsilon_{\mu\nu\alpha\beta}F^{\alpha\beta}/2$). A very important aspect of this result is that the anomalous contribution to the measure is a pure phase, both in Minkowski and Euclidean space.

---

[27]See App. E and Sect. 5 of [55] for details.

The change in the measure under the axial transformation will give an "anomalous" contribution to the WI (5.10). Considering only the axial transformation, it becomes,

$$\partial_\mu J_5^\mu = 2m\langle\bar\psi\gamma^{D+1}\psi\rangle - 2i\mathcal{A}, \tag{5.26}$$

giving, after putting $m = 0$ and going back to Minkowski space, the axial anomalies,

$$D = 2 \quad : \qquad \partial_\mu J_5^\mu = 2\mathcal{A}(x) = \frac{1}{2\pi}\varepsilon^{\mu\nu}F_{\mu\nu}\,,$$

$$D = 4 \quad : \qquad \partial_\mu J_5^\mu = 2\mathcal{A}(x) = \frac{1}{8\pi^2}F_{\mu\nu}\tilde{F}^{\mu\nu}\,. \tag{5.27}$$

These findings reproduce the earlier results (2.18) and (4.11).

We close this part with a comment on the nature of the anomaly relations. In the above derivation they appear as relations between vacuum expectation values of derivatives of currents, and external classical gauge field. However, from the Ward identity (5.10) we learn that we can trade an insertion of a divergence of the axial current for an insertion of the anomaly $\mathcal{A}$ also in a general amplitude. This means that the anomaly relations (5.27) *etc.,* also hold as operator equations. There is a more direct way to reach this conclusion by using point-splitting [2]. In this connection it should also be mentioned that there is also a purely Hamiltonian approach to anomalies [56].

# 6    4D anomalies at work – Dirac and Weyl materials

At least since the discovery of graphene, condensed matter physicists have been interested in finding systems in more than 1d where fermions have relativistic dispersion, which happens when the Fermi level is at a band crossing.[28] For a review of such "Dirac materials", see [58]. Graphene, for example, has a band structure with two such "Dirac points", where the dispersion relation takes the form of "Dirac cones". When the Fermi energy is tuned to these Dirac points, the density of states vanishes, and then the system is called a semi-metal. At low energies this is described by a Dirac equation, thus quasiparticles are gapless (or massless, if we use the high energy language), but by adding symmetry breaking terms, we can open up a gap. The gapless Dirac equation has modes of both chiralities, but there are also materials where the chiralities are separated so that a Dirac cone splits in two "Weyl cones", and these Weyl semi-metals, have many intriguing properties. The chiral and axial anomalies we discussed so far are important for understanding the physics of these systems, and in this section we shall discuss a couple of the many interesting phenomena, namely the chiral magnetic effect, and anomalous transport. But before doing so, we should get some understanding on the mixed anomalies obtained by including both nonvanishing vector $A$ and axial $A_5$ gauge backgrounds.

## 6.1    Mixed anomalies and consistent – covariant currents

In the absence of a mass term, the two chiralities are decoupled, and the path integral calculations of previous sections can be extended to give the anomalies in the presence of both $A$ and $A_5$ fields, or equivalently chiral and antichiral components $A_\pm = A \pm A_5$. One finds [59]:

$$D = 2 \quad : \qquad \partial_\mu J_\pm = \pm\frac{1}{4\pi}\varepsilon^{\mu\nu}F_{\mu\nu}\,,$$

$$D = 4 \quad : \qquad \partial_\mu J_\pm^\mu = \pm\frac{1}{16\pi^2}F_\pm\tilde{F}_\pm\,. \tag{6.1}$$

---

[28]The history is in fact longer and is briefly outlined in [57].

Rewritten in terms of vector and axial currents and field strengths, they read:

$$D = 2 \quad : \qquad \partial_\mu J^\mu = \frac{1}{2\pi} \varepsilon^{\mu\nu} F_{5\mu\nu}, \qquad\qquad \partial_\mu J_5^\mu = \frac{1}{2\pi} \varepsilon^{\mu\nu} F_{\mu\nu},$$

$$D = 4 \quad : \qquad \partial_\mu J^\mu = \frac{1}{4\pi^2} F_5 \tilde{F}, \qquad\qquad \partial_\mu J_5^\mu = \frac{1}{8\pi^2}(F\tilde{F} + F_5 \tilde{F}_5), \qquad (6.2)$$

(with *e.g.* $F\tilde{F} \equiv F_{\mu\nu}\tilde{F}^{\mu\nu}$). That the electric charge is not conserved when both backgrounds are present is troublesome, and we shall see how this issue can be resolved in various physical settings.

In the next section, we discuss the realization of the mixed anomalies in the quantum spin Hall effect, that can be thought of a time-reversal invariant generalization of the quantum Hall effect. In this case, the non-conservation of currents, both vector and axial, at the edge is compensated by the anomaly inflow from the bulk of the system, thus causing no problem at all. This model will give us some intuition about the physical meaning of $A_5$ backgrounds.

The Weyl semi-metals will then be analyzed in the following section. In these 3d systems, there is no compensating bulk in higher dimension and vector current conservation should be verified. In such a case, another kind of anomalous currents can be defined, that do not obey the inflow relation. They are called "consistent" because they can be obtained by the variation of the 3d effective action $\Gamma[A, A_5]$. The anomalies discussed so far in these lectures are instead "covariant", because they have the correct gauge transformations, (are invariant in the Abelian case), and in fact are related by inflow to bulk currents that are themselves covariant. This terminology is conventionally used in the literature, especially in discussing non-Abelian gauge theories [60].

The difference between covariant and consistent currents has both physical and geometrical meaning and it is unrelated to the freedom of adding local terms to the effective action. In some physical settings, like the spectral flow and index theorem, the covariant anomalies should be used, while in other situations, such as for the response of isolated chiral systems, the consistent ones are appropriate. The discussion of properties of the two kinds of currents is rather technical and is presented in App. D. In the following we simply list the consistent anomalies and compare them to the covariant ones already given in (6.2).

Consistent currents are denoted by $j_\mu$ and $j_{5\mu}$ and read,

$$D = 2 \quad : \qquad j^\mu = J^\mu + \Delta J^\mu, \qquad\qquad \Delta J^\mu = -\frac{1}{\pi}\varepsilon^{\mu\nu} A_{5\nu},$$

$$j_5^\mu = J_5^\mu, \qquad\qquad\qquad\qquad\qquad\qquad (6.3)$$

$$D = 4 \quad : \qquad j^\mu = J^\mu + \Delta J^\mu, \qquad\qquad \Delta J^\mu = -\frac{1}{4\pi^2}\varepsilon^{\mu\nu\alpha\beta} A_{5\nu} F_{\alpha\beta},$$

$$j_5^\mu = J_5^\mu + \Delta J_5^\mu, \qquad\qquad \Delta J_5^\mu = -\frac{1}{12\pi^2}\varepsilon^{\mu\nu\alpha\beta} A_{5\nu} F_{5\alpha\beta}, \qquad (6.4)$$

where the pieces $\Delta J_\mu, \Delta J_{5\mu}$ being added to the covariant currents $J_\mu, J_{5\mu}$ are called Bardeen-Zumino terms [60]. It is apparent that these modifications cannot be realized by adding local terms to $\Gamma[A, A_5]$, as anticipated.

The corresponding consistent anomalies read, using (6.2), (6.3) and (6.4):

$$D = 2 \quad : \qquad \partial_\mu j^\mu = 0, \qquad\qquad \partial_\mu j_5^\mu = \frac{1}{2\pi}\varepsilon^{\mu\nu} F_{\mu\nu}, \qquad (6.5)$$

$$D = 4 \quad : \qquad \partial_\mu j^\mu = 0, \qquad\qquad \partial_\mu j_5^\mu = \frac{1}{8\pi^2}\left(F\tilde{F} + \frac{1}{3}F_5\tilde{F}_5\right). \qquad (6.6)$$

Note the relevant feature that the consistent vector current is conserved in the mixed case too. The comparison with (6.2) shows that the two kinds of anomalies coincide for vanishing $A_5$, thus explaining why the consistent currents were not introduced in the earlier analyses.

## 6.2  Spectral flow in the quantum spin Hall effect

Topological insulators in $(2+1)$ dimensions are topological phases of matter sharing properties with the quantum Hall effect, in that they have a gapped bulk and gapless edge excitations. However, they respect time reversal (TR) symmetry. Later we shall discuss them in more detail, but in this section we shall consider a special case which is easy to analyze given what we already know. This is the quantized spin Hall effect which can be modeled by two independent quantum Hall liquids with opposite chirality and also opposite spin polarizations. TR invariance is achieved since the two component are mapped into each other by time-reversal, which flips both chirality and spin.

On the boundary there are two massless Weyl fermions, a chiral one with spin-up and an antichiral with spin-down along some direction, *e.g.* orthogonal to the plane. As a result there are no charge currents, but the two edges carry net spin currents with opposite chirality, as illustrated in Fig. 13. Together the chiral modes form a $(1+1)$-dimensional Dirac fermion on each edge, and the system possesses a $U(1)_Q \times U(1)_S$ symmetry, where the second, axial, part corresponds to the conserved spin current, and is responsible for the quantized spin Hall effect.

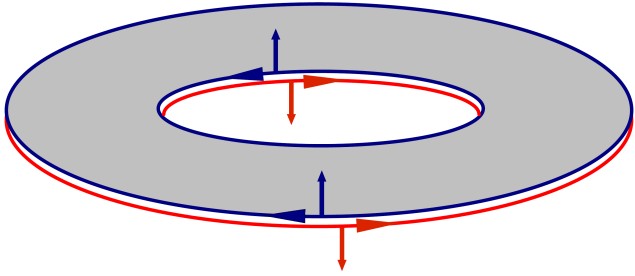

Figure 13: Edge excitations in the quantum spin Hall effect. On the outer edge, right-moving spin-down modes are drawn in red, left-moving spin-up ones in blue. Velocities are reversed in the inner edge.

Before studying the spectral flow in this system, let us recall that of the quantum Hall effect in Sect. 3.4. There we discussed the Laughlin flux argument in the geometry of an annulus with two edges and a radial Hall current, exemplifying the anomaly inflow mechanism. Upon adiabatically adding a flux quantum, an electron is pulled out of the Fermi sea, say on the inner edge, and another is pushed down into the sea on the outer edge, altogether corresponding to the displacement of one charge by the bulk $\nu = 1$ Hall current (see (3.36)).

On each edge, the anomalous charge non-conservation is compensated by the flux of the bulk current $J_r$ obtained from the Chern-Simons action $S_{CS}$ (3.42). In formulas,

$$\text{Anomaly}: \quad \dot{Q}_L \;=\; \frac{1}{2\pi} \oint \varepsilon^{\mu\nu} \partial_\mu A_\nu, \qquad \mu, \nu = 1, 2 \,,$$

$$\text{Inflow}: \quad \dot{Q}_L \;=\; \oint J_r = \oint \frac{\delta S_{CS}}{\delta A_r} = \frac{1}{2\pi} \oint \varepsilon^{r\mu\nu} \partial_\mu A_\nu \,, \qquad (6.7)$$

where the integrals are over the spatial boundary. The first equation follows from the $(1+1)$-dimensional theory, the second expresses the continuity of the bulk current as it reaches the boundary.

Referring back to our study of the fermion spectrum in Sect. 2.1, we recall that the induced boundary electric field $E_x$ is due to the constant potential $A_x(t)$, which acts as a chemical potential and shifts the energy levels in opposite ways on the two edges/chiralities; $\mu_L = A_x$,

$\mu_R = -A_x$. It follows that the spectral flow can also be seen as due to an imbalance of the Fermi level across the system.

The vector and axial backgrounds, $A = (A_L + A_R)/2$ and $A_5 = (A_L - A_R)/2$, act on the two chiralities/edges with different signs. For example, inserting a $A_5$ flux in the quantum Hall effect would lead to the same charge variation on both edges, leading to no net Hall current. Equivalently, one can see this as the simultaneous lowering/raising of the chemical potential on both edges $\mu_L = \mu_R = A_{5x}$.

We now turn to the quantum spin Hall system, described as a pair of Hall states (see Fig. 13). One copy, say that of spin-up electrons (in blue), is subject to the usual spectral flow when its background field $A_+$ is adiabatically switched on. For the other copy, with spin-down electrons (in red), the corresponding flow is associated to $A_-$. Turning to the bulk, the Chern-Simons theory is now,[29]

$$S_{TI} = \frac{1}{4\pi} \int A_+ dA_+ - A_- dA_- = \frac{1}{\pi} \int A dA_5 , \tag{6.8}$$

where we recall that $A_\pm = A \pm A_5$. This action is time-reversal invariant by mapping $A_+ \leftrightarrow A_-$, or equivalently by the corresponding transformations of vector and axial fields. The anomaly inflows are given by the radial currents,

$$J_r = \frac{1}{\pi} \varepsilon^{r\mu\nu} \partial_\mu A_{5\nu}, \qquad\qquad J_{5r} = \frac{1}{\pi} \varepsilon^{r\mu\nu} \partial_\mu A_\nu . \tag{6.9}$$

These currents cancel the mixed anomalies of the edge fermions given by the (6.2), using the inflow relations (6.7). Specifically (see Fig. 14):

- The $A$ field creates a chiral anomaly in the Dirac theory at each edge, opposite on the two edges, corresponding to non-conservation of axial charge, $\Delta Q_{5L} = -\Delta Q_{5R}$ (*i.e.* a relative imbalance of chemical potentials). The bulk axial (or spin) current $J_{5r}$ compensates for this effect between the two edges.

- The $A_5$ field creates a vector Hall current $J_r$ that balances the charge non-conservation since $\Delta Q_L = -\Delta Q_R$. Actually, electrons of both chiralities are pushed down the Fermi sea at one edge and pulled up at the other one.

- The value $\nu = 2$ of the conductance is correct since these adiabatic processes always transport pairs of electrons.

In conclusion, the spectral flow in the quantized spin Hall system exemplifies the action of mixed anomalies, in the presence of both vector and axial backgrounds.

There is a close similarity between the quantized spin Hall system and the 2$d$ topological insulator. In fact, often the distinction is not made, since in real materials, the spin is not a good quantum number, and the $U(1)_Q \times U(1)_S$ symmetry is explicitly broken to $U(1)_Q \times \mathbb{Z}_2$ by time-reversal invariant interactions that do not conserve spin $S_z$, such as the Rashba term, and disorder [61]. So rather than having a $\mathbb{Z}$ valued topological number, which you can think of as number of filled Landau levels, or layers, there is only a $\pm 1$ number related to $\mathbb{Z}_2$, which corresponds to the spin parity $(-1)^{2S}$. That this symmetry is present even when no component of the spin is conserved is highly non-trivial and at the heart of the discovery of topological insulators. We shall come back to this in later sections.

---

[29]The action is expressed in terms of differential forms, see App. E.

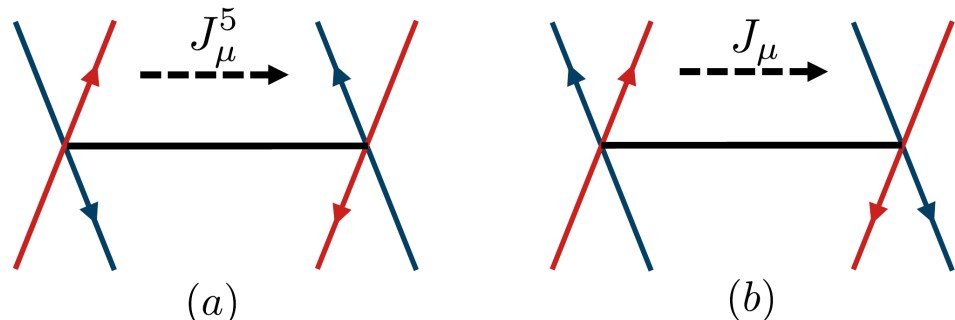

Figure 14: Radial view of the annulus geometry (black line) with energy spectra of right-moving (red) and left-moving (blue) edge modes in the quantum spin Hall effect. The arrows show the spectral flow under adiabatic changes of: $(a)$ the $A$ field; $(b)$ the $A_5$ field. The compensating bulk currents are indicated.

## 6.3   Mixed anomalies and Weyl semi-metals

In this section we shall learn how chiral anomalies are crucial for understanding the electromagnetic response of Weyl semi-metals (WSM). This is not the place for any deep explanation of the physics and material science of WSMs, but we still have to set the stage by providing some basics of these very interesting systems.

### 6.3.1   Weyl semi-metals – a primer

The WSMs are 3d cousins of graphene, but with distinct and interesting new properties. The first prediction of a WSM was in certain iridate materials with rather complicated lattice structure, but it was subsequently found experimentally in clean crystals of TaAs [57], which is a simpler compound. Still the band structure is involved, and several steps are needed to extract an effective low energy model simple enough to be described here. Details about how that is done can be found *e.g.* in [62], where you can learn under which conditions a Dirac cone can split into Weyl cones, each having only one chirality. An important point is that you cannot have both TR and inversion symmetry, but if you keep one of these symmetries the Weyl points come in pairs related by symmetry.

A simple and instructive model, that will suffice to illustrate the importance of anomalies, was given in [63]. They started from a model on a cubic lattice for a 3d topological insulator and added symmetry breaking terms to get non-degenerate Weyl points, that are shifted both in energy and momentum. Assuming isotropy, the effective theory for the modes close to Weyl points located at $\vec{p} = \pm\vec{b}$, and shifted in energy by $\pm\rho\, b_0$, is,

$$H_W = \pm\rho\, b_0 \pm \mathrm{v}\vec{\sigma}\cdot(\vec{p}\mp\vec{b}) \,, \tag{6.10}$$

where $\rho$ is the density, v a characteristic velocity, and the sign denotes the helicity of the Weyl point.[30] The Dirac cones in these Weyl points are depicted in Fig. 15.

WSMs, together with the quantum Hall and spin Hall effects discussed earlier, are examples of topological phases of matter, to be discussed in more detail in Sect. 9. A very useful concept to understand these phases is that of Berry phase which we already alluded to in Sect. 3.4. In general, during a slow, adiabatic variation, an eigenstate of a quantum system acquires a

---

[30]Since in real materials, there are typically several Weyl points, we have a bona fide semi metal only when $b_0 + \mu = 0$ for all Weyl points where $\mu$ is the relevant chemical potential.

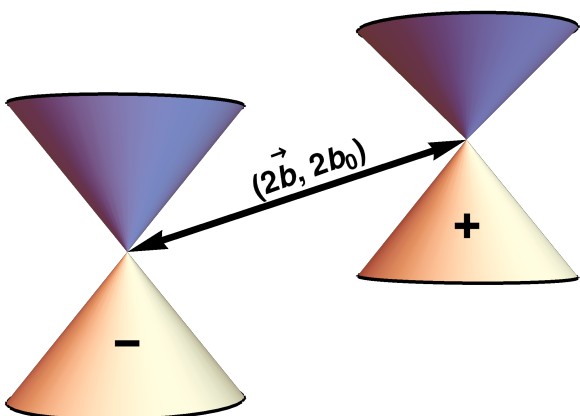

Figure 15: Dirac cones around Weyl points in a Weyl semi-metal.

phase that consists of two components [64]. One is a dynamical contribution, that depends on the energy spectrum, the other just depends on the trajectory defined by the Hamiltonian in the parameter space. Because the latter does not depend on the details of the evolution, it is a geometric or even topological quantity. This phase is expressed in terms of the integral of a gauge connection, the Berry connection $\mathcal{A}$, that is originating from the parameter variation of one-particle states during evolution. It is like a particle moving in an external electromagnetic field, but this is generated inside the quantum problem. As we will see, non-vanishing Berry phase and Berry curvature often characterizes the presence of a topological phases of matter.

We now explain why the WSM is a topological state of matter, by analogy with the more familiar case of the integer QHE. In that case, the topology is coded in the map from the Brillouin zone to the space of one particle states, $\{\vec{k} \in BZ\} \rightarrow \{\psi_{\vec{k}} \in U\}$, where $U$ is one patch of the Hilbert space. More precisely, one first defines the Berry potential by,

$$\mathcal{A}_{k_i}(\vec{k}) = -i\langle \psi_{\vec{k}} | \partial_{k_i} | \psi_{\vec{k}} \rangle , \tag{6.11}$$

and then the Berry field strength by, $\mathcal{B} = \epsilon^{ij} \partial_{k_i} \mathcal{A}_{k_j}$. The first Chern number of $\mathcal{B}$ is given by (cf. (2.28)),

$$I = \frac{1}{2\pi} \int_{BZ} d^2k \, \mathcal{B} , \tag{6.12}$$

and since the Brillouin zone is a closed surface (a torus) we can, in analogy with an ordinary magnetic field $B$ in the presence of monopoles, conclude that the integral must be an integer number. As originally shown in [65], the Hall conductance is given by $\sigma_H = I \, e^2/h$. This way of defining topology is based on having a gapped system, and cannot be directly applied to a WSM. What we can do, however, is to consider an integral not over the full zone, but over a 2d surface enclosing a Weyl point. Since the energy is finite everywhere except at the Weyl points this is a well defined object, which is actually easy to calculate by the following analogy: The Hamiltonian $H_W$ you can think of as a momentum space version of the Hamiltonian $H_B = -\mu \vec{S} \cdot \vec{B}$ for a particle with magnetic moment $\mu$ and spin $\vec{S}$ in a magnetic field. Changing the direction of $\vec{B}$ adiabatically provides the archetypical example of a geometrical phase, and Berry showed that this phase only depends on the value $m$ of the spin projection on $\vec{B}$ and equals $m\Omega$, where $\Omega$ is the solid angle subtended by the loop traced by $\vec{B}$ [64]. For a full sphere, $\Omega = 4\pi$ so for a spin half particle corresponding to the two level Hamiltonian $H_W$, the integrated Berry flux is $\pm 2\pi$, where the sign is given by the helicity. Thus the total flux of a pair of Weyl points with different helicity vanish, meaning that they can annihilate into a trivial state if they are made to collide.

### 6.3.2 Electromagnetic response

We are now prepared to study the response of a WSM to electromagnetic fields, and let us consider the simplest case with only two Weyl points. Using the Weyl representation (4.17) for the Dirac matrices, the two cones can be described by the Dirac action,

$$S = \int dx \, \bar{\psi}(i\slashed{\partial} + \slashed{A} + \slashed{A}_5 \gamma^5)\psi \, , \tag{6.13}$$

where $A_{5\mu} = (\rho b_0, \vec{b})$, and we take $v = 1$ and $e = 1$, as usual. The shift in energy and momentum between the Weyl points enters the Dirac theory as the axial gauge potential $A_5$. More importantly, by applying magnetic fields and/or strain it is possible to have a space dependent axial potential, and thus non zero axial field strengths. We shall not dwell here on how this can come about, but details and more references can be found *e.g.* in [66].

Now consider a situation where $A_5 = 0$, but with non-vanishing and parallel electric and magnetic fields, so $\vec{E} \cdot \vec{B} \neq 0$. Because of the chiral anomaly (6.1) electric charge will not be conserved in the separate chiral channels, but taken together, the electric charge is conserved. A more interesting situation occurs when there is a non vanishing axial magnetic field, $\vec{B}_5$ and a parallel electric field $\vec{E}$. In this case, the result (6.2) and a corresponding variant of the "infinite hotel" calculation in Sect. 4.4, seem to show that the electric charge is not conserved,

$$\partial_\mu J^\mu = \frac{1}{4\pi^2} \vec{E} \cdot \vec{B}_5 \, , \tag{6.14}$$

which is troublesome.

Let us see how the issue is resolved in this context. An axial gauge potential $A_5$ is certainly present since without it there are no Weyl cones with non degenerate spectra. But to get an axial magnetic field, $\vec{B}_5$ the potential must be space dependent. This can happen if the stress in the material is inhomogeneous, but we shall look at a simpler realization which is a boundary between a Weyl metal and a Dirac metal where two Weyl cones have merged [66]. Consider a slab of a WSM with sharp boundaries, $x \in (-a, a)$. Further assume that $\vec{A}_5 = b[\Theta(x+a) - \Theta(x-a)]\hat{y}$, where $\Theta$ is the Heaviside step function, and a constant electric field $\vec{E} = E_z \hat{z}$. The axial magnetic field becomes,

$$B_5 = b \left( \delta(x+a) - \delta(x-a) \right) \hat{z} \, . \tag{6.15}$$

Using this in (6.14) we get the explicit expression,

$$\partial_\mu J^\mu = \frac{E_z b}{4\pi^2} \left( \delta(x-a) - \delta(x+a) \right), \tag{6.16}$$

which shows that the electric charge is not conserved at the boundaries. The total electric charge is conserved, but that is not good enough, since electric charge must be conserved locally.

The situation is somehow similar to that of the spin Hall effect in Fig. 14 (b) (see Sect. 6.2), with a charge unbalance between the two cones. Actually, there is a current flowing within the system itself, not through an external bulk, because deep inside the Fermi sea the bands of the two cones reconnect, similar to the case of the Luttinger model in Fig. 2. How this ultra-violet feature can be modeled in our effective theory which is valid near the Fermi surface? The answer is found by using the consistent currents (6.4) introduced in Sect. 6.1, instead of covariant ones in (6.14). Here are their important features:

- They describe isolated systems without inflow from outside, as described in App. D.

- The vector current is conserved also in the case of mixed anomalies, see (6.6).

- The Bardeen-Zumino term modifying the vector current in (6.4) re-establishes local charge conservation.

Let us evaluate this Bardeen-Zumino term $\Delta J^\mu$ in our geometry. Its value is non vanishing inside the slab,

$$\Delta J^x = -\frac{1}{4\pi^2} A_{5y} E_z, \qquad x \in (-a, a), \qquad (6.17)$$

and zero outside, and the resulting divergence $\partial_x \Delta J^x$ indeed cancels the anomaly (6.16). Therefore, this current correctly describes the charge compensation between the two Weyl cones. It is an interesting possibility that the difference between covariant and consistent currents (6.17) might originate by a shift of normal-ordering conditions for the ground state in the region $x \in (-a, a)$.

There is lot more to be said about the role of the mixed anomalies in the study of transport properties of WSMs. An important issue is to what extent the simple effective models just described reflect the physics of real materials associated to concrete lattice models. An example of this are the recent studies [67–69], which also include an $E_5$ field and thus can probe the full expression of the anomaly (6.6). Another example is [70], which presents a model calculation where torsion gives rise to a uniform axial magnetic field in a WSM wire, and goes on to studying detail how the effect of the bulk anomaly in the presence of an electric field is canceled by modes concentrated on the surface. Refs. [71, 72] contains later work on the effects of torsion.

## 6.4 The chiral magnetic effect

The chiral magnetic effect was originally proposed in the context of ultra-relativistic heavy ion collisions [73], but later also generalized to Dirac materials and then experimentally discovered [74]. It predicts the appearance of an electric current when there is an imbalance between the chemical potential of the right and left moving modes. Such an imbalance can be modeled by a non-zero axial potential $A_{50} = (\mu_R - \mu_L)/2 \equiv \mu_5$, as discussed in Sect. 6.2.

Assume that we subject a material where $A_{50}$ can be finite to a constant magnetic field $\vec{B}$, and also to a constant axial electric field $\vec{E}_5$. The mixed anomaly (6.2) is again,

$$\dot{Q} = \frac{\text{Vol}}{4\pi^2} \vec{B} \cdot \vec{E}_5, \qquad (6.18)$$

and this expression holds in the frame defined by the magnetic field where there is no current flowing. We learned in Sect. 6.1 that this unacceptable anomaly can be eliminated by the Bardeen-Zumino term $\Delta J^\mu$ in (6.4), that provides the full current, in this case:

$$j^i \equiv \Delta J^i = \frac{1}{2\pi^2} \epsilon^{0ijk} A_{50} \partial_j A_k = \frac{\mu_5}{2\pi^2} B^i. \qquad (6.19)$$

For a detailed explanation for why a non-zero $\mu_5$ can occur in a quark gluon plasma, or in a Dirac material, we refer to the original papers, and here we shall just outline the argument given in *e.g.* [74]. Assuming that we apply a weak electric field, $\vec{E}$, again in the same reference frame as before, we get the axial anomaly,

$$\dot{\rho}_5 = \frac{1}{4\pi^2} \vec{E} \cdot \vec{B} - \frac{\rho_5}{\tau}, \qquad (6.20)$$

where we also added a Drude-like term that describe relaxation to the equilibrium with a time constant $\tau$. Solving (6.20) gives the steady state value $\rho_5 = \frac{1}{4\pi^2} \vec{E} \cdot \vec{B} \tau$, and combining this with the thermodynamic relation between $\rho_5$ and $\mu_5$, gives the chiral magnetic linear response,

$$j^i_{CME} \sim \tau B^i \vec{B} \cdot \vec{E}. \qquad (6.21)$$

Note that this expression does not depend on $\vec{E}_5$, which should be thought of as a probe field that is taken to zero at the end of the calculation.[31] For $\vec{E} \parallel \vec{B}$ we get an enhanced longitudinal conductivity $\sim B^2$. This contribution is in addition to the usual Ohmic term, and the characteristic quadratic dependence on the magnetic field was experimental observed in [74].

# 7  Anomalies and index theorems

In the perturbative derivation of the anomalies, we glossed over an important question: Are there corrections due to higher order diagrams? From the path integral derivation one would be tempted to say no, since adding interactions will not change the fermionic measure. The infinite hotel method, using spectral flow and point-splitting did involve the Hamiltonian, so there it is not clear how anomalies could be changed by interactions. Not long after the discovery of the axial anomaly, it was proven that the only effect of interactions is to replace bare charges and masses with renormalized ones, a result know as the Adler-Bardeen theorem [75] .

There is, however, a much more general and elegant way to show that the anomalies we have discussed, as well as their generalization to non-Abelian gauge groups, and curved geometries, are not changed by interactions. It follows from deep results in algebraic topology, but unfortunately the original literature is not very accessible to non mathematicians. The important results for us are the index theorems of Atiyah and Singer and Atiyah, Patodi and Singer. These theorems link properties of differential operators to topological properties of gauge fields, a relation anticipated in Sect. 2.4. As we now show, we have already indirectly derived a special case of one of these theorems, and after explaining this point we shall just state some general results.

## 7.1  Definition of the index of the Dirac operator

First we recall the definition of chiral projectors and the Dirac operator in even dimensional spacetime,

$$P_{\pm} = \frac{1}{2}(1 \pm \gamma^{D+1}) ; \qquad \slashed{D} = \gamma^{\mu}(\partial_{\mu} - iA_{\mu}), \qquad (7.1)$$

where we use Euclidean signature, and the same gamma matrices $\gamma_E^{\mu}$ as in Sect. 4.1, but will not write the subscript $E$ in what follows. Next define,

$$D = \slashed{D}P_{+} , \qquad D^{\dagger} = -P_{+}\slashed{D} = -\slashed{D}P_{-} , \qquad (7.2)$$

where we recalled that $\slashed{D}$ is anti-Hermitian. Since $P_{\pm}$ are projection operators they define two subspaces, $\mathcal{H}_{\pm}$ so that for D=4,

$$\psi_{\pm} \in \mathcal{H}_{\pm}, \qquad \text{obeying} \qquad P_{\pm}\psi_{\pm} = \psi_{\pm}, \qquad \gamma^5\psi_{\pm} = \pm\psi_{\pm}. \qquad (7.3)$$

With these definitions we get,

$$\gamma^5 D^{\dagger}\psi_{-} = -\gamma^5\slashed{D}P_{-}\psi_{-} = \slashed{D}\gamma^5 P_{-}\psi_{-} = -\slashed{D}P_{-}\psi_{-} = D^{\dagger}\psi_{-} , \qquad (7.4)$$

and similarly $\gamma^5 D\psi_{+} = -D\psi_{+}$, so the operators $D$ and $D^{\dagger}$ gives a one-to-one mapping between the states in two subspaces, $\mathcal{H}_{\pm}$, except for the zero modes of any of these operators. The index

---

[31]An alternative derivation of this relation is given by [73].

of the Dirac operator is defined [1] as,[32]

$$\text{ind}(\slashed{D}) = \# \text{ zero modes of } D - \# \text{ zero modes of } D^\dagger . \tag{7.5}$$

In order to calculate the index, it will be useful to work with the Hermitian operators $D^\dagger D$ and $DD^\dagger$. Obviously any zero mode of $D$ is also a zero mode of $D^\dagger D$ and the same for the pair $D^\dagger$ and $DD^\dagger$. It is also not too hard to prove that the index can also be re-expressed in terms of the quadratic operators so we have the alternative definition,

$$\text{ind}(\slashed{D}) = \# \text{ zero modes of } D^\dagger D - \# \text{ zero modes of } DD^\dagger . \tag{7.6}$$

## 7.2 Calculation of the index

Since $D^\dagger D$ is a Hermitian operator on $\mathcal{H}_+$ it can be diagonalized with real eigenvalues, $\lambda_n^+$, and similarly for $DD^\dagger$ on $\mathcal{H}_-$. Since the non-zero modes $\lambda^\pm \neq 0$ are matched in pair, we get the following formal expression for the index,

$$\text{ind}(\slashed{D}) = \sum_{n \in \mathcal{H}_+} 1 - \sum_{n \in \mathcal{H}_-} 1 = \int dx \left( \sum_{n \in \mathcal{H}_+} \psi_n^{+\star}(x)\psi_n^+(x) - \sum_{n \in \mathcal{H}_-} \psi_n^{-\star}(x)\psi_n^-(x) \right), \tag{7.7}$$

whose proof is left as an exercise. In this expression, $\psi_n^\pm$ are the eigenfunctions associated to $\lambda_n^\pm$, similarly to those in Sect. 5.3. Now we regularize the above expressions as follows,

$$\begin{aligned}
\text{ind}(\slashed{D}) &= \int dx \left( \sum_{n \in \mathcal{H}_+} \psi_n^{+\dagger}(x)\gamma^5 e^{-\frac{D^\dagger D}{M^2}} \psi_n^+(x) + \sum_{n \in \mathcal{H}_-} \psi_n^{-\dagger}(x)\gamma^5 e^{-\frac{DD^\dagger}{M^2}} \psi_n^-(x) \right) \\
&= \text{Tr}\left[ \gamma^5 e^{-\frac{(i\slashed{D})^2}{M^2}} \right]_{\mathcal{H}},
\end{aligned} \tag{7.8}$$

where we used $D^\dagger D \psi_n^+ = -\slashed{D}\slashed{D}\psi_n^+$ and $DD^\dagger \psi_n^- = -\slashed{D}\slashed{D}\psi_n^-$ to get the last equality. The trace is both over the gamma matrices, and the full Hilbert space $\mathcal{H} = \mathcal{H}_+ \oplus \mathcal{H}_-$. As in Sect. 5.3 we used a gauge invariant cutoff, and note that since the nonzero eigenvalues of $(i\slashed{D})^2$ comes in pairs, the expression is independent of $M^2$. This means that we can evaluate the regularized sum for any $M^2$, and in particular for the limit $M^2 \to \infty$. But this is precisely the calculation we already did in Sect. 5.3, the trace in (7.8) is given by the integrated anomaly (5.21), so we immediately get the result,

$$\text{ind}(\slashed{D})_{D=4} = \frac{1}{16\pi^2} \int dx \, F^{\mu\nu}(x)\tilde{F}_{\mu\nu}(x). \tag{7.9}$$

This is a special case of the Atiyah-Singer theorem. There are in fact a number of index theorems, but this is the only one that will concern us here. Similarly we can extract the index in D=2 from (5.23) ,

$$\text{ind}(\slashed{D})_{D=2} = \frac{1}{4\pi} \int dx \, \epsilon^{\mu\nu} F_{\mu\nu}. \tag{7.10}$$

Now comes an important point. The index is by definition an integer, but it is not obvious that this is the case for the right-hand sides of (7.9) and (7.10). We have however not been precise enough in defining the eigenvalue problem for the operators. To do so, we assume that the field strength vanishes on a circle at infinity, which, after identifying this circle as a

---

[32]In mathematics the number of zero modes of an operator defines the kernel of that operator, so in this language the definition of the index is: $\text{ind}(i\slashed{D}) = \text{dimension of kernel of } D - \text{dimenson of kernel of } D^\dagger$.

point, is the same as defining the theory on a sphere. Thus, the index is given by the flux of the magnetic field through a closed surface, which is a topological quantity, as discussed in Sect. 2.4.

Another way to understand (7.10) is to recall that a non-zero magnetic flux through a closed surface $\Sigma$ is obtained by enclosing a number of Dirac monopoles, each attached to a flux tube of strength $2\pi$, since it should be invisible for a particle of unit charge [1]. Assume that there is only one monopole, and integrate over an infinitesimal circle $S_\epsilon$ around the Dirac string, such that $\Sigma$ covers the whole sphere in the limit when the radius of the $S_\epsilon$ goes to zero. Note that this does not depend on the shape of the surface, which proves that the index (7.10) is a topological quantity.

## 7.3   The Atiyah-Singer theorem for the Dirac operator in general dimensions

We will now state the general result for the index of the Dirac operator defined on an even dimensional closed (and orientable) curved manifold, and for this we need some notation. In case you are not familiar with field theory on curved backgrounds and some basics notions of differential geometry, you can skip this section and just note the final result, (7.15) for the 4D Dirac index. App. E may nonetheless help you with some basic notions and formulas.

The Dirac operator on a closed curved manifold coupled to a non-Abelian vector field $A_\mu$, is defined by,

$$\slashed{D} = e_a^\mu \gamma^a (\partial_\mu + i\omega_\mu^{ab}\sigma_{ab} - iA_\mu) \,, \tag{7.11}$$

where $e_a^\mu$ is the vierbein field $\omega_\mu^{ab}$ the spin connection, $\sigma_{ab} = i[\gamma_a, \gamma_b]/4$ and $A_\mu \equiv A_\mu^\alpha T^\alpha$ with $T^\alpha$ generators of the appropriate representation of the gauge group.[33]

The Chern characters, $Ch_n(F)$ are defined by the generating function [55]:

$$Ch(F) = \mathrm{Tr}\, e^{F/2\pi} = \sum_{n=0} \frac{1}{n!} \mathrm{Tr}\left(\frac{F}{2\pi}\right)^n \,, \tag{7.12}$$

where $F$ is the field strength two form $F = \frac{1}{2}F_{\mu\nu}dx^\mu \wedge dx^\nu$ and the trace is on the gauge indices.

For a curved space the $\hat{A}$-genus is defined by,

$$\hat{A} = 1 + \frac{1}{48\pi^2}\mathrm{Tr}\,R \wedge R + \dots \,, \tag{7.13}$$

where $R$ is the Riemann curvature two form $R_b^a = \frac{1}{2}R_{\mu\nu,\ b}^{\ \ \ a}dx^\mu \wedge dx^\nu$, that is obtained from the spin connection $\omega_{\ b}^a$ and is also a matrix valued two form. In this case the trace acts on the Lorentz indices.

The Atiyah-Singer theorem gives a formula for the index of the Dirac operator,

$$\mathrm{ind}(\slashed{D}) = \int_{\mathcal{M}_D} \hat{A}(R) \wedge Ch(F) = \int_{\mathcal{M}_D} \Omega_D \,, \qquad (D\ \text{even}), \tag{7.14}$$

where $\mathcal{M}$ is D dimensional closed manifold with D even, and the last equality defines the index density $\Omega_D$. Using this together with (7.12) gives the 2D index (prove this!), and also using (7.13), and writing out the indices, we have,

$$\mathrm{ind}(\slashed{D})_{D=4} = \int dx\, \epsilon^{\mu\nu\sigma\lambda}\left(\frac{1}{32\pi^2}\mathrm{Tr}\,F_{\mu\nu}F_{\sigma\lambda} + \frac{1}{192\pi^2}\mathrm{Tr}\,R_{\mu\nu}R_{\sigma\lambda}\right) = P - \hat{A}, \tag{7.15}$$

---

[33]Our notations for differential forms and calculus on curved surfaces are summarized in App. E.

where we specialized to a $U(1)$ connection, and introduced the notation $P$ for the first term which is (also) called the instanton number. The term $\hat{A}$ is the corresponding gravitational instanton number. We should mention that this result can be obtained by more pedestrian methods. Just as the electromagnetic vector potential couples to the electromagnetic current, the metric couples to the energy-momentum tensor. Thus the second term in (7.15) can be obtained from a triangle graph with two insertions of $T^{\mu\nu}$, and it can also be obtained using Fujikawa's method as shown in his original paper [51].

Using the theory of connections on fiber bundles, and characteristic classes, one can prove that both terms in this expression are topological, *i.e.* they cannot be changed by small deformation of the gauge field or the metric, and that they are properly normalized to be integers. This generalizes the result for the 2D index that we discussed above, and amounts to a general proof that the anomaly in any even dimension is a topological object.[34] The observant reader might have spotted a loop hole in this argument – we have not proven that we cannot add a local counter term that could change the anomaly. At a pedestrian level, this can be excluded just by enumerating all operators with the right dimensionality. This method is very cumbersome in the non-Abelian case, and there is a more sophisticated and general way to reach the same conclusion introduced by Ref. [76] (for a detailed explanation, see [1]).

As a final comment, we mention that in higher dimensions there are also mixed anomalies which contain both gauge fields and gravity. Expanding (7.14) to next order we get the 6D index density,

$$\Omega_6 = \frac{1}{48\pi^3}\operatorname{Tr} F^3 + \frac{1}{384\pi^3}\operatorname{Tr} F \operatorname{Tr} R^2 \,. \tag{7.16}$$

We can now give the result for the axial anomaly in a general even dimension and on a curved space. Recalling the formulas (5.19) and (5.27) which *mutatis mutandis* holds in any even dimension, we get,

$$\partial_\mu J_5^\mu = 2\mathcal{A}(x) = 2\Omega_D \qquad\qquad D = 2, 4, \ldots. \tag{7.17}$$

## 8    Gravitational anomalies and responses

Introducing a coupling to a background metric $g_{\mu\nu}$ gives another way to probe quantum systems and to calculate important response functions. In this section we shall explore this with an emphasis on the consequences of gravitational anomalies.

The most direct way to introduce a metric is to consider an essentially two-dimensional material on a curved surface. This could be achieved by growing films on top of curved substrates, see *e.g.* Refs. [77] and [78]. Another possibility is to deform the lattice geometry by applying strain or inserting dislocations. Furthermore, as already discussed in Sect. 3.9, gravitational fields can mimic thermal gradients and allow us to calculate thermal response.

The defining feature of a gravitational background is that it couples to all form of matter; it is not specific to fermions, nor does it distinguish chiralities. This is a great advantage when studying the response of neutral matter, and it reveals new aspects of the violation of classical conservation laws. The gravitational (3.21) and trace (3.22) anomalies can again be understood, using the perturbative and path-integral methods of the previous sections, as a consequence of the renormalization of loops or the integration measure. Also here, one anomaly can be traded for the other by adding finite counterterms to the effective action.

---

[34]Note, incidentally, that there is no gravitational contribution in 2D (see (7.10)) because the Riemann tensor has one independent scalar component only, and there is no counterpart to a pseudoscalar quantity.

For example, in the case of non-chiral theories like Dirac fermions ($c = \bar{c} = 1$), one can choose to cancel the gravitational anomaly and be left only with the trace anomaly (3.22). Integrating it again gives a topological invariant, the Euler characteristic (3.86), that generalizes to any even dimension [42]. In chiral theories with $c \neq \bar{c}$, the standard choice in conformal field theory is to cancel the trace, see (3.78). As discussed in Sect. 3.9, this non-conserved stress tensor acquires an anomalous transformation law which is responsible for the thermal Hall current.

The coupling to gravity is more technically involved than that to (Abelian) gauge fields, and here we we shall only consider 2d theories. Higher dimensions are discussed in the context of index theorems in Sect. 7 and global anomalies in Sect. 10. The physical consequences of gravitational anomalies in higher-dimensional topological states of matter is a topic of active research, and we refer to the more advanced review [79] for recent achievements.

In what follows, we first explain in more detail how heat flow can be related to gravity, and then analyze the gravitational bulk-boundary correspondence by extending the results in Sect. 3.9. We shall also use a metric to introduce a time dependent strain in a quantum Hall system, as a mean to calculate the so-called Hall viscosity. Our notations and conventions for curved space calculus are given in App. E.

## 8.1 Heat transport from gravitational response

We start by giving a general argument, due to Luttinger, which connects temperature gradients with gravitational potentials [80]. For this, consider a bar of material subject to both a temperature gradient and a gravitational field in the $y$-direction, as illustrated in Fig. 16, and let us first think of this as an insulating crystal where the only excitations are phonons, and the only transport is that of heat. The bottom is at temperature $T_2$, and the top at $T_1$, with $T_1 = T_2 + \Delta T < T_2$. Now consider a fluctuation that transfers the heat $\delta Q = \delta E$ from the hot to the cold region. The entropy changes in the two regions are $\delta S_1 = \frac{\delta E}{T_1}$ and $\delta S_2 = -\frac{\delta E}{T_2}$, respectively. This gives a change in the free energy,

$$\delta_T F \sim -T(\delta S_1 + \delta S_2) \sim \frac{T_1 - T_2}{T} \delta E = \frac{\Delta T}{T} \delta E, \tag{8.1}$$

where we assumed that the temperature gradient is small, so $|\Delta T| \ll T \equiv (T_1 + T_2)/2$. Since a positive $\delta E$ decreases the free energy, this correctly gives the direction of the heat flow from hot to cold regions. Luttinger sets out to understand such a flow in a Hamiltonian context. Naively, this is not possible, since there is no Hamiltonian description of diffusive processes, but he came up with a clever trick: Due to the equivalence principle, an energy density couples to the gravitational potential by the term,

$$F_g = \int d\mathbf{r} \, \mathcal{H}(\mathbf{r}) \frac{1}{c^2} \Phi(\mathbf{r}) \,, \tag{8.2}$$

where $\mathcal{H}$ is the energy density, and $\Phi(y) = g y$ the gravitational potential, $g$ being the force per unit of mass. Lifting the energy $\delta E$ in the gravitational potential will give an energy contribution to the free energy,

$$\delta F_g = \frac{1}{c^2} \left( \Phi(y_1) - \Phi(y_2) \right) \delta E = \frac{1}{c^2} \Delta \Phi \, \delta E \,. \tag{8.3}$$

Luttinger's insight was that in the equilibrium situation, where there is no energy flow, we must have $\delta_T F + \delta_g F = 0$, so,

$$\frac{\Delta T}{T} = -\frac{1}{c^2} \Delta \Phi \qquad \Rightarrow \qquad \frac{\partial_y T}{T} = -\frac{1}{c^2} \partial_y \Phi \,, \tag{8.4}$$

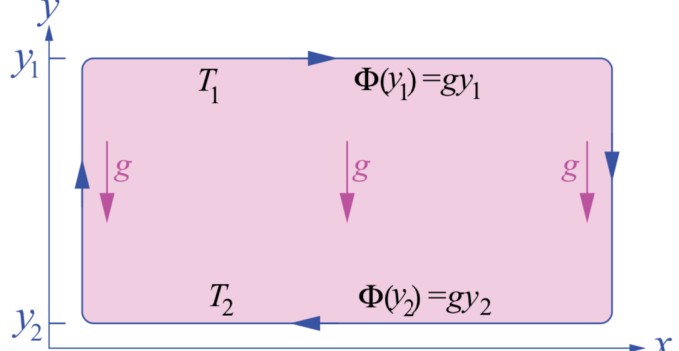

Figure 16: Quantum Hall bar placed in a gravitational field along the $y$ direction, with upper edge kept at temperature $T_1$ and lower one at $T_2$ ($T_1 < T_2$). The chiral edge modes are indicated by blue arrows. figure from [81].

where the implication holds for small gradients in $T$ and $\Phi$. This allows us to calculate the heat flow by an equivalent gravitational problem.

## 8.2 Heat flow in quantum Hall liquids

The previous Luttinger argument is not immediately applicable to the quantum Hall effect since the bulk is gapped and the heat should be transported by the gapless edge excitations as shown in Fig. 16. That Luttinger's relation (8.4) nevertheless applies to this case was shown by Stone in [81], and in the rest of this section we follow this paper rather closely. In particular, we will show that although a constant gravitational field will not amount to a bulk heat flow, there is nevertheless a gravitational bulk-edge connection related to an anomaly inflow caused by "tidal" effects that occur for non-constant fields.

### 8.2.1 Edge transport in a gravitational field

From Sect. 3.9 we recall the result for the thermal current,

$$J_T = (c - \bar{c}) \frac{\pi}{12} k_B^2 T^2 \,, \tag{8.5}$$

so, again referring to Fig. 16, this current is larger on the lower horizontal edge than on the upper one, $J_{T_2} > J_{T_1}$. But this leads to an apparent contradiction, since they are connected by the vertical edges, and the heat flow should be conserved.

The solution to this puzzle lies in the different definitions of time at the upper and lower edges due to the gradient of $\Phi(y)$, and the associated gravitational red and blue shifts of the modes on the vertical edges. To calculate these shifts, we note that the wave equation for the chiral edge modes, $\varphi$, depend on the metric as, $(\sqrt{g_{00}}\, \partial_t \pm v_F \partial_x)\varphi = 0$, and thus $\omega \propto 1/\sqrt{g_{00}}$. By Wien's displacement law, the Planck distribution peaks at a frequency $\omega \propto 1/T$, so for the frequency shift to match the temperature difference we must have,

$$\frac{\omega(y_1)}{\omega(y_2)} = \sqrt{\frac{g_{00}(y_2)}{g_{00}(y_1)}} = \frac{T_1}{T_2} \,. \tag{8.6}$$

In the limit of weak fields, the Newtonian approximation to the metric is,[35]

$$ds^2 = g_{00} dt^2 - \delta_{ij} dx^i dx^j \,, \qquad \text{with} \qquad g_{00}(y) \approx 1 + \frac{2\Phi(y)}{c^2} \,. \tag{8.7}$$

---

[35]See [82] for the precise derivation of the gravitational red shift (8.6) for the metric (8.7).

Using this, and also assuming that the temperature gradient is small, (8.6) reproduces the Luttinger relation (8.4).

The equilibrium is upheld since the edge waves on left vertical edge are red shifted, and those on the right vertical edge are blue shifted, ensuring that the energy that is flowing from one horizontal edge is "arriving" at the temperature of the receiving edge. This gives a qualitative explanation for why different currents flow on the two horizontal edges, even though they are connected by the vertical edges. Thus we conclude, with Stone, that the Luttinger picture also holds in the quantum Hall case, where the heat transport is only along the edges [81].

This results may seem at variance with the case of the bulk Hall current, but in the following we shall nevertheless find a bulk action whose variation adsorbs the anomaly at the edge, thus extending the anomaly inflow to the gravitational case.

### 8.2.2   Gravitational bulk-edge connection in the quantum Hall effect

Having seen how a chiral anomaly on a quantum Hall edge is canceled by a bulk flow due to the lack of gauge invariance of the Chern-Simons action, it is natural to ask if there is a related way to cancel the gravitational and trace anomalies. For this, we define two gravitational Chern-Simons actions, expressed in terms of the (matrix valued) Christoffel one-form, $\Gamma^\mu_\nu = \Gamma^\mu_{\nu\sigma} dx^\sigma$, and the spin connection one-form, $\omega^a_b = \omega^a_{b\mu} dx^\mu$, respectively:

$$S_{CS}[\Gamma] = \frac{c}{96\pi} \int_{\mathcal{M}} \text{Tr}\left( \Gamma d\Gamma + \frac{2}{3}\Gamma^3 \right), \tag{8.8}$$

$$S_{CS}[\omega] = \frac{c}{96\pi} \int_{\mathcal{M}} \text{Tr}\left( \omega d\omega + \frac{2}{3}\omega^3 \right), \tag{8.9}$$

where the trace is over the matrix indices, and where we, for simplicity, take $\bar{c} = 0$ from now on. These actions include cubic terms as needed for non-Abelian gauge field, and they are, for closed manifolds $\mathcal{M}$, invariant under the pertinent gauge transformations, diffeomorphisms and local Lorentz transformations respectively (see App. E).

Since we are considering gravitational backgrounds without torsion, the variables $\Gamma$ and $\omega$ are closely related, $\Gamma_\mu = e^{-1}(\omega_\mu + \partial_\mu)e$, and one can show that the two Chern-Simons actions (8.8) and (8.9) only differ by a boundary term. It turns out, however, that the variation of the actions define two different 3D stress tensors,

$$\delta S_{CS}[\Gamma] = -\frac{1}{2} \int \sqrt{g}\, \mathbf{T}_{\mu\nu}\, \delta g^{\mu\nu}, \qquad \delta S_{CS}[\omega] = - \int e\, \widetilde{\mathbf{T}}^a_\mu\, \delta e^\mu_a, \tag{8.10}$$

where $\mu, \nu, a = 0, 1, 2$. The first one is symmetric with respect to its indices, while the second one is, in general, not. This can be corrected for but let us focus on $S_{CS}[\Gamma]$, which is traceless and covariantly conserved (there is no anomaly in odd spacetime dimensions).

Consider the half space $x_2 < 0$, with a non-trivial metric at the boundary $x_2 = 0$,

$$ds^2 = (dx^2)^2 + g_{\alpha\beta}(x^0, x^1)dx^\alpha dx^\beta, \qquad \alpha, \beta = 0, 1. \tag{8.11}$$

The flux of momentum at the edge is then,

$$\dot{P}^\alpha = \int dx_1 \sqrt{g}\, \mathbf{T}^{2\alpha}, \qquad \alpha, \beta = 0, 1, \tag{8.12}$$

which should be compared to the the anomalous conservation law of the two-dimensional tensor $T_{\alpha\beta}$ introduced in Sect. 3.3,

$$\dot{P}^\alpha = \int dx_1 \sqrt{g} D_\beta T^{\beta\alpha}, \qquad \alpha, \beta = 0, 1. \tag{8.13}$$

We now evaluate the expressions (8.12) and (8.13) using the expressions, respectively, from the bulk and boundary theories and then verify that they match. At the $x_2 = 0$ boundary, the 3D tensor obtained from $S_{CS}[\Gamma]$ reads,

$$\mathbf{T}^{2\alpha} = -i \frac{c}{96\pi} \frac{1}{\sqrt{g}} \epsilon^{2\alpha\beta} \partial_\beta \mathcal{R}, \qquad \alpha, \beta = 0, 1, \tag{8.14}$$

where $\mathcal{R}$ is the two-dimensional curvature.[36] Here you should recall that (6.18) shows that the bulk flow of the electric current proportional to the field strength. Thus we would naively expect that the energy-momentum flow would by proportional to the curvature $\mathcal{R}$, while (8.14) shows that it depends on its derivative. Thus a constant curvature gives no contribution; the flow is a tidal effect. Also note that the curvature is zero for a constant gravitational field, so the heat flow discussed in Sect. 8.2.1 cannot be explained as a bulk effect.

The gravitational anomaly of the edge theory requires some discussion: we start from the general expressions (3.78) parameterized by the central charges $(c, \bar{c})$,

$$\textbf{(I)}: \qquad D^z T_{zz} = \frac{c}{48\pi} \partial_z \mathcal{R}, \qquad D^{\bar{z}} T_{\bar{z}\bar{z}} = \frac{\bar{c}}{48\pi} \partial_{\bar{z}} \mathcal{R}, \qquad T_{z\bar{z}} = T_{\bar{z}z} = 0. \tag{8.15}$$

Local counterterms, which are polynomials in the metric and its derivatives, can be added to the effective action, changing the value of the stress-tensor trace $T^\alpha_\alpha = g^{z\bar{z}}(T_{z\bar{z}} + T_{\bar{z}z})$, while keeping it symmetric $T_{z\bar{z}} = T_{\bar{z}z}$, as explained in Ref. [15]. They can only depend on the scalar curvature $\mathcal{R}$, which is the unique local, covariant quantity that can parameterize the trace anomaly in two dimensions. We may take,

$$\textbf{(IIa)}: \qquad T^\alpha_\alpha = -\frac{\lambda}{48\pi} \mathcal{R}, \qquad i.e. \qquad T_{z\bar{z}} = -\frac{\lambda}{96\pi} g_{z\bar{z}} \mathcal{R}, \qquad (\lambda = c + \bar{c}), \tag{8.16}$$

where the value of the proportionality constant $\lambda$ will become clear momentarily. Using (8.16) and the symmetry property $T_{z\bar{z}} = T_{\bar{z}z}$, the anomalous conservation laws in (8.15) can be transformed into,

$$D^z T_{zz} + D^{\bar{z}} T_{\bar{z}z} = \frac{c - \bar{c}}{96\pi} \partial_z \mathcal{R}, \qquad D^{\bar{z}} T_{\bar{z}\bar{z}} + D^z T_{z\bar{z}} = -\frac{c - \bar{c}}{96\pi} \partial_{\bar{z}} \mathcal{R}. \tag{8.17}$$

In the case of equal chiralities $c = \bar{c}$, the right-hand sides of these equations vanish, *i.e.* $D_\alpha T^{\alpha\beta} = 0$, and thus the gravitational anomaly (8.15) has been transformed into the trace anomaly (8.16), a possibility mentioned in Sect. 3.3.

For $c \neq \bar{c}$ there is no choice of the proportionality constant $\lambda$ in (8.16) or any further counterterms that would completely eliminate the gravitational anomaly [15]. Choosing $\lambda = c + \bar{c}$ is nonetheless convenient, because it makes the anomaly equations (8.17) to acquire the following covariant form,

$$\textbf{(IIb)}: \qquad D_\beta T^{\beta\alpha} = -i \frac{c - \bar{c}}{96\pi} \frac{1}{\sqrt{g}} \epsilon^{\alpha\gamma} \partial_\gamma \mathcal{R}, \qquad \alpha, \beta, \gamma = 0, 1. \tag{8.18}$$

The expressions **(I)** and **(II)** in (8.15) – (8.18) give two equivalent forms of the anomaly. As usual, what really counts is the interplay between the two conservation laws.

We are now ready to compare the edge anomaly (for $\bar{c} = 0$) with the bulk flow given by the three-dimensional Chern-Simons action $S_{CS}[\Gamma]$ (8.8). Inserting the two-dimensional (8.18) and three-dimensional (8.14) tensors in the respective flow equations (8.12) and (8.13), gives identical expressions which proves the asserted bulk-edge connection [81].

---

[36] The $i$ in front of the anomaly comes from $iS_{CS}[\Gamma]$, which is omitted in (8.8), but is important here for later comparison with Euclidean conformal field theory expressions.

For edge theories with both chiralities, $\bar{c} \neq 0$, the coupling constant of gravitational Chern-Simons action $S_{CS}[\Gamma]$ (8.8) should be changed to $(c - \bar{c})$, according to the bulk-boundary correspondence just discussed. Note that under time reversal transformations, this effective action changes sign and $c$ is exchanged with $\bar{c}$ as expected (see App. B for further discussion of edges with both chiralities).

As already pointed out, the coupling to gravity is a useful tool to characterize neutral boundary degrees of freedom in higher dimensional topological states. An important example are the Majorana fermions describing topological superconductors, and in Sect. 10 we shall indeed classify these states by using global gravitational anomalies.

### 8.3 Strain and viscosity

In elasticity theory [83], deformations are expressed in terms of the displacement field $u_i(x_j)$ as follows. On a lattice, we can think of discretized coordinates $x_i = a n_i$ representing the atom positions, with $a$ the lattice spacing and $n_i \in \mathbb{Z}$. After a deformation, their positions are displaced to $x'_i = x_i + u_i(x)$. This extends to continuous media by letting $x_i$ to be a real number.

The deformation clearly changes the distances between atoms, as measured in terms of the Euclidean metric $g_{ij} = \delta_{ij}$. The infinitesimal form of the new distance is given by,

$$dl^2 = \frac{\partial x'_k}{\partial x_i} \frac{\partial x'_k}{\partial x_j} dx_i dx_j = \left( \delta_{ij} + \partial_i u_j + \partial_j u_i + \partial_i u_k \partial_j u_k \right) dx_i dx_j = g_{ij} dx_i dx_j \, . \qquad (8.19)$$

Neglecting the quadratic term in $u_j$ for small deformations, the expression (8.19) can be interpreted as defining a non-trivial metric for distances between the original points. Note that is has the same form as a diffeomorphism of the flat metric: thus, the response to mechanical strain can be described by coupling the system to a nontrivial metric. However, unlike general relativity, diffeomorphisms do change the physical system, *i.e.* forces are not only of tidal type.

The response of quantum Hall states to gravitational backgrounds has been extensively investigated [84–87]. The Chern-Simons effective action has been extended by coupling it to a metric, leading to the Wen-Zee terms [88–90]. Other methods have also been developed, such as explicit wave-function constructions [91, 92], hydrodynamic theory [93, 94] and the $W_{\infty}$ symmetry [95].

### 8.4 The Wen-Zee term

In quantum Hall liquids there are some effects that are neither purely topological, nor depend on the details of the interactions, but only on the geometry of the surface on which the liquid resides. The reason is that both electrons and quasiparticles are characterized not only by a charge, but also by an intrinsic "orbital spin", which is an internal angular momentum associated with the particles, and is due both to their cyclotron motion in the magnetic field, and their interaction. When these spins move on a curved surface, they pick up Berry phases due to the coupling to the spin connection of the manifold.

The addition of geometrical terms to the Chern-Simons action has been discussed by Fröhlich and collaborators [84, 88], and by Wen and Zee [89]. In order to motivate them, we recall the canonical way to couple fermions to gravity (see App. E),

$$\Delta \mathcal{L} = \omega_\mu^{ab} \bar{\psi} \gamma^\mu \frac{1}{4} [\gamma_a, \gamma_b] \psi, \qquad a, b = 0, 1, 2 \, . \qquad (8.20)$$

In the non relativistic limit we consider a spatial metric $g_{ij} = g_{ij}(t, x^i)$, while the other components are $g_{0i} = 0, g_{00} = 1$. This reduces the local Lorentz symmetry to the rotation in

the plane $O(2)$ with Abelian connection $\omega_\mu = \varepsilon_{012}\omega_\mu^{12}$, and the interaction (8.20) reduces to $\omega_\mu \bar{\psi}\gamma^\mu\psi$, corresponding to the current coupling to another Abelian background $\omega_\mu$. This can be introduced in the hydrodynamics effective theory (3.40) by the substitution,

$$j^\mu A_\mu \rightarrow j^\mu \left(A_\mu + s\omega_\mu\right), \tag{8.21}$$

where the constant $s$ is the orbital spin of the excitation in question. Using the notation of forms, *i.e.* $a = a_\mu dx^\mu$ etc., we get,

$$S[a, A, \omega] = -\frac{1}{4\pi\nu}\int a\,da + \frac{1}{2\pi}\int (A + s\omega)\,da\,, \tag{8.22}$$

where $\nu = 1/k$ is the filling fraction. Integration over $a_\mu$ yields the response action,

$$S_{\text{ind}}[A, g] = \frac{\nu}{4\pi}\int \left(A\,dA + 2s\,\omega\,dA + s^2\,\omega\,d\omega\right) + \frac{c}{96\pi}\int \text{Tr}\left(\Gamma d\Gamma + \frac{2}{3}\Gamma^3\right). \tag{8.23}$$

The last term is the one canceling the gravitational anomaly on the edge as described earlier, and does not follows by the naive evaluation of the Gaussian integral over $a_\mu$. It was first found by an explicit calculation for the case of a $n$ filled Landau level of non-interacting electrons [90], and later by taking into account the so called framing anomaly [96]. This rather subtle effect is discussed in Sect. 11.

The first term in the response action (8.23) is the standard one for the Hall current; the other terms are called, respectively, the Wen-Zee term,

$$S_{WZ}[A, g] = \frac{\nu s}{2\pi}\int \omega\,dA\,, \tag{8.24}$$

and the gravitational Wen-Zee term,

$$S_{gWZ}[g] = \frac{\nu s^2}{4\pi}\int \omega\,d\omega\,. \tag{8.25}$$

The action (8.23) is not the full story. There are other, non-geometrical, terms that are local and gauge invariant and depend on details of the microscopic Hamiltonian [90, 97], but these non-universal contribution will be of no concern here.

The concept of orbital spin $s$ is relevant not only for quantum Hall liquids, but also for paired states like superconductors. In fractional states, $s$ depends on the interactions but for completely filled Landau levels it is particularly simple. In the $i$-th level it is,

$$s_i = \frac{2i+1}{2}, \qquad i = 0, \ldots, n-1\,. \tag{8.26}$$

This expression is obtained by computing the total angular momentum $M_i$ of the $i-th$ Landau level densely filled with $N$ electrons, starting from the origin, and then using the formula,

$$M_i = \frac{N^2}{2} - Ns_i\,, \tag{8.27}$$

whose term linear in $N$ gives the "intrinsic" angular momentum.

## 8.5 Hall viscosity

We now review the physical consequences of the Wen-Zee action (8.24) which involves three terms,

$$S_{WZ} = \frac{\nu s}{2\pi} \int A d\omega = \frac{\nu s}{2\pi} \int d^3x \left( \frac{\sqrt{g}}{2} A_0 \mathcal{R} + \epsilon^{ij} \dot{A}_i \omega_j + \sqrt{g} B \omega_0 \right), \tag{8.28}$$

where the dot indicates the time derivative, $i, j = 1, 2$ are spatial indices, $\mathcal{R}$ is the curvature of the spatial metric and $\mathcal{B}$ the magnetic field.[37]

The variation of $S_{WZ}$ with respect to $A_0$ gives an additional contribution to the density that is expressed in terms of the spatial curvature. The total number of electrons $N$ for a closed spatial geometry, *e.g.* the sphere, is obtained by integrating the density, leading to the result,

$$N = \int d^2x \sqrt{g} \rho = \frac{\nu}{2\pi} \int d^2x \ \sqrt{g} \left( B + \frac{s}{2} \mathcal{R} \right) = \nu N_\phi + \nu s \chi, \tag{8.29}$$

where $N_\phi$ is the number of magnetic fluxes through the surface and $\chi$ is its Euler characteristic (3.86).

Equation (8.29) shows that the usual relation, $N = \nu N_\phi$, between electrons and fluxes is modified on a curved surface by a $O(1)$ correction, called the "shift", $\mathcal{S} = s\chi$. For the sphere, this is $\mathcal{S} = 2s$, and its value is obtained from the total angular momentum of the wavefunction in this geometry using (8.27); for example, $s = p/2$ for $\nu = 1/p$ [89]. The shift is an important quantity since it can be calculated numerically and is used to distinguish between competing model wave functions having the same filling fraction.

Next, varying the action (8.28) with respect to the spatial gauge background gives the Hall current with an additional term,

$$J^i = \frac{1}{\sqrt{g}} \frac{\delta S[A, g]}{\delta A_i} = \frac{1}{\sqrt{g}} \frac{\nu}{2\pi} \epsilon^{ij} \left( E^j + s E_{(g)}^j \right), \qquad E_{(g)}^i = \partial_i \omega_0 - \partial_0 \omega^i, \tag{8.30}$$

which defines the "gravi-electric" field $E_{(g)}^i$.

We now compute the induced stress tensor by varying the Wen-Zee action with respect to the metric. For small fluctuations around flat space, $g_{ij} = \delta_{ij} + \delta g_{ij}$, we should relate the variation of the metric to that of the zweibeins $\delta e_j^a$. First, we must choose a gauge for the local O(2) symmetry: the zweibeins $e_i^a$, with $a, i = 1, 2$, can be taken to be symmetric matrices, *i.e.* $e_i^a = \delta_i^i$ to lowest order. It follows that the variation of the metric is, $\delta g_{ij} = \delta e_j^a \delta_{ai} + \delta e_i^a \delta_{aj} = 2\delta e_{ij}$. Using formulas explained in [95], the Wen-Zee action (8.28) can be rewritten to quadratic order in $\delta g_{ij}$ as follows,

$$S_{WZ} = \frac{\nu s}{4\pi} \int d^3x \left( A_0 \mathcal{R} + \epsilon^{ij} \dot{A}_i \Gamma_{j,kl} \epsilon^{kl} - \frac{B_0}{4} \epsilon^{ij} \delta g_{ik} \delta \dot{g}_{jk} \right), \tag{8.31}$$

where $B_0$ is the constant magnetic field.

The induced stress tensor to leading order in the metric is therefore,

$$T_{ij} = -\frac{2}{\sqrt{g}} \frac{\delta S}{\delta g^{ij}} = -\frac{\eta_H}{2} \left( \epsilon_{ik} \dot{g}_{kj} + \epsilon_{jk} \dot{g}_{ki} \right), \tag{8.32}$$

with

$$\eta_H = \frac{\rho_0 s}{2} = \frac{\nu s B_0}{4\pi}. \tag{8.33}$$

---

[37]For a detailed discussion, see [95]. Curved space expressions are explained in the Appendix of this work and also in App. E.

The quantity $\eta_H$ is a transport coefficient, called Hall viscosity, which parameterizes the response of the fluid to a time-dependent shear. It is only present in two dimensional systems that violate time reversal symmetry, and describes a non-dissipative viscous force orthogonal to the fluid velocity (see Fig. 17), as first discussed by Avron, Seiler and Zograf [85, 98].[38]

Since $s$ is the coupling constant of an action of Chern-Simons type, it describes a geometric response of the system that is independent of the local dynamics. Furthermore, in compact geometries it parameterizes a topological quantity (cf. (8.29)), although it cannot be associated to a anomaly of the edge theory. In Ref. [13], it was shown that $s$ amounts to a shift in the momentum of edge excitations, that is a kind of Casimir effect, or chemical potential change. This quantity can be unambiguously measured in edge transport experiments.

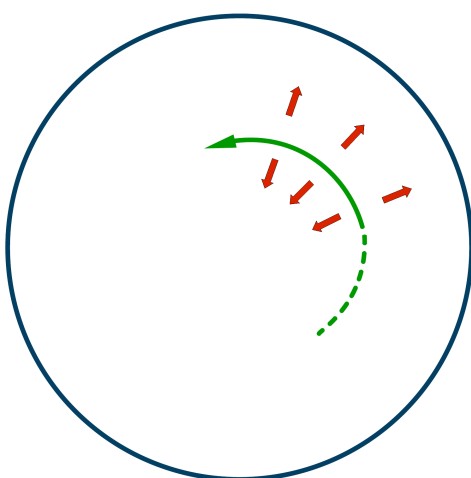

Figure 17: Illustration of the Hall viscosity in a circular droplet of electrons: a counter-clockwise stirring of the fluid in the bulk of the droplet causes an orthogonal force (red arrows) (figure from [95]).

We remark that the stress tensor (8.32) is first order in time derivatives, and thus corresponds to a non-covariant force. This is expected, since the Wen-Zee action (8.28) is only invariant under time-independent coordinate reparameterization. Let us choose the conformal gauge for the metric at $t = 0$, $g_{ij}(0, x) = \sqrt{g}\,\delta_{ij}$, and represent the deformations of the fluid by time-dependent coordinate changes, $\delta x^i = u^i(t, x)$. These can be divided into: (i) Conformal transformations, for which $\partial g_{ij} = \partial_i u_j + \partial_j u_i = \delta_{ij}\,\partial_k u^k$ as explained in Sect. 3.3; (ii) Area-preserving diffeomorphisms, obeying $\partial_k u^k = 0$ and thus keeping the determinant of the metric $g$ constant.

Upon substitution into (8.32), one finds that the stress tensor is expressed in terms of area-preserving diffeomorphisms only, causing the shear. These can be parameterized by a scalar function $w(t, x)$, with the result,

$$T_{ij} = \eta_H \left( 2\partial_i \partial_j - \delta_{ij} \partial^2 \right) \dot{w}, \qquad \delta x^i = u^i = \varepsilon^{ij}\,\partial_j w(t, x). \qquad (8.34)$$

In conclusion, the orthogonal force in Fig. 17 is due to the deformations of the fluid given by time-dependent area-preserving diffeomorphisms.

---

[38]The relation (8.33) between the Hall viscosity and the orbital spin $s$ is actually valid in general [86]. For a non-trivial example at $\nu = 2/5$ see [99].

# 9 Effective theories of symmetry protected topological states

The last twenty years has seen an enormous progress in our understanding of various topological states of matter.[39] We have already discussed quantum Hall and quantum spin Hall states, as well as Weyl semi-metals, but there are other systems, such as the topological insulators and topological superconductors. In the remaining parts of these lectures we shall outline how anomalies have become an important tool for classifying and understanding these states. To begin, we give a brief introduction to symmetry protected topological states of non-interacting fermionic systems (See the first chapter in Ref. [100] for some complementary material and Refs. [101], [102], [103] for a thorough presentation). We shall limit our discussion to systems where the electronic excitations are gapped.[40]

Next, we describe the effective theories of topological insulators and superconductors, and explain the nature of their gapless boundary states. We also discuss the stability argument by Fu, Kane and Mele that was instrumental to understand topological insulators. Finally we mention the possibility of having gapped boundary states that are themselves topologically non-trivial.

In this and the following sections, you will gradually encounter the different types of anomalies that are associated to these topological states and learn how they are used to characterize them also in presence of interactions.

## 9.1 Introduction – the ten-fold classification

Usual non-topological gapped systems are featureless, or "frozen", when probed at energies below the gap. In topological states, there are instead global effects, usually captured by a topological effective theory, like massless boundary excitations, and non-trivial responses to varying a gauge background or space geometry. The archetypical example is the quantum Hall effect, and the properties captured by the Chern-Simons effective theory. In the following we shall outline how this kind of physics extends to a wide range of systems not necessarily involving a strong external magnetic field or breaking of parity and time-reversal symmetries.

We have already emphasized that the integer Hall effect can be understood in terms of free electrons, while interactions are crucial in the case of the fractional effect. The new topological states that have been discovered and classified are close in spirit to the integer case in the sense that free-fermion models capture their main aspects. They are referred to as "symmetry protected topological" (SPT) states and have gapless edge states, but no fractionalization [104, 105]. They are different from the "topologically ordered" (TO) states which are all fractionalized, and also have degenerate ground states when defined on spaces with non-trivial topology. The best known example is the $p$-fold degeneracy of the $\nu = 1/p$ quantum Hall state on a torus, where the concept of topological order was first introduced [26]. Yet another feature of the TO states is that they are "long-range entangled", which is manifested in a universal term in the ground-state entanglement entropy [104]. An SPT state is instead "short-range entangled" [106].

There has been several proposals for fractional, and thus TO, versions of the new topological states. These are based both on effective field theories and explicit band theory models, but so far they have not yet been observed experimentally [107, 108].

The SPT states can be described by band models of non-interacting fermions characterized by the presence/absence of the symmetries of time reversal ($T$) and charge conjugation ($C$) (or particle-hole, symmetry). The integer quantum Hall liquids are SPT states with no

---

[39]Or, equivalently, "topological phases of matter".
[40]By only considering electronic excitations we disregard the gapless phonons that are present in all crystal materials.

symmetries.[41] By definition, two states have the same TO if one can continuously transform the Hamiltonian, and thus the ground-state wave function, of one to that of the other without closing the gap. For the SPT states the same holds but with the restriction that the Hamiltonian must respect some symmetries during the whole transformation. Breaking a symmetry can bring them into different SPT phases or trivial insulators.

| class\$\backslash d$ | T | C | S | 0 | 1 | 2 | 3 | 4 | 5 | 6 | 7 |
|---|---|---|---|---|---|---|---|---|---|---|---|
| A | 0 | 0 | 0 | $\mathbb{Z}$ | 0 | $\mathbb{Z}$ | 0 | $\mathbb{Z}$ | 0 | $\mathbb{Z}$ | 0 |
| AIII | 0 | 0 | 1 | 0 | $\mathbb{Z}$ | 0 | $\mathbb{Z}$ | 0 | $\mathbb{Z}$ | 0 | $\mathbb{Z}$ |
| AI | $+$ | 0 | 0 | $\mathbb{Z}$ | 0 | 0 | 0 | $2\mathbb{Z}$ | 0 | $\mathbb{Z}_2$ | $\mathbb{Z}_2$ |
| BDI | $+$ | $+$ | 1 | $\mathbb{Z}_2$ | $\mathbb{Z}$ | 0 | 0 | 0 | $2\mathbb{Z}$ | 0 | $\mathbb{Z}_2$ |
| D | 0 | $+$ | 0 | $\mathbb{Z}_2$ | $\mathbb{Z}_2$ | $\mathbb{Z}$ | 0 | 0 | 0 | $2\mathbb{Z}$ | 0 |
| DIII | $-$ | $+$ | 1 | 0 | $\mathbb{Z}_2$ | $\mathbb{Z}_2$ | $\mathbb{Z}$ | 0 | 0 | 0 | $2\mathbb{Z}$ |
| AII | $-$ | 0 | 0 | $2\mathbb{Z}$ | 0 | $\mathbb{Z}_2$ | $\mathbb{Z}_2$ | $\mathbb{Z}$ | 0 | 0 | 0 |
| CII | $-$ | $-$ | 1 | 0 | $2\mathbb{Z}$ | 0 | $\mathbb{Z}_2$ | $\mathbb{Z}_2$ | $\mathbb{Z}$ | 0 | 0 |
| C | 0 | $-$ | 0 | 0 | 0 | $2\mathbb{Z}$ | 0 | $\mathbb{Z}_2$ | $\mathbb{Z}_2$ | $\mathbb{Z}$ | 0 |
| CI | $+$ | $-$ | 1 | 0 | 0 | 0 | $2\mathbb{Z}$ | 0 | $\mathbb{Z}_2$ | $\mathbb{Z}_2$ | $\mathbb{Z}$ |

Figure 18: The tenfold classification of symmetry protected topological phases of non-interacting fermions. See the text for explanation of axis labels and symbols.

The classification of SPT states in any space dimension $d$ is shown in Fig. 18. We first explain the content of this table and then discuss how it is derived. Columns $T, C, S$ indicate the symmetries of each class, where $S = TC$ is called "chiral" or reflection symmetry of the energy spectrum with respect to positive and negative values. Both time reversal and charge conjugation are antiunitary maps that square to $T^2 = \pm 1$, and $C^2 = \pm 1$. The alternatives are denoted as $(+)$ and $(-)$ in the table, and the third option $(0)$ means that there is no symmetry. This would imply nine classes, but when neither $T$ nor $C$ is present there are nevertheless two alternatives for the third entry $S$. The labels in the first column refer to the classification of maximally symmetric spaces due to the mathematician E. Cartan. These space are similar to spheres in the sense that all point are equivalent. The correspondence between the two classifications is by the theory of localization, to be discussed later.

The integer Hall effect has no symmetries and appears in the first row in the table, labeled by (A); here $\mathbb{Z}$ denotes the possible values of the Hall conductance in units of $e^2/h$. As explained earlier, this integer is the first Chern class as calculated from the Berry phase (or alternatively, the monopole charge in the spectral flow of edge states). Correspondingly, the integer sets listed in the other cases give the possible values of other characteristic topological invariants (with $(0)$ indicating no topology). In more physical terms, these integers count the number of branches of boundary excitations.

The topological insulators, already briefly introduced in Sect. 6.2, belong to class AII, are protected by TR symmetry that squares to minus one, and have index $\mathbb{Z}_2$ in both two and three dimensions, meaning that their boundary excitations have one branch at most – later we shall explain how this comes about. An interesting feature of the classification is that the same pattern repeats itself in any dimension, shifted one step down, with period two for the

---

[41]It is fair to ask why a state without any symmetry should be called SPT. Some argue that it is protected by charge conservation. We consider this a matter of semantics.

first two rows and eight for the remaining ones. This feature is related to the properties of fermions in different dimensions.

The ten-fold classification was originally obtained using three approaches:[42]

- The first, referred to as "topological band theory", studies general translation invariant lattice models using the mathematics of K-theory [110].

- The second analyzes Anderson localization in non-interacting electron systems, and finds a relation to the classification of random matrices and diffusion equations on maximally symmetric manifolds [111].

- The third considers the representations of the $T$, $C$ and $S$ symmetries acting on fermion spinors. In fact, Ref. [112] showed that for any SPT state, one can find a "representative" massive Dirac theory in the continuum with the same topological properties as those of more realistic microscopic lattice Hamiltonians.

In these lectures, we do not explain these results, but describe another approach that uses effective field theories and anomalies (initiated in Ref. [113]). Using the continuum theories associated to SPT states by the third method above, we identify the associated massless boundary excitations and use their characteristic anomalies to detect and classify them.

Actually the use of anomalies has major advantages: They are exactly known even in the presence of interactions, and moreover are related to topological quantities that are insensitive to small changes in the Hamiltonian. Therefore, a classification based on anomalies also applies to interacting systems. This is obviously very important since the electrons in all real material are interacting. In later sections, we shall provide examples where the original ten-fold classification breaks down by introducing cleverly chosen interactions, which turn the SPT state into a trivial insulator without closing the gap. In these cases, anomalies give the correct answer for the stable topological phases with interactions. We shall come back to this in Sect. 10.4.

### 9.1.1 Some historical remarks

Here we describe some hallmarks in the development that led to the discovery of topological states of matter beyond the quantum Hall effect. For comprehensive reviews see *e.g.* [114] and [115].

*The Chern Insulator.*

Haldane constructed a model of two-dimensional lattice fermions that has gapless edge excitations and a quantized Hall conductance, in absence of an external magnetic field [116]. The lattice is hexagonal, as in graphene, but the Hamiltonian has additional next-to-nearest neighbor couplings with nontrivial phase factors that break TR invariance. One can think of them as staggered fluxes that average to zero over a unit cell. Haldane found that the Berry curvature is non-vanishing over a range of coupling constants, and integrates to the Chern invariant and thus a quantized Hall conductance $\sigma_H = \pm e^2/h$ [65].

In more physical terms, the effect of the external magnetic field is traded for phase dependent couplings in the Hamiltonian, resulting into a topologically non-trivial Berry connection, a gauge connection in parameter space instead of real space.

Regarding the effective field theory description, the low-energy spectrum of the Haldane model includes two Dirac fermions in $(2+1)$ dimensions, that break TR symmetry when their masses obey $m_1 + m_2 \neq 0$. In this system, there is a "global" anomaly, a new kind of anomaly affecting the discrete symmetries of parity and time reversal, to be discussed in Sect. 9.2.2. It

---

[42]For a review, see [109].

implies a low-energy Chern-Simons effective theory with coupling $k = \pm 1$. This field theory result confirms the presence of the Hall current and edge excitations.

*The quantum spin Hall effect.*

This system was already introduced in Sect. 6.2 in a rather formal way by considering a pair of quantum Hall system whose edge excitations have opposite chiralities and spins, so as to obtain a TR invariant system. It can be actually realized in some materials in presence of strong spin-orbit coupling, of the form $(\mathbf{E} \times \mathbf{p}) \cdot \mathbf{S}$.[43] When the electric field $\mathbf{E}$ is confined to a plane and linear in the $(x, y)$ coordinates, one obtains the TR invariant Hamiltonian,

$$H = \frac{1}{2m} (\mathbf{p} - \mathbf{A} \sigma_z)^2 , \qquad \mathbf{A} = \frac{B}{2} (-y, x, 0) , \qquad (9.1)$$

which actually represents the two opposite Hall systems, respectively for spin-up and spin-down electrons [118].

The quantum spin Hall effect is interesting because TR symmetry makes it rather different from the quantum Hall effect. The Hall conductivity as well as the Berry phase vanish in this case, but there is a non-vanishing spin Hall conductivity given by the difference of currents of the two kinds of electrons.

*Topological insulators.*

One would naively think that edge excitations of opposite chirality can interact and create a gap, thus making the quantum spin Hall system a trivial insulator. However, such interactions are not possible without breaking time-reversal symmetry, because only half of the possible scattering channels are present, given that the signs of velocity and spin are paired.[44] Still, there can be TR invariant interactions that causes spin-flips and violate spin current conservation.

The question of stability of topological insulators, *i.e.* realistic TR invariant gapped systems with topological features (class AII), was addressed and solved by Kane, Mele and Fu in a series of papers [61, 119, 120]. They introduced a $\mathbb{Z}_2$-valued topological quantity which generalizes the Berry phase, and measures the spin parity $(-1)^{2S} = (-1)^F$ (*i.e.* the fermion number parity) of the boundary modes. This sign is the remnant of the $U(1)_S$ symmetry of the spin Hall effect, once spin flip transitions are introduced while preserving TR invariance. These authors also generalized the Laughlin flux insertion argument to explain how this index characterizes the TR invariant topological phase in the presence of TR invariant interaction. In Sect. 9.3 we shall discuss these results further.

*Experimental observation of topological insulators and the tenfold classification.*

Topological insulators were observed in two and three space dimensions in 2007 and 2008, respectively, and their gapless boundary excitations have been observed by measuring responses and photoelectric emission [114]. At about the same time, the classification of the SPT phases was achieved, using the methods outlined above [115].

## 9.2 Effective response action for 4D topological insulators

In this section we analyze SPT states with an unbroken electromagnetic $U(1)$ symmetry. We shall derive the bulk topological effective theories and the associated boundary dynamics. As seen already, the response of a system is coded in the effective response action $\Gamma[A]$, and we show that in 4D (as well as in any even dimension), this contains a piece which encodes an "anomalous response" to external fields, and that it implies the existence of "protected"

---

[43] An electric field is not necessary to generate this state. The intrinsic spin-orbit coupling in heavy atoms, and the presence of a substrate can also lead to the quantum spin Hall effect. Actually, the interplay of these factors has led to the observation of conductive edge states even at room temperature [117].

[44] For a more detailed explanation of this, see Sect. II C of [115].

surface modes. As for the physical significance of this result, we already mentioned that the Dirac continuum theory is just a representative theory for extracting the topological properties of more realistic models.

### 9.2.1 The $\theta$ term and the Chern-Simons boundary action

We will now apply the techniques we used to derive chiral and axial anomalies in even space-time dimensions to calculate possible topological terms in the electromagnetic response action. The starting point is the representative massive Dirac action,

$$S_E = \int dx\, \bar{\psi}\left(i\slashed{\partial} + \slashed{A} - M_0(\cos\theta + i\sin\theta\,\gamma^{D+1})\right)\psi\,, \tag{9.2}$$

and you can find the derivation of this action from topological band theory in *e.g.* [112]. In (9.2) $\theta$ is a constant angle, and we note that changing it from 0 to $\pi$ amounts to switching sign of the mass $M_0$. Under the axial symmetry transformation $\psi \to \exp(-i\frac{\xi}{2}\gamma^{D+1})\psi$, theta changes as $\theta \to \theta - \xi$. We can now calculate the change in the effective action $\Gamma_\theta[A]$ as we vary $\theta$. But this is precisely the calculation that we did in Sect. 5.3 to get the anomaly,

$$\delta\Gamma_\theta[A] = 2i \int dx\, \frac{\delta\theta}{2}\mathcal{A}\,, \tag{9.3}$$

where $\mathcal{A}$ is the anomaly given by (5.21). This can immediately be integrated to give the "theta term" in the effective action, which in 4D becomes,

$$\Gamma_\theta[A] - \Gamma_0[A] = i\theta\,\frac{1}{16\pi^2}\int_{\mathcal{M}} dx\, F\tilde{F} = i\theta P = i\theta\,\mathrm{ind}(\slashed{D})\,. \tag{9.4}$$

Since for closed 4D manifolds $\mathcal{M}$ the index in (9.4) is an integer, the partition function $\exp(i\Gamma_\theta[A])$ is a $2\pi$-periodic function of $\theta$. The lesson we get from the above calculation is that SPT states in even dimensions can have a theta term in the effective action. Also note that symmetries will impose constraints on $\theta$. Under time reversal, $i\Gamma$ changes sign, so for TR invariant states, the only allowed values are $\theta \in \{0, \pi\}$, modulo $2\pi$. If we further assume that $\theta = 0$ describes a trivial insulator equivalent to the vacuum, we also have a non-trivial, TR invariant, SPT phase for $\theta = \pi$ which is the 3d topological insulator.

An important point about $\Gamma_\theta[A]$ is that it is a topological invariant: Its variation with respect to $A$ vanishes, so it does not give any current response, and also no contribution to Lagrangian equations of motion. This is, however, true only if the theory is defined on a closed manifold. In reality, any topological insulator has boundaries, and then $\Gamma_\theta[A]$ is no longer a constant. To see this we write,

$$\begin{aligned}
\Gamma_\theta[A] &=& i\frac{\theta}{8\pi^2}\int_{\mathcal{M}} dx\, \partial_\mu \epsilon^{\mu\nu\sigma\lambda}A_\nu\partial_\sigma A_\lambda \\
&=& i\frac{\theta}{8\pi^2}\int_{\partial\mathcal{M}} dx\, \epsilon^{\nu\sigma\lambda}A_\nu\partial_\sigma A_\lambda\,.
\end{aligned} \tag{9.5}$$

and in the last equality we recognize the 3D Chern-Simons action with level $k = \theta/2\pi$. Taking $\theta = \pm\pi$, for topological insulators, we have $k = \pm 1/2$. We recall from Sect. 3.5.2 that this is effective action for a quantum Hall state with Hall conductivity corresponding to $\nu = \pm 1/2$.

This result raises two questions. First, a Hall state on the boundary would seem to violate the TR symmetry. Second, the sign of the Chern-Simons term differ depending on whether $\theta$ is $\pi$ or $-\pi$ although $\theta$ is required to be defined only modulo $2\pi$. To resolve these issues, we

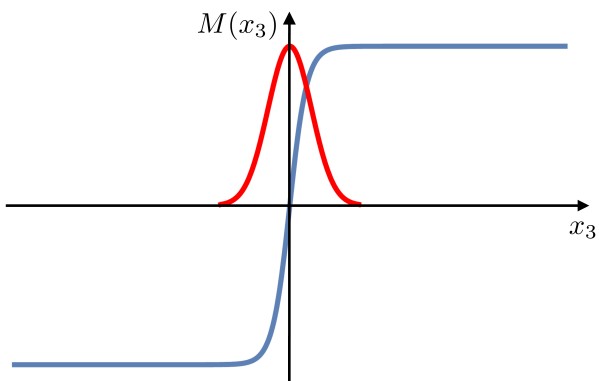

Figure 19: Mass profile $M = M(x_3)$ (blue line) at the boundary of a topological insulator and wavefunction of the Dirac zero mode (red line).

will consider a regularized version of the interface, and model it by taking the mass in (9.2) to have a kink profile $M(x_3)$ such that $\lim_{x\to\pm\infty} M(x_3) = \pm M_0$ (see Fig. 19). This amounts to taking $\theta = \pi$ in the left half space and $\theta = 0$ in the right half space, with the boundary at $x_3 = 0$. Furthermore, we shall explicitly break TR invariance by adding a small Pauli coupling with magnetic moment $\mu$. The reason for this is twofold. It corresponds to an interesting physical situation when a TI is subjected to a weak magnetic field, and it also provides an infrared cutoff that allows for a well defined perturbative calculation. The Pauli term for a magnetic field in the $x^3$ direction, *i.e.* perpendicular to the surface, is,

$$H_p = -\mu B \psi^\dagger \Sigma^3 \psi = \mu B \bar\psi \gamma^5 \gamma^3 \psi \,, \tag{9.6}$$

where we used the relation $\Sigma^i = i\varepsilon_{ijk}[\gamma^j, \gamma^k]/4 = \gamma^5 \gamma^0 \gamma^i$ for the Dirac spin vector [44].

We now look for solutions to the Dirac equations that are localized at the boundary. We first note that the Dirac equation (9.2) admits a static (zero energy) solution localized at $x_3 = 0$ (see Fig. 19),

$$[i\gamma^3 \partial_3 - M(x_3)]\psi(x_3) = 0 \,, \tag{9.7}$$

as originally shown by Jackiw and Rebbi in $(1+1)$-dimensions [121]. This allows for the dimensional reduction of the 4D fermion to the 3D boundary, as follows.

We use the following basis for the $\gamma$ matrices,

$$\gamma^0 = \begin{pmatrix} 0 & \sigma_3 \\ \sigma_3 & 0 \end{pmatrix}, \quad \gamma^1 = i\begin{pmatrix} 0 & \sigma_1 \\ \sigma_1 & 0 \end{pmatrix}, \quad \gamma^2 = i\begin{pmatrix} 0 & \sigma_2 \\ \sigma_2 & 0 \end{pmatrix}, \quad \gamma^3 = i\begin{pmatrix} 1 & 0 \\ 0 & -1 \end{pmatrix}, \tag{9.8}$$

and seek for a general solution with separation of variables,

$$\psi = \chi(x_\alpha)\phi(x_3), \qquad \phi = N\begin{pmatrix} \chi \\ 0 \end{pmatrix}\exp\left(-\int^{x_3} dx' M(x')\right), \tag{9.9}$$

where $\alpha = 0, 1, 2$. Upon substituting it in the Dirac equation (9.2), putting $A_\mu = 0$ and adding the Pauli term (9.6), we get,

$$\mathcal{L} = \begin{pmatrix} 0, & \chi^\dagger \sigma_3 \end{pmatrix}\begin{pmatrix} 0 & i\sigma_3\partial_0 - \sigma_1\partial_1 - \sigma_2\partial_2 - m \\ i\sigma_3\partial_0 - \sigma_1\partial_1 - \sigma_2\partial_2 - m & 0 \end{pmatrix}\begin{pmatrix} \chi \\ 0 \end{pmatrix}, \tag{9.10}$$

where $m = -\mu B$. Identifying the 3D gamma matrices, $\gamma^\alpha = (\sigma_3, i\sigma_1, i\sigma_2)$, $\alpha = 0, 1, 2$, and $\bar{\chi} = \chi^\dagger \sigma_3$ we get the 3D Dirac equation,

$$\mathcal{L} = \bar{\chi} \left( i\slashed{\partial} + \slashed{A} - m \right) \chi \,, \tag{9.11}$$

where we reintroduced the electromagnetic field in the gauge $A_3 = 0$. Note that the TR breaking Pauli term appear as a mass in the (2+1)D Dirac boundary theory. At this point we make a slight digression and derive the effective response action for this theory.

### 9.2.2 Chern-Simons action from Dirac fermions in 2+1 dimensions

The most direct way to extract the electromagnetic effective action $\Gamma[A]$ is to start from the Lagrangian (9.11) and simply calculate the Feynman diagram in Fig. 10 with a suitable regularization as was originally done in Ref. [122], and we will return to this in Sect. 10. As an illustrative alternative, we shall here use the same idea as in the calculation of the axial anomaly in Sect. 4.4, *i.e.* solve the Dirac equation exactly in a constant magnetic and then invoke gauge and Lorentz invariance to find the general result.

We take the following $(2 + 1)$-dimensional Dirac $\alpha$-matrices,

$$(\beta, \alpha^x, \alpha^y) = (\sigma^3, -\sigma^2, \sigma^1), \tag{9.12}$$

use the complex coordinates,

$$z = \sqrt{\frac{eB}{2}}(x + iy), \tag{9.13}$$

and the notation $\partial = \partial_z$ and $\bar{\partial} = \partial_{\bar{z}}$, as in Sect. 3.1. In the symmetric gauge, $\mathbf{A} = \frac{B}{2}(y, -x)$, where $B$ is a constant magnetic field in the negative $z$-direction, the Hamiltonian for the relativistic Landau problem becomes,

$$
\begin{aligned}
H &= \vec{\alpha} \cdot (\vec{p} + e\vec{A}) + \beta m \\
&= \sqrt{eB} \left( \begin{array}{cc} \mu & \frac{1}{\sqrt{2}}(\partial - \bar{z}) \\ -\frac{1}{\sqrt{2}}(\bar{\partial} + z) & -\mu \end{array} \right) = \sqrt{eB} \left( \begin{array}{cc} \mu & a^\dagger \\ a & -\mu \end{array} \right),
\end{aligned} \tag{9.14}
$$

where $[a, a^\dagger] = 1$, and we defined $\mu = m/\sqrt{eB}$. In terms of the number operator states $a^\dagger a |n\rangle = n|n\rangle$, we easily find the following Landau level spectrum for the $m = 0$ case,

$$|\Psi_0\rangle = \left( \begin{array}{c} |0\rangle \\ 0 \end{array} \right); \qquad\qquad\qquad E_0 = 0 \,, \tag{9.15}$$

$$|\Psi_{n\pm}\rangle = \frac{1}{\sqrt{2}} \left( \begin{array}{c} |n\rangle \\ \pm |n-1\rangle \end{array} \right); \qquad\qquad E_{n\pm} = \pm\sqrt{neB} \,, \qquad n > 0 \,. \tag{9.16}$$

Thus there are infinitely many positive and negative levels which are paired by the charge-conjugation (particle-hole) symmetry. The exception is the zero-mode which is self-conjugate. The solution for $m \neq 0$ is easily found by observing that $H^2$ is a diagonal matrix. The $|\Psi_0\rangle$ remains an eigenstate with the energy $E_0 = m$, while the rest of the spectrum has energies $E_{n\pm} = \pm\sqrt{neB + m^2}$, for $n > 0$. We should also remember that, just as explained in Sect. 3.1, all the Landau levels are highly degenerate.

The current operator $j^\mu = e : \bar{\psi}\gamma^\mu\psi :$ includes the normal ordering to subtract the contribution of the filled Dirac sea, while respecting charge conjugation symmetry. In a static magnetic field the spatial components of the current vanish: recalling that $\gamma^0 = \beta$, we get the following expression for the charge density,

$$\langle j^0 \rangle = \frac{e}{2} \langle \Omega| \left( \psi^\dagger\psi - \psi\psi^\dagger \right) |\Omega\rangle. \tag{9.17}$$

This expectation value is charge conjugation symmetric and receives contributions from the states in the $E_0 = m$ level only. Setting the Fermi energy to zero, we find that for $m < 0$ all states in this level are all filled, and the expectation value comes entirely from the first term in (9.17), while for $m > 0$ only the second term contributes. Recalling from Sect. 3.1 that the density of a Landau level is $\rho_0 = eB/2\pi$, we get the ground state charge density,

$$\langle j^0 \rangle = -\frac{m}{|m|} \frac{1}{2} \frac{e^2}{2\pi} B \ . \tag{9.18}$$

To understand this result you should recall (or prove!) that a Dirac mass term in (2+1)d violates both TR invariance and parity.[45] A way to realize this is to first note that in 2d the group of spatial rotations is just O(2), so the spin of a fermion is a pseudoscalar number fixed to $S = \pm 1/2$. Contrary to the 4D case the mass cannot change sign by a chiral rotation. Furthermore the sign of the spin and mass are related by $S = \text{sign}(m)/2$, and the antiparticles will have the opposite sign.[46] Since the TR operation changes the sign of angular momentum, the mass term breaks TR invariance.

Just as in the arguments based on the infinite hotel, we can now invoke Lorentz and gauge invariance to conclude,

$$\langle j^\mu \rangle = \frac{1}{i} \frac{\delta}{\delta A_\mu} \Gamma[A] = -\frac{m}{|m|} \frac{e^2}{4\pi} \varepsilon^{\mu\nu\sigma} \partial_\nu A_\sigma \,, \qquad \left( F_{\mu\nu} = \text{const.} \right) \,, \tag{9.19}$$

for constant electric and magnetic fields. Integrating this expression we get the Chern-Simons response action,

$$\Gamma[A] = \frac{m}{|m|} \frac{e^2}{8\pi} \int d^3x \, \varepsilon^{\mu\nu\rho} A_\mu \partial_\nu A_\rho \,, \tag{9.20}$$

with coupling $k = \text{sign}(m)/2$. You should note that even though we had to introduce a mass to make this calculation well defined, the final result only depends on its sign.

Finally note that (9.20) does not include Maxwell terms, nor higher derivatives and powers of $F_{\mu\nu}$ that do not contribute to the current (9.19) for $F^{\mu\nu} = \text{const.}$ [124]. These additional terms are non-anomalous and do not enter in the following discussions.

### 9.2.3 Boundary action for the 3d topological insulator

We are now prepared to revisit the questions raised in 9.2.1 about the nature of the boundary states. First we note that, not unexpectedly, regularizing with a small Pauli term $\sim B$ determines the chirality of the boundary, or equivalently, the sign of the Chern-Simons action. But we are left with another serious problem. Both the argument leading to (9.5) and the effective action (9.20) give a Chern-Simons term with a half integer level number, in contradiction with the expected integer value for a theory of free fermions. So the problem is now whether these results should be added to give an integer level number, or are they two ways to calculate the same thing, in which case the level number is indeed half integer. This question was addressed in a paper by Mulligan and Burnell [125], who did the full calculation of the two point function taking into account both bulk and boundary effects. They used the special mass profile,

$$M(x) = M_0 \tanh(M_0 x) \,, \tag{9.21}$$

---

[45]Parity in 2d is a reflection *e.g.* $(x, y) \to (-x, y)$ since the transformation $(x, y) \to (-x, -y)$ amounts to a $\pi$ rotation.

[46]To prove this, you can *e.g.* work out the total angular momentum operator for 3d Dirac particles or derive the Zeeman term in a non-relativistic limit [123].

where the full fermion propagator can be found analytically. The calculations are cumbersome, and we only present the result for the effective boundary action which is $\Gamma[A] = k\, S_{CS}[A]$ where the level number is,

$$k = \frac{m}{2|m|} - \frac{1}{\pi} \arctan\left(\frac{m}{M_0}\right). \tag{9.22}$$

In the physical limit $m \ll M_0$ we retain the result from 9.20 while in general, the level number is not quantized.[47] Although this calculation was done using the special profile (9.21), there is no reason to believe that the result would be qualitatively different for a more general profile.

In summary we found that there is indeed a $\nu = 1/2$ quantum Hall effect on the boundary, with the chirality set by the weak magnetic field. One should not add the contributions from the $\theta$-term in the bulk action to the one originating from the boundary fermion, but consider them as alternative derivations of the same effect.

With this said, there are still problems left. How come that we can get a $\nu = 1/2$ state starting from free fermions, and is there any way to handle the boundary theory for the case when $m$ is strictly zero, and where we would not expect any TR non-invariant terms? To address these questions we must learn about the parity anomaly, which is an example of a global anomaly. This will be the subject of Sect. 10.

Not surprisingly, one can also use anomalies to extract the thermal response of 3d TIs by coupling to gravity and using (8.8). We will not pursue this here, but refer to [79] for a good explanation with references to the original literature.

## 9.3 Stability of topological insulators – the flux insertion argument

We have already stressed that the strength of the anomaly approach to classifying SPT states is that it is topological and should thus be robust against interactions as long as they respect the symmetry. In the case of topological insulators there is, as we will now discuss, a set of arguments for the stability which predates the analysis based on anomalies but is very close in spirit and eventually fits into it, as will become clear later.

The question, originally analyzed by Kane and Mele, is whether TR symmetric interactions added to the boundary fermions can gap them leading to the decay into trivial insulators. We have already mentioned that a TR symmetric mass term can be introduced for a pair of fermions, while a single mode is anomalous and protected. A general answer beyond quadratic Hamiltonians and band theory can be given using a flux insertion argument that is closely related to the spectral flow we have have discussed earlier [61, 119, 120]. The outcome of this analysis is that the $\mathbb{Z}_2$ characterization of topological band insulators in two and three space dimensions (class AII in Fig. 18) remains valid in the presence of interactions.

### 9.3.1 The Kramers theorem in time reversal invariant systems

The action of the TR symmetry operation on states is represented by the antiunitary operator $\mathcal{T} = UK$, where $U$ is unitary and $K$ is the complex conjugation, that also maps bra into kets. On a state $|S\rangle$ of spin $S$, the square of the transformation gives,

$$\mathcal{T}^2|S\rangle = (-1)^{2S}|S\rangle = (-1)^F|S\rangle, \tag{9.23}$$

where the sign can be called spin parity or fermion parity. For fermionic states (*i.e.* $2S$ odd integer), $\mathcal{T}$ squares to minus one: in the case of Dirac fermions, using Euclidean gamma matrices $\gamma^\mu = (\sigma^1, \sigma^2, \sigma^3)$, the time-reversal operator is $\mathcal{T} = i\sigma^2 K$, that satisfies $\mathcal{T}^2 = -1$.

---

[47]The opposite limit $M_0 \ll m$ where $k \to 0$ is unphysical since the bulk gap vanish and there is no distinction between the topological and normal insulator.

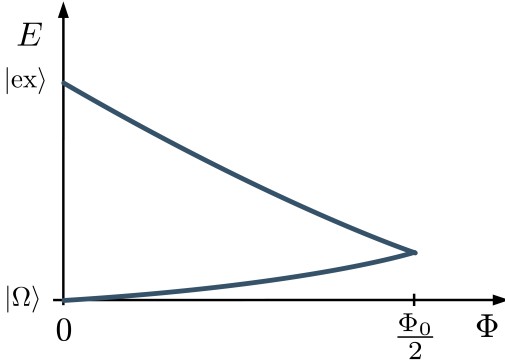

Figure 20: Energy levels of the ground state and an excited state as a function of the flux inserted, showing the Kramers degeneracy at half flux quantum.

Let us consider a system with TR invariant Hamiltonian, $\mathcal{T}\mathcal{H}\mathcal{T}^{-1} = \mathcal{H}$. Kramers theorem [126] uses antiunitarity of $\mathcal{T}$ and $\mathcal{T}^2 = -1$ to prove that the following half-integer spin states,

$$|S\rangle \qquad \text{and} \qquad |S'\rangle = \mathcal{T}|S\rangle, \qquad (2S \text{ odd}), \qquad (9.24)$$

called a Kramers pair, are orthogonal,

$$\langle S|S'\rangle = \langle S|\mathcal{T}|S\rangle = 0. \qquad (9.25)$$

Furthermore, the following relations holds for any Hermitian TR invariant operator $\mathcal{O}$, *i.e.* obeying $\mathcal{T}\mathcal{O}\mathcal{T}^{-1} = \mathcal{O}$,

$$\langle S'|\mathcal{O}|S\rangle = 0, \qquad \langle S'|\mathcal{O}|S'\rangle = \langle S|\mathcal{O}|S\rangle, \qquad (2S \text{ odd}). \qquad (9.26)$$

These imply that the degeneracy of the Kramers pair cannot be lifted by any TR invariant interaction (Kramers degeneracy).

Let us now discuss the breaking of this symmetry by the addition of a magnetic field. We consider geometries that allow flux insertions, like the two-dimensional annulus in Sect. 3.4. The Hamiltonian has the following properties in the presence of fluxes,[48]

$$\mathcal{T} H[\Phi] \mathcal{T}^{-1} = H[-\Phi], \qquad H[\Phi + \Phi_0] = H[\Phi], \qquad (9.27)$$

where the second relation follows from the Byers-Yang theorem, which implies that the spectrum returns to itself after adding one flux quantum. From these relations, it follows that there are discrete flux values, $\Phi = n\Phi_0/2$, $n \in \mathbb{Z}$, for which the system is TR invariant also in the presence of a magnetic field.

### 9.3.2 The flux insertion argument and the $\mathbb{Z}_2$ invariant

We first consider the quantum spin Hall effect in two dimensions introduced in Sect. 6.2. This is a rather idealized model of a topological insulator, because $S_z$ conservation is in general broken to $(-1)^{2S} = (-1)^F$ by spin-flip transitions. However, it is simple, and actually sufficient for the sake of the following argument that only appeals to fermion parity conservation.

In Sect. 6.2 we discussed the spectral flow in the quantum spin Hall effect and showed that the addition of a flux quantum causes positive and negative spin electrons to move in opposite

---

[48]For any compact space direction that allows flux insertion.

directions, leading to a spin Hall current with $\nu = 2$. On a the outer edge of the annulus (see Fig. 13), an edge excitation, $|1\rangle = \Theta[\Phi_0]|\Omega\rangle$, is created, where $\Theta[\Phi]$ is the flux-insertion operator and $|\Omega\rangle$ the ground state. The quantum numbers of $|1\rangle$ obtained by the flows are $\Delta Q = Q^\uparrow - Q^\downarrow = 1 - 1 = 0$ and $\Delta S_z = 1/2 - (-1/2) = 1$ (a similar excitation appears in the inner edge with opposite spin).

Let us now consider what happens at half flux insertion. This creates a neutral $S_z = 1/2$ excitation, $|1/2\rangle = \Theta[\Phi_0/2]|\Omega\rangle$, on the outer edge. Since $\Phi_0/2$ is a TR invariant flux value, we can use the Kramers theorem[49] and conclude that this state is degenerate with another state of opposite spin $|-1/2\rangle = \mathcal{T}|1/2\rangle$, as shown in Fig. 20. The two states are easily found in the massless free fermion spectrum, for example, by recalling that a half flux changes the standard antiperiodic boundary conditions into periodic ones, but of course, the result of the flux insertion is the same in interacting theories.

We can now follow the state $|-1/2\rangle$ backward to vanishing flux and show that this is an excited state $|\text{ex}\rangle$, because it is orthogonal to $|\Omega\rangle$ (see Fig. 20),

$$\langle\Omega|\text{ex}\rangle = \langle\Omega|\Theta[-\Phi_0/2]|-1/2\rangle = \langle\Omega|\Theta[\Phi_0/2]^\dagger \, \mathcal{T} \, \Theta[\Phi_0/2]|\Omega\rangle = 0 \,, \qquad (9.28)$$

where we used (9.25) for the last equality. Since the work done on the system by inserting one flux goes to zero as $O(1/R)$ in the limit of large radius $R$ of the annulus, the energy of $|\text{ex}\rangle$ at $\Phi = \Phi_0/2$ and $\Phi = 0$ is also $E_\text{ex} = O(1/R)$. We conclude that there is no gap for edge excitations in the thermodynamic limit $R \to \infty$.

Therefore, we have proven that TR symmetry keeps a single edge fermion gapless. In the case of two fermion modes at the edge, the state created by half flux has spin one and Kramers theorem cannot be invoked. Thus an even number of fermions can acquire a gap.

Note that the argument does not change in the presence of edge interactions that respects TR symmetry. Furthermore, it does not require $S_z$ to be a good quantum number, but only charge and fermion parity conservation. We conclude that the $\mathbb{Z}_2$ characterization of stable topological insulators is valid for general interacting systems [61].

The above argument extends to three-dimensional topological insulators by considering the $(2+1)$-dimensional boundary fermions, in the geometry $\mathcal{M}_3 = \mathbb{R} \times S^1 \times S^1$ and inserting half flux in each hole of the two-torus [120].

In the next chapter, we shall see that this flux insertion argument, which is a discrete variant of the Laughlin proof of the integer Hall conductivity (Sect. 3.4) nicely relates to the spectral flow of global anomalies.

## 9.4 Electromagnetic response of topological superconductors

In this section we shall derive the electromagnetic response action of 3d topological superconductors, which are states where the electrons form spin-triplet pairs. As of today, there is no confirmed experimental candidates, but the B-phase of $^3$He is an example of a triplet pairing state of neutral fermions. The time-reversal invariant superconductors have very interesting properties: They have a relativistic dispersion relation and gapless 2d Majorana fermions on their surfaces. For a recent review with a comprehensive list of references, see [128]. Because of the Meissner effect the electromagnetic fields are screened in a superconductor, so the most natural would be to study thermal response by coupling to gravity and using the index theorem. We shall not cover this topic but refer to the review [79].

One can nonetheless derive an effective theory for the electromagnetic field [129]. This was originally done in [129] using an extension to a (4+1)D theory and later in [130] using

---

[49]A local version of the Kramers theorem at each edge is necessary as described in [127]; here we follow closely their presentation.

the representative Dirac theory,

$$\mathcal{L} = \bar{\Psi}_\pm \left( i\slashed{\partial} \pm \slashed{A}\gamma^5 - \Delta e^{\pm i\theta_\pm \gamma^5} \right) \Psi_\pm . \tag{9.29}$$

which can be derived from an appropriate non-relativistic Bogoliubov-de Gennes (BdG) Hamiltonian for a triplet-paired superconductor [128]. Here $\Psi$ is a Majorana fermion, and $\theta_\pm$ are the phases of the superconducting order parameter corresponding to the two helicities. Note that in the representative theory, the electromagnetic potential couples to the axial current.

As for the TI, the requirement of TR invariance is that the angles $\theta_\pm$ can only take the values 0 or $\pi$ modulo $2\pi$. Superficially we have a description similar to that of a TI, and we would expect that integrating out the fermions gives a topological $\theta$-term, $\sim \theta \int F\tilde{F}$, in the response action. A crucial difference, however, is that in the TI case, $\theta$ is a material parameter that is determined by the band structure, while in the TSC it is the dynamical phase of the order parameter, and the statement that $\theta$ takes a certain value should be interpreted as referring to the expectation value. The full action is then found by integrating the fermions [129, 130],

$$
\begin{aligned}
S_{\text{eff}} &= \int d^4x \left[ -\frac{1}{4} F_{\mu\nu} F^{\mu\nu} - \frac{1}{192\pi^2} (\theta_+ - \theta_-) F_{\mu\nu} F_{\sigma\lambda} \right. \\
&\quad \left. + \frac{\rho_+}{2} \left( \partial_\mu \theta_+ - 2eA_\mu \right)^2 + \frac{\rho_-}{2} \left( \partial_\mu \theta_- - 2eA_\mu \right)^2 \right],
\end{aligned}
\tag{9.30}
$$

where, in addition to the topological and Maxwell terms, there is also the usual kinetic terms $\sim |\vec{D}\theta_\pm|^2$ that give the Meissner effect. Also note that in order to have a topological term, you need the difference $\theta_+ - \theta_-$ to be an odd integer times $\pi$. As compared with the TI case, the $\theta$-term has an extra factor $1/2$ because of the Majorana condition, and another factor of $1/3$ since the $A_\mu$ couples to the axial current so the relevant Feynman diagrams have three $\gamma_5$ vortices (this is explained in App. D.3.1).[50]

Because of the Meissner effect, we only expect interesting electromagnetic phenomena on boundaries and vortex lines (the superconductivity is type II). The simplest geometry to study is that of a planar interface between a trivial and topological superconductor, and here we encounter a potential contradiction – by the same argument that led to (9.5), it looks like there would be a $\nu = 1/12$ quantum Hall effect on the boundary. This, however, cannot be correct since the Majorana particles in the boundary Majorana theory are neutral [128]. To resolve this puzzle we can again to do a full 4D calculation of the two-point function using the same boundary profile (9.21) as in the TI case, and we also regularize the infrared divergences in the same way by adding a small term $m\bar{\Psi} i\gamma^5\Psi$, and thus take as a starting point the (Euclidean) Lagrangian,

$$\mathcal{L} = \bar{\Psi}_\pm \left[ \gamma^\mu \partial_\mu + M(x) + i\gamma^5 m \right] \Psi_\pm , \tag{9.31}$$

where $m$ is a chirally twisted Majorana mass. Comparing with (9.29), the gap function is $\Delta = \sqrt{M^2(x) + m^2}$ and the twist angle $\theta = \text{sign}(m) \tan^{-1}(m/M(x))$.

The explicit calculation in [130] does give a Chern-Simons term but with a level number, $k \sim m/M_0$, which, contrary to the TI result (9.22), vanish in the limit $m \to 0$. The lesson here is that one must be careful with applying results that are valid for infinite, or compactified, spaces to systems with boundaries. In the case of the $\theta$-term, the naive procedure leading to (9.5) turned out to be correct in the TI case, but not for the topological superconductor.

## 9.5 Higher dimensional topological insulators and descent relations

In Sect. 9.1 we learned that SPT states exist in different dimensions, and that the classification table in Fig. 18 shows regular patterns. Some of these regularities are due to the relations existing between anomalies in different dimensions.

---

[50]The factor of 1/3 is missing in [129].

The formulas for the topological insulators in Sect. 9.2.1 were derived for the special case of 4D, but it should be clear that, *mutatis mutandis*, everything generalizes to an arbitrary even dimensional space-time D. Actually, anomalies and effective actions in different dimensions are related by the following set of equations, called "descent relations" [55]. We start from the integrated anomaly $\Omega_D(A)$ in even $D$ (7.14) and use the Stokes theorem[51] to relate it to a Chern-Simons action $\Omega_{D-1}^{(0)}(A)$ generalizing (9.5),

$$\int_{\mathcal{M}_D} \Omega_D(A) = \int_{\partial \mathcal{M}_D} \Omega_{D-1}^{(0)}(A) \,. \tag{9.32}$$

Next, the infinitesimal gauge transformation $\delta_\lambda$ of the latter,

$$\delta_\lambda \int_{\mathcal{M}_{D-1}} \Omega_{D-1}^{(0)}(A) = \int_{\partial \mathcal{M}_{D-1}} \Omega_{D-2}^{(1)}(A, \lambda) \,, \tag{9.33}$$

defines the Wess-Zumino-Witten action $\Omega_{D-2}^{(1)}(A, \lambda)$, which is the integrated form of the (D-2)-dimensional anomaly, as described in App. D.4.

## 9.6 Symmetry enriched topological phases

We have learned that the existence of a Chern-Simons theory with half-integer level is the characteristic feature for the boundary theory of topological insulators. It turns out that this low-energy response can also be realized by massive $(2+1)$-dimensional theories that are themselves topological, as experience with quantum Hall states tell us. In such cases, called "symmetry enriched" topological phases (SET), the boundary fermions are strongly interacting and other degrees of freedom may be present. [131–133]

Several theories have been proposed that realize such close relatives of $\nu = 1/2$ Hall states; note, however, that TR symmetry is not explicitly broken by the mass gap, and the Chern-Simons response is due to the anomaly. For example, anyon excitations may appear in TR invariant pairs, having opposite charges. These states have not yet been observed experimentally, but are are not purely academic: in particular, the TR invariant version of the quantum Hall "Pfaffian state" [40], dubbed T-Pfaffian, is a candidate for realizing anyons with non-Abelian fractional statistics, a possible platform for decoherent-free quantum computations [41].

## 10 Global anomalies

The anomalies we studied so far manifested themselves as non-conserved currents, corresponding to gauge non-invariance of the partition function in the presence of background fields. These anomalies can be exactly determined from the response to weak fields calculated in a low order perturbative expansion. We showed that the anomalies in 2D signal the existence of topological states in 3D, the connection being the symmetry restoration due to the flow of a conserved quantity, charge or energy-momentum, from bulk to edge.

It is natural to ask if there is a similar connection between a topological state in 4D and an anomaly in 3D, but from what we learned so far, this does not seem likely, since the anomalies we discussed are present only in even D. It turns out, however, that there is a different class of anomalies that do appear in odd dimensions. These are the global, or nonperturbative, anomalies, that imply that in the quantum theory one cannot preserve all classical discrete symmetries and, at the same time, invariance under large, or global, gauge transformations.

---

[51]See App. E.

In this case, the bulk-boundary compensation cannot be explained in terms of flow of a current, but there is an analogous phenomenon in that symmetries are restored in the partition function for the whole system.

In this section, we first discuss the inconsistencies in a 3D massless fermionic theory when considered in isolation, and the origin of the $\mathbb{Z}_2$ parity anomaly (being also a TR anomaly). More precisely we show that the partition function cannot be regularized in a way that is both globally gauge invariant and invariant under TR. We start by rederiving the effective Chen-Simons action (9.20) using a method that can also deal with the $m = 0$ case.

Next we show that the partition function of a Chern-Simons action with half integer level number is not globally gauge invariant, and then that the same is true for a theory with an odd number of massless fermions.

These problems are then resolved by attaching the 4D topological bulk. We derive the general form of the bulk-boundary partition function in the presence of both gauge and gravitational backgrounds by using the Atiyah-Patodi-Singer index theorem. Later we also treat the case of topological superconductors that have Majorana fermions on the boundary.

## 10.1   The parity anomaly in 2+1 dimensions

### 10.1.1   Anomalous effective action

Massive and massless 3D Dirac theory is of broad relevance for condensed matter physics, since it describes both genuine two-dimensional crystals such as graphene, and boundary layers in semiconductor heterostructures that support quantum Hall liquids. In Sect. 9.2.2 we derived the Chern-Simons effective action (9.20) by studying the boundary fermionic modes of topological insulators: We referred to it as anomalous, since it breaks the rule that the Hall conductance of free fermions must be integer in units of $e^2/h$. Its violation of time reversal and parity was found to persist in the massless limit where the 3D Dirac action is symmetric, signaling a clash between classical and quantum symmetries, although the case of strict vanishing mass could not be described.

We now rederive the Chern-Simons action for the isolated 3D theory and explain the origin of the anomaly [124]. The effective action can be obtained by expanding the fermionic determinant to quadratic order in the gauge field as in (4.4), leading to,

$$S_{\text{eff}}[A] = \frac{1}{2} \int \frac{d^3k}{(2\pi)^3} A_\mu(k)\Pi_{\mu\nu}(k,m)A_\nu(-k) \;+\; O\left(A^3\right) . \tag{10.1}$$

The corresponding Feynman diagram (see Fig. 10) has a linear ultraviolet divergence in three dimensions (cf. (4.5)), that can be regularized by the Pauli-Villars method.[52] This amounts to adding another spinor field with mass $M$ to the theory[53], but with Bosonic statistics, that contributes to (10.1) by an analogous expression with a minus sign, subtracting the ultraviolet infinities. In the limit $M \to \infty$ the additional field decouples leaving a finite result for the polarization tensor $\Pi_{\mu\nu}$,

$$\Pi_{\mu\nu}(k,m) \to \Pi_{\mu\nu}^{reg}(k,m) = \lim_{M\to\infty} \left(\Pi_{\mu\nu}(k,m) - \Pi_{\mu\nu}(k,M)\right) . \tag{10.2}$$

The calculation of the regularized (Euclidean) Feynman diagram gives,

$$\begin{aligned}
\Pi_{\mu\nu}(k,m) &= \frac{1}{4\pi} k_\alpha \epsilon^{\alpha\mu\nu} \left( \frac{m}{|m|} \frac{\arctan(x)}{x} - \frac{M}{|M|} \right) \\
&\quad - \left(k^2 \delta_{\mu\nu} - k_\mu k_\nu\right) \frac{1}{8\pi|k|} \left( \frac{1}{x} - \frac{1-x^2}{x^2} \arctan(x) \right), \qquad x = \frac{|k|}{2|m|}.
\end{aligned} \tag{10.3}$$

---

[52]The Pauli-Villars regularization is explained in more detail *e.g.* in [134].
[53]Not to be confused with the 4D mass of previous section.

Note that the Pauli-Villars method (10.2) employs a gauge-symmetry breaking momentum cut-off at an intermediate stage, but this cancels out in the final result, that is (locally) gauge invariant since $k^\mu \Pi_{\mu\nu}(k, m) = 0$. The first term in the expression (10.3) is odd in momentum and breaks parity and time reversal symmetries. Technically it originates from the non-vanishing gamma matrix trace, $\text{Tr}[\gamma^\alpha \gamma^\beta \gamma^\delta] = 2i\varepsilon^{\alpha\beta\delta}$ .

In the limit $|m| \to \infty$, the expression (10.3) becomes,

$$\Pi_{\mu\nu}(k, m) = \frac{1}{4\pi} k_\alpha \epsilon^{\alpha\mu\nu} \left( \frac{m}{|m|} - \frac{M}{|M|} \right) - \left( k^2 \delta_{\mu\nu} - k_\mu k_\nu \right) \frac{1}{12\pi|m|} , \qquad (|k| \ll |m|). \quad (10.4)$$

The result depends on the choice of sign for the Pauli-Villars mass $M$. A natural choice for the isolated 3D Dirac theory is to require a vanishing effective action in the limit $m \to \infty$ which fixes $\text{sgn}(M) = \text{sgn}(m)$, but, and more importantly, for either choice, the level number of the Chern-Simons term is integer.

The result for the massless theory is found by taking $x \to \infty$ in (10.3),[54]

$$\Pi_{\mu\nu}(k, 0) = -\frac{1}{4\pi} k_\alpha \epsilon^{\alpha\mu\nu} \frac{M}{|M|} - \left( k^2 \delta_{\mu\nu} - k_\mu k_\nu \right) \frac{1}{16|k|}, \qquad (m = 0), \qquad (10.5)$$

and the first term in (10.5) gives the Chern-Simons action,

$$S_{CS}[A] = \pm \frac{i}{8\pi} \int d^3x \, \varepsilon^{\mu\nu\rho} A_\mu \partial_\nu A_\rho , \qquad (10.6)$$

so the parity and time reversal symmetries are broken at the quantum level even though the classical theory preserves these symmetries – this is the parity anomaly of $(2+1)$-dimensional fermions [122, 135].

The analysis of the isolated 3D theory can be related to the discussion of the topological insulator boundary in Sect. 9.2, where the same theory appeared as an effective low energy description for $|k| \ll M_0$, where $M_0$ is the bulk gap. In that realization, $M_0$ played the role of a 3D ultraviolet cutoff, but we also saw that a different regularization can be used in the full 4D theory [125], such that the Pauli-Villars subtraction is not needed in (10.4) for $m \neq 0$. Actually, the topological Chern-Simons theory with half-integer level is the correct low energy theory in this setting with a physically motivated breaking of TR invariance.

An equivalent regularization choice, which is available in the full 4D bulk-boundary system, is to include the 3D Pauli-Villars regulator in (10.4), but at the same time cancel it by adding a theta term (9.4) with $\theta = \pi$. This approach extends to the $m = 0$ case, and is appealing since it implies a bulk-boundary anomaly cancellation. However, more aspects need to be discussed before reaching any definite conclusion about the fate of the boundary anomaly in the full theory, and this will be clarified in Sect. 10.2.2.

### 10.1.2  Global gauge non-invariance

Above we learned, that integrating out a massless Dirac fermion gives a Chern Simons action with level number $\pm 1/2$, which violates parity. But imagine that we did not know about the free fermion origin of this effective theory, is there any fundamental reason for not accepting such an effective action? It turns out that there is such a reason – the lack of invariance of

---

[54]The first term can in fact be easily be obtained by expanding the Feynman integral to leading order in $k$ without calculating the full $\Pi_{\mu\nu}$. You might, and rightly so, worry about infrared divergences in these perturbative expressions. A more complete analysis in [124] addresses this problem by summing all one loop graphs in the presence of a constant background field. The result for the Chern-Simons term is the same as in our simplified treatment, but the non topological part of the action has a non-analytic contribution $\sim |^\star F|^{3/2}$ where $^\star F^\mu = \frac{1}{2} \epsilon^{\mu\nu\sigma} F_{\nu\sigma}$.

the theory under large gauge transformations. We now explain this other aspect of the parity anomaly .

We shall calculate the change of the effective action $\Gamma[A]$ under a general gauge transformation, and to do this we introduce an additional parameter $\tau$ varying in the interval $\tau \in [-\mathcal{T}, \mathcal{T}]$, such that the gauge field is adiabatically transformed as follows,

$$A_\mu(x, \tau): \qquad A_\mu(x, -\mathcal{T}) \to A_\mu(x, \mathcal{T}) = U^{-1}(x)\big(A_\mu(x, -\mathcal{T}) + \partial_\mu\big)U(x), \qquad (10.7)$$

where $x \equiv x^\mu$ and $U(x) = \exp(i\phi(x))$ is the large $U(1)$ gauge transformation.[55] The change in the effective action (10.1) comes entirely from the Chern-Simons term (10.6). We can consider $\tau$ as an additional coordinate and assume the corresponding field $A_4(x) = 0$, so we can write, using Minkowski metric,

$$
\begin{aligned}
S_{CS}[A(\mathcal{T})] - S_{CS}[A(-\mathcal{T})] &= \int_{-\mathcal{T}}^{\mathcal{T}} d\tau \frac{k}{4\pi} \int_{\mathcal{M}_3} d^3x \, \partial_\tau \big(\varepsilon^{4\mu\nu\rho} A_\mu \partial_\nu A_\rho\big) \\
&= \frac{k}{16\pi} \int_{\mathcal{M}_4} d^4x \, \varepsilon^{\mu\nu\rho\sigma} F_{\mu\nu} F_{\rho\sigma} \qquad (\mu, \nu, \rho, \sigma = 1, 2, 3, 4) \\
&= 2\pi k P = 2\pi k n, \qquad n \in \mathbb{Z}, \qquad (10.8)
\end{aligned}
$$

where we recalled the definition (7.15) of the instanton number $P$, that takes integer values for compact manifolds. That $\mathcal{M}_4$ is compact can be motivated by first taking $\mathcal{M}_3$ to be compact, say the three-torus $S^1 \times S^1 \times S^1$, and then $\mathcal{M}_4 = [-\mathcal{T}, \mathcal{T}] \times \mathcal{M}_3$ can also be considered compact by identifying the extrema $\tau = \pm\mathcal{T}$, where for $\mathcal{T} \to \infty$ the field strength vanishes and the $A_\mu(x, \pm\mathcal{T})$ differ by a large gauge transformation, as discussed in Sect. 2.4 for the monopole field in 2D. This compactified generalized cylinder is called a "mapping torus" [136].

From (10.8), it follows that for $k = \pm 1/2$, and $P$ odd, $\Delta\Gamma[A] = \pm\pi$, so $Z = e^{i\Gamma[A]}$ is not invariant but changes sign. This discrete non-invariance is as unacceptable as the infinitesimal change associated with anomalies of continuous symmetries. Note that this argument relates the 3D global anomaly to the integrated 4D chiral anomaly (cf. Sect. 4.4, (4.22)).

### 10.1.3 Spectral flow analysis and the Euclidean hotel

The sign change under large gauge transformations can be directly understood from the behavior of the Dirac spectrum and the related determinant which we express as,

$$Z[A] = \int d\mu \, e^{-\int dx \, \bar\psi i \slashed{D} \psi} = \det(\mathcal{D}) = \prod_i \lambda_i, \qquad (10.9)$$

where $d\mu$ is the fermionic path integral measure (5.13) and $\mathcal{D} = i\slashed{D}$ is an Hermitian operator with real eigenvalues $\lambda_i$. This would seem to imply that $Z$ is real, in accordance with TR invariance, but since the determinant requires a regularization, this conclusion is to naive.

To proceed, take two copies of the theory so $Z^2 = \prod_i \lambda_i^2$. From Sect. 9.2.2 we recall that although just adding a mass breaks TRI, by taking two copies with opposite spin, *i.e.* mass, one can couple them so as to preserve this symmetry.[56] Now, since we have a invariant mass term, we can regularize the theory using the Pauli-Villars method, as we did earlier. Therefore, the renormalized $Z^2$ is finite, and both TR and gauge invariant. This means that the only possible ambiguity in $Z$ is a sign, and if this "anomalous sign" is present, the quantum theory cannot be defined in a way that respects both symmetries.

---

[55]Note that the interpolating field $A_\mu(x, \tau)$ at any intermediate $\tau$ value is not a gauge transform of $A_\mu(x, -\mathcal{T})$.

[56]For a discussion of what kind of perturbation in a physical system that will give rise to such a mass term see *e.g.* [115].

Indeed, we will find that some $\lambda_i$ in (10.9) change from being positive to negative (or vice versa) under an adiabatic variation of the background (10.7), thus changing the sign of the determinant.[57] Note that the Dirac spectrum is not symmetric with respect to $\lambda_i \to -\lambda_i$ since there is no chiral symmetry in 3D, otherwise all sign changes would occur in pairs leaving the determinant invariant.

We now investigate the spectral flow of the Euclidean Dirac operator, in close analogy with that of the Hamiltonians in the 1d and 3d infinite hotels in Sects. 2.3 and 4.4. We shall first obtain the flow analytically and then give a qualitative explanation.

It will be useful to introduce the following auxiliary evolution equations associated to the Dirac operator [124] (using $\gamma^\mu = \sigma^\mu$ in 3D Euclidean space),

$$\partial_\tau \psi_\pm(x,\tau) = \pm i\slashed{D}(\tau)\psi_\pm(x,\tau) = \pm i\sigma^\mu\left(\partial_\mu - iA_\mu(\tau)\right)\psi_\pm(x,\tau)\,, \qquad (10.10)$$

which can be solved in an adiabatic approximation by taking $\psi_\pm = f_\pm(\tau)\phi_\lambda(x,\tau)$, where $f_\pm(\tau)$ are functions, and $\phi_\lambda(x,\tau)$ are bicomponent spinors diagonalizing the Dirac equation $i\slashed{D}(\tau)\phi_\lambda(x,\tau) = \lambda(\tau)\phi_\lambda(x,\tau)$ in the $\tau$-varying gauge background $A_\mu(\tau,x^i)$. The solutions are,

$$\partial_\tau f_\pm = \pm\lambda(\tau)f_\pm(\tau)\,, \qquad f_\pm(\tau) = f_\pm(-\mathcal{T})\exp\left(\pm\int_{-\mathcal{T}}^{\tau} d\tau'\lambda(\tau')\right)\,, \qquad (10.11)$$

and you should notice the analogy with the Jackiw-Rebbi localized surface state discussed in Sect. 9.2.1. Thus, the solutions are normalizable if and only if $\lambda(\tau)$ change sign during the adiabatic evolution: $f_+(\tau)$ is normalizable for $\lambda(\tau)$ going from positive to negative values and vice versa for $f_-(\tau)$. The next step is to recast the equations (10.10) as an eigenvalue equation for an auxiliary four-dimensional Euclidean Dirac operator,

$$\slashed{D}_4\Psi = \left[\gamma^4\partial_\tau + \gamma^\mu\left(\partial_\mu - iA_\mu\right)\right]\Psi = 0\,, \qquad (10.12)$$

where we identify $\tau \equiv x^4$, use the gauge $A_4 = 0$ and take,

$$\Psi = \begin{pmatrix}\psi_+\\\psi_-\end{pmatrix}\,, \qquad \gamma_i = \begin{pmatrix}0 & i\sigma_i\\-i\sigma_i & 0\end{pmatrix}\,, \qquad \gamma_4 = \begin{pmatrix}0 & 1\\1 & 0\end{pmatrix}\,, \qquad \gamma_5 = \begin{pmatrix}1 & 0\\0 & -1\end{pmatrix}\,, \qquad (10.13)$$

Note that in this formulation, $\psi_\pm$ are the chiral components of the 4D theory. So now we have a way to tell when one of the eigenvalues in (10.9) changes sign; it happens every time there is a zero mode for the 4D auxiliary theory, and, taking signs properly into account, this is nothing but the index $\mathrm{ind}(\slashed{D}) = n_+ - n_-$. Finally, we can use the index theorem of Sect. 7 for a flat space, stating that, $\mathrm{ind}(\slashed{D}) = P$, to infer that $Z$ changes sign between two 3D gauge configurations which interpolate by a 4D instanton. This is precisely the result obtained in the previous section for the Chern-Simons action with a half integer level, and confirms that the partition function is not invariant under large gauge transformations that are "realized" by an odd number of instantons.

An intuitive picture of this spectral flow follows from 3d infinite hotel analysis in Sect. 4.4 (see (4.17) and (4.18)). The Hamiltonian described there is block diagonal, each block being precisely the 3D Euclidean Dirac operator $\pm i\slashed{D}$ in (10.10). Thus, that energy spectrum is the double of the present Dirac spectrum,[58] obtained by reflecting it with respect to $\lambda = 0$. The time evolution of the spectrum in Sect. 4.4 corresponds to the $\tau$ variation considered here. It is apparent from Fig. 12 that the flow amounts to two levels switching sign. One of these levels

---

[57]Clearly this cannot happen if the spectrum is fully gapped, so there cannot be any problem in the massive theory.

[58]Note that the Dirac spectrum is not itself symmetric, as stressed earlier.

could correspond to an eigenvalue of $i\slashed{D}$ and the other of $-i\slashed{D}$, and in this case, $Z = \det(i\slashed{D})$ changes sign. Another possibility is that they both belong to the spectrum of either $i\slashed{D}$ or $-i\slashed{D}$ leaving the determinant unchanged. Since there is no chiral symmetry, this second option would, however, require an accidental degeneracy. The first option is indeed confirmed since the instanton number is computed to be 1 in (4.22), in agreement with the above result from the index theorem.[59]

In conclusion we have shown the sign change of the 3D partition function under large gauge transformations, both studying the effective action and the Dirac determinant. Note that we made use of an auxiliary 4D compact space and 4D Dirac equation.

**Historical note:** The first occurrence of a global anomaly was discovered by Witten in non-Abelian gauge theory with $SU(2)$ symmetry, leading to an inconsistent theory with an odd number of chiral fermions carrying the fundamental (doublet) representation of the gauge group [138]. The same phenomenon was later observed in 3D Abelian gauge theory with odd number of Dirac fermions as described here [135] [124]. It is rather remarkable that another global anomaly in non-Abelian gauge theories has been discovered only recently [139], following the renewed interest in this subject in relation to topological matter studies.

## 10.2 Global anomalies, bulk-boundary cancellation and index theorems

In this section, we find the expression for the partition function for the bulk-boundary combined system, and see how this lets us understand how TR and gauge invariance are recovered. The analysis makes use of the index theorem introduced in Sect. 7, but extended to systems with boundaries. We first consider the topological insulators, and later the topological superconductors. Our presentation is a shortened version of that in [136] and [140], and we use essentially the same notation.

### 10.2.1 The Atiyah-Patodi-Singer $\eta$ invariant

The study of the 3D Dirac determinant will not only incorporate the gauge transformation properties found earlier, but also give an explicit formula for the (regularized) partition function in terms of the geometry and topology of the manifold and the background $U(1)$ field.

We start from the path integral (10.9), coupling the fermions not only to the $U(1)$ gauge field, but also to the metric $g_{\mu\nu}$:

$$Z_\psi = \int d\mu\, e^{-\int dx \sqrt{g}\, \bar{\psi} i\slashed{D}\psi} = \det\mathcal{D} = \prod_i \lambda_i \,, \tag{10.14}$$

where $g = \det(g_{\mu\nu})$ and the Dirac operator is given in (7.11).

We now let both the gauge field and the metric depend on a parameter $\tau$, $(A_\mu(\tau), g_{\mu\nu}(\tau))$, which interpolates between an initial and final configuration as discussed above. When $\tau$ changes, so do the eigenvalues $\lambda_i$, and in particular a number of them can pass through zero.

We regularize $Z$ by adding a Pauli-Villars fermion with mass $M > 0$ and opposite statistics that cancels the ultraviolet divergences [134]. Recalling from (4.1) that the mass is imaginary in the Euclidean Dirac equation, the partition function for the regulator field is $Z_{PV} = 1/\det(\mathcal{D} + iM)$, giving the regularized expression,

$$Z_\psi \rightarrow \frac{Z_\psi}{Z_{PV}} = \prod_i \frac{\lambda_i}{\lambda_i + iM} \,. \tag{10.15}$$

For $M \rightarrow \infty$, we factor out a large positive mass-dependent factor from the determinant and are left with a product of eigenvalues times the phase $i$ or $-i$ depending on the sign of $\lambda_i$. We

---

[59]Another explicit description of this flow can be found in the appendix of Ref. [137].

get,

$$Z_\psi = |Z_\psi| \exp\left(-i\frac{\pi}{2}\sum_i \text{sign}(\lambda_i)\right) = |Z_\psi| \exp\left(-i\frac{\pi}{2}\eta\right), \tag{10.16}$$

where,

$$\eta = \lim_{s \to 0} \sum_i \text{sign}(\lambda_i)|\lambda_i|^{-s}, \tag{10.17}$$

defines the so-called Atiyah-Patodi-Singer (APS) η invariant.[60]

The partition function (10.16) is finite and well-defined but suffers from two problems. First, it is not gauge invariant for large gauge transformation: whenever an eigenvalue $\lambda_i$ changes sign under the spectral flow, η will change with ±2 giving the phase change $e^{\pm i\pi}$ in $Z_\psi$ found earlier. Secondly, $Z$ is not real, because the regularized value of η (10.17) is not integer in general. This violates TR symmetry, that acts by complex conjugation on $Z_\psi$. Note also that, had we chosen a Pauli-Villars field with negative mass $M$, we would have obtained the conjugate expression $Z_\psi = |Z_\psi| \exp(i\pi\eta/2)$. We recover the same sign ambiguity of the effective action (9.20), with the mass being that of the regulator. Note also that in the case of the square of the partition function, regularized by taking two TR invariant opposite-sign masses, the phases would cancel out, leading to the harmless $Z_\psi^2 = |Z_\psi|^2$, as commented on earlier.

### 10.2.2   Bulk-boundary anomaly cancellation in topological insulators

We have already pointed out that the effective action $\Gamma_\pi[A]$ in (9.4) is TR invariant when the integral is over a compact 4D space. In reality, topological insulators have a boundary, and $\Gamma_\pi[A]$ is not a multiple of $2\pi$: in this case, TR invariance is restored by the anomalous contribution coming from the surface fermions (see (9.5) and (9.20)).

In Euclidean quantum field theory, anomalous terms are always given by the imaginary part of the effective action [15], so we should focus on the phase of the partition function. The bulk part $Z_b$ is obtained by substituting the expression (7.15) for the index of $\slashed{D}$ in the presence of both $(A_\mu, g_{\mu\nu})$ into the effective action (9.4),

$$Z_b = |Z_b| \exp\left(\pm i\pi(\mathcal{P} - \hat{\mathcal{A}})\right), \tag{10.18}$$

where $\mathcal{P}$ is the Abelian theta term and $\hat{\mathcal{A}}$ is the corresponding gravitational expression. They differ from $P$ and $\hat{A}$ in (7.15) because they are not integrated over all spacetime due to the presence of a boundary. We now combine this with the regularized surface partition function, (10.16), to get the total expression,[61]

$$Z_{TRI} = |Z_b||Z_\psi| \exp\left[\pm i\pi\left(\mathcal{P} - \hat{\mathcal{A}} - \frac{\eta}{2}\right)\right] = |Z_{TRI}|(-1)^{\mathcal{J}}. \tag{10.19}$$

In the last term of this equation, we used the Atiyah-Patodi-Singer theorem to express the phase in the square brackets as the index of the Dirac operator $\mathcal{J}$ on non-compact 4D manifolds with specific boundary conditions.[62] Since this is an integer, the total partition function (10.19) is real and does preserve TR invariance. We have thus demonstrated the bulk-boundary cancellation of the anomaly.

---

[60]This infinite sum can be made convergent by putting a power law suppression factor; this is usually called a zeta-function regularization [1]. Alternatively one can use a Gaussian cutoff as in the path integral calculation of the axial anomaly in Sect. 5.3.

[61]The gravitational contribution is included in these equations, but is not relevant in the discussion of this section.

[62]For a relatively simple derivation of this theorem, see the appendix of Ref. [141].

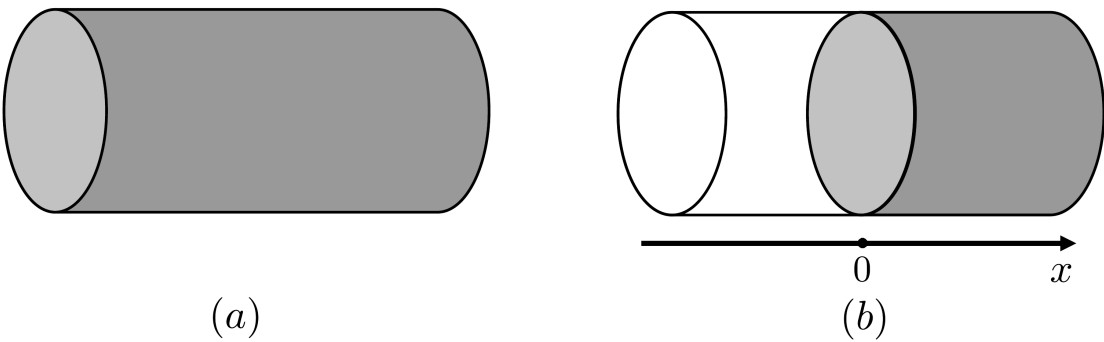

Figure 21: Simplified picture of a four-dimensional spacetime geometry with one unbounded coordinate $-\infty < x < \infty$ and three compact ones: $(a)$ The topological insulators is filling the whole space; $(b)$ It is located in $x > 0$ and separated from the empty space in $x < 0$ by a boundary at $x = 0$.

This is a both remarkable and comforting result. Comforting, since, absent external magnetic fields, or magnetic impurities, the microscopic physics is TR invariant, and we have shown, in great generality, that the effective theory, properly defined, preserve this symmetry. That the proof goes via the use of deep mathematical results is the remarkable part.

The lack of invariance of $Z_\psi$ under large gauge transformations is also resolved within the full partition function $Z_{TRI}$: this point needs some further explanation which is based on a thought experiment laid out in Sect. 2.A.5 of [136].

Let us consider the geometry in Fig. 21 which is that of a generalized cylinder $\mathcal{M}_4 = \mathbb{R} \times \mathcal{M}_3$, where $x \in \mathbb{R}$ is a spatial coordinate and $\mathcal{M}_3$ is compact, *e.g.* a three-torus (only one circle of $\mathcal{M}_3$ is shown in the figure). Panel $(a)$ show a space-filling topological insulator, where we identify the boundaries at $x = \pm\infty$. This is an example of the so-called mapping torus discussed already in Sect. 10.1.2, which is a compact Euclidean spacetime. We can then, just as above, use the Atiyah-Singer index $J = \text{ind}(\slashed{D}_4)$ for the 4D Dirac operator on compact manifolds (7.9) discussed in Sect. 7 to obtain the partition function,

$$Z_b = \exp\left(\pm i\left(P - \hat{A}\right)\right) = (-1)^J . \tag{10.20}$$

This quantity is clearly gauge invariant and its sign measures the parity of the number of instantons inside the system.

Panel $(b)$ in Fig. 21 shows the physical situation of having a topological insulator filling the $x > 0$ space, and supporting massless fermions at the $x = 0$ boundary. The other boundary at $x = +\infty$ is disregarded assuming that the gauge field vanish there. The partition function is now $Z_{TRI}$ in (10.19), expressed in terms of the APS index $\mathcal{J}$. This is rather different from the index $J$, since it involves the terms $\mathcal{P}$ and $\hat{A}$, which are integrated in the half spacetime $x > 0$ only; in addition, there is the boundary part $\eta$ that plays the role of a generalized Chern-Simons term.

Contrary to $J$, the quantity $\mathcal{J}$ is not a topological invariant, because its value depends on where the instanton background is "mostly" located. Upon tuning the gauge field, this can be squeezed in the region $x < 0$, so that $\mathcal{J}$ vanishes; conversely, it can be moved to $x > 0$ inside the topological insulator, leading to $\mathcal{J} = 1$ which amounts to a sign change of $Z_{TRI}$. The transition between these two discrete values of $\mathcal{J}$ can be understood as as follows: The $\eta$ function depends smoothly on the gauge field but it has $\pm 2$ discontinuities when one Dirac eigenvalue switch sign, as discussed earlier. The $\mathcal{P}$ and $\hat{A}$ terms actually cancel the smooth

part of $\eta/2$, and one finds [42, 141],

$$\frac{1}{2}\text{disc}(\eta[A(s)])_{s_1} = \mathcal{J}, \tag{10.21}$$

where $s$ is a parameter moving the instanton along $x$, and $s_1$ the value at which an eigenvalue vanishes and the discontinuity takes place.[63] Note that the partition function $Z_{TRI}$ is nonetheless continuous because the determinant in $|Z_\psi|$ vanishes at the point $s_1$ due to the zero mode precisely at the point where the eigenvalue changes sign.

We now discuss this jump of $\eta$, *i.e.* of $\mathcal{J}$, at the level of effective actions. Using earlier results in 10.1.3, we know that the eigenvalue sign switch occurs when the $3D$ gauge background at $x = 0$ changes by a large gauge transformation. Let us verify that this indeed takes place under the displacement of the instanton from $x < 0$ to $x > 0$.

We start by observing that the instanton number $P$ is a total derivative (cf. (9.5)) and thus non-vanishing only when its boundary Chern-Simons terms do not cancel each other. In Fig. 21(a), the value $P = 1$ is due to the backgrounds $A_\mu(\pm)$ at $x = \pm\infty$ being different by a large gauge transformation $U$ (see (10.8)),

$$\pi P = \frac{1}{2}\left(S_{CS}[A(+)] - S_{CS}[A(-)]\right) = \frac{1}{2}\left(S_{CS}[A(+)] - S_{CS}[A^U(+)]\right) = \pi, \tag{10.22}$$

where $A_\mu^U$ is the gauge transformed of $A_\mu$. As in the case of the monopole in Sect. 2.4, the demand that the gauge transformation $U$ is well defined, implies that $P$ is quantized as an integer.

In Fig. 21 (b), the quantity $\mathcal{P}$ is defined as the integral over the region $x > 0$ and its complement $\overline{\mathcal{P}}$ for $x < 0$, so that $P = \mathcal{P} + \overline{\mathcal{P}}$. They can be expressed in terms of the Chern-Simons action at $x = 0$, $S_{CS}[A(0)]$ as,

$$\pi\mathcal{P} = \frac{1}{2}\left(S_{CS}[A(+)] - S_{CS}[A(0)]\right), \qquad \pi\overline{\mathcal{P}} = \frac{1}{2}\left(S_{CS}[A(0)] - S_{CS}[A(-)]\right). \tag{10.23}$$

When the instanton is placed at $x \ll 0$, one has $\mathcal{P} = 0$ and $\overline{\mathcal{P}} = 1$; thus $A_\mu(0) = A_\mu(+)$, that vanishes by our earlier assumption. As the instanton moves to the $x > 0$ region, the gauge field at $x = 0$ grows and eventually accumulate a large gauge transformation, $A_\mu(0) \to A_\mu(-) = A_\mu^U(+)$, leading to $\mathcal{P} = 1$ and a corresponding spectral flow.

Therefore, the large gauge transformation at $x = 0$ is responsible for the non-vanishing instanton number $\mathcal{P}$ inside the topological insulator. We conclude that what was seen as an unphysical gauge-dependent sign of the 3D boundary partition function $Z_\psi$ becomes a gauge-invariant physical effect for $Z_{TRI}$ describing the full system (10.19): This partition function is just counting the number of instantons inside the TI modulo two. This completes the proof of gauge invariance for this expression.

You may have noted that during the instanton displacement, $\mathcal{P}$ changes smoothly while $\mathcal{J}$ jumps. We can elaborate on this, by observing that (10.19) and (10.21) imply (disregarding $\hat{A}(R)$),

$$\pi\mathcal{P} - \frac{1}{2}S_{CS} = \pi\mathcal{J}. \tag{10.24}$$

Said differently, the Chern-Simons action $S_{CS}[A]$ is the smooth part of the function $\eta[A]$ [136].

---

[63]In case of several singularities they should be summed over in (10.21).

## 10.3 Topological superconductors and Majorana boundary modes

In a superconductor the gauge field acquires a gap, and magnetic fields are expelled due to the Meissner effect. Also, as explained in 9.4, the boundary modes are Majorana fermions and carry no $U(1)$ charge [98].[64] We now analyze their TR anomaly and the related bulk-boundary cancellation, paralleling the case of topological insulators. The charge neutrality means that the global anomaly is manifested by symmetry violations related to certain background geometries.

A Majorana fermion should be thought of as a real fermion, meaning that $\psi$ and $\bar\psi$ are not independent two-component spinors, but (in Euclidean space) are related by $\bar\psi_\alpha = \psi^\beta \varepsilon_{\beta\alpha}$, where $\alpha$ and $\beta$ are spinor indices. The Euclidean action for a 3D Majorana fermion is,

$$S_{Mj} = \int dx\, \bar\psi i \slashed{D} \psi = \psi^\gamma \varepsilon_{\gamma\alpha} i \slashed{D}^\alpha_\beta \psi^\beta = \psi^\gamma \mathrm{D}_{\gamma\beta} \psi^\beta\,. \tag{10.25}$$

We used the notation of [136] where spinor indices are raised and lowered using the $\varepsilon$ tensor.[65]

Before continuing, we briefly recall how to evaluate path integrals for real, as opposed to complex, fermions. A general quadratic action is $S = \sum_{\alpha\beta} \psi^\alpha \Lambda_{\alpha\beta} \psi^\beta$ where $\psi^\alpha$ are real Grassmann numbers, and $\Lambda_{\alpha\beta}$ an antisymmetric $2n \times 2n$ matrix. Using the rules of Grassmann variable integration, we get:

$$\int \prod_{\alpha=1}^{2n} d\psi_\alpha \exp(S_{MJ}) = \frac{1}{2^n n!} \sum_{\{i_k,j_k\}} \varepsilon_{i_1 j_1 i_2 j_2 \dots i_n j_n} \Lambda_{i_1 j_1} \Lambda_{i_2 j_2} \cdots \Lambda_{i_n j_n} \equiv \mathrm{Pf}(\Lambda)\,, \tag{10.26}$$

where the last equality defines the Pfaffian of the antisymmetric matrix $\Lambda$. To understand the logic of this formula you should notice that in the expansion of the exponential you get a string of elements like $\psi^1 \Lambda_{13} \psi^3 \psi^2 \Lambda_{27} \psi^7 \dots$, where each $\psi^i$ should occur once, and only once, for the integral not to vanish. Thus we get all possible such strings, and by a few examples you can convince yourself that the normalization and the sign in (10.26) is correct.[66] Note the identity $\mathrm{Pf}(\Lambda)^2 = \det \Lambda$ for antisymmetric matrices $\Lambda$.

With our conventions for the Euclidean gamma matrices $\gamma^\mu = (\sigma^1, \sigma^2, \sigma^3)$, the time-reversal operator is $\mathcal{T} = i\sigma^2 K$, where $K$ denotes complex conjugation. It satisfies $\mathcal{T}^2 = -1$, and $[\mathcal{T}, i\slashed{D}] = 0$. Thus the spectrum of $i\slashed{D}$ is doubly degenerate with eigenvalues $\lambda_n$ and pairs of eigenfunctions which we combine in the spinor $(\chi_\lambda, \mathcal{T}\chi_\lambda)^{\mathrm{T}}$. Acting on these spinors, the operator $\mathrm{D}_{\alpha\beta} = \varepsilon_{\alpha\gamma} \mathcal{D}^\gamma_\beta$ becomes,

$$\mathrm{D} = \begin{pmatrix} 0 & -\lambda \\ \lambda & 0 \end{pmatrix}\,, \tag{10.27}$$

and the Pfaffian for this single mode is just $\lambda$. Including the contributions from all modes we get,

$$\mathrm{Pf}(\mathrm{D}) = \widetilde{\prod_i} \lambda_i\,, \tag{10.28}$$

where the product $\widetilde{\prod}$ is defined to only pick one $\lambda_i$ from each degenerate pair.

---

[64] They do have a conserved fermion parity $(-1)^F$.

[65] In this reference you can find more details about the properties of Majorana fermions in Euclidean space-time, and a recent thorough discussion of this topic is [142].

[66] You might have encountered the Pfaffian in the theory of superconductivity, where it gives the BCS wave function for a fixed number of particles. There the Pfaffian, which is a sum of strings of pairs, $A_{13}A_{27}\dots$, tells that the wave function is built from Cooper pairs.

In the case of Dirac fermions in Sect. 10.1.3, we showed how a large gauge transformation resulted in a sign change of the determinant. One can similarly study how Pf(D) changes under a large diffeomorphism (here there is no gauge field), that interpolate between two different metrics $g_{\mu\nu}(0) \to g_{\mu\nu}(1)$, as some parameter $s$ is changed from 0 to 1. The spectral flow gives Pf(D) $\to (-1)^{I/2}$ Pf(D), where $I$ is the index taking all modes, $i.e.$ both $\gamma_i$ and $-\gamma_i$ into account. $I$ is related to the index for Dirac fermions on a compact manifold introduced in (10.20) by $I = 2J$ when both are defined, and is always even in 4D. Its value is the gravitational analog of the instanton number and is given by the corresponding theta term $\hat{A}(R)$ for compact manifolds with $\theta = \pi$.

It turns out that $I = 0$ for the geometry of the mapping torus relevant for the spectral flow discussed in Sect. 10.1.3; thus, there is no anomaly for large gauge transformations (diffeomorphism invariance for gravity) of the 3D Majorana partition function [136].

Nonetheless, there is a violation of TR invariance as in the Dirac case. To se this, we again regularize Pf(D) in a gauge-invariant way using the Pauli-Villars subtraction method to get,

$$Z_\psi^{Mj} \to \frac{Z_\psi^{Mj}}{Z_{PV}} = \text{Pf(D)} = \widetilde{\prod} \frac{\lambda_i}{\lambda_i \pm iM} \, . \tag{10.29}$$

We can now take over the expression (10.16) if we define $\eta$ using the sum $\widetilde{\sum}$ which again picks only one eigenvalue in each degenerate pair. We have,

$$Z_\psi^{Mj} = |Z_\psi^{Mj}| \exp\left(\mp i \frac{\pi}{2} \widetilde{\sum}_i \text{sign}(\lambda_i)\right) = |Z_\psi^{Mj}| \exp\left(\mp i \frac{\pi}{2}\eta\right) = |Z_\psi^{Mj}| \exp\left(\mp i \frac{\pi}{4}\eta\right), \tag{10.30}$$

where the last expression is in terms of the quantity $\eta = 2\eta$, whose sum runs over all eigenvalues. Since $\eta$ is a non-trivial complex number in general, the boundary partition function violates TR invariance. How this function depends on the geometry of the manifold is subtle, and will be discussed in the next section.

We now proceed to show that, as in the TI case, the TR invariance is restored when the boundary is combined with the bulk. For this we need the Majorana version of the bulk partition function $Z_b$ (10.18). There is an extra factor of a half due to integration of Majorana instead of Dirac fermions, and no gauge coupling, so we get,

$$Z_b^{Mj} = |Z_b^{Mj}| \exp\left(\pm i \frac{\pi}{2}\hat{A}(R)\right) . \tag{10.31}$$

Again combining boundary and bulk, we rely on the APS index theorem for a manifold with boundary to obtain,

$$Z_{TRI}^{Mj} = |Z_\psi^{Mj}||Z_b^{Mj}| \exp\left[\pm i \frac{\pi}{2}\left(\hat{A}(R) - \frac{1}{2}\eta\right)\right] = |Z_{TRI}^{Mj}|(-1)^{\mathcal{I}/2} . \tag{10.32}$$

where $\mathcal{I}$ is the index of the Dirac operator on a manifold with boundary, taking all modes into account. Since $\mathcal{I}$ is integer, the expression (10.32) shows the cancellation between bulk and boundary TR breaking terms, leading again to a real total partition function.

We already mentioned that $\mathcal{I} = 0$ for the mapping torus, but $\mathcal{I}/2$ can be odd in general on a compact 4D "spin manifold". We elaborate on this in Sect. 10.5.

## 10.4 Anomalies and interacting topological superconductors

At the end of Sect. 9.1 we emphasized that the classification of SPT states based on anomalies is robust, and that there are cases where it differs from the original 10-fold classification based of free fermions. In this section we will expand on this with some concrete examples.

Our first example is the 4D TR invariant topological superconductor that we just discussed in some detail. It belongs to class DIII (see Fig. 18) and is characterized by an integer topological number $\nu \in \mathbb{Z}$, meaning that there can be any number of boundary gapless Majorana modes. Rather surprisingly, it was shown in [143] that a symmetry preserving interaction can gap 16 of them, thus causing a reduction $\mathbb{Z} \rightarrow \mathbb{Z}_{16}$. An analogous breakdown of the 10-fold classification was found earlier in the 2D TSC, the so called Kitaev chain (class BDI in Fig. 18), where Fidkowski and Kitaev showed that the $\mathbb{Z}$ index is reduced to $\mathbb{Z}_8$ by a suitable interaction [144, 145].

Next we show how the global gravitational anomaly of Majorana fermions in 3D, discussed in the previous section, also leads to the $Z_{16}$ symmetry, and we conclude by discussing the interacting Kitaev model.

### 10.4.1 $\mathbb{Z}_{16}$ – the anomaly perspective

Fermions in 3D do not have any notion of chirality, but nevertheless are of two kinds corresponding to non-equivalent representations of the gamma matrices. In the massive case, we already saw that fermions with positive and negative masses cannot be mapped into each other. In the massless case this is reflected in that there are two different ways the TR transformation can act [140]. In terms of Majorana fermions $\psi_\pm$, these are $\mathcal{T}\psi_+ = \gamma_0 \psi_+$ and $\mathcal{T}\psi_- = -\gamma_0 \psi_-$. Since these can be gapped in pairs, the number of massless modes at the boundary of TSC is given by $\nu = n_+ - n_- \in \mathbb{Z}$.

In the interacting case, however, 16 fermions can be gapped while respecting TR symmetry. This suggests the existence of a an anomaly for $\nu \bmod 16$, that manifests itself as an obstruction to define a TR symmetric 3D partition function. We already saw that the phase of the boundary $Z_\psi^{Mj}$ (10.30) is in general a complex number, but to find out what values are actually possible, we have to discuss a class of background geometries that describe unorientable manifolds.

A well-known unorientable surface is the Möbius strip, where one cannot in a consistent way define the orientation of a curve, or equivalently the direction of the normal vector.[67]. Since time reversal transformations change the orientation, a TR symmetric theory does not bother about orientation and can thus be defined also on unorientable manifolds.

Using a Pauli-Villars regulator for the bulk (gapped) 4D TSC, similar to the one used for the 3D Majorana theory in (10.29), one can compute the partition function as $Z_{4D} = \exp(-i\nu\eta/2)$, where the $\eta$-function is defined on the compact 4D manifold. In the special case of an orientable manifold this result equals $(-1)^{\mathcal{I}/2}$, but in general one must use the formula involving $\eta/2$.[68]

Now mathematicians have proved that for the general, not necessarily orientable, four-dimensional manifolds of interest, $\exp(i\eta/2)$ is a $16^{th}$ root of unity, which implies that the partition function is periodic in $\nu$ with a period 16. Or put differently, the topological number characterizing a 4D TSC is an element of $\mathbb{Z}_{16}$ rather than of $\mathbb{Z}$.

It is fair to ask whether this analysis has any relevance for physics. After all, there are not a lot of unorientable materials around, at lest not in 3d. But before rejecting the whole exercise as purely academic, one should recall that similar excursions into physically non-realizable manifolds can be quite fruitful. A well-known example is the integer QHE on a torus. A real QH sample always have boundaries since our labs have no supply of the magnetic monopoles that would be needed to get a net flux out of a closed 2d surface. However, there is no fundamental physical principle that prevent us from having magnetic monopoles, so a microscopic theory of electrons on a torus enclosing monopoles is perfectly consistent. This implies that any correct low energy theory should also be such that it *can* be consistently defined on a torus, and this in

---

[67]The generalization of the concept to higher dimensional spaces is less intuitive [42].
[68]For a complete discussion, see sections IV.C, IV.E and App. A.3 in [136].

turn implies that the level number $k$ in front of the CS action (3.41) must be integer, meaning that the Hall conductance is quantized. In the same vein, there are no principles that forbid us from studying TIs and TSCs on unorientable surfaces. In the case of the TIs which are described by $\mathbb{Z}_2$ topological numbers, this does not add anything, but for TSCs we might hope that these rather technical properties of the partition function may be captured by some observables in more mundane geometries. A final remark is that another interesting physical relation exists between non-trivial geometries and properties of the entanglement spectrum [146].

### 10.4.2   $\mathbb{Z} \to \mathbb{Z}_8$ in the interacting TR invariant Kitaev chain

The above argument is certainly rather formal, so to illustrate that it does capture an important aspect of the physics of TSCs, we now outline the concrete 1d example given in Ref. [144] for how the symmetry class reduction $\mathbb{Z} \to \mathbb{Z}_8$ is caused by interactions. We use the notation from this paper that you should consult for further details.

First we give a short summary of the relevant properties of the Kitaev chain, following Kitaev's original paper [147]. The model is a 1d BdG lattice superconductor with a Hamiltonian,

$$H_K = \sum_{j=1}^{N} \left[ -w(a_j^\dagger a_{j+1} + a_{j+1}^\dagger a_j) - \mu\left(a_j^\dagger a_j - \frac{1}{2}\right) + \Delta a_j a_{j+1} + \Delta^\star a_{j+1}^\dagger a_j^\dagger \right], \qquad (10.33)$$

where $\Delta$ is the complex (proximity) superconducting order parameter, and for simplicity we shall take it to be real. The $N$ annihilation and creation operators satisfy $\{a_i, a_j^\dagger\} = \delta_{ij}$, and can be rewritten in terms of $2N$ real Majorana operators,

$$c_{2j-1} = a_j + a_j^\dagger, \qquad\qquad c_{2j} = -i(a_j - a_j^\dagger), \qquad (10.34)$$

satisfying $c_i = c_i^\dagger$ and $\{c_i, c_j\} = 2\delta_{ijj}$. Two points in the parameter space $(w, \mu, \Delta)$ are particularly easy to analyze. The first is $\Delta = w = 0$, $\mu > 0$, where $H_K$ becomes,

$$H_{\text{triv}} = -\mu \sum_{j=1}^{N}\left(a_j^\dagger a_j - \frac{1}{2}\right) = -\frac{\mu}{2}\sum_{j=1}^{N} i c_{2j-1} c_{2j}, \qquad (10.35)$$

so the ground state $|\Psi_0\rangle_{\text{triv}}$ is clearly the one where all the sites are occupied, as illustrated in the top panel of Fig. 22. The second parameter set is $\Delta = w > 0$ and $\mu = 0$ which gives the Hamiltonian,

$$H_{\text{top}} = 2w \sum_{j=1}^{N-1}\left(\tilde{a}_j^\dagger \tilde{a}_j - \frac{1}{2}\right) = w \sum_{j=1}^{N-1} i c_{2j} c_{2j+1}, \qquad (10.36)$$

were a new set of fermion operators, with standard properties, are defined by:

$$\tilde{a}_j = \frac{1}{2}(c_{2j} + i c_{2j+1}), \qquad\qquad \tilde{a}_j^\dagger = \frac{1}{2}(c_{2j} - i c_{2j+1}), \qquad\qquad j = 1, \dots, N-1. \quad (10.37)$$

The corresponding ground state, $|\Psi_0\rangle_{\text{top}}$, is annihilated by the $N-1$ fermion operators $\tilde{a}_j$ but note that $H_{\text{top}}$ does not depend on the two operators $c_1$ and $c_{2N}$, which can be combined to the single fermion $b = (c_1 + i c_{2N})/2$. Thus the ground state is doubly degenerate, since the fermion state corresponding to $b$ can be filled or empty. In the bottom panel of Fig. 22, we illustrate this ground state with the two Majoranas edge modes. Such a degenerate edge state is characteristic of a topological state, as seen earlier, so we shall refer to the Hamiltonians $H_{\text{triv}}$ and $H_{\text{top}}$ and the related ground states as trivial and topological, respectively. In Kitaev's

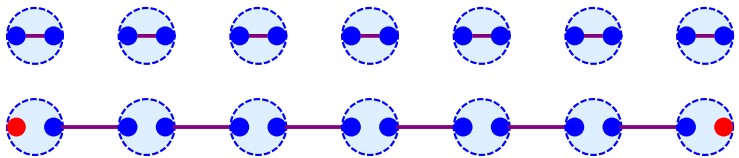

Figure 22: Top panel: Ground state of $H_{\text{triv}}$. Bottom panel: Ground state of $H_{\text{top}}$. The small blue circles are the bounded Majorana fermions and the small red circles the free ones.

paper it is shown that the two phases correspond to extended regions of parameter space, these two special points being the easiest to analyze.

Next we generalize the Kitaev chain in two ways. First we shall extend to a multicomponent model with $n$ identical copies of the TR invariant chain, and label the fermions by $c_i^\alpha$, where $\alpha = 1, \ldots, n$. Notice that if we have two copies in the topological state, by adding the terms $ic_1^1 c_1^2$ and $ic_{2N}^1 c_{2N}^2$ to the Hamiltonian, we can gap out the end Majorana modes and get a trivial model. The same can be done for any even number of copies. Thus, we have obtained the $\mathbb{Z}_2$ classification of 1d models in absence of TR symmetry (class D in Fig. 18).

The second generalization introduce TR invariance, by demanding that $\mathcal{T} c_j \mathcal{T}^{-1} = c_j$ on even sites and $\mathcal{T} c_j \mathcal{T}^{-1} = -c_j$ on odd sites. Thus both $ic_{2j-1} c_{2j}$ and $ic_{2j} c_{2j+1}$ are TR invariant, and in general this is true for any quadratic term $M_{ij} ic_j c_k$ where the sites $j$ and $k$ have different parity and $M_{ij}$ is real and symmetric.

For this TR invariant model we cannot gap out the edge Majoranas by adding terms like $ic_1^1 c_1^2$, since they are not TR invariant, but Fidkowski and Kitaev showed that this is possible if one allows for other interactions. Their strategy was to consider the Hamiltonian,

$$H_{\text{ext}} = H_{\text{triv}} + H_{\text{top}} + W , \tag{10.38}$$

where $W$ is interaction term that has no matrix elements between $|\Psi_0\rangle_{\text{triv}}$ and $|\Psi_0\rangle_{\text{top}}$.

Again take $\Delta = w$ and consider the following adiabatic transformation in the $(\mu, w)$ space, $(\mu, 0) \rightarrow (0, 0) \rightarrow (0, w)$, where $w$ is kept zero in the first part and $\mu$ in the second part. Absent $W$, the system remain gapped during this full evolution, except at $w = \mu = 0$ where the Hamiltonian is identically zero and the gap vanish, signaling a change of phase. The crucial insight in Ref. [144] is now to consider interaction terms of the form $c_i^{\alpha_1} c_i^{\alpha_2} c_i^{\alpha_3} c_i^{\alpha_4}$, which are Hermitian and cannot mix the states $|\Psi_0\rangle_{triv}$ and $|\Psi_0\rangle_{top}$. The latter follows since $c_i$, by (10.34) and (10.37), is a linear combination of the fermion creation and annihilation operators, and the two ground states are eigenstates of the corresponding number operators.

The question is now if we can choose $W$ so that the gap is maintained under the evolution through the point $w = \mu = 0$. If so, for some number $N$ of copies, the states with $n = 1$ and $n = N$ are topologically the same. The obvious first try is $n = 4$, where there is only one possible choice for the interaction on each site, namely $W_i = A_i c_i^1 c_i^2 c_i^3 c_i^4$. It is not hard to show that this term alone has a doubly degenerate state independent of the sign of $A_i$, meaning that at the point $w = \mu = 0$, where the kinetic term vanish, the system has a large ground state degeneracy and no gap. By a group theoretical analysis, it was proven in [144] that there is a way to continuously connect $|\Psi_0\rangle_{\text{triv}}$ and $|\Psi_0\rangle_{\text{top}}$ without closing a gap for $n = 8$, meaning that the free fermion $\mathbb{Z}$ classification is reduced to $\mathbb{Z}_8$. We shall not here try to explain the analysis, which is rather involved, but Fidkowski and Kitaev also give the following heuristic argument in favor of $\mathbb{Z}_8$: Divide the system in two parts with 4 chains each, and add in both groups the above interaction that will give doubly degenerate ground states. This system can now be viewed as two spin 1/2 particles, so if we could construct an interaction that favors the singlet, and has a gap to the triplet, we would expect the gap to remain even when the kinetic

term vanish. For how to construct such a term we refer to Ref. [144], which also contains a derivation of the result using conformal field theory. In a later paper [145], the same authors used matrix product state techniques to extended their work, both by a more detailed analysis of the Majorana chain, and by providing a classification of all interacting 1d fermion systems.

## 10.5 Global anomalies and the classification of SPT states

In previous sections, we found a general pattern by which quantum field theory anomalies are related to topological phases; a boundary (gapless) quantum field theory displays an anomaly and a bulk massive theory "adsorbs" it, restoring the symmetry. For systems in even d, such as quantum Hall states, the anomalies are perturbative; in odd d, as for topological insulators, they are global and the bulk action is a purely topological theta term.

In a sense, the former, perturbative, anomalies are "simple", since the same pattern repeats in any even dimension, involving generalized bulk Chern-Simons theories and non-conserved boundary currents, as discussed in Sect. 9.5. The study of global anomalies in odd d is more difficult, since one would need a complete list of topological actions (theta terms) and topological partition functions, including those related to discrete symmetries and the associated gauge fields.[69] Actually, also in even d, a further global anomaly could be included, since it does not hamper the perturbative pattern.

Thus, the classification of interacting topological phases can be recast as the purely mathematical problem of finding topological quantum field theories (TQFT) in any dimension. A complete solution of this problem is still missing, but different approaches have been proposed for attacking it. In this section we present some basic elements of the approach based on "cobordism invariance" [148], which recovers known topological theories in low dimension and encompasses results obtained by other methods.

We shall discuss TQFTs defined on compact manifolds, assuming that the bulk-boundary correspondence can be understood afterwards. Topological theories are characterized by partition functions that are just phases, $Z_{\mathcal{M}}[A, g] = \exp(i\phi_{\mathcal{M}}[A, g])$, as in the earlier examples (10.20) and (10.31) of TIs and TSCs respectively. These partition functions obey a composition rule when a manifold $\mathcal{M}$ is obtained by gluing together two other manifolds $\mathcal{M}_1$ and $\mathcal{M}_2$. This "connected sum" is obtained by cutting a hole in each of them, and then gluing them along the created boundaries [149]. The composition rule is,

$$Z_{\mathcal{M}} = Z_{\mathcal{M}_1} Z_{\mathcal{M}_2}, \tag{10.39}$$

corresponding to just summing the respective effective actions. Actually, we already used this property for the instanton number in Sect. 10.2.2, when we wrote $P = \mathcal{P} + \hat{\mathcal{P}}$ (cf. (10.22) and (10.23)).

Note that (10.39) may not hold in the presence of ground state degeneracy as in the fractional quantum Hall effect. The topological theories we are considering here have a single ground state on a compact manifold, and they are referred to as "invertible" TQFTs. Here you should recall the discussion of TO contra SPT phases in Sect. 9.1; only the latter will be discussed here.

Partition functions of TQFTs depend on the manifold $\mathcal{M}$ and its properties, but possess some invariances in the sense that they are the same for all manifolds that can be related by certain rules, which thus define equivalence classes. We can characterize manifolds by the following properties:

---

[69]Gauge symmetry can be generalized to discrete groups by using finite transformations as for ordinary gauge theories on a lattice.

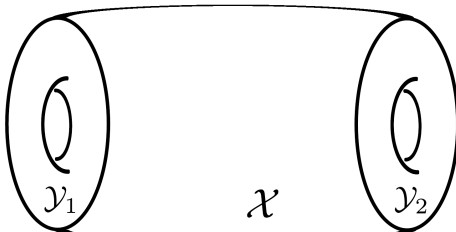

Figure 23: Two-dimensional compact manifolds $\mathcal{Y}_1$ and $\mathcal{Y}_1$ are boundaries of the 3D manifold $\mathcal{X}$.

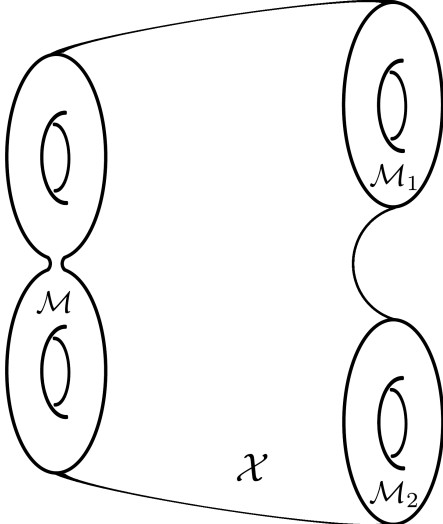

Figure 24: The 3D manifold $\mathcal{X}$ with boundaries $\mathcal{M}_1$, $\mathcal{M}_2$ and their connected sum $\mathcal{M}$.

- In cases of fermionic systems, the possibility to unambiguously define spinors, *i.e.* to consistently assign the minus sign associated to $2\pi$ rotations. This is called a "spin structure" and the manifold a "spin manifold".

- For TR invariant theories, we should also consider unorientable manifolds, that possess generalized spin structures called Pin$^{\pm}$ structures, depending on how TR invariance is realized: $\mathcal{T}^2 = 1$ (Pin$^-$ structure) and $\mathcal{T}^2 = (-1)^F$ (Pin$^+$ structure).

- A smooth gravitational background should be defined and, for charged states, a $U(1)$ gauge field.

Manifolds with these properties can be divided into equivalence classes according to the so-called cobordism invariance: two D-dimensional compact manifolds $\mathcal{Y}_1$ and $\mathcal{Y}_2$ are bordism equivalent, or cobordant, if there exist a (D+1)-dimensional manifold $\mathcal{X}$ that joins them, *i.e.* the two manifolds are disjoint boundaries of it (see Fig. 23). The structures listed above, defined on $\mathcal{Y}_1$ and $\mathcal{Y}_2$, are supposed to extend smoothly to $\mathcal{X}$. These equivalence classes are elements of the so-called bordism groups that are known in the mathematical literature.

TQFT partition functions are naturally invariant under cobordism, in the sense that $Z_{\mathcal{Y}_1} = Z_{\mathcal{Y}_2}$ if $\mathcal{Y}_1$ and $\mathcal{Y}_2$ are bordism equivalent. This property was first conjectured in [148] and then proved in [149].

| class\$d$ | T | C | S | 0 | 1 | 2 | 3 | 4 | 5 | 6 | 7 |
|---|---|---|---|---|---|---|---|---|---|---|---|
| A | 0 | 0 | 0 | $\mathbb{Z}$ | 0 | $\mathbb{Z}$ | 0 | $\mathbb{Z}$ | 0 | $\mathbb{Z}$ | 0 |
| | | | | $\mathbb{Z}_2^2$ | $\mathbb{Z}_2^2$ | $\mathbb{Z} \times \mathbb{Z}_8$ | 0 | 0 | 0 | $\mathbb{Z}^2 \times \mathbb{Z}_{16}$ | 0 |
| BDI | + | + | 1 | $\mathbb{Z}_2$ | $\mathbb{Z}$ | 0 | 0 | 0 | $2\mathbb{Z}$ | 0 | $\mathbb{Z}_2$ |
| | | | | $\mathbb{Z}^2$ | $\mathbb{Z}_8$ | 0 | 0 | 0 | $\mathbb{Z}_{16}$ | 0 | $\mathbb{Z}_2^2$ |
| D | 0 | + | 0 | $\mathbb{Z}_2$ | $\mathbb{Z}_2$ | $\mathbb{Z}$ | 0 | 0 | 0 | $2\mathbb{Z}$ | 0 |
| | | | | $\mathbb{Z}_2$ | $\mathbb{Z}_2$ | $\mathbb{Z}$ | 0 | 0 | 0 | $\mathbb{Z}^2$ | 0 |
| DIII | − | + | 1 | 0 | $\mathbb{Z}_2$ | $\mathbb{Z}_2$ | $\mathbb{Z}$ | 0 | 0 | 0 | $2\mathbb{Z}$ |
| | | | | 0 | $\mathbb{Z}_2$ | $\mathbb{Z}_2$ | $\mathbb{Z}_{16}$ | 0 | 0 | 0 | $\mathbb{Z}_2 \times \mathbb{Z}_{32}$ |

Figure 25: Topological classes of the free-fermion tenfold classification (black, see Fig. 18) compared with the results of cobordism invariance (red) for topological theories of interacting fermions.

The above equivalence can be restated in the following way. Let $\mathcal{W}$ be a $D$-dimensional compact manifold that is the boundary of a $(D + 1)$-dimensional manifold $\mathcal{X}$, to which geometric structures extend without obstructions. The partition function $Z_{\mathcal{W}}$ is then said to be a cobordism invariant if it is trivial, *i.e.* if

$$Z_{\mathcal{W}} = 1, \qquad \text{for} \qquad \mathcal{W} = \partial \mathcal{X}, \qquad \text{(cobordism invariance)}. \qquad (10.40)$$

In the previous case of two boundaries, we take $\mathcal{W} = \mathcal{Y}_1 \cup \mathcal{Y}_2$ (disjoint union), and apply (10.40) to obtain $Z_{\mathcal{Y}_1}^{-1} Z_{\mathcal{Y}_2} = 1$, as anticipated. The inversion relating one boundary to the other can be understood as due to orientation, but it is actually more general. It follows from unitarity of the TQFT, as one can understand by thinking of the extra dimension in Fig. 23 as time and the whole picture as a kind of quantum mechanical amplitude [136].

Let us verify that the relation (10.40) is fulfilled by the partition function of a a 3d topological insulator. For this, consider the instanton number on the 4D manifold $\mathcal{W}$, smoothly extend the gauge field to the 5D space $\mathcal{X}$, and use Stokes theorem to find,

$$\int_{\mathcal{W}} F^2 = \int_{\mathcal{X}} d F^2 = 0. \qquad (10.41)$$

The last result holds since the anomaly density $\Omega_4 \propto F^2$ is a closed form. Thus, for this simple TQFT, cobordism invariance is just a consequence of Stokes theorem.

The composition rule of partition functions (10.39) also follows from cobordism invariance: Consider the $(D+1)$-dimensional manifold $\mathcal{X}$ in the form of a "pair of pants", with boundary $\mathcal{Y}$ given by three disconnected parts, $\mathcal{M}_1$, $\mathcal{M}_2$ and their connected sum $\mathcal{M}$ (see Fig. 24). In this case, taking into account the inversion rule, (10.40) implies, $Z_{\mathcal{Y}} = Z_{\mathcal{M}}^{-1} Z_{\mathcal{M}_1} Z_{\mathcal{M}_2} = 1$, which is equivalent to (10.39).

With this background, it is hopefully clear that the identification of topological partition functions as cobordism invariants allows one to associate them to elements of the bordism groups of manifolds, and classify them by borrowing results from the mathematical literature [148].

The results of this analysis are summarized in Fig. 25: Four of the ten classes of the free fermion classification in Fig. 18 are displayed, with the corresponding groups written in black, indicating the possible numbers of massless boundary modes. Below each class, the bordism

groups of manifolds are given in red. Very importantly, we see that symmetry reductions $\mathbb{Z} \to \mathbb{Z}_8$ for the Kitaev chain in 1d (class BDI), and $\mathbb{Z} \to \mathbb{Z}_{16}$ for 3d topological insulators (class DIII), are captured by the cobordism analysis, which clearly is powerful enough to correctly classify the interacting phases. It also shows that there are symmetry enhancements, indicating topological phases of interacting fermions completely different from band systems. Also note that the periodicity in space dimension modulo eight is lost.

A virtue of the cobordism approach is that it reproduces results obtained by other methods, such as "group cohomology" [150], that will not be described here. A drawback is that the cobordism method is not constructive, in that it does not give explicit expressions form of the partition function/effective action for the various bordism classes.[70] The group cohomology method, on the other hand, is constructive and can analyze topological phases in the presence of additional, non-generic symmetries. We hope that this brief description has given at least a flavor of this mathematically advanced subject, which reveals deep relations between geometrical properties and physical phenomena.

# 11 Chern-Simons theory and the framing anomaly

In our study of quantum Hall liquids we encountered Chern-Simons terms in two related, but distinct, appearances. In (3.42) as an effective action $S[A]$ for the electromagnetic field that encodes the Hall response, and in (3.40) as the kinetic term in the effective action (3.40) for the hydrodynamical field $a_\mu$ that parameterizes the current. We also showed how $S[A]$ could be obtained by integrating out the hydrodynamic field, but deferred the calculation of the gravitational part of the full Wen-Zee action (8.23) to this section. It is worth noticing that there is yet another condensed matter context where dynamical Chern-Simons terms appear, namely as actions for the "statistical gauge fields" that are employed in microscopic mean field theories for the quantum Hall effect based on composite bosons [151] or composite fermions [12]. In these theories, both electrons and quasiparticles are charge-flux composite, and during braiding they will pick up Aharonov-Bohm type phases that endows them with the correct integer or fractional statistics.

We now proceed to the derivation of gravitational Chern-Simons action by evaluating the path-integral of the hydrodynamic field. We consider the action (3.40), which has no explicit coupling to the metric, and write it as

$$S_{CS}[A,\tilde{A}] = -\frac{k}{4\pi} \int d^3x \, \epsilon^{\mu\nu\sigma} A_\mu \partial_\nu A_\sigma + \frac{k}{2\pi} \int d^3x \, \epsilon^{\mu\nu\sigma} A_\mu \partial_\nu \tilde{A}_\sigma \qquad (11.1)$$

$$\equiv -\frac{k}{4\pi} \int A \wedge dA + \frac{k}{2\pi} \int A \wedge d\tilde{A}.$$

To simplify comparison with [152], which we will follow rather closely, we shall in this section change notation and use $A_\mu$ for the hydrodynamic Chern-Simons field, $-\frac{e}{k}\tilde{A}_\mu$ for the background electromagnetic field, and also write the wedge products explicitly in the differential forms (cf. App. E). Furthermore, we will also assume some basic knowledge of the Faddeev-Popov method to quantize gauge theories [1].

From (11.1) follows the classical equations of motion $dA = d\tilde{A}$, and the effective action $S[\tilde{A}]$ is obtained by integrating out the $A_\mu$ field. We define $Q$ as the quantum field fluctuating around the mean field,

$$Q_\mu = A_\mu - \tilde{A}_\mu \,, \qquad (11.2)$$

---

[70]Of course, known low-dimensional theories have been shown to fit the scheme [148].

and write the path integral as,

$$Z[\tilde{A}] = e^{iS[\tilde{A}]} = \exp\left(i\frac{k}{4\pi}\int \tilde{A}\wedge d\tilde{A}\right)\int \mathcal{D}Q \exp\left(i\frac{k}{4\pi}\int Q\wedge dQ\right). \qquad (11.3)$$

Naively, we might think that the problem is now solved, since the path integral is independent of $A$, but there is more to the story. For the theory to be topological, it should not depend on the metric. Superficially this looks obvious, since the Lagrangian (11.1) does not, but for this to be true also for (11.3), we must establish that the path integral is also independent of the metric. And here comes the problem: there is no way to regularize the theory without introducing a metric. This is intuitively clear, since any regularization involves changing the short distance behavior, and this does not make any sense without a way to measure lengths.

For a TQFT to be interesting, it is not enough to have a metric independent partition function, one must also be able to calculate correlation functions of metric independent operators. In gauge theories these operators are the Wilson loops,

$$W[C] = \text{Tr}\left[\mathcal{P}\exp\left(i\oint dx^i A_i\right)\right], \qquad (11.4)$$

where $\mathcal{P}$ denotes path ordering, and the average is defined with respect to the action in (11.1).

Below we show that both the partition function, and the correlation functions $\langle W[C_1]\ldots W[C_n]\rangle$ do depend on the metric, but that this dependence can be compensated for by adding a gravitational Chern-Simons counter term to the action. Still, however, there remains a subtle dependence on the geometry that is referred to as the "framing anomaly".

How to properly quantize the path integral in (11.3) was shown in the classic paper by Witten [152], and we shall now summarize some of its key results. Another useful technical reference is the later extension in [153] and also chapter 5 in the lecture notes [154].

Following these papers, we generalize (11.3) to the case of a compact simple non-Abelian gauge group, *e.g.* $SU(N)$, and use the corresponding Chern-Simons action,

$$\begin{aligned} S[A] = kI[A] &= \frac{k}{4\pi}\int_{\mathcal{M}}\text{Tr}\left(A\wedge dA + \frac{2}{3}A\wedge A\right) \qquad (11.5)\\ &= kI[\tilde{A}] + \frac{k}{4\pi}\int_{\mathcal{M}}\text{Tr}(Q\wedge DQ)\,, \end{aligned}$$

where, as in Sect. 7, $A$, $Q$ and $\tilde{A}$ are Lie algebra valued one forms, $A = \tilde{A} + Q$ and $\mathcal{M}$ is a closed manifold. $D$ is the covariant derivative with respect to the $\tilde{A}$ background which is an extremal point of the CS action. In [155], a homotopy argument was used to show that for the gauge groups $SU(N)$, the level number $k$ must be an integer in order for (11.5) to be well defined.

In [152] the CS theory is quantized in two ways. The most general method uses a canonical framework, but here we will describe the other method based on a saddle point evaluation of the path integral (11.5). This calculation, which is valid for large $k$, amounts to calculating a functional determinant somewhat similarly to the ones we have already encountered. At the end we comment on the relationship between the two methods.

As is always the case for gauge theories, the naive path integral (11.5) includes gauge copies, so to be properly defined one must pick a gauge for the quantum field $Q$. A natural choice is the covariant background field gauge $D_\mu Q^\mu = [\partial_\mu + \tilde{A}_\mu, g^{\mu\nu}Q_\nu] = 0$ which explicitly depends on the metric. The essence of the Faddeev-Popov method is to integrate the gauge condition over the gauge volume, and include the appropriate Jacobian as a path integral over ghost fields. The appropriate Lagrangian that implement this is,

$$\mathcal{L}_{\text{gf}} = \frac{k}{4\pi}\int_{\mathcal{M}}\text{Tr}\left(\phi D_\mu Q^\mu + \bar{c}D_\mu D^\mu c\right), \qquad (11.6)$$

where $\phi$ is a scalar multiplier field implementing the gauge condition, and $c$ and $\bar{c}$ are the Grassmann valued ghost fields. With this we can write the path integral as,

$$Z[\tilde{A}] = e^{ikI[\tilde{A}]} \int \mathcal{D}Q \, \mathcal{D}\phi \, \mathcal{D}c \, \mathcal{D}\bar{c} \, \exp\left(\frac{ik}{4\pi} \int_{\mathcal{M}} \text{Tr}\left(Q \wedge DQ + \phi \star D \star Q + \bar{c} \star D \star Dc\right)\right)$$
$$\equiv e^{ikI[\tilde{A}]} \mu[\tilde{A}], \tag{11.7}$$

where $\phi$ is a 3-form and $\star$ is the (metric dependent) Hodge operator that maps $k$ forms to $3 - k$ forms.[71]

It is a non-trivial mathematical task to evaluate this path integral, and we shall just outline how to proceed and then state the result. The first observation is that (after a rescaling with $\sqrt{k/2\pi}$, detailed in [153]), the bosonic part of the action can be expressed as,

$$(H \cdot L\_H) = \text{Tr}\left(Q \wedge DQ + \phi \star D \star Q\right), \tag{11.8}$$

where the inner product is defined by,

$$(\phi \cdot \phi') = \int_{\mathcal{M}} \text{Tr}\left(\phi \wedge \star \phi'\right), \tag{11.9}$$

$H = (Q, \phi)$, and $L\_ = \star D + D\star$ is a self adjoint operator which exists in 3 dimensions and maps odd forms into odd forms. The expression (11.8) is thus rewritten $(H \cdot L\_H) = Q \cdot L\_Q + Q \cdot L\_\phi + \phi \cdot L\_Q + \phi \cdot L\_\phi$. In the fermionic part we recognize $\star D \star D = \triangle$ as the Laplacian. We can now formally write the result of the Gaussian integrations in terms of determinants which can be computed by the methods discussed in Sects.10.2.1, *i.e.* by finding the eigenvalues of $\triangle$ and $L\_$ and regularizing the relevant products. Doing so one gets,

$$\mu[\tilde{A}] = \frac{\text{Det} \triangle}{\sqrt{\text{Det} L\_}} = \sqrt{\tau} \, \exp\left(-\frac{i\pi}{4} \eta(\tilde{A})\right). \tag{11.10}$$

The quantity $\tau$, appearing in the absolute value, is the Ray-Singer analytic torsion which is a topological invariant. The proof of the second identity requires mathematics beyond this exposition, but it can be traced from the references given above. The phase is proportional to $\eta(\tilde{A}) \equiv \eta(L\_(\tilde{A}))$ where the $\eta$ invariant was introduced in Sect. 10.2.1. Using the APS index theorem and specializing to the gauge group $SU(N)$ gives the relation,

$$\eta(\tilde{A}) = \eta(0) + \frac{2N}{\pi} I[\tilde{A}], \tag{11.11}$$

so the second term in the RHS will just renormalize the coefficient of $I[\tilde{A}]$ in (11.7). The first term is more troublesome; it is not topologically invariant, but potentially depends on the metric. For $A = 0$ the theory is just $d_G$ copies of the purely gravitational theory where $d_G$ is the dimension of the gauge group $G$, so writing $\eta(0) = d_G \eta_g$ the interesting phase factor is $\exp(id_G \pi \eta_g/4)$. One can now again refer to the APS index theorem that in this case states that the combination,

$$-\frac{1}{4}\eta_g + \frac{1}{24\pi} I[g], \tag{11.12}$$

is a topological invariant. Here $I[g]$ is the gravitational Chern-Simons term (8.9),

$$I[g] = \frac{1}{4\pi} \int_M \text{Tr}\left(\omega d\omega + \frac{2}{3}\omega \wedge \omega \wedge \omega\right), \tag{11.13}$$

---

[71]See App. E for the properties of differential forms used in this analysis.

which is normalized to be $2\pi\mathbb{Z}$ for a closed manifold. It is now clear how one can make $Z[\tilde{A}]$ independent of the metric, namely by the substitution,

$$\exp\left(-id_G\frac{\pi}{4}\eta_g\right) \rightarrow \exp\left(id_G\left(-\frac{\pi}{4}\eta_g + \frac{1}{24}\frac{1}{2\pi}I[g]\right)\right), \tag{11.14}$$

which amounts to adding a metric dependent counter term.

It turns out, however, that although the above expression does not depend on the metric, $I[g]$ will depend on another property of the manifold, namely how one globally define the coordinate system. Mathematically this is related to how one "trivializes" the tangent bundle, or equivalently the frame bundle. In fact, $I[g]$ will change by a multiple of $2\pi$ under a global "twist" just as the $U(1)$ CS term can be changed by a non-trivial global gauge transformation as discussed in Sect. 10.1.3. This remaining dependence on the geometry,

$$Z \rightarrow Z\exp\left(is\frac{2\pi d_G}{24}\right), \tag{11.15}$$

under a unit $s$ twist, is the framing anomaly. Note, however, that for a given framing, all correlation functions are topological invariants.[72]

As already mentioned, the above analysis was based on a saddle point approximation, which in general is appropriate only for large $k$. For the special case of an Abelian theory, *i.e.* a collection of $U(1)$ fields it is exact since the integral is Gaussian. In Ref. [152] Witten also treats the general non-Abelian case using canonical quantization. The result of that analysis is a formula very similar to (11.15) but with $d_G$ replaced by the central charge $c(k) = kd_G/(k+2N)$ of the two dimensional current algebra of the group $G = SU(N)$ at level $k$.

The gravitational Chern-Simons term (11.13) obtained by functional integration of the hydrodynamic field completes the derivation of the Wen-Zee effective action of quantum Hall fluids discussed in Sect. 8.4 (see (8.23)). Its coupling constant clearly becomes the conformal central charge in the case of interacting fermions. In this way the effective theory directly generalizes to general Abelian quantum Hall liquids, but also to non-Abelian ones, (for details on this you should read [96]).

# 12 To conclude – some important points

We conclude by stressing a number of important points made in these lectures.

1. Anomalies occur when a quantum field theory cannot be quantized so that all the classical symmetries are preserved.

2. In a perturbative anomaly, a local gauge invariance is violated, and the related current is not locally conserved.

3. In a global anomaly, the partition function $Z$ is not invariant under non trivial global gauge transformations, and typically some discrete symmetries, such as parity and time reversal, are violated.

4. The perturbative anomalies occur in even space-time dimensions, and can be calculated by various methods, the most important being:

   - Analysis of spectral flow

---

[72]For this to be true the Wilson loops (11.4) must be modified by a "framing" which turns it into a ribbon, as discussed in [152].

- Path integrals – the Fujikawa method
- Perturbation theory

5. Currents coupled to dynamical gauge fields, be they fundamental or emergent, cannot be anomalous, since this makes the theory inconsistent. This puts important constraints on theories, an important example being the particle content of the Standard Model of fundamental interactions.

6. Anomalies have a Janus face. In the infrared they appear as symmetry violating spectral flows, and in the ultraviolet as symmetry violating regularization procedures.

7. An anomaly in a fundamental theory, will also be present in a corresponding effective low energy theory, that can be either bosonic or fermionic, and these "'t Hooft anomaly matching conditions" put important restrictions on model building.

8. A theory that can be formulated on a lattice is anomaly free. An important consequence is the phenomenon of fermion doubling where anomalies cancel between different Fermi points (or, more generally, Fermi surfaces). Cancellations also occur between bulk and boundary and between different boundaries of the system.

9. In condensed matter physics, anomalies provide a powerful tool for classifying symmetry protected topological states of matter by relating them to topological invariant exact quantities that are not changed by interactions.

10. Even in condensed matter physics, it is often useful to consider theories on curved manifolds. The related gravitational anomalies can be used to classify states of neutral fermions, as well as provide a tool to exactly compute thermal response functions.

## Acknowledgements

AC would like to thank Alexander Abanov, Domenico Seminara and Paul Wiegmann for interesting discussions on anomalies, and Nigel Cooper for reading and commenting the draft. THH thanks Jens Bardarson and Parameswaran Nair for valuable comments on parts of the manuscript. AC thanks Nordita, Stockholm, for hospitality and the G. Galilei Institute for Theoretical Physics, Arcetri, for running the 2021 program "Topological properties of gauge theories and their applications to high-energy and condensed-matter physics". RA and THH thank Cristiane Morais Smith and the Institute for Theoretical Physics at Utrecht University, where a shorter version of these lectures was given. RA acknowledge funding from CAPES, NWO, CNPq, and the Knut and Alice Wallenberg Foundation.

## A   Notations and conventions

Coordinates in Minkowski space are denoted by $x = (t, \vec{r}) = (x^0, x^i)$ and their metric is $\eta_{\mu\nu} = diag(1, -1, \ldots, -1)$. Most of the time we will put $c = \hbar = 1$, and use $\sigma^i$, $i = 1, 2, 3$ for the Pauli matrices. Space-time dimension is denoted by D, and space dimension by d, so D=d+1. For Euclidean spacetime we write *e.g.* 4D, while for Minkowski (3+1)d etc.

The $(1 + 1)$-dimensional gamma matrices are given by,

$$\gamma^0 = \begin{pmatrix} 0 & -i \\ i & 0 \end{pmatrix} = \sigma^2 \, , \qquad \gamma^1 = \begin{pmatrix} 0 & i \\ i & 0 \end{pmatrix} = i\sigma^1 \, , \qquad \text{(A.1)}$$

and satisfy $\{\gamma^\mu, \gamma^\nu\} = 2\eta^{\mu\nu}$, so $\gamma^3 = \gamma^0 \gamma^1 = \sigma^3$. In Euclidean space, the metric is $\delta^{\mu\nu}$ and the gamma matrices are Hermitian, satisfying, $\{\gamma_E^\mu, \gamma_E^\nu\} = 2\delta^{\mu\nu}$, with $\mu, \nu = 1, \dots, D$.

In 2D Euclidean space, we take $\gamma_E^\mu = \sigma^\mu$, $\mu = 1, 2$ and we introduce (omitting the subscript $E$ whenever there is no risk of confusion),

$$\gamma^3 = -i\gamma^1 \gamma^2 = \sigma^3 \qquad \text{implying} \qquad \text{Tr}[\gamma^3 \gamma^\mu \gamma^\nu] = 2i\epsilon^{\mu\nu}, \tag{A.2}$$

and we also have the useful relation,

$$\gamma^3 \gamma^\mu = i\epsilon^{\mu\nu} \gamma_\nu, \tag{A.3}$$

In 3D space, we similarly take $\gamma^\mu = \sigma^\mu$, $\mu = 1, 2, 3$.

In (3+1) dimensions, we consider the following Weyl representation of gamma matrices,

$$\gamma_i = \begin{pmatrix} 0 & i\sigma_i \\ -i\sigma_i & 0 \end{pmatrix}, \qquad \gamma_4 = \begin{pmatrix} 0 & 1 \\ 1 & 0 \end{pmatrix}, \qquad \gamma_5 = \begin{pmatrix} 1 & 0 \\ 0 & -1 \end{pmatrix}. \tag{A.4}$$

The matrix $\gamma^5$ obeys,

$$\gamma^5 = -\gamma^1 \gamma^2 \gamma^3 \gamma^4, \qquad (\gamma^5)^2 = 1 \quad \text{and} \quad \text{Tr}\left[\gamma^5 \gamma^\mu \gamma^\nu \gamma^\sigma \gamma^\lambda\right] = -4\epsilon^{\mu\nu\sigma\lambda}, \tag{A.5}$$

and anticommutes with all the other $\gamma^\mu$. It defines spinors of positive or negative chirality depending on its eigenvalue $\pm 1$. In odd dimension this is not such matrix and we cannot define chirality of spinors and a axial current.

The Euclidean Dirac action is written in terms of the Hermitian operator, $i\slashed{D}_E = \gamma_E^\mu(i\partial_\mu + A_\mu)$, as follows,

$$S_E = \int dx_E \, \bar{\psi}(i\slashed{D}_E + im)\psi, \tag{A.6}$$

where $dx_E$ is the D-dimensional measure.

We use $A$ to denote the vector gauge potentials and $A_5$ to denote the axial gauge potentials. All quantities with a index 5 are the axial ones, regardless of the spacetime dimension, and with a $+$ ($-$) index are the ones with positive (negative) chirality. Using this convention, the axial current will be denoted by $J_5$ and a gauge field by $A_-$ for negative chirality, for example.

Currents are obtained by functional derivative of the effective action with respect to the backgrounds, $J^\mu = \delta S[A, A_5]/\delta A_\mu$ and similarly for $J_5^\mu$. The relation between the Euclidean $(E)$ and Minkowskian $(M)$ actions, $S^{(M)} = iS^{(E)}$, gives a quick way to map the corresponding currents, e.g. $J_\mu^{(M)} = iJ_\mu^{(E)}$.

# B  Multicomponent Chern-Simons theory of edge excitations

In this appendix we show how the chiral edge theory can be extended to edges with several modes of both chiralities. To motivate the construction we start with a brief introduction to the physics of the quantum Hall states in the so called Jain series.

## B.1  Effective bulk theory of Jain states

Besides the Laughlin series $\nu = 1/(2q + 1)$, the most stable plateaus are observed at filling fractions:

$$\nu = \frac{n}{2pn \pm 1}, \qquad n, p = 1, 2, \dots. \tag{B.1}$$

The theory for these Hall states has been developed by Jain, generalizing Laughlin approach [12]. The basic physical picture involves a correspondence between integer and fractional Hall effects. Let us analyze the inverse filling fraction, that gives the number of flux quanta per particle:

$$\frac{1}{\nu} = 2q \pm \frac{1}{n} = 2q \pm \frac{1}{\nu^*} \, . \tag{B.2}$$

These values correspond to integer fillings $\nu^* = n$, if $2q$ fluxes per particles could be removed, leaving a residual magnetic field $B^* = B - 2q\Phi_0\rho_o$. The Jain theory assumes that the $2q$ fluxes disappear because they are bound to each electron forming a composite state, called composite fermion. Therefore, the experimentally observed filling fractions (3.81) can be interpreted as integer Hall states of composite fermions, that completely fill effective Landau levels for the magnetic field $B^*$. Note that the fermionic statistics is not hampered by adding an even number of fluxes.

Flux attachment is precisely implemented in wave-function constructions for the ground state and excitations, and finds remarkable numerical and experimental confirmations [12]. For example, low-energy local excitations have been observed with the expected properties of the composite fermion, *i.e.* weakly interacting and moving as if they were in the field $B^*$. Furthermore, effective field theories have been formulated that implement Jain flux attachment [25].

Jain as well as Laughlin theories have not yet been derived from a study of the microscopic dynamics, but several analytic and numerical results confirm their predictions. The extensive phenomenology based on the composite fermion picture let us argue that the structure of several branches is correct [12].

## B.2 Multi-component Chern-Simons theory

The Chern-Simons effective theory introduced in Sect. 3.5 cannot describe the Jain states, because its Hall conductivities $\nu = 1/p$ do not match the values (B.1) for $n > 1$. Since these states can been interpreted as integer Hall states of composite fermions, $\nu^* = n$, it is rather natural to generalize the effective low-energy theory by including several field components [26].

The current of matter excitations is written (3.39):

$$j^\mu = \frac{1}{2\pi}\varepsilon^{\mu\nu\rho}\partial_\nu \sum_{i=1}^{n} t_i a_\rho^i, \tag{B.3}$$

where the Abelian gauge fields $a_\mu^i$ are labeled by $i = 1, \ldots, n$ and $t_i$ are their charges. This kind of generalization is rather natural for extending the earlier analysis to several non-interacting filled Landau levels, *i.e.* $\nu = n$, but can also describe fractional cases as follows. The multi-component Abelian Chern-Simons action is:

$$S_{\text{eff}} = \frac{1}{4\pi}\int d^3x \, K_{ij}\varepsilon^{\mu\nu\rho}a_\mu^i\partial_\nu a_\rho^j + \frac{1}{2\pi}\int d^3x \, t_i\varepsilon^{\mu\nu\rho}A_\mu\partial_\nu a_\rho^i. \tag{B.4}$$

In this expression, the summation over the components $i, j = 1, \ldots, n$ is assumed, and they are mixed through the symmetric matrix $K_{ij}$ of couplings. It turns out that both $t_i$ and $K_{ij}$ should take integer values. Upon repeating the analysis of the equations of motion done in Sect. 3.5.2, one finds the values of the Hall conductivity and the spectrum of charge and statistics of anyons in this theory (excitations are specified by a set of integers $l_i$, $i = 1, \ldots, n$):

$$\nu = t_i K_{ij}^{-1} t_j, \qquad Q = t_i K_{ij}^{-1} l_j, \qquad \frac{\theta}{\pi} = l_i K_{ij}^{-1} l_j. \tag{B.5}$$

The multicomponent theory is able to describe many filling fractions and spectra. The Jain series (B.1) are obtained by choosing the form of $K_{ij}$ and $t_i$ that is completely symmetric over the permutation of the $n$ fluids, namely [26]:

$$t_i = 1, \qquad K_{ii} = 2q \pm 1, \qquad K_{ij} = 2q, \qquad i \neq j = 1, \dots, n. \tag{B.6}$$

Working out the inverse matrix $K^{-1}$, and inserting it in the first of (B.5), one verifies that the filling fractions (B.1) are obtained.

The corresponding theory at the edge is given by the multicomponent generalization of the chiral boson described in Sect. 3.6. The action involves the scalar fields $\varphi^i$, $i = 1, \dots, n$, and reads (cf. (3.48)):

$$S_{\text{edge}} = -\frac{1}{4\pi} \int_C d^2x \, K_{ij} \partial_x \varphi^i \partial_t \varphi^j + V_{ij} \partial_x \varphi^i \partial_x \varphi^j, \tag{B.7}$$

where $K$ is the earlier matrix and $V$ is a positive-definite symmetric matrix of velocities, not encoding any universal feature.

Let us remark that the simplest one-component effective theory (bulk Chern-Simons and edge boson) uniquely identifies the Laughlin Hall states. However, its multicomponent generalization predicts far more Hall states than the Jain series, upon changing the integer parameters in the symmetric matrix $K$ [156]. These states are not observed experimentally. A solution to this puzzle has been found in the works [11] [157] [158], by relying on a property of the two-dimensional incompressible fluids made by Laughlin and Jain states. These are characterized by constant bulk density $\rho_0$ and fixed particle number $N = \rho \mathcal{A}$, thus constant area $\mathcal{A}$. It follows that excitations are generated by deformations of the fluid that keep the area constant: these are the area-preserving transformations of spatial coordinates, obeying the so-called $W_\infty$ symmetry. It was shown that the Jain states are characterized by special 'minimal' realizations of this symmetry [159], involving reduced multiplicities, as *e.g.* a single electron excitation instead of $n$ of them as given by the spectrum (B.5), (B.6). These minimal theories are associated to non-trivial representations of the $W_\infty$ algebra.

## C  Detailed derivation of 3D infinite hotel

The free Dirac Hamiltonian is $H = \vec{\alpha} \cdot \vec{p}$, so in the presence of a potential it becomes $H = \vec{\alpha} \cdot (\vec{p} + \vec{A})$. Using the $\gamma$ matrices defined in (4.17), we get:

$$\alpha_i \equiv \gamma^0 \gamma^i = \begin{pmatrix} \sigma_i & 0 \\ 0 & -\sigma_i \end{pmatrix}, \tag{C.1}$$

so that the Hamiltonian in matrix form becomes

$$H = \vec{\alpha} \cdot (\vec{p} + \vec{A}) = \begin{pmatrix} \vec{\sigma} \cdot (\vec{p} + \vec{A}) & 0 \\ 0 & -\vec{\sigma} \cdot (\vec{p} + \vec{A}) \end{pmatrix}. \tag{C.2}$$

In the Weyl representation, each block describes a specific helicity of the wavefunctions, such we can write the Hamiltonian as the direct sum $H_+ \oplus H_-$, such that the upper block of the Hamiltonian is related to the states with positive helicity ($\vec{\sigma} \cdot \hat{p} = 1$) and the lower one with negative helicity ($\vec{\sigma} \cdot \hat{p} = -1$).

To diagonalize this Hamiltonian, we square it and use the relations $\sigma_i \sigma_j = \delta_{ij} + \iota \epsilon^{ijk} \sigma_k$, to get,

$$\sigma_i \sigma_j (p_i + A_i)(p_j + A_j) = p^2 + 2\vec{A} \cdot \vec{p} + A^2 + (\vec{\nabla} \times \vec{A})_z \sigma_z = p_z^2 + (\vec{q} + \vec{A})^2 - B\sigma_z, \tag{C.3}$$

where we used that,

$$\epsilon^{ijk}\left(A_i p_j + p_i A_j\right) = \epsilon^{ijk}\left[A_i, p_j\right] = i\epsilon^{ijk}\partial_j A_i = -iB^k = iB\,\delta^{k,z}. \tag{C.4}$$

Thus,

$$H^2 = \left[p_z^2 + \left(\vec{q} + \vec{A}\right)^2 - B\sigma_z\right] \otimes 1_2, \tag{C.5}$$

where $1_2$ is the $2 \times 2$ identity . This Hamiltonian is a sum of three different terms:

$\mathbf{p_z^2}$ Momentum along $z$ is a good quantum number. This part of the wavefunction will be plane waves along $z$, $e^{\pm i p_z z}$.

$(\mathbf{q}+\mathbf{A})^2$ This will lead to Landau levels, as discussed in Sect. 3.1. The energies will be given by the harmonic oscillator levels $(2n+1)B$, $n \in \mathbb{N}$, and the wavefunctions are highly degenerated.

$\mathbf{B}\sigma_\mathbf{z}$ The energies will be $Bs$, where $s = \pm 1$ is the eigenvalue of $\sigma_z$. The wavefunctions are spinors $\chi_1 = \begin{pmatrix} 1 \\ 0 \end{pmatrix}$ and $\chi_{-1} = \begin{pmatrix} 0 \\ 1 \end{pmatrix}$.

Then the eigenvalues of $H^2$ are given by $\left(E_{p_z,n,s}\right)^2 = p_z^2 + (2n+1)B + sB$, so that the energies becomes $E_{p_z,n,s} = \pm\sqrt{p_z^2 + (2n+1)B + sB}$ and they are shown for some values of $n$ in Fig. 12. For $n = 0$ and $s = -1$, the dispersion relation is linear.

# D   Covariant and consistent mixed anomalies

We have at several occasions referred to the freedom of adding local counterterms to the effective action, and in this appendix we extend the discussion about this. It turns out that it is also possible to add local terms to the currents themselves, independently of the action, according to definite rules that have geometric and physical meaning. There are in fact two kinds of currents, called "consistent" and "covariant", and correspondingly two types of anomalies. The consistent/covariant distinction is important in the case of mixed Abelian anomalies, and more prominently for non-Abelian gauge anomalies.

Recall that we used three methods to calculate anomalies: the spectral flow and path integral calculations gave us the currents written as $J^\mu$ and $J_5^\mu$ in the main text – we shall see that they are gauge invariant. In the non-Abelian case they transform covariantly under gauge transformations which motivates the name "covariant" which is used also in the Abelian case. The currents evaluated using perturbation theory are in general different, and will be denoted by $j^\mu$ and $j_5^\mu$ and named "consistent", because they are obtained from an effective action $\Gamma[A,A_5]$ and satisfy certain consistency conditions. We first analyze the simpler 2D case before turning to the 4D case which is of relevance for Weyl semi-metals in Sect. 6.3.

## D.1   The 2D case

The path integral method gives the covariant anomalies (6.2) when both vector and axial vector gauge fields are present (mixed anomalies),

$$\partial_\mu J^\mu = \frac{1}{\pi}F_5, \qquad\qquad \partial_\mu J_5^\mu = \frac{1}{\pi}F, \tag{D.1}$$

where we used the notation $F = \frac{1}{2}\varepsilon^{\mu\nu}F_{\mu\nu}$ and similarly for $F_5$. The spectral flow method yields the same results, see Sect. 6.2.

In Sect. 4, we discussed the perturbative calculation of the effective action $\Gamma[A, A_5]$ (4.4), leading by differentiation to the following currents,[73]

$$j^\mu = \Pi^{\mu\nu}(iA_\nu - \varepsilon_{\nu\alpha}A_5^\alpha) = \left(c\eta^{\mu\nu} - \frac{q^\mu q^\nu}{q^2}\right)(iA_\nu - \varepsilon_{\nu\alpha}A_5^\alpha)\,, \tag{D.2}$$

$$j_5^\mu = \varepsilon_{\mu\alpha}\Pi^{\alpha\nu}(A_\nu + i\varepsilon_{\nu\alpha}A_5^\alpha)\,. \tag{D.3}$$

As explained in the text, the value of the constant $c$ depends on the renormalization conditions we choose for the theory.

From the above currents, we get the anomalies,

$$\partial_\mu j^\mu = \frac{c-1}{\pi}(-F_5 + i\partial \cdot A)\,, \qquad \partial_\mu j_5^\mu = \frac{c}{\pi}(F - i\partial \cdot A_5)\,, \tag{D.4}$$

so for the particular values of $c$ picked in Sect. 4.1 we reproduce the results (4.7) and (4.8) when either $A$ or $A_5$ vanishes. In the presence of both backgrounds, the expressions (D.4) are not invariant under neither vector nor axial gauge transformations as opposed to the covariant anomalies (D.1) obtained from spectral flow or path integrals.

To understand what is going on we revisit the freedom to add counterterms to $\Gamma[A, A_5]$ [1]. These are local polynomials in fields and derivatives and in 2D we only have three possible terms with dimension equal to two that are compatible with Lorenz invariance: $A_\mu A^\mu$, $A_{5\mu}A_5^\mu$ and $\varepsilon_{\mu\nu}A^\mu A_5^\nu$. The first term has in fact already been used, and correspond to fixing the value of $c$. If we choose to maintain the vector gauge invariance we take $c = 1$ (corresponding to having a transverse $\Pi^{\mu\nu}$); we can furthermore eliminate the term $\partial_\mu A_5^\mu$ in (D.4) by adding the counterterm $iA_5^\nu A_{5\mu}/(2\pi)$, with the result,

$$\partial_\mu j^\mu = 0\,, \qquad \partial_\mu j_5^\mu = \frac{1}{\pi}F\,. \tag{D.5}$$

The only remaining freedom by the term $\frac{c'}{\pi}\varepsilon^{\mu\nu}A_\mu A_{5\nu}$ leads to the anomalies,

$$\partial_\mu j^\mu = \frac{c'}{\pi}F_5\,, \qquad \partial_\mu j_5^\mu = \frac{1-c'}{\pi}F\,. \tag{D.6}$$

Putting $c' = 1/2$ almost reproduces (D.1), but differs by a significant factor of 2. It is clear that no choice of counterterms will allow us to reproduce the path integral result. Therefore, in the mixed case, the "consistent" currents $j^\mu$ and $j_5^\mu$ that are obtained by the effective action are definitely different from the covariant currents $J^\mu$ and $J_5^\mu$.

## D.2   Gauge invariance of currents and Bardeen-Zumino terms

How to reconcile the results (D.1), and (D.5)? The first answer to this problem was given in the context of four-dimensional non-Abelian gauge theories by Bardeen and Zumino [60]. They noticed that the *currents* (not the action!) can be redefined by adding local terms. Let us begin by showing how this can be done and then explain the rationale behind the modifications.

The consistent vector current in (D.5) can be changed by adding a "Bardeen-Zumino term" $\Delta j_\mu = \varepsilon_{\mu\nu}A_5^\nu/\pi$ while keeping the axial current unchanged, as follows:

$$\tilde{j}^\mu = j^\mu + \frac{1}{\pi}\epsilon^{\mu\nu}A_{5\nu}\,, \qquad \text{implying} \qquad \partial_\mu\tilde{j}^\mu = \frac{1}{\pi}F_5\,, \tag{D.7}$$

$$\tilde{j}_5^\mu = j_5^\mu\,, \qquad \text{implying} \qquad \partial_\mu\tilde{j}_5^\mu = \frac{1}{\pi}F\,. \tag{D.8}$$

---

[73]Recalling that in Minkowski space the currents acquire a factor of $(i)$.

The new currents $\tilde{j}$ and $\tilde{j}_5$ have the same anomalies (D.1) that were obtained using spectral flow or path integrals and are identified with the covariant expressions , $\tilde{j}^\mu = J^\mu$ and $\tilde{j}_5^\mu = J_5^\mu$. One can convince oneself that the modifications (D.7) (D.8) cannot be obtained by adding counterterms to the effective action $\Gamma[A, A_5]$.

Let us now explicitly check the gauge invariance of the currents $j^\mu$ and $j_5^\mu$. The expressions (D.2), (D.3) with the choice of counterterms leading to the anomalies (D.5) become:

$$j^\mu = \frac{1}{\pi}\left(\eta^{\mu\nu} - \frac{q^\mu q^\nu}{q^2}\right)(iA_\nu - \varepsilon_{\nu\alpha}A_5^\alpha) , \tag{D.9}$$

$$j_5^\mu = \frac{1}{\pi}\varepsilon^{\mu\nu}A_\nu - \frac{1}{\pi}\varepsilon_{\mu\alpha}\frac{q^\mu q^\nu}{q^2}(A_\nu + i\varepsilon_{\nu\alpha}A_5^\alpha) . \tag{D.10}$$

Under gauge transformations $\delta_\lambda A_\mu = \partial_\mu\lambda$ and $\delta_\xi A_{5\mu} = \partial_\mu\xi$, their variations are:

$$\delta_\lambda j^\mu = \delta_\lambda j_5^\mu = \delta_\xi j_5^\mu = 0, \qquad \delta_\xi j^\mu = -\frac{1}{\pi}\varepsilon^{\mu\alpha}\partial_\alpha\xi \neq 0 . \tag{D.11}$$

Thus, the consistent current $j_\mu$ is not invariant under axial gauge transformations, but note that the Bardeen-Zumino term (D.7) is precisely the correction needed to restore the invariance. We have then shown that the so-called covariant currents $J_\mu$ and $J_{5\mu}$ are indeed gauge invariant, as advertised earlier.

We now discuss gauge invariance from the point of view of the effective action $\Gamma[A, A_5]$. The functional variations are defined by,

$$\delta_\lambda = \int dx\, \partial_\mu\lambda\frac{\delta}{\delta A_\mu} , \qquad \delta_\xi = \int dx\, \partial_\mu\xi\frac{\delta}{\delta A_{5\mu}} , \tag{D.12}$$

leading to,

$$G(\lambda) \equiv \delta_\lambda\Gamma = \int dx\, \partial_\mu\lambda\frac{\delta\Gamma}{\delta A_\mu} = -\int dx\, \lambda\, \partial_\mu j^\mu , \qquad G(\xi) \equiv \delta_\xi\Gamma = -\int dx\, \xi\, \partial_\mu j_5^\mu , \tag{D.13}$$

where we used the notation $G$ as in [1], and the currents $j^\mu$ and $j_5^\mu$ are those obtained from the perturbative calculation. Of course, in the presence of anomalies, the effective action is not invariant, but its variations obey the consistency conditions,

$$\delta_{\lambda_1}\delta_{\lambda_2}\Gamma - \delta_{\lambda_2}\delta_{\lambda_1}\Gamma = \delta_{[\lambda_1,\lambda_2]}\Gamma , \tag{D.14}$$

since they must obey the composition rules of the gauge group. Specializing to the axial case, the definitions (D.13) of the anomalies, directly yields,

$$\delta_{\lambda_1}G(\lambda_2) - \delta_{\lambda_2}G(\lambda_1) = G([\lambda_1,\lambda_2]), \tag{D.15}$$

which is the celebrated Wess-Zumino consistency condition [160].

In our Abelian case, the right-hand side of (D.15) vanishes, and the same holds for its axial counterpart. The left side also vanish since the anomalies (D.5) are expressed in terms of field strengths only, so the Wess-Zumino conditions are satisfied for both symmetries. Given that the anomalies obey the consistency conditions, the corresponding currents $j_\mu, j_{5\mu}$ are also called consistent.

We now reconsider the gauge invariance of the currents themselves. We have:

$$\delta_\lambda j^\mu(x) = \delta_\lambda\frac{\delta}{\delta A_\mu(x)}\Gamma = \frac{\delta}{\delta A_\mu(x)}\delta_\lambda\Gamma , \qquad \delta_\xi j^\mu(x) = \frac{\delta}{\delta A_\mu(x)}\delta_\xi\Gamma , \tag{D.16}$$

where we used that the two functional derivatives commute. Analogous expressions hold for the axial current. Upon evaluating the variation (D.16) using (D.13) together with the consistent anomalies (D.5), one reobtains the result (D.11) from the explicit calculations, showing that $j_\mu$ is not gauge invariant. This second argument for the gauge transformation of currents will be useful later in the 4D case. Note that this derivation was only possible since the consistent anomalies were derived from an effective action; it could not have been used for the covariant anomalies.

Summarizing the discussion so far, we have shown that there are two kinds of currents and anomalies in the mixed Abelian case: the covariant currents obey gauge invariance but cannot be obtained from an effective action, while the opposite holds for the consistent currents. These same features are found in non-Abelian gauge theories even in presence of a single background gauge field [60].

Below, we shall give another, geometrical, explanation for the difference between consistent and covariant currents and the origin of Bardeen-Zumino terms using the so-called Wess-Zumino-Witten action. This will also clarify why the covariant currents absorbs the anomaly inflow, while the consistent ones do not.

### D.3   The 4D case

We now extend the previous analysis to the 4D case. Recall the perturbative result (4.10) for the three-point function,

$$
\begin{aligned}
q_\mu \Pi^{\mu\nu\lambda} &= \frac{-i}{8\pi^2}(2c)\epsilon^{\nu\lambda\alpha\beta}k_\alpha p_\beta \,, \\
k_\nu \Pi^{\mu\nu\lambda} &= \frac{-i}{8\pi^2}(1-c)\epsilon^{\lambda\mu\alpha\beta}p_\alpha q_\beta \,, \\
p_\lambda \Pi^{\mu\nu\lambda} &= \frac{-i}{8\pi^2}(1-c)\epsilon^{\mu\nu\alpha\beta}q_\alpha k_\beta \,,
\end{aligned}
\tag{D.17}
$$

where the momentum $q^\mu$ flows into the vertex with the $\gamma^5$, and $c$ is the free parameter. In the main text, we took $c = 1$, since this enforces vector gauge invariance. Another possibility is $c = 1/3$ which treats the vertices symmetrically. For chiral fermions coupled to independent gauge fields $A_+$ and $A_-$, this choice is *required* by Bose symmetry for the currents to be obtained by taking functional derivatives of an effective action. Evaluating all the diagrams in Fig. 11, and recalling that $A_\pm = A \pm A_5$ so $J_\pm = \frac{1}{2}(J \pm J_5)$, the chiral 4D consistent anomalies become,

$$
\partial_\mu j_\pm^\mu = \pm \frac{1}{48\pi^2} F_\pm \tilde{F}_\pm \,,
\tag{D.18}
$$

with the notation $F\tilde{F} = F_{\mu\nu}\tilde{F}^{\mu\nu}$, or equivalently,

$$
\partial_\mu j^\mu = \frac{1}{12\pi^2} F_5 \tilde{F} \,, \qquad\qquad \partial_\mu j_5^\mu = \frac{1}{24\pi^2}(F\tilde{F} + F_5 \tilde{F}_5) \,.
\tag{D.19}
$$

As in the 2D case, we can modify anomalies by adding local terms to the effective action. Let us consider the expression,

$$
\Delta\Gamma = \frac{1}{6\pi^2} \int dx\, \epsilon^{\mu\nu\sigma\lambda} A_\mu A_{5\nu} \partial_\sigma A_\lambda \,,
\tag{D.20}
$$

that leads to the form,

$$
\partial_\mu j_5^\mu = \frac{1}{8\pi^2}(F\tilde{F} + \frac{1}{3}F_5 \tilde{F}_5) \,, \qquad\qquad \partial_\mu j^\mu = 0 \,,
\tag{D.21}
$$

so that the vector current is conserved.

### D.3.1 Relation between consistent and covariant anomalies in 4D

The expressions (D.18) and (D.19) should be compared with the covariant anomalies derived by the path-integral method,

$$\partial_\mu J_\pm^\mu = \pm \frac{1}{16\pi^2} F_\pm \tilde{F}_\pm \,, \tag{D.22}$$

or in terms of vector and axial currents,

$$\partial_\mu J^\mu = \frac{1}{4\pi^2} F_5 \tilde{F} \,, \qquad\qquad \partial_\mu J_5^\mu = \frac{1}{8\pi^2} (F\tilde{F} + F_5 \tilde{F}_5) \,. \tag{D.23}$$

The 4D consistent and covariant anomalies differ by a factor of 3 which is the counterpart of the factor of 2 that we encountered in the 2D case. This again shows that the covariant anomalies cannot be obtained by an effective action. For vanishing $A_5$, however, the equations (D.23) and (D.21) are identical, so the difference is only present in the mixed case, just as in 2D.

We now derive the gauge transformations of the consistent currents (D.21), using, *mutatis mutandis* the same method as in the 2D case. Again, the gauge transformations of the currents are obtained from the variations (D.16) with (D.13) evaluated with the consistent anomalies (D.21). We find the following violations of gauge invariance,

$$\delta_\lambda j_\mu = \delta_\lambda j_{5\mu} = 0 \,,$$
$$\delta_\xi j^\mu = -\frac{1}{4\pi^2} \varepsilon^{\mu\nu\rho\sigma} \partial_\nu \xi F_{\rho\sigma} \,, \tag{D.24}$$
$$\delta_\xi j_5^\mu = -\frac{1}{12\pi^2} \varepsilon^{\mu\nu\rho\sigma} \partial_\nu \xi F_{5\rho\sigma} \,.$$

From this it is not hard to see that we can make the currents gauge invariant by adding Bardeen-Zumino terms as,

$$\tilde{j}^\mu = j^\mu + \frac{1}{4\pi^2} \varepsilon^{\mu\nu\rho\sigma} A_{5\nu} F_{\rho\sigma} = J^\mu \,, \tag{D.25}$$

$$\tilde{j}_5^\mu = j_5^\mu + \frac{1}{12\pi^2} \varepsilon^{\mu\nu\rho\sigma} A_{5\nu} F_{5\rho\sigma} = J_5^\mu \,. \tag{D.26}$$

The anomalies of these currents agree with those of the currents $J_\mu$ and $J_{5\mu}$ in (D.23), so up to terms with zero divergence, we can identify $J_\mu \equiv \tilde{j}_\mu$ and $J_{5\mu} \equiv \tilde{j}_{5\mu}$, which proves that $J_{5\mu}$ and $J_\mu$ are indeed covariant. The relation (D.25) between the conserved consistent current $j_\mu$ and the non-conserved covariant one $J_\mu$ is used in the discussion of Weyl semi-metals in Sect. 6.3.2.

## D.4 Anomaly inflow and the Wess-Zumino-Witten action

In this section we discuss the relation between covariant and consistent currents and anomalies from a geometric perspective, closely following the exposition in Ref. [81].

The first observation is that the anomaly inflow provides a definition for the covariant anomalies. We first explain this point in the simpler case of a chiral two-dimensional theory at the edge of a quantum Hall liquid. In the main text, we showed that charge non-conservation in the chiral theory is compensated by a current flowing from an extra dimension, describing the attached bulk. The three-dimensional inflow current is obtained by varying the Chern-Simons action $S_{CS}[A]$ (3.42) which gives,

$$J_{(3D)}^\rho = \frac{\delta}{\delta A_\rho} S_{CS}[A] = \frac{1}{2\pi} \varepsilon^{\rho\mu\nu} \partial_\mu A_\nu, \qquad \mu, \nu, \rho = 1, 2, 3, \tag{D.27}$$

$$J_{(3D)}^3 = \partial_\alpha J_{(2D)}^\alpha = \frac{1}{2\pi} \varepsilon^{3\mu\nu} \partial_\mu A_\nu, \qquad \alpha = 1, 2 \,. \tag{D.28}$$

where $x^3$ is the coordinate perpendicular to the edge.

The equation (D.28) can be considered as a *definition* of the covariant current $J^\alpha_{(2D)}$ and its anomaly: since the bulk current $J^\rho_{(3D)}$, is itself invariant covariant, so is the corresponding edge current. We have thus shown that the covariant anomaly can also be derived from an effective action, albeit one that is defined in one extra dimension. This inflow argument extends to mixed anomalies in $3D$ topological insulators, as discussed in Sect. 6.2.

We now turn to the consistent anomaly. As stressed earlier, it can be derived from the effective action $\Gamma[A]$ defined in the ambient dimension which here is two. This action can be computed, *e.g.* by a perturbative expansion, but we now show that it can also be partially reconstructed from the knowledge of the anomaly using a universal method.

The idea is to integrate the infinitesimal gauge variation $\delta_\lambda \Gamma[A]$ in (D.13) over a finite range as follows:

$$S_{WZW}[A, \lambda] = \int_0^1 dt \int dx \, \frac{d\bar{\lambda}}{dt} \, \partial_\mu j^\mu[A^{\bar{\lambda}(t)}] = \Gamma[A] - \Gamma[A^\lambda], \qquad (D.29)$$

where, $\bar{\lambda}(t) = \bar{\lambda}(x, t)$ for $t \in [0, 1]$ is a parametric variation of the gauge transformation from $\bar{\lambda}(0) = 0$ to a finite value $\lambda(x) = \bar{\lambda}(x, 1)$, and the consistent anomaly in the integrand is evaluated on the gauge transformed background $A^{\bar{\lambda}}$ which equals $A_\mu + \partial_\mu \bar{\lambda}$ in the Abelian case.

The quantity $S_{WZW}[A, \lambda]$ is called the Wess-Zumino-Witten action; it is not itself the effective action but its gauge variation, so it depends on two fields. It is clear that the anomaly only allows a partial reconstruction of $\Gamma[A]$, whose full expression is usually unknown. Nonetheless, the part given by the Wess-Zumino-Witten action is exact and has a very general form, possessing a number of remarkable properties [160] [76]:

- $S_{WZW}[A, \lambda]$ obeys the Wess-Zumino consistency conditions and its variation gives, by definition, the consistent anomaly.

- $S_{WZW}[A, \lambda]$ can be written as a local expression in one more dimension, upon interpreting the parameter $t$ as the extra coordinate; however, any variation with respect to its variables $A$ and $\lambda$ turns out to be a total derivative and is thus a local expression in the original dimension.

- Owing to general properties of anomalies, the form of $S_{WZW}$ is known and given by,

$$S_{WZW}[A, \lambda] = S_{CS}[A^\lambda] - S_{CS}[A] . \qquad (D.30)$$

  In this relation, the Chern-Simons action is defined on a three-dimensional space with a boundary, on which the two-dimensional anomalous theory is defined.

In particle physics applications, the addition of one dimension is a technical trick, and the ambiguities in this extension are shown to be harmless due to the integer quantization of the Chern-Simons coupling constant [76]. In condensed matter systems such as the quantum Hall effect, the extra dimension is physical and the Chern-Simons action is implementing the anomaly inflow discussed earlier, so that the relation (D.30) simply expresses the overall gauge invariance.

Summarizing, we have established that the Chern-Simons theory gives the covariant anomaly and the Wess-Zumino-Witten action the consistent one, and we also found the relation (D.30) between the two actions. We then can explain the geometrical origin of the Bardeen-Zumino terms, relating the two currents. Before carrying a detailed analysis, let us anticipate the result: It turns out that while varying the Chern-Simons action to obtain the bulk current, there is a (often disregarded) boundary term. Once this is taken into account, it gives a correction to the boundary (consistent) current that exactly equals the Bardeen-Zumino term, thus matching the inflowing (covariant) current.

### D.4.1 The 2D case

We recall from Sect. 6.2 the Chern-Simons action for the anomaly inflow in the mixed case,

$$S_{CS}[A_+, A_-] = \frac{1}{4\pi} \int_{\mathcal{M}} A_+ dA_+ - A_- dA_- , \tag{D.31}$$

where we use chiral backgrounds $A_+$ and $A_-$, whose gauge variations are $A_\pm \to A_\pm + d\lambda_\pm$, and $\mathcal{M}$ is the three-dimensional geometry with the two-dimensional boundary $\partial\mathcal{M}$. According to (D.30), the corresponding Wess-Zumino-Witten action is:

$$S_{WZW}[A_\pm, \lambda_\pm] = \frac{1}{4\pi} \int_{\mathcal{M}} d\lambda_+ dA_+ - d\lambda_- dA_- = \frac{1}{4\pi} \int_{\partial\mathcal{M}} \lambda_+ dA_+ - \lambda_- dA_- . \tag{D.32}$$

Note that this action is explicitly two-dimensional, a simplification occurring in the Abelian case. This expression should be compared with the full effective action $\Gamma[A, A_5]$ in (4.4) and (4.6), which is nonlocal and only approximately known.

The gauge variation of $S_{WZW}$ gives the consistent anomalies,

$$\partial_\mu j^\mu_{(2D)\pm} = \pm \frac{1}{4\pi} \varepsilon^{\mu\nu} \partial_\mu A_{\pm\nu} , \tag{D.33}$$

in agreement with earlier findings (cf. (D.6) for $c' = 1/2$), and differ by a factor of 2 compared to the covariant anomalies (D.1).

Next we compute the bulk current by varying the Chern-Simons action (D.31), being careful with the boundary terms,

$$\delta S_{CS}[A_+, A_-] = \frac{1}{4\pi} \int_{\mathcal{M}} \delta A_+ dA_+ + A_+ d(\delta A_+) - (A_+ \leftrightarrow A_-) \tag{D.34}$$

$$= \frac{1}{2\pi} \int_{\mathcal{M}} \delta A_+ dA_+ + \frac{1}{4\pi} \int_{\partial\mathcal{M}} \delta A_+ A_+ - (A_+ \leftrightarrow A_-) , \tag{D.35}$$

where the corresponding terms with $A_+$ replaced by $A_-$ are not written. From the variation of the first and second term, we read the quantities,

$$J^3_{(3D)\pm} = \pm \frac{1}{2\pi} \varepsilon^{3\mu\nu} \partial_\mu A_{\pm\nu} = \partial_\alpha J^\alpha_{(2D)\pm} , \tag{D.36}$$

$$\Delta j^\alpha_{(2D)\pm} = \pm \frac{1}{4\pi} \varepsilon^{\alpha\beta} \partial_\alpha A_{\pm\beta} , \tag{D.37}$$

defined on $\mathcal{M}$ and $\partial\mathcal{M}$, respectively.

The expression (D.36) shows the inflow of the bulk current giving the expected form of covariant currents (D.1). The quantity (D.37) coming from the boundary part of (D.35) reproduces the Bardeen-Zumino terms that relate the consistent and covariant currents, namely $j^\mu_{(2D)\pm} + \Delta j^\mu_{(2D)\pm} = J^\mu_{(2D)\pm}$, providing the missing factor of 2 between the corresponding anomalies, respectively (D.33) (*i.e.* (D.6) with $c' = 1/2$) and (D.1).

We conclude that covariant anomalies are relevant when the two-dimensional theory is part of a higher dimensional system, as for edge states in topological systems, the covariant currents being unambiguously defined by the bulk theory.

For isolated two-dimensional theories there is an unresolved ambiguity in which kind of currents to use, and different methods (*e.g.* perturbative versus path-integral) may yield one kind of anomaly or the other, sometimes "adsorbing" the boundary term (D.37) (Bardeen-Zumino) coming from the "outside". Further physical input is necessary for choosing the proper currents.

### D.4.2 The 4D case

The previous analysis of Chern-Simons and Wess-Zumino-Witten actions directly extends to four dimensions. We first write the five-dimensional action that realizes the anomaly inflow as,

$$S_{CS}[A_+, A_-] = \frac{1}{24\pi^2} \int_{\mathcal{M}} A_+ dA_+ dA_+ - A_- dA_- dA_- , \tag{D.38}$$

where $\mathcal{M}$ is now a five-dimensional manifold with a four-dimensional boundary $\partial\mathcal{M}$, and varying this action gives,

$$
\begin{aligned}
\delta S_{CS}[A_+, A_-] &= \frac{1}{24\pi^2} \int_{\mathcal{M}} \delta A_+ dA_+ dA_+ + 2A_+ dA_+ d(\delta A_+) - (A_+ \leftrightarrow A_-) \tag{D.39} \\
&= \frac{1}{8\pi^2} \int_{\mathcal{M}} \delta A_+ dA_+ dA_+ + \frac{1}{12\pi^2} \int_{\partial\mathcal{M}} \delta A_+ A_+ dA_+ - (A_+ \leftrightarrow A_-) . \tag{D.40}
\end{aligned}
$$

The first term in this equation gives the bulk currents,

$$J^5_{(5D)\pm} = \pm \frac{1}{8\pi^2} \varepsilon^{5\mu\nu\rho\sigma} \partial_\mu A_{\pm\nu} \partial_\rho A_{\pm\sigma} = \partial_\alpha J^\alpha_{(4D)\pm}, \tag{D.41}$$

whose inflow match the covariant anomalies (D.22). The second, boundary part gives the following Bardeen-Zumino terms,

$$\Delta j^\mu_{(4D)\pm} = \pm \frac{1}{12\pi^2} \varepsilon^{\mu\nu\rho\sigma} A_{\pm\nu} \partial_\rho A_{\pm\sigma} . \tag{D.42}$$

The Wess-Zumino-Witten action is obtained by the gauge variation of (D.38), according to the relation (D.30), and reads,

$$S_{WZW}[A_\pm, \lambda_\pm] = \frac{1}{24\pi^2} \int_{\partial\mathcal{M}} \lambda_+ dA_+ dA_+ - \lambda_- dA_- dA_- . \tag{D.43}$$

This action in turns gives the consistent anomalies (D.18),

$$\partial_\mu j^\mu_{(4D)\pm} = \pm \frac{1}{24\pi} \varepsilon^{\mu\nu\rho\sigma} \partial_\mu A_{\pm\nu} \partial_\rho A_{\pm\sigma} , \tag{D.44}$$

that are off by a factor of three with respect to the covariant ones (D.41), in agreement with earlier analyses. The addition of the Bardeen-Zumino terms (D.42) establishes the map between the two kinds of currents, $j^\mu_{(4D)\pm} + \Delta j^\mu_{(4D)\pm} = J^\mu_{(4D)\pm}$, and similarly for the axial ones, thus connecting the anomalies (D.41) and (D.44). Note that the earlier Bardeen-Zumino terms (D.25), (D.26) are different from those in (D.42), because they related the covariant anomalies (D.41) to the other form (D.21) of the consistent anomalies, equivalent up to counterterms in $\Gamma[A, A_5]$.

In conclusion, the relation (D.30) between Wess-Zumino-Witten and Chern-Simons actions clarifies the geometrical meaning of the two types of anomalous currents and provides a general method for deriving Bardeen-Zumino terms.

# E Elements of differential geometry and curved space calculus

This appendix give a summary of the tools used for describing topological and geometrical properties of gauge and gravitational backgrounds. This is just a guide to more comprehensive and rigorous presentations that can be found in [42, 134, 161]. We start by introducing manifolds and differential forms, then discuss the metric structure, gauge fields and fermions on curved backgrounds. In the final part we derive some consequences of the trace anomaly in two-dimensional conformal field theory.

## E.1  Manifolds and differential forms

A manifold $\mathcal{M}$ is basically a $n$-dimensional surface that *locally* looks like flat space, namely $\mathbb{R}^n$. Around each point there is a well-defined neighborhood of other points, that can be described by an open set $U_{(i)}$ called patch, and mapped it into a open subset of $\mathbb{R}^n$. The map associates an element $P$ in the manifold to an element $\left(x^1(P), x^2(P), ..., x^n(P)\right)$ of $\mathbb{R}^n$ called the coordinates of $P$. As every neighborhood can be mapped into $\mathbb{R}^n$, we can carry over known tools of calculus.

Several open subsets are needed to cover $\mathcal{M}$ and they should be consistent on overlapping regions. Two patches $U_{(1)}, U_{(2)}$ give equivalent descriptions on their intersection if there exists a transition function $f_{(1,2)}$ between $x^\mu_{(1)}(P)$ and $x^\mu_{(2)}(P)$, *i.e.* $x^\mu_{(1)} = f_{(1,2)}(x^\mu_{(2)})$, corresponding to a change of coordinates on $\mathbb{R}^n$. The properties of these transitions functions specify the structures that are build on $\mathcal{M}$. The first one is the differentiable structure, the possibility to draw smooth curves on the manifold, define derivatives and the tangent space at each point $P$, as we shall see momentarily. For this to work, the transition functions $f_{(1,2)}$ should be differentiable functions. More structures will be needed later.

Derivatives on the manifold are defined through the tangent vector to a curve, specified by a function $x^\mu(\lambda)$ parameterized by $\lambda$. Therefore, a function $f(x^\mu)$ will depend implicitly on $\lambda$ along the curve and we can define the directional derivative,

$$\frac{d}{d\lambda} = \frac{\partial x^\mu}{\partial \lambda} \partial_\mu. \tag{E.1}$$

In this expression, $d/d\lambda$ is the natural definition of a vector on the manifold with basis spanned by $\partial_\mu$ and components $(\partial x^\mu / \partial \lambda)$ [42] .

We write a vector as $V = V^\mu \partial_\mu$, where derivatives are the basis elements and $V^\mu$ its components; the differential $df = \partial_\mu f \, dx^\mu$ is instead an element of the dual space with basis $dx^\mu$. Then, the directional derivative of a function $f$, $V[f] = V^\mu \partial_\mu f$, can be considered as an inner product. Vectors and dual vectors satisfy,

$$(dx^\mu, \partial_\nu) = \frac{\partial x^\mu}{\partial x^\nu} = \delta^\mu_\nu. \tag{E.2}$$

The differential $df = \partial_\mu f \, dx^\mu$ is also called a differential form of degree one, or 1-form. A generic form can be written as $\omega = \omega_\mu dx^\mu$. From the basis of vectors and forms one can build the $(n, p)$-rank tensors,

$$T = T^{\mu_1 ... \mu_n}_{\nu_1 ... \nu_p} \partial_{\mu_1} \cdots \partial_{\mu_n} dx^{\nu_1} \cdots dx^{\nu_p}. \tag{E.3}$$

The tangent space and its dual at one point are invariant under linear coordinate transformations of the group $GL(n, \mathbb{R})$. The tensors (E.3) carry representations of this group and in this context their lower (upper) indices are called covariant (contravariant).

A special class is given by the totally antisymmetric $(0, n)$-tensors that are associated to higher differential forms. Let us start with the definition of the exterior (wedge) product $\wedge$ of $n$ dual vectors as their totally antisymmetric tensor product $\otimes$,

$$dx^{\mu_1} \wedge dx^{\mu_2} \wedge \cdots \wedge dx^{\mu_n} = \sum_{\sigma \in \mathcal{S}_n} \text{sign}(\sigma) \, dx^{\mu_{\sigma(1)}} \otimes dx^{\mu_{\sigma(2)}} \otimes \cdots \otimes dx^{\mu_{\sigma(n)}}, \tag{E.4}$$

where the sum extends to permutations of $n$ elements and includes their sign. This product of differentials is just the $n$-dimensional volume element.

An important property of differential forms is that they can be used without introducing any metric on $\mathcal{M}$ and thus are suited for describing topological quantities that are metric independent, as encountered in the text. Examples will be analyzed later.

We define the 2-forms and $p$-forms as follows,

$$F = \frac{1}{2!}F_{\mu\nu}\,dx^\mu \wedge dx^\nu, \qquad\qquad H = \frac{1}{p!}H_{\mu_1\cdots\mu_p}\,dx^{\mu_1} \wedge \cdots \wedge dx^{\mu_p}. \qquad\text{(E.5)}$$

In a $D$-dimensional manifold, the form of degree $D$ has only one component, because the antisymmetric basis vector is unique up to permutations; this is called the top form. Forms of higher degree vanish.

The exterior product of forms follows from the properties of the basis (E.4): it is written $A \wedge B$, or simply $AB$. The components of the product of a $p$-form $A$ and a $q$-form $B$ are given by,

$$\frac{1}{p!q!}(A \wedge B)_{\mu_1\cdots\mu_{p+q}} = \frac{1}{(p+q)!}\left(A_{\mu_1\cdots\mu_p}B_{\mu_{p+1}\cdots\mu_{p+q}} \pm \text{permutations}\right). \qquad\text{(E.6)}$$

Note the property,

$$A \wedge B = (-1)^{p+q}B \wedge A. \qquad\text{(E.7)}$$

The exterior derivative $d$ of a $p$-form is defined by generalizing the earlier derivative of a function (0-form) $df = \partial_i f\,dx^i$. The operator $d$ is a map from a $p$-form $A$ to a $(p+1)$-form $dA$, defined by,

$$dA = \frac{1}{p!}\left(\partial_\nu A_{\mu_1\cdots\mu_p}\right)dx^\nu \wedge dx^{\mu_1} \cdots \wedge dx^{\mu_p}. \qquad\text{(E.8)}$$

The distribution rule for the exterior derivative is,

$$d(A \wedge B) = dA \wedge B + (-1)^p A \wedge dB, \qquad\text{(E.9)}$$

where $A$ is a $p$ form, that can be memorized by saying the $d$ is a 1-form to be carried close to the quantity on which it is acting.

## E.2   Metric, connection and curvature

We now define the notion of distance on the manifold. Given a point $x^\mu$ on $\mathcal{M}$, the distance to a close point $x^\mu + dx^\mu$ is given by $ds$, where,

$$ds^2 = g_{\mu\nu}(x)dx^\mu dx^\nu, \qquad\text{(E.10)}$$

with $g_{\mu\nu}(x)$ the metric tensor. As discussed several times in the text, metric backgrounds are not only relevant for general relativity but also for studying the response of quantum systems to strain, dislocations and disorder, to model thermal gradients and expose the effects of gravitational anomalies.

The metric provides the inner product (scalar product) of vectors

$$(U \cdot V) = g_{\mu\nu}U^\mu V^\nu, \qquad\text{(E.11)}$$

and, consequently, a way to upper and lower indices of components of vectors and tensors. If the metric has only positive eigenvalues it is called a Riemannian metric, while a pseudo-Riemannian metric has both positive and negative eigenvalues, as in the case of Lorentzian signature. The inner product (E.11) is positive definite for Riemannian metrics.

The tangent vectors to the curves on $\mathcal{M}$ passing through the point $x^\mu(P)$ form a vector space that is called the tangent space $T(P)$. In order to define derivatives of vectors, one needs a way to relate the tangent spaces at $x^\mu$ and $x^\mu + dx^\mu$, and transport vectors between

them. This is specified by the connection $\Gamma_{\mu\lambda}^{\nu}$, that gives the variation $\delta_\Gamma V^\nu = u^\mu \Gamma_{\mu\lambda}^{\nu} V^\lambda$ of the vector $V^\nu$ under the transport along a curve of tangent $u^\mu$ connecting the two points. This should be added to the flat-space variation to obtain the derivative on the manifold,

$$D_\mu V^\nu = \partial_\mu V^\nu + \Gamma_{\mu\lambda}^{\nu} V^\lambda \,. \tag{E.12}$$

A natural choice of connection $\Gamma$ is that realizing the parallel transport, under which the angle between the tangent to the curve $u$ and $V$ remains constant. Otherwise said, the parallel transport of the tangent vector $u^\alpha$ vanishes: this condition is equivalent to the geodesic equation for the metric,

$$u^\alpha D_\alpha u^\mu = 0, \qquad \frac{du^\mu}{d\tau} + \Gamma_{\nu\lambda}^{\mu} u^\nu u^\lambda = 0 \,, \tag{E.13}$$

where $u^\alpha = d\xi^\alpha/d\tau$ is the tangent to the curve $\xi^\alpha(\tau)$.

The parallel transport identifies the Levi-Civita connection: the corresponding derivative is called the covariant derivative, that is compatible with the metric because it obeys,

$$D_\mu g_{\nu\lambda}(x) = 0 \,. \tag{E.14}$$

This condition ensures that the parallel transport of the scalar product of two vectors vanishes, so that the covariant derivative reduces to the flat-space one,

$$D_\mu(A^\alpha g_{\alpha\beta} B^\beta) = \partial_\mu(A^\alpha g_{\alpha\beta} B^\beta) \,. \tag{E.15}$$

In order to prove this statement, one should use the relation $D_\mu(A^\alpha B_\alpha) = \partial_\mu(A^\alpha B_\alpha)$ to obtain the other covariant derivative $D_\mu V_\nu = \partial_\mu V_\nu - \Gamma_{\mu\nu}^{\lambda} V_\lambda$, apply it to (E.14) and solve for $\Gamma$ under the assumption of symmetric lower indices, $\Gamma_{\nu\lambda}^{\mu} = \Gamma_{\lambda\nu}^{\mu}$, as in (E.13). One obtains the explicit expression of $\Gamma$ in terms of the metric, the so-called Christoffel symbol,

$$\Gamma_{\mu\lambda}^{\nu} = \frac{1}{2} g^{\nu\rho} \left( \partial_\mu g_{\lambda\rho} + \partial_\lambda g_{\mu\rho} - \partial_\rho g_{\mu\lambda} \right) \,. \tag{E.16}$$

This expression checks (E.15). Note also that covariant derivatives obey the Leibniz rule, $D_\mu(A^\alpha B_\alpha) = (D_\mu A^\alpha) B_\alpha + A^\alpha(D_\mu B_\alpha)$.

We now consider the parallel transport on a sphere along parallels and meridians: starting from a point and moving in the two directions on the sides of a quadrangle reaches the opposite point with two different results. The infinitesimal difference of parallel transports on the two paths is given by the commutator of covariant derivatives,

$$\left[ D_\mu, D_\nu \right] V^\alpha = R_{\mu\nu,\ \beta}^{\ \ \alpha} V^\beta \,. \tag{E.17}$$

that defines the curvature, or Riemann tensor $R_{\mu\nu,\alpha\beta}$. Such difference is $\delta_R V^\alpha = \delta\Sigma^{\mu\nu} R_{\mu\nu,\ \beta}^{\ \ \alpha} V^\beta$, where $\delta\Sigma^{\mu\nu}$ is the infinitesimal square. One can show that $\delta_R V^\alpha$ is a rotation, thus the Riemann tensor is antisymmetric on the second pair of indices too.

We remark that the connection and curvature can be associated to matrix-valued differential forms, as follows,

$$\Gamma_{\lambda}^{\nu} = \Gamma_{\mu\lambda}^{\nu} dx^\mu, \qquad R_{\ \beta}^{\alpha} = \frac{1}{2} R_{\mu\nu,\ \beta}^{\ \ \alpha} \, dx^\mu \wedge dx^\nu = \left( d\Gamma + \Gamma^2 \right)_{\ \beta}^{\alpha} \,. \tag{E.18}$$

having used (E.17) to write the last expression for $R_{\ \beta}^{\alpha}$.

### E.3   Gauge fields

The wavefunction $\Psi(x)$ introduce an additional complex vector space at each point of the manifold. The same occurs when considering a charged quantum field $\phi$. This new vector space is called a "fiber" on the manifold and the collection of fibers is a fiber bundle.[74] The connection relating such vector spaces at different points is the gauge field $A_\mu(x)$ and the covariant derivative is defined according to the usual expressions. The correspondence with the gravitational connection (E.12) is better seen in the case of a non-Abelian gauge group $G$: the field $\phi^i$ transforms under a representation $\mathcal{R}(G)$ of the group, with $i = 1, \ldots, \dim \mathcal{R}(G)$, and the covariant derivative is,

$$(D_\mu \phi)^i = \partial_\mu \phi^i + i A_\mu^a (T_a)^i_{\ j} \phi^j, \qquad a = 1, \ldots, \dim G, \qquad (E.19)$$

where $(T_a)^i_{\ j}$ are generators obeying the Lie algebra of the group in that representation. These generators are usually omitted, to write $A_\mu^a (T_a)^i_{\ j} \equiv (A_\mu)^i_j$.

The curvature associate to the gauge connection is clearly the field strength, $F_{\mu\nu} = [D_\mu, D_\nu]/i$, in close analogy with (E.17). Note that this formula applies both to Abelian and non-Abelian fields.

### E.4   Reparameterization invariance and integrals of forms

The presence of a metric (E.10) introduces an invariance under general coordinate transformations, $x'^\mu = x'^\mu(x^\alpha)$, that is also called general covariance or diffeomorphism invariance. Gravitational forces possess this symmetry which can be viewed as a kind of gauge invariance. Effective field theories can naturally be generalized to curved space where they are written in covariant form, unless specified differently, as in the case of the elasticity effects discussed in Sect. 8.5.

The length element is invariant under reparameterizations,

$$ds^2 = g_{\mu\nu}(x) dx^\mu dx^\nu = \left( g_{\mu\nu}(x) \frac{dx^\mu}{dx'^{\mu'}} \frac{dx^\nu}{dx'^{\nu'}} \right) dx'^{\mu'} dx'^{\nu'} = g'_{\mu'\nu'}(x') dx'^{\mu'} dx'^{\nu'}, \quad (E.20)$$

and vectors transform accordingly, *e.g.* $V'^{\mu'}(x') = (dx'^{\mu'}/dx^\mu) V^\mu(x)$.

Reparameterization invariant actions involve covariant scalar quantities made up from the fields, the metric and their covariant derivatives. The integration measure is given by the invariant volume element,

$$dV = \sqrt{|g|} dx^1 \wedge dx^2 \wedge \cdots \wedge dx^D, \qquad (E.21)$$

where $g = \det(g_{\mu\nu})$, and under reparameterizations (E.20), the transformation of the determinant cancels the Jacobian factor from the volume. Another form of the same relation is given by the covariant totally antisymmetric tensor. For example in 2D,

$$\frac{1}{\sqrt{|g|}} \varepsilon^{\mu\nu} A_\mu B_\nu = \frac{1}{\sqrt{|g'|}} \varepsilon^{\mu'\nu'} A'_{\mu'} B'_{\nu'}. \qquad (E.22)$$

Using these results we can write for example the Maxwell action on a curved manifold as,

$$S_{YM} = -\frac{1}{4} \int_{\mathcal{M}} d^D x \sqrt{|g|} F_{\mu\nu} F_{\alpha\beta} g^{\mu\alpha} g^{\nu\beta}. \qquad (E.23)$$

---

[74]For a precise definition of fiber bundles, see [42].

In the case of the Chern-Simons action, we similarly have

$$S_{CS} = \frac{k}{4\pi} \int_{\mathcal{M}} d^3x \ \varepsilon^{\mu\nu\rho} A_\mu \partial_\nu A_\rho \ , \tag{E.24}$$

and remark that the $\sqrt{|g|}$ factors from the measure and the epsilon tensor cancel out.

This indicates that the Chern-Simons action can be naturally rewritten in terms of differential forms that are independent of the metric structure. The non-Abelian case can be discussed without much more effort. The gauge connection and field strength are associated to the following forms,

$$A = A_\mu dx^\mu, \qquad F = dA + A^2 = \frac{1}{2} F_{\mu\nu} dx^\mu \wedge dx^\nu \ , \tag{E.25}$$

that are matrix valued in a representation $\mathcal{R}(G)$, as in (E.19).

The Chern-Simons form is,

$$\omega_{CS} = AdA + \frac{2}{3} A^3, \tag{E.26}$$

and it obeys $d\mathrm{Tr}\,\omega_{CS} = \mathrm{Tr}\,F^2$ [42].

The integral of a differential form of maximal degree (top form) is given by the simple integration of its unique independent component. For the 3-form $\theta$ in three dimensions, for example,

$$\int_{\mathcal{M}} \theta = \int_{\mathcal{M}} \frac{1}{3!} \theta_{ijk} dx^i \wedge dx^j \wedge dx^k = \int_{\mathcal{M}} \frac{1}{3!} \varepsilon^{ijk} \theta_{ijk} dx^1 dx^2 dx^3 \ . \tag{E.27}$$

In the case of the Chern-Simons form (E.26), the integral is,

$$S_{CS} = \frac{k}{4\pi} \int \mathrm{Tr}\left( AdA + \frac{2}{3} A^3 \right) \ . \tag{E.28}$$

The gauge indices of the form are resolved by taking the trace over the gauge indices, obeying $\mathrm{Tr}(T^a T^b) = C(\mathcal{R})\delta^{ab}$, where the normalization factor[75] $C(\mathcal{R})$ depends on the representation. Note that (E.28) reproduces (E.24) in the Abelian case, where $A^3$ vanishes.

In $D$-dimensional space, the integration of a $k$-form on a surfaces with $k < D$, such as a two-sphere in $\mathbb{R}^3$, can be defined by first rewriting $\omega$ as a top form in $k$ dimensions using the parameterization of the surface and then applying the earlier definition (E.27) [42]. In the case of the sphere, the parameterization $x^\mu(z^\alpha)$, where $\mu = 1, 2, 3$ and $\alpha = 1, 2$, can be given by the polar coordinates $(\varphi, \theta)$. The two form $F$ can then be rewritten in term of surface differentials $(dz^1 = d\varphi, dz^2 = d\theta)$ as follow,

$$F = \frac{1}{2} F_{\mu\nu} \, dx^\mu \wedge dx^\nu = \frac{1}{2} \left( F_{\mu\nu} \frac{dx^\mu}{dz^\alpha} \frac{d^\nu}{dz^\beta} \right) dz^\alpha \wedge dz^\beta = \frac{1}{2} \hat{F}_{\alpha\beta} \, dz^\alpha \wedge dz^\beta \ . \tag{E.29}$$

In the last expression, $\hat{F}$ is a top form on the sphere and can be integrated using (E.27). Note that the metric is again not necessary.

It turns out that the Maxwell action can also be written in terms of differential forms, although the metric dependence is unavoidable. We introduce the Hodge dual of a $p$-form $\omega$ as the following $(D-p)$-form $^*\omega$,

$$^*\omega = \frac{\sqrt{|g|}}{p!(D-p)!} \omega_{\mu_1 \dots \mu_p} \varepsilon^{\mu_1 \dots \mu_p}{}_{\mu_{p+1} \dots \mu_d} dx^{\mu_{p+1}} \wedge \dots \wedge dx^{\mu_D} \ , \tag{E.30}$$

---

[75]The normalization of (E.28) is such that the action is gauge invariant for integer $k$ values [155]; for example, $C(\mathcal{R}) = 1/2$ for the fundamental representation of $SU(N)$.

where indices in epsilon tensor are raised with the help of the metric.

If two forms $A$ and $B$ have equal degree $p$, then $A \wedge {}^*B$ is a top form that can be integrated in $D$ dimensions. The integral defines the inner product of forms,

$$(A \cdot B) \equiv \int A \wedge {}^*B = \frac{1}{p!} \int A_{\mu_1 \cdots \mu_p} B^{\mu_1 \cdots \mu_p} \sqrt{|g|} d^D x, \qquad (\text{E.31})$$

that is symmetric and positive-define for Riemannian manifolds. In particular, the Yang-Mills action can be written,

$$S_{YM} = -\frac{1}{2} \int \text{Tr}[F \wedge {}^*F]. \qquad (\text{E.32})$$

The inner product (E.31) can be used to define the adjoint derivative $d^\dagger$ by $(A, dB) = (d^\dagger A, B)$; one finds the equivalent expression when acting on a $p$ form,

$$d^\dagger = (-1)^{np+n+\alpha} {}^*d{}^*, \qquad (\text{E.33})$$

where $\alpha$ is equal to 1(0) for Riemannian (Lorentzian) manifolds.

From $d$ and $d^\dagger$, we define the Laplacian acting on $p$ forms,

$$\Delta = \left(d + d^\dagger\right)^2 = dd^\dagger + d^\dagger d = (-1)^{np+n+\alpha} (d{}^*d{}^* + {}^*d{}^*d). \qquad (\text{E.34})$$

### E.5 The first Chern class and the monopole charge

A differential form $\omega$ is said closed if $d\omega = 0$, and exact if $\omega = d\Lambda$. Stokes theorem is particularly simple once expressed in terms of forms, where it reads,

$$\int_{\mathcal{M}} \omega = \int_{\partial \mathcal{M}} \Lambda, \qquad \text{for} \qquad \omega = d\Lambda. \qquad (\text{E.35})$$

Familiar expressions in $\mathbb{R}^3$ can be recovered by working out examples and using earlier definitions.

We now discuss some further mathematical aspects of topological invariant local functionals of the gauge field, like the first Chern class. This is the integral of the field strength two-form $F$ on a closed two-dimensional surface $\mathcal{M}$,

$$I[A] = \frac{1}{2\pi} \int_{\mathcal{M}} F = \frac{1}{4\pi} \int_{\mathcal{M}} \varepsilon^{\mu\nu} F_{\mu\nu} d^2 x = \frac{1}{2\pi} \int \vec{B} \cdot d\vec{S} = n \in \mathbb{Z}, \qquad (\text{E.36})$$

where $\vec{B}$ is the magnetic field orthogonal to the surface. Using $F = dA$ and the Stokes theorem (E.35), one would conclude that this integral vanishes since the surface has no boundary. However, the monopole field is singular, due to the Dirac string emanating from inside the surface and piercing it at one point. It follow that the relation $F = dA$ is not well defined on the whole surface and (E.35) cannot be applied immediately.

We consider the sphere and cover it with two patches, corresponding to the Northern and Southern hemispheres, respectively $D^{(+)}$ and $D^{(-)}$, that overlap at the equator $S^1 = D^{(+)} \cap D^{(-)}$. Let us consider two vector potentials for the monopole: $A^{(+)}$ is well defined in the upper hemisphere because the string is sent downward, and $A^{(-)}$ is regular in the lower patch, having moved the string upward. We then have well defined expressions: $F = dA^{(+)}$ in $D^{(+)}$ and $F = dA^{(-)}$ in $D^{(-)}$. At the equator, the two forms should give the same $F$, thus the two potentials differ by a gauge transformation, $A^{(+)} - A^{(-)} = d\chi$, as one can check from the explicit expressions [42].

The Stokes theorem (E.35) applied to $F$ defined in two patches gives,

$$\int_{\mathcal{M}} F = \int_{D^{(+)}} dA^{(+)} + \int_{D^{(-)}} dA^{(-)} = \int_{S^1} A^{(+)} - A^{(-)}$$

$$= \int_{S^1} d\chi = \chi(2\pi) - \chi(0) = 2\pi n, \tag{E.37}$$

having paid attention to boundary orientations. The result is nonvanishing because the gauge transformation $\Lambda(\varphi) = e^{i\chi(\varphi)}$ is nontrivial: it is a map from the equator, $\varphi \in S^1$, into the group, $\Lambda(\varphi) \in U(1) \equiv S^1$, that winds around that circle. Such maps fall into equivalence classes under smooth deformations that form the additive homotopy group $\Pi_1(U(1)) = \mathbb{Z}$. The representative element of the $n$-th class is the gauge transformation $U = e^{in\varphi}$ leading to monopole charge $n$ in (E.37).

In conclusion, the quantization of the monopole charge follows from the periodicity of the large $U(1)$ gauge transformation on the equator of the sphere: this is a well-defined transition function for the gauge field at patch overlaps. A similar argument was used in Sect. 2.4 for the monopole inside the cylinder $S^1 \times [-T, T]$: the field values at $\pm T$ where also found to differ by a large gauge transformation.

## E.6 Vielbeins, spin connection and fermions

The definition of spinors on manifolds requires additional elements to be introduced because they carry a double-valued representation of Lorentz transformations: a $2\pi$ rotation returns a minus sign. This feature is not present in tensors (E.3), that have integer spin only, thus one should introduce a new "basis of vectors" on which Lorentz transformations can act, and an associated connection for their parallel transport.[76] In so doing, we introduce additional degrees of freedom and a new gauge symmetry, that of local Lorentz transformations, as we shall see momentarily.

The new vectors $\hat{e}_a$ are linear combinations of the tangent vectors introduced at the beginning of this appendix,

$$\hat{e}_a = e_a^{\ \mu}(x)\partial_\mu, \tag{E.38}$$

where $\det(e_a^\mu) > 0$ for keeping orientation and $a = 1, \ldots, D$ is the index counting these vectors. We require that the new basis is orthonormal at any point $x^\mu$, as it were flat Euclidean (resp. Minkowskian) space,

$$(\hat{e}_a \cdot \hat{e}_b) = g_{\mu\nu} e_a^{\ \mu} e_b^{\ \nu} = \delta_{ab}, \qquad (\text{resp.} = \eta_{ab}). \tag{E.39}$$

It follows that the Lorentz group acts by, $e_a^\mu(x) \to e_b^\mu(x)\Lambda^b_{\ a}(x)$, where $\Lambda^b_{\ a}(x)$ is the local Lorentz transformation. The index $a$ is called a Lorentz index and denoted by Latin letters, while we keep Greek symbols for spacetime indices.

The matrices $e_a^\mu$ are called vielbeins, where *viel* comes from the German word for many; depending on dimension, they are also named zweibein for 2D and vierbein for 4D. Further names for $e_a^\mu$ are local frame vectors and, in 4D, tetrads. Their indices can be lowered and raised using the corresponding metrics. Note the relations,

$$e^a_{\ \nu} e_b^{\ \nu} = \delta^a_b, \qquad g_{\mu\nu} = e^a_{\ \mu} e^b_{\ \nu} \delta_{ab}, \tag{E.40}$$

showing that $e^a_{\ \mu}$ is the inverse[77] of $e_a^{\ \mu}$ and parameterizes the dual vectors, $\hat{\theta}^a = e^a_\mu dx^\mu$. Note that all tensors can be transformed into the Lorentz basis, *e.g.* $V^a = e^a_\mu V^\mu$.

---

[76]In this section, we follow closely the discussion of Sect. 12 in [161].

[77]Sometimes vielbein and its inverse are given different names to avoid confusion, as *e.g.* $(e^a_\mu, e_a^\mu) \to (e^a_\mu, E_a^\mu)$.

The parallel transport of Lorentz vectors is specified by the spin connection $\omega^a_{\mu b}$ by,

$$D_\mu V^a = \partial_\mu V^a + \omega^a_{\mu b} V^b \,. \tag{E.41}$$

The spin connection causes an infinitesimal Lorentz transformation on the index $a$, so it obeys $\omega_{\mu,ab} = -\omega_{\mu,ba}$. The corresponding one-form is $\omega^a_b = \omega^a_{\mu b} dx^\mu$.

The compatibility of the parallel transport with the metric is specified by the following condition,

$$D_\mu e^a_\nu = \partial_\mu e^a_\nu - \Gamma^\lambda_{\mu\nu} e^a_\lambda + \omega^a_{\mu b} e^b_\nu = 0 \,, \tag{E.42}$$

that extends the earlier condition (E.14).

The parallel transport of a Lorentz vector on a infinitesimal square determines again the Riemann curvature,

$$[D_\mu, D_\nu] V^a = R_{\mu\nu,}{}^a{}_b V^b \,, \tag{E.43}$$

that is now expressed in terms of the spin connection only,

$$R_{\mu\nu,}{}^a{}_b = \partial_\mu \omega^a_{\nu b} + \omega^a_{\mu c} \omega^c_{\nu b} - (\mu \leftrightarrow \nu) \,. \tag{E.44}$$

In analogy with (E.18), the curvature can be written as the Lorentz-valued two-form, $R^a_b = (d\omega + \omega^2)^a_b$.

Having established a local frame analogous to flat space, we can consider Dirac fields in it. The usual gamma matrices obey[78] $\left\{ \gamma^a, \gamma^b \right\} = 2\eta^{ab}$, where $a, b$ are the local Lorentz indices. The covariant derivative acting on spinors $\psi$ should be consistent with Lorentz transformations, thus it takes the form,

$$D_a \psi = e^\mu_a \left( \partial_\mu - i\omega^{bc}_\mu \sigma_{bc} \right), \tag{E.45}$$

where $\sigma_{ab} = i[\gamma_a, \gamma_b]/4$ is the generator of Lorentz transformations on spinors [1].

Thus the expression of the Dirac action in curved space is given by,

$$S[\psi, e^a_\mu] = \int_{\mathcal{M}} d^D x \, e \, \bar\psi \, (i\gamma^a D_a - m) \psi, \tag{E.46}$$

where $e = \det(e^a_\mu)$.

A few words of explanation are needed at this point. We started by explaining that the connection $\Gamma$ and the metric $g$ are independent concepts, but eventually the requirement that they are compatible among themselves implied the condition (E.14). This in turn lead to the expressions (E.16) and (E.18) respectively, for the connection $\Gamma = \Gamma(g)$ and the curvature $R(g)$ in terms of the metric. In this subsection, we introduced vielbeins $e$ and spin connection $\omega$, that are also independent quantities in general; similarly, the compatibility condition (E.42) can be shown to be sufficient to determine $\omega(e)$ and $R(e)$, given that $\Gamma(e)$ and $g(e)$ follow from (E.16) and (E.40) [161]. In particular, it is found that $\Gamma^\rho_{\mu\nu} = e^\rho_a \partial_\mu e^b_\nu + e^\rho_a \omega^a_{\mu b} e^b_\nu$.

It turns out that the curved space calculus in the $(e, \omega)$ variables, called Einstein-Cartan or first-order formulation, is more general and closer to gauge theory than the $(g, \Gamma)$ formulation. The introduction of Dirac fermions and corresponding localized spin sources also provides an extension of general relativity, leading to torsion $T^\lambda_{\mu\nu} = \Gamma^\lambda_{\mu\nu} - \Gamma^\lambda_{\nu\mu} \neq 0$, and effects not completely accounted by the metric. Actually, the transport of a vector around an infinitesimal square (E.17) results into a rotation and a translation, $\delta V^\alpha = \delta\Sigma^{\mu\nu}(R_{\mu\nu,}{}^\alpha{}_\beta V^\beta + T^\alpha_{\mu\nu})$, the latter being given by the torsion. This is the continuum formulation of a dislocation on the lattice. In

---

[78]We are assuming that the manifold is Minkowskian.

these lectures, we never discuss the effects of torsion and consider the metric and vielbein formulations as equivalent descriptions.

In absence of torsion, the Levi-Civita connection $\Gamma(g)$ is given by (E.16) and the spin connection $\omega(e)$ follows by solving (E.42) and the relation between $\Gamma$ and $\omega$ [161],

$$\omega_\mu^{ab} = \frac{1}{2} e^{\nu[a} \partial_{[\mu} e_{\nu]}^{b]} - \frac{1}{2} e^{\rho[a} e^{\sigma b]} e_{\mu c} \partial_\rho e_\sigma^c , \tag{E.47}$$

where square brackets means antisymmetrization of indices, *i.e.* $A_{[\alpha} B_{\beta]} = A_\alpha B_\beta - A_\beta B_\alpha$.

## E.7 The example of the sphere

To make the previous discussion more concrete, we derive the geometric properties of a familiar manifold. Consider a sphere in $\mathbb{R}^3$: its metric can be obtained by the parameterization of the surface given by the polar coordinates, $x^i(z^\alpha) = (r \sin\theta \cos\phi, r \sin\theta \sin\phi, r \cos\theta)$, where $i = 1, 2, 3$, $\alpha = 1, 2$ and $z^\alpha = (\theta, \phi)$, as follows,

$$\begin{aligned}
ds^2 &= \delta_{ij} dx^i dx^j = \delta_{ij} \frac{dx^i}{dz^\alpha} \frac{dx^j}{dz^\beta} dz^\alpha dz^\beta \\
&= g_{\alpha\beta}(\theta, \phi) \, dz^\alpha dz^\beta = r^2 d\theta^2 + r^2 \sin^2\theta d\phi^2 , \qquad \alpha, \beta = \theta, \phi .
\end{aligned} \tag{E.48}$$

Using (E.39) and (E.40), we can obtain the following expression of the zweibeins, after choosing a gauge condition for the Lorentz frame, $a, b = 1, 2$,

$$\begin{aligned}
g_{\alpha\beta} &= e_\alpha^a \delta_{ab} e_\beta^b , \\
e_\theta^1 &= e_{\theta 1} = \left(e^{\theta 1}\right)^{-1} = r , \qquad e_\phi^2 = \left(e^{\phi 2}\right)^{-1} = r \sin\theta ,
\end{aligned} \tag{E.49}$$

with the other components being equal to zero.

Using (E.47) and $\partial_\gamma e_\beta^a = r \cos\theta \delta^{a2} \delta_{\gamma\theta} \delta_{\beta\phi}$, we obtain that $\omega_\gamma^{ab} = -\epsilon^{ab} \cos\theta \delta_{\gamma\phi}$. From (E.44) follows the Riemann tensor,[79] $R_{\gamma\beta,ab} = \sin\theta \epsilon_{\gamma\beta} \epsilon_{ab}$. Finally, one finds the scalar curvature of the sphere

$$\mathcal{R} = R_{\gamma\beta,ab} e^{\gamma a} e^{\beta b} = \frac{2}{r^2}. \tag{E.50}$$

## E.8 Trace anomaly in conformal field theory

The constant $c$ appearing in the trace anomaly,

$$D^\mu T_{\mu\nu} = 0 , \qquad T_\mu^\mu = -\frac{c}{24\pi} \mathcal{R} , \tag{E.51}$$

is very important in conformal theory because it parameterizes several physically relevant quantities. Since the anomaly manifests itself on curved metric backgrounds, it is convenient to relate it to observables that have a non-vanishing flat space limit, $g_{\mu\nu} \to \delta_{\mu\nu}$. Let us write the path-integral expression of an observable $\mathcal{O}$,

$$\langle \mathcal{O}(x) \rangle_g = \frac{1}{Z} \int \mathcal{D}\phi \, e^{-S[\phi, g]} \, \mathcal{O} , \tag{E.52}$$

where the integration is carried out over some fields $\phi$ in the presence of the background denoted by $g$, and where $Z$ is the path integral without the insertion of $\mathcal{O}$. Under an infinitesimal change of the metric, we have the identity:

$$\frac{\delta}{\delta g^{\mu\nu}(y)} \langle \mathcal{O}(x) \rangle_g = \frac{\sqrt{g}}{2} \langle \mathcal{O}(x) T_{\mu\nu}(y) \rangle_g , \tag{E.53}$$

---

[79]Notice that the antisymmetric symbol is $\epsilon_{\gamma\beta} = \delta_{\gamma\theta} \delta_{\beta\phi} - \delta_{\gamma\phi} \delta_{\beta\theta}$.

where the insertion of the stress tensor follows from the variation of the action inside the path integral using the definition (3.19).

We consider (E.53) for $\mathcal{O} = T_\mu^\mu$ and evaluate the metric variation of the trace anomaly (E.51) on the left-hand side by using the formula $\delta \mathcal{R} = \left( g_{\mu\nu} D^2 - D_\mu D_\nu \right) \delta g^{\mu\nu}$ (See [162], appendix B). Taking the limit to flat space, we get:

$$\langle T_\alpha^\alpha(x) T_{\mu\nu}(y) \rangle = -\frac{c}{12\pi} \left( \partial_\mu \partial_\nu - \delta_{\mu\nu} \partial_\rho^2 \right) \delta^{(2)}(x-y) , \tag{E.54}$$

where $\partial_\alpha = \partial / \partial x^\alpha$ and $\delta^{(2)}(x)$ is the two-dimensional Dirac delta function. The trace anomaly partially determines the stress-tensor two-point function in Euclidean space. Note also that the flat-space limit is non-vanishing because the scalar curvature $\mathcal{R}$ is approximately linear in the metric. In four dimensions the trace anomaly is quadratic in the curvature for dimensional reasons, and one would need two derivatives in (E.53) for getting a non-vanishing limit, leading to informations on the stress-tensor three-point function [163].

Two-dimensional conformal invariance determines the stress tensor two-point function up to a constant [164] by,

$$\langle T_{\alpha\beta}(x) T_{\mu\nu}(y) \rangle = \lambda \left( \partial_\alpha \partial_\beta - \delta_{\alpha\beta} \partial_\rho^2 \right) \left( \partial_\mu \partial_\nu - \delta_{\mu\nu} \partial_\rho^2 \right) G(x-y) , \tag{E.55}$$

where $G(x-y) = -\left[ \log(x-y)^2 \right]/(4\pi)$ is the two-dimensional massless scalar Green function obeying $\partial_\rho^2 G(x-y) = \delta^{(2)}(x-y)$. Note that (E.55) has the correct scale dimension 4 and the index structure is appropriate for enforcing transversality due to $T_{\mu\nu}$ conservation (E.51). The comparison between (E.55) and (E.54) gives $\lambda = c/(48\pi)$.

The correlator (E.55) is only non-vanishing for $x \neq y$ for the traceless components of the stress tensor, that are given by $T_{zz}$ and $T_{\bar{z}\bar{z}}$, using the complex notation introduced in Sect. 8.2. One finds,

$$\langle T_{zz}(z) T_{zz}(0) \rangle = -\frac{c}{12} \partial_z^4 \log(z\bar{z}) = \frac{c}{2z^4} , \qquad \langle T_{\bar{z}\bar{z}}(z) T_{\bar{z}\bar{z}}(0) \rangle = \frac{c}{2\bar{z}^4} , \tag{E.56}$$

where we also rescaled $T_{\mu\nu} \to 2\pi T_{\mu\nu}$ as is customary in the CFT literature.

We can now use the result (E.56) for checking the Virasoro algebra (3.63). We need the expression of the Virasoro generators $L_n$ as Fourier modes of the stress tensor on the unit circle $z = \exp(i\theta)$ [7], as well as the ground-state conditions (3.28), respectively given by:

$$T_{zz}(z) = \sum_{-\infty}^{\infty} \frac{L_n}{z^{n+2}} , \qquad L_n |\Omega\rangle = \langle \Omega | L_{-n} = 0, \quad n = -1, 0, 1, \dots . \tag{E.57}$$

Note that the $L_n$ generators were introduced earlier in (3.24) and (3.60) as the Fourier modes of the Hamiltonian density $\mathcal{H} = T_{zz} + T_{\bar{z}\bar{z}}$, for chiral theories where actually $T_{\bar{z}\bar{z}}$ vanished.

Another derivation of the stress tensor two-point function is obtained by using the Virasoro algebra (3.63) together with the properties (E.57), as follows,

$$\langle \Omega | T_{zz}(z) T_{zz}(0) |\Omega\rangle = \lim_{w \to 0} \sum_{n,m=2}^{\infty} \langle \Omega | [L_n, L_{-m}] |\Omega\rangle \frac{w^{m-2}}{z^{n+2}} = \langle \Omega | [L_2, L_{-2}] |\Omega\rangle \frac{1}{z^4} = \frac{c}{2z^4} . \tag{E.58}$$

This expression matches the form (E.56) given by the trace anomaly. This completes the proof that the Virasoro central charge $c$ and the coefficient of the trace anomaly are equal, as claimed in Sect. 3.3

Another interesting property parameterized by the central charge is the anomalous transformation law of the stress tensor under conformal transformations. This result is used in the derivation of the thermal Hall current in Sect. 3.9. Let us briefly discuss this point.

As mentioned in Sect. 3.3, analytic coordinate transformations in two dimensions change the metric by an overall (conformal) factor,

$$z = z(w), \qquad ds^2 = dz\,d\bar{z} = \left|\frac{dz}{dw}\right|^2 dw\,d\bar{w} = 2g_{w\bar{w}}\,dw\,d\bar{w}, \qquad (E.59)$$

namely $g_{\mu\nu}(w) = |dz/dw|^2 g_{\mu\nu}(z)$.

The transformation of the traceless part of stress tensor found to be

$$T_{ww}(w)\,dw^2 = T_{zz}(z)\,dz^2 + \frac{c}{12}\left[\frac{z'''(w)}{z'(w)} - \frac{3}{2}\left(\frac{z''}{z'}\right)^2\right]dw^2\,, \qquad (E.60)$$

(a corresponding expression for $T_{\bar{z}\bar{z}}$ involves $\bar{c}$). One recognizes the usual covariant transformation of a two-index tensor and the additional term given by the central charge times the quantity between square brackets called Schwarzian derivative $\{z, w\}$. The derivation of (E.60) is rather technical and can be found in Sect. 4 of Ref. [162]: in short, it is based on the calculation of the Wess-Zumino-Witten action by integration of the trace anomaly (E.51), using the analogue of (D.29). Upon a metric variation, this action then gives the full expression of the stress tensor, not only its trace part, more precisely its variation between the two geometries (E.60) related by the conformal map $z = z(w)$.

The result (E.60) is used in Sect. 3.9 to obtain the stress tensor in the cylinder geometry with periodic time representing the thermal state. As described in [36], the map between the complex $z$-plane and the $w$-cylinder is:

$$z(w) = \exp\left[\frac{i2\pi w}{v\beta}\right], \qquad\qquad w = v\tau + ix\,, \qquad (E.61)$$

where the Euclidean time $\tau$ has period $\beta = 1/k_B T$, related to temperature $T$, and $v$ is the Fermi velocity. Insertion of the map (E.61) into (E.60) yields the ground-state value,

$$\langle T_{ww}(w)\rangle_{\text{cyl}} = \langle T_{zz}(z)\rangle_{\text{plane}} + \frac{\pi^2 c}{6v^2\beta^2} = \frac{\pi^2 c}{6v^2\beta^2}\,, \qquad (E.62)$$

being normal-ordered to zero in the plane. This result is used to compute the thermal current and energy (3.80) given in the text (after restating the $1/2\pi$ factor and suitable powers of the Fermi velocity).

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
