# Peer review of "Quantum Field Theory Anomalies in Condensed Matter Physics"

_SciPost Physics Lecture Notes_

## Round 1 · Referee Report · Anonymous (Referee 1) · 2022-6-20

Strengths

1) Very timely 2) Covers almost all aspects 3) tries to be as pedagogical as possible 4) despite 3 aims at sufficient rigor

Weaknesses

1) at times a bit terse, understanding of some passages might depend on prior knowledge. However, it must be said that due to the wealth of material covered this seems almost unavoidable (unless one is willing to write a book of 500+ pages)

Report

These lecture notes on quantum field theory anomalies in condensed matter physics are very timely. I enjoyed reading (and learning from) them very much!
I think they will become an indispensable resource for theoretically inclined students and researchers in condensed matter physics. I strongly recommend to publish them.

That said, I found a few details in the manuscript that probably can be addressed
or even improved.

Requested changes

1) On page 5 in the 4th paragraph the authors write that the low energy response is "geometric", I wonder if this is the right word. After all also the dissipative thermal conductivity can be seen as a "geometric" response following Luttinger. The distinguishing feature is that it is robust (insensitive to microscopic details). So probably "topological" is a better word?

2) I find the discussion of the gauge field on page 9 confusing. Is it really true that after fixing $A_0 =0$ one can assume $A_x$ to be constant in space? As the authors emphasize the gauge fields are external and do not have to obey any equations of motion (except the topological Bianchi identity, which is trivial in D=2). Why does one have to chose a gauge at all if only the fermions are quantized here?

3) In the footnote 3 the authors state that anti-periodic boundary conditions form fermions are standard. I think one could chose either if one works strictly on a circle. Anti-periodic boundary conditions seem most natural if the circle is the boundary of a disc.

4) Something went wrong in the normalization in (2.29)? It seems to contradict (2.28).

5) In (3.35) the eigenvalue of the charge operator should be "n".

6) Chapter 6: I think there are some misprints: -) below equation (6.13) the axial vector (gauge) field should be $(b_0, \vec b)$ -) The normalization of (6.13) should be $\frac{1}{2\pi^2}$ (also (6.16) and (6.17) )

7) Section 6.4 on the Chiral Magnetic Effect: I think the discussion suffers from the problem that the authors only write the contribution due to the Bardeen-Zumino term. But also the covariant current contributes (that is really the non-trivial term). What happens is that the two contributions exactly cancel if $A_{5,0} = \mu_5$. In fact there is a theorem (due to Bloch apparently) stating that the electric (DC) current must vanish in equilibrium. (a modern field theoretic formulation is Phys.Rev.D 92 (2015) 8, 085011.) This cancellation is discussed in ref. [66]. Since this led to some confusion in the early discussions of the effect in Weyl semimetals it would be good to mention this.

8) Probably the authors can expand a bit what the dots mean in equation (7.13). I understand that this is an introductory lecture on Index theorems but still a few words on what the dots are should be added.

9) In the discussion in the first paragraph on page 59 I would suggest to add a citation to the paper by Kimura Prog.Theor.Phys. 42 (1969) 1191-1205 which appeared already in 1969 and computes the gravitational contribution to the axial anomaly. This is remarkable and the paper has much less citations than it deserves.

10) In the discussion of the 4D topological insulator the axial anomaly in Euclidean space is used in (9.3) and (9.4). It is stated that the partition function is a phase and therefore we have the $2\pi$ periodicity in the theta-angle. But then the authors discuss Time Reversal symmetry. It would be good if the authors could clarify what is meant here by time reversal when one is implicitly working in Euclidean signature.

  • validity: top
  • significance: top
  • originality: top
  • clarity: high
  • formatting: perfect
  • grammar: excellent

Author:  Rodrigo Arouca  on 2022-08-03  [id 2707]

(in reply to Report 1 on 2022-06-20)

We thank the referees for the throughout analysis of our lectures and their comments and suggestions. Please find our replies and corresponding changes in the attached file "refs_reply.pdf".

Attachment:

refs_reply_ZEUUIKL.pdf

---

## Round 1 · Referee Report · Anonymous (Referee 2) · 2022-7-12

Strengths

1- Pedagogical introduction to the topic. 2- Covers a broad range of phenomena, including often neglected aspects of anomalies. 3- Aims to offer different perspectives on each phenomenon

Weaknesses

1- At times a bit convoluted: Since the authors want to illuminate several phenomena from different points of view, the aim of some sections is not always immediately obvious. 2- Internal references to topics discussed later are at times missing, cf. point "1" of the requested changes

Report

In these lecture notes, the authors discuss a wide range of anomalies in quantum field theory and their relevance in condensed-matter physics. The authors aim for a pedagogical introduction to this topic. Starting with the axial anomaly in 1+1d, they first provide an intuitive explanation for the axial anomaly before introducing the condensed matter realization, and two complementary points of view from quantum field theory: perturbation theory (including the triangle diagram) and a Fujikawa-style path integral approach. The authors also discuss gravitational anomalies, SPT phases, global anomalies and their consequences for interacting systems.

These lecture notes are well-written, comprehensive, on a high technical level and (in my eyes) pedagogical. They do not only provide an excellent introduction to the topic, but also a valuable reference for readers already familiar with some of the topics discussed here, such as me. Thanks to this excellent summary of the field’s state of the art, I have learned more about the connection to fractional quantum Hall states (Sec. 3.5), gravitational anomalies (Sec. 8), and the connection to interacting topological superconductors (Sec. 10.4).

Thanks to consistent progress in the discovery of topological materials and efforts to identify signatures that originate in their topological nature, the subject covered in these notes is certainly of ongoing interest to the research community. Apart from minor points that I suggest to change below, these notes are a correct, systematic and intelligible presentation of anomalies and their consequences in condensed matter physics.

Requested changes

I do not have any objections against publication in SciPost Lecture Notes, but only a few additional remarks the authors might want to consider.

1- At times, the authors could be more concrete when referring to methods and results presented later in this manuscript. For example, the divergent term in Eq. (2.16) is simply subtracted without any explanation. I suppose this becomes more obvious when regularizing the integral, which is done much later in the manuscript. Similarly, on page 25, the authors promise to return to the anomaly inflow “at several occasions in the context of the so called bulk-boundary correspondence in topological states of matter.” 2- In Sec. 2.2, the authors write that A5 is a “non-dynamical ‘emergent gauge potential’,” which does not need to be conserved. Maybe it would be good to specify at this point that it can often (generally?) be directly measured, e.g., as the Hall conductance in Weyl semimetals, i.e., its value has physical consequences, and it is thus fundamentally different from a gauge potential. 3- In Eq. (6.10), the role of the density ρ is not clear to me: It is a first-quantized Hamiltonian, so the density should not appear here. 4- In Eq. (6.12) and above, it would make sense to switch to a vector notation of the Berry curvature. In the 2D example, the integral over the field strength is of course identical to the surface integral over the vector field, however, is it only the vector field integration that generalizes to the 3D case. 5- The authors emphasize that this paper “is not a review paper, but a set of lecture notes covering a wide range of topics” and that they thus “have not attempted to provide a comprehensive bibliography,” which is fine. Nevertheless, I have two suggestions for additional papers the authors might want to cite: [A] and Ref. [70] by Pikulin and et al. appeared simultaneously and discuss similar ideas. [B] covers most aspects of Sec. 9.2.1 and could be maybe introduced as an additional reference for interested readers.

[A] Adolfo G. Grushin, Jörn W. F. Venderbos, Ashvin Vishwanath, and Roni Ilan Phys. Rev. X 6, 041046 (doi:10.1103/PhysRevX.6.041046) [B] Pavan Hosur, Shinsei Ryu, and Ashvin Vishwanath Phys. Rev. B 81, 045120 (doi:10.1103/PhysRevB.81.045120)

  • validity: top
  • significance: high
  • originality: ok
  • clarity: high
  • formatting: perfect
  • grammar: excellent

Author:  Rodrigo Arouca  on 2022-08-03  [id 2706]

(in reply to Report 2 on 2022-07-12)

We thank the referees for the throughout analysis of our lectures and their comments and suggestions. Please find our replies and corresponding changes in the attached file "refs_reply.pdf".

Attachment:

refs_reply.pdf

---

## Editorial Decision

resubmitted